# An $\varepsilon$-Best-Arm Identification Algorithm
# for Fixed-Confidence and Beyond

**Marc Jourdan**
marc.jourdan@inria.fr

**Rémy Degenne**
remy.degenne@inria.fr

**Emilie Kaufmann**
emilie.kaufmann@univ-lille.fr

Univ. Lille, CNRS, Inria, Centrale Lille, UMR 9189-CRIStAL, F-59000 Lille, France

## Abstract

We propose EB-TC$_\varepsilon$, a novel sampling rule for $\varepsilon$-best arm identification in stochastic bandits. It is the first instance of Top Two algorithm analyzed for approximate best arm identification. EB-TC$_\varepsilon$ is an *anytime* sampling rule that can therefore be employed without modification for fixed confidence or fixed budget identification (without prior knowledge of the budget). We provide three types of theoretical guarantees for EB-TC$_\varepsilon$. First, we prove bounds on its expected sample complexity in the fixed confidence setting, notably showing its asymptotic optimality in combination with an adaptive tuning of its exploration parameter. We complement these findings with upper bounds on its probability of error at any time and for any error parameter, which further yield upper bounds on its simple regret at any time. Finally, we show through numerical simulations that EB-TC$_\varepsilon$ performs favorably compared to existing algorithms, in different settings.

## 1 Introduction

In pure exploration problems, the goal is to answer a question about a set of unknown distributions (modelling for example the efficacy of a treatment) from which we can collect samples (measure its effect), and to provide guarantees on the candidate answer. Practitioners might have different pre-defined constraints, e.g. the maximal budget might be fixed in advance or the error made should be smaller than a fixed admissible error. However, in many cases, fixing such constraints in advance can be challenging since a "good" choice typically depends on unknown quantities. Moreover, while the budget is limited in clinical trials, it is often not fixed beforehand. The physicians can decide to stop earlier or might obtain additional fundings for their experiments. In light of those real-world constraints, regardless of its primal objective any strategy for choosing the next treatment should ideally come with guarantees on its current candidate answer that hold at any time.

We formalize our investigations in the well-studied stochastic bandit model [4, 29], in which a learner interacts sequentially with an environment composed of $K \in \mathbb{N}$ arms, which are unknown distributions $(\nu_i)_{i \in [K]}$ with finite means $(\mu_i)_{i \in [K]}$. At each stage $n \in \mathbb{N}$, the learner chooses an arm $I_n \in [K]$ based on the samples previously observed and receives a sample $X_{n,I_n}$, random variable with conditional distribution $\nu_{I_n}$ given $I_n$. It then proceeds to the next stage. An algorithm for the learner in this interaction is specified by a *sampling rule*, a procedure that determines $I_n$ based on previously observed samples. Formally, the sampling rule defines for all $n \in \mathbb{N}$ a function from $([K] \times \mathbb{R})^{n-1}$ to the probability distribution on $[K]$, which is measurable with respect to the $\sigma$-algebra $\mathcal{F}_n := \sigma(\{I_t, X_{t,I_t}\}_{t \in [n-1]})$. We call that $\sigma$-algebra *history* before $n$.

**Identification tasks**     We focus on best arm identification (BAI). In that task, the goal of the algorithm is to find which of the arms has the largest mean, and to do so with a small probability of error, as

37th Conference on Neural Information Processing Systems (NeurIPS 2023).

quickly as possible. If several arms have means very close to the maximum, finding the one with the highest mean might be difficult. However in practice we are often satisfied by any good enough arm, in the sense that its mean is greater than $\mu_\star - \varepsilon$, where $\mu_\star = \max_{i \in [K]} \mu_i$. This is the $\varepsilon$-BAI task. Our results can also be adapted to the multiplicative $\varepsilon$-BAI task, in which all means are non-negative and we want to find an arm with mean $\mu_i \geq (1 - \varepsilon)\mu_\star$ [19] (see Appendix I for details).

Now that we have a (for now informal) goal, we need to complement the sampling rule with a recommendation rule that specifies which arm is the candidate returned by the algorithm for the best arm. We follow [5] and define that rule for all stages: for all $n \in \mathbb{N}$, we denote by $\hat{\imath}_n$ this $\mathcal{F}_n$-measurable function from $([K] \times \mathbb{R})^{n-1}$ to $[K]$. We call *algorithm* the combination of a sampling and a recommendation rule.

**Performance criteria**   There are several ways to evaluate the performance of an algorithm for $\varepsilon$-BAI. Let $\mathcal{I}_\varepsilon(\mu) = \{i \in [K] \mid \mu_i \geq \mu_\star - \varepsilon\}$ be the set of $\varepsilon$-good arms. The *probability of $\varepsilon$-error* or the recommendation at $n$ is defined as $\mathbb{P}_\nu(\hat{\imath}_n \notin \mathcal{I}_\varepsilon(\mu))$. Introduced in [1], the expected *simple regret* is defined as $\mathbb{E}_\nu[\mu_\star - \mu_{\hat{\imath}_n}]$, and is independent of any parameter $\varepsilon$. Based on those notions, several setting are studied in the bandit identification literature.

- Fixed confidence: we augment the algorithm with a *stopping rule*, a stopping time $\tau_{\varepsilon,\delta}$ with respect to the history of samples and we impose that the algorithm should be $(\varepsilon, \delta)$-PAC. That is, its probability of $\varepsilon$-error at $\tau_{\varepsilon,\delta}$ must satisfy $\mathbb{P}_\nu(\tau_{\varepsilon,\delta} < +\infty, \hat{\imath}_{\tau_{\varepsilon,\delta}} \notin \mathcal{I}_\varepsilon(\mu)) \leq \delta$. The parameter $\delta$ is known to the algorithm. An algorithm is judged based on its expected sample complexity $\mathbb{E}[\tau_{\varepsilon,\delta}]$, the expected number of samples it needs to collect before it can stop and return a good arm with the required confidence.

- Fixed budget: we run the algorithm until a predefined time $T$ and we evaluate it based on the probability of error at $T$. This setting has been mostly studied for $\varepsilon = 0$ [1, 25], but [41] present the first bounds for $\varepsilon > 0$ for an algorithm that is actually agnostic to this value.

- Simple regret minimization: we evaluate the expected simple regret at $T$ [5, 41].

Simple regret is typically studied in an anytime setting: [5] contains upper bounds on the simple regret at time $n$ for any $n \in \mathbb{N}^*$. Similarly, [23] propose the *anytime exploration* setting, in which they control the error probability $\mathbb{P}(\hat{\imath}_n \neq i_\star)$ for exact best arm identification. Interestingly, the authors build on an algorithm for the fixed-confidence setting, LUCB [24], whose sampling rule depends on the risk parameter $\delta$, which they replace by a sequence $\delta_n$. The algorithm that we study in this paper, motivated by the fixed-confidence $\varepsilon$-BAI problem, will already be *anytime*, which means that it does not depend on a given final time $T$ or a confidence level $\delta$. We shall analyze its sample complexity in the fixed confidence setting but thanks to the anytime property we will also be able to prove guarantees on its probability of $\varepsilon$-error for every $\varepsilon \geq 0$ and its simple regret at any time.

**Additional notation and assumption**   We denote by $\mathcal{D}$ a set to which the distributions of the arms are known to belong. We suppose that all distributions in $\mathcal{D}$ are 1-sub-Gaussian. A distribution $\nu_0$ is 1-sub-Gaussian if it satisfies $\mathbb{E}_{X \sim \nu_0}[e^{\lambda(X - \mathbb{E}_{X \sim \nu_0}[X])}] \leq e^{\lambda^2/2}$ for all $\lambda \in \mathbb{R}$. For example, all distributions bounded in $[-1, 1]$ are 1-sub-Gaussian. Let us denote by $i^\star(\mu) := \arg\max_{i \in [K]} \mu_i$ the set of arms with largest mean (i.e. $i^\star(\mu) = \mathcal{I}_0(\mu)$). Let $\Delta_i := \mu_\star - \mu_i$ denote the sub-optimality gap of arm $i$. We denote by $\triangle_K \subset \mathbb{R}^K$ the simplex of dimension $K - 1$.

**Fixed-confidence $\varepsilon$-best-arm identification**   Let $\varepsilon \geq 0$ and $\delta \in (0, 1)$ be fixed error and confidence parameters. In the *fixed-confidence $\varepsilon$-BAI* setting [31, 13, 35, 16], the probability of error of an algorithm is required to be less than $\delta$ for all instances $\nu \in \mathcal{D}^K$. That requirement leads to an asymptotic lower bound on the expected sample complexity on any instance.

**Lemma 1** ([10]). *For all $(\varepsilon, \delta)$-PAC algorithms and all instances $\nu_i = \mathcal{N}(\mu_i, 1)$ with $\mu \in \mathbb{R}^K$,* $\liminf_{\delta \to 0} \frac{\mathbb{E}_\nu[\tau_{\varepsilon,\delta}]}{\log(1/\delta)} \geq T_\varepsilon(\mu)$ *where* $T_\varepsilon(\mu) = \min_{i \in \mathcal{I}_\varepsilon(\mu)} \min_{\beta \in (0,1)} T_{\varepsilon,\beta}(\mu, i)$ *with*

$$T_{\varepsilon,\beta}(\mu, i)^{-1} = \max_{w \in \triangle_K, w_i = \beta} \min_{j \neq i} \frac{1}{2} \frac{(\mu_i - \mu_j + \varepsilon)^2}{1/\beta + 1/w_j} . \tag{1}$$

We say that an algorithm is asymptotically (resp. $\beta$-)optimal if its sample complexity matches that lower bound, that is if $\limsup_{\delta \to 0} \frac{\mathbb{E}_\nu[\tau_{\varepsilon,\delta}]}{\log(1/\delta)} \leq T_\varepsilon(\mu)$ (resp. $T_{\varepsilon,\beta}(\mu) = \min_{i \in \mathcal{I}_\varepsilon(\mu)} T_{\varepsilon,\beta}(\mu, i)$). Note

that the expected sample complexity of an asymptotically $1/2$-optimal algorithm is at worst twice higher than that of any asymptotically optimal algorithm since $T_{\varepsilon,1/2}(\mu) \le 2T_\varepsilon(\mu)$ [34].

The asymptotic characteristic time $T_\varepsilon(\mu)$ is of order $\sum_{i=1}^K \min\{\varepsilon^{-2}, \Delta_i^{-2}\}$. It is computed as a minimum over all $\varepsilon$-good arms $i \in \mathcal{I}_\varepsilon(\mu)$ of an arm-specific characteristic time, which can be interpreted as the time required to verify that $i$ is $\varepsilon$-good. Each of the times $\min_{\beta \in (0,1)} T_{\varepsilon,\beta}(\mu, i)$ correspond to the complexity of a BAI instance (i.e. $\varepsilon$-BAI with $\varepsilon = 0$) in which the mean of arm $i$ is increased by $\varepsilon$ (Lemma 9). Let $w_{\varepsilon,\beta}(\mu, i)$ be the maximizer of (1). In [16], they show that $T_\varepsilon(\mu) = T_{\varepsilon,\beta^\star(i^\star)}(\mu, i^\star)$ and $T_{\varepsilon,\beta}(\mu) = T_{\varepsilon,\beta}(\mu, i^\star)$, where $i^\star \in i^\star(\mu)$ and $\beta^\star(i^\star) = \arg\min_{\beta \in (0,1)} T_{\varepsilon,\beta}(\mu, i^\star)$. For $\varepsilon = 0$, a similar lower bound to Lemma 1 holds for all $\delta$ [15]. Lower bounds of order $\sum_{i=1}^K \Delta_i^{-2} \log \log \Delta_i^{-2}$ (independent of $\delta$, but with a stronger dependence in the gaps) were also shown [17, 6, 38, 7]. Note that the characteristic time for $\sigma$-sub-Gaussian distributions (which does not have a form as "explicit" as (1)) is always smaller than the ones for Gaussian having the same means and variance $\sigma^2$.

A good algorithm should have an expected sample complexity as close as possible to these lower bounds. Several algorithms for ($\varepsilon$-)BAI are based on modifications of the UCB algorithm [24, 17, 14]. Others compute approximate solutions to the lower bound maximization problem and sample arms in order to approach the solution [15, 11, 39]. Our method belongs to the family of Top Two algorithms [34, 36, 21], which select at each time two arms called leader and challenger, and sample among them. It is the first Top Two algorithm for the $\varepsilon$-BAI problem (for $\varepsilon > 0$).

**Any time and uniform $\varepsilon$-error bound**   In addition to the fixed-confidence guarantees, we will prove a bound on the probability of error for any time $n$ and any error $\varepsilon$, similarly to the results of [41]. That is, we bound $\mathbb{P}_\nu(\hat{\imath}_n \notin \mathcal{I}_\varepsilon(\mu))$ for all $n$ and all $\varepsilon$. This gives a bound on the probability of error in $\varepsilon$-BAI, and a bound on the simple regret of the sampling rule by integrating: $\mathbb{E}_\nu[\mu_\star - \mu_{\hat{\imath}_n}] = \int \mathbb{P}_\nu(\hat{\imath}_n \notin \mathcal{I}_\varepsilon(\mu)) \mathrm{d}\,\varepsilon$.

The literature mostly focuses on the fixed budget setting, where the time $T$ at which we evaluate the error probability is known and can be used as a parameter of the algorithm. Notable algorithms are successive rejects (SR, [1]) and sequential halving (SH, [25]). These algorithms can be extended to not depend on $T$ by using a doubling trick [23, 41]. That trick considers a sequence of algorithms that are run with budgets $(T_k)_k$, with $T_{k+1} = 2T_k$ and $T_1 = 2K\lceil \log_2 K \rceil$. Past observations are dropped when reaching $T_k$, and the obtained recommendation is used until the budget $T_{k+1}$ is reached.

## 1.1   Contributions

We propose the EB-TC$_{\varepsilon_0}$ algorithm for identification in bandits, with a slack parameter $\varepsilon_0 > 0$, originally motivated by $\varepsilon_0$-BAI. We study its combination with a stopping rule for fixed confidence $\varepsilon$-BAI (possibly with $\varepsilon_0 \ne \varepsilon$) and also its probability of error and simple regret at any time.

- EB-TC$_{\varepsilon_0}$ performs well empirically compared to existing methods, both for the expected sample complexity criterion in fixed confidence $\varepsilon$-BAI and for the anytime simple regret criterion. It is in addition easy to implement and computationally inexpensive in our regime.

- We prove upper bounds on the sample complexity of EB-TC$_{\varepsilon_0}$ in fixed confidence $\varepsilon$-BAI with sub-Gaussian distributions, both asymptotically (Theorem 1) as $\delta \to 0$ and for any $\delta$ (Theorem 2). In particular, EB-TC$_\varepsilon$ with $\varepsilon > 0$ is asymptotically optimal for $\varepsilon$-BAI with Gaussian distributions.

- We prove a uniform $\varepsilon$-error bound valid for any time for EB-TC$_{\varepsilon_0}$. This gives in particular a fixed budget error bound and a control of the expected simple regret of the algorithm (Theorem 3 and Corollary 1).

## 2   Anytime Top Two sampling rule

We propose an anytime Top Two algorithm, named EB-TC$_{\varepsilon_0}$ and summarized in Figure 1.

**Recommendation rule**   Let $N_{n,i} := \sum_{t \in [n-1]} \mathbb{1}(I_t = i)$ be the number of pulls of arm $i$ before time $n$, and $\mu_{n,i} := \frac{1}{N_{n,i}} \sum_{t \in [n-1]} X_{t,I_t} \mathbb{1}(I_t = i)$ be its empirical mean. At time $n > K$, we recommend the Empirical Best (EB) arm $\hat{\imath}_n \in \arg\max_{i \in [K]} \mu_{n,i}$ (where ties are broken arbitrarily).

1: **Input:** slack $\varepsilon_0 > 0$, proportion $\beta \in (0,1)$ (only for fixed proportions).

2: Set $\hat{\imath}_n \in \arg\max_{i \in [K]} \mu_{n,i}$, $B_n = \hat{\imath}_n$ and $C_n \in \arg\min_{i \neq B_n} \frac{\mu_{n,B_n} - \mu_{n,i} + \varepsilon_0}{\sqrt{1/N_{n,B_n} + 1/N_{n,i}}}$.

3: Set $\bar{\beta}_{n+1}(B_n, C_n) = (T_n(B_n, C_n)\bar{\beta}_n(B_n, C_n) + \beta_n(B_n, C_n))/T_{n+1}(B_n, C_n)$ with [**fixed**] $\beta_n(i,j) = \beta$ or [**IDS**] $\beta_n(i,j) = N_{n,j}/(N_{n,i} + N_{n,j})$ and $T_{n+1}(B_n, C_n) = T_n(B_n, C_n) + 1$.

4: Set $I_n = C_n$ if $N^{B_n}_{n,C_n} \leq (1 - \bar{\beta}_{n+1}(B_n, C_n))T_{n+1}(B_n, C_n)$, otherwise set $I_n = B_n$.

5: **Output:** next arm to sample $I_n$ and next recommendation $\hat{\imath}_n$.

Figure 1: EB-TC$_{\varepsilon_0}$ algorithm with **fixed** or **IDS** proportions.

## 2.1 Anytime Top Two sampling rule

We start by sampling each arm once. At time $n > K$, a Top Two sampling rule defines a leader $B_n \in [K]$ and a challenger $C_n \neq B_n$. It then chooses the arm to pull among them. For the leader/challenger pair, we consider the Empirical Best (EB) leader $B_n^{\text{EB}} = \hat{\imath}_n$ and, given a slack $\varepsilon_0 > 0$, the Transportation Cost (TC$_{\varepsilon_0}$) challenger

$$C_n^{\text{TC}\varepsilon_0} \in \arg\min_{i \neq B_n^{\text{EB}}} \frac{\mu_{n,B_n^{\text{EB}}} - \mu_{n,i} + \varepsilon_0}{\sqrt{1/N_{n,B_n^{\text{EB}}} + 1/N_{n,i}}} \ . \tag{2}$$

The algorithm then needs to choose between $B_n$ and $C_n$. In order to do so, we use a so-called *tracking* procedure [15]. We define one tracking procedure per pair of leader/challenger $(i,j) \in [K]^2$ such that $i \neq j$, hence we have $K(K-1)$ independent tracking procedures. For each pair $(i,j)$ of leader and challenger, the associated tracking procedure will ensure that the proportion of times the algorithm pulled the leader $i$ remains close to a target average proportion $\bar{\beta}_n(i,j) \in (0,1)$. At each round $n$, only one tracking rule is considered, i.e. the one of the pair $(i,j) = (B_n, C_n)$.

We define two variants of the algorithm that differ in the way they set the proportions $\bar{\beta}_n(i,j)$. *Fixed* proportions set $\bar{\beta}_n(i,j) = \beta$ for all $(n,i,j) \in \mathbb{N} \times [K]^2$, where $\beta \in (0,1)$ is fixed beforehand. Information-Directed Selection (*IDS*) [40] defines $\beta_n(i,j) = N_{n,j}/(N_{n,i} + N_{n,j})$ and sets $\bar{\beta}_n(i,j) := T_n(i,j)^{-1} \sum_{t \in [n-1]} \mathbb{1}\left((B_t, C_t) = (i,j)\right)\beta_t(i,j)$ where $T_n(i,j) := \sum_{t \in [n-1]} \mathbb{1}\left((B_t, C_t) = (i,j)\right)$ is the selection count of arms $(i,j)$ as leader/challenger.

Let $N^i_{n,j} := \sum_{t \in [n-1]} \mathbb{1}\left((B_t, C_t) = (i,j),\ I_t = j\right)$ be the number of pulls of arm $j$ at rounds in which $i$ was the leader. We set $I_n = C_n$ if $N^{B_n}_{n,C_n} \leq (1 - \bar{\beta}_{n+1}(B_n, C_n))T_{n+1}(B_n, C_n)$ and $I_n = B_n$ otherwise. Using Theorem 6 in [12] for each tracking procedure (i.e. each pair $(i,j)$) yields Lemma 2 (proved in Appendix H).

**Lemma 2.** *For all $n > K$, $i \in [K]$, $j \neq i$, we have $-1/2 \leq N^i_{n,j} - (1 - \bar{\beta}_n(i,j))T_n(i,j) \leq 1$.*

The TC$_{\varepsilon_0}$ challenger seeks to minimize an empirical version of a quantity that appears in the lower bound for $\varepsilon_0$-BAI (Lemma 1). As such, it is a natural extension of the TC challenger used in the T3C algorithm [36] for $\varepsilon_0 = 0$. In earlier works on Top Two methods [34, 32, 36], the choice between leader and challenger is randomized: given a fixed proportion $\beta \in (0,1)$, set $I_n = B_n$ with probability $\beta$, otherwise $I_n = C_n$. [20] replaced randomization by tracking, and [40] proposed IDS to define adaptive proportions $\beta_n(B_n, C_n) \in (0,1)$. In this work we study both fixed proportions with $\beta = 1/2$ and adaptive proportions with IDS. Empirically, we observe slightly better performances when using IDS (e.g. Figure 7 in Appendix J.2.1). While [20] tracked the leader with $K$ procedures, we consider $K(K-1)$ independent tracking procedures depending on the current leader/challenger pair.

**Choosing $\varepsilon_0$** [21] shows that EB-TC (i.e. EB-TC$_{\varepsilon_0}$ with slack $\varepsilon_0 = 0$) suffers from poor empirical performance for moderate $\delta$ in BAI (see Appendix D.3 in [21] for a detailed discussion). Therefore, the choice of the slack $\varepsilon_0 > 0$ is critical since it acts as a regularizer which naturally induces sufficient exploration. By setting $\varepsilon_0$ too small, the EB-TC$_{\varepsilon_0}$ algorithm will become as greedy as EB-TC and perform poorly. Having $\varepsilon_0$ too large will flatten differences between sub-optimal arms, hence it will behave more uniformly. We observe from the theoretical guarantees and from our experiments that it is best to take $\varepsilon_0 = \varepsilon$ for $\varepsilon$-BAI, but the empirical performance is only degrading slowly for $\varepsilon_0 > \varepsilon$. Taking $\varepsilon_0 < \varepsilon$ leads to very poor performance. We discuss this trade-off in more details in our experiments (e.g. Figures 5, 6 and 7 in Appendix J.2.1). When tackling BAI, the limitation of EB-TC can be solved by adding an implicit exploration mechanism in the choice of the leader/challenger

pair. For the choice of leader, we can use randomization (TS leader [34, 32, 36]) or optimism (UCB leader [20]). For the choice of the challenger, we can use randomization (RS challenger [34]) or penalization (TCI challenger [21], KKT challenger [40] or EI challenger [32]).

**Anytime sampling rule**   EB-TC$_{\varepsilon_0}$ is independent of a budget of samples $T$ or a confidence parameter $\delta$. This anytime sampling rule can be regarded as a stream of empirical means/counts $(\mu_n, N_n)_{n>K}$ (which could trigger stopping) and a stream of recommendations $\hat{i}_n = i^\star(\mu_n)$. These streams can be used by agents with different kinds of objectives. The fixed-confidence setting couples it with a stopping rule to be $(\varepsilon, \delta)$-PAC. It can also be used to get an $\varepsilon$-good recommendation with large probability at any given time $n$.

## 2.2   Stopping rule for fixed-confidence $\varepsilon$-best-arm identification

In addition to the sampling and recommendation rules, the fixed-confidence setting requires a stopping rule. Given an error/confidence pair, the GLR$_\varepsilon$ stopping rule [15] prescribes to stop at the time

$$\tau_{\varepsilon,\delta} = \inf \left\{ n > K \mid \min_{i \neq \hat{i}_n} \frac{\mu_{n,\hat{i}_n} - \mu_{n,i} + \varepsilon}{\sqrt{1/N_{n,\hat{i}_n} + 1/N_{n,i}}} \geq \sqrt{2c(n-1,\delta)} \right\} \quad \text{with} \quad \hat{i}_n = i^\star(\mu_n), \quad (3)$$

where $c : \mathbb{N} \times (0,1) \to \mathbb{R}_+$ is a threshold function. Lemma 3 gives a threshold ensuring that the GLR$_\varepsilon$ stopping rule is $(\varepsilon, \delta)$-PAC for all $\varepsilon \geq 0$ and $\delta \in (0,1)$, independently of the sampling rule.

**Lemma 3** ([27]). *Let $\varepsilon \geq 0$ and $\delta \in (0,1)$. Given any sampling rule, using the threshold*

$$c(n,\delta) = 2\mathcal{C}_G(\log((K-1)/\delta)/2) + 4\log(4 + \log(n/2)) \quad (4)$$

*with the stopping rule* (3) *with error/confidence pair* $(\varepsilon, \delta)$ *yields a* $(\varepsilon, \delta)$*-PAC algorithm for sub-Gaussian distributions. The function* $\mathcal{C}_G$ *is defined in* (23). *It satisfies* $\mathcal{C}_G(x) \approx x + \log(x)$.

# 3   Fixed-confidence theoretical guarantees

To study $\varepsilon$-BAI in the fixed-confidence setting, we couple EB-TC$_{\varepsilon_0}$ with the GLR$_\varepsilon$ stopping rule (3) using error $\varepsilon \geq 0$, confidence $\delta \in (0,1)$ and threshold (4). The resulting algorithm is $(\varepsilon, \delta)$-PAC by Lemma 3. We derive upper bounds on the expected sample complexity $\mathbb{E}_\nu[\tau_{\varepsilon,\delta}]$ both in the asymptotic regime of $\delta \to 0$ (Theorem 1) and for finite confidence when $\varepsilon = \varepsilon_0$ (Theorem 2).

**Theorem 1.** *Let $\varepsilon \geq 0$, $\varepsilon_0 > 0$ and $(\beta, \delta) \in (0,1)^2$. Combined with GLR$_\varepsilon$ stopping* (3)*, the EB-TC$_{\varepsilon_0}$ algorithm is $(\varepsilon, \delta)$-PAC and it satisfies that, for all $\nu \in \mathcal{D}^K$ with mean $\mu$ such that $|i^\star(\mu)| = 1$,*

$$\textbf{[IDS]} \limsup_{\delta \to 0} \frac{\mathbb{E}_\nu[\tau_{\varepsilon,\delta}]}{\log(1/\delta)} \leq T_{\varepsilon_0}(\mu) D_{\varepsilon,\varepsilon_0}(\mu) \quad and \quad \textbf{[fixed } \beta\textbf{]} \limsup_{\delta \to 0} \frac{\mathbb{E}_\nu[\tau_{\varepsilon,\delta}]}{\log(1/\delta)} \leq T_{\varepsilon_0,\beta}(\mu) D_{\varepsilon,\varepsilon_0}(\mu),$$

*where $D_{\varepsilon,\varepsilon_0}(\mu) = (1 + \max_{i \neq i^\star} (\varepsilon_0 - \varepsilon)/(\mu_\star - \mu_i + \varepsilon))^2$.*

While Theorem 1 holds for all sub-Gaussian distributions, it is particularly interesting for Gaussian ones, in light of Lemma 1. When choosing $\varepsilon = \varepsilon_0$ (i.e. $D_{\varepsilon_0,\varepsilon_0}(\mu) = 1$), Theorem 1 shows that EB-TC$_{\varepsilon_0}$ is asymptotically optimal for Gaussian bandits when using IDS proportions and asymptotically $\beta$-optimal when using fixed proportions $\beta$. We also note that Theorem 1 is not conflicting with the lower bound of Lemma 1, as shown in Lemma 11 in Appendix C. Empirically we observe that the empirical stopping time can be drastically worse when taking $\varepsilon_0 < \varepsilon$, and close to the optimal one when $\varepsilon_0 > \varepsilon$ (Figures 5, 6 and 7 in Appendix J.2.1).

Until recently [40], proving asymptotic optimality of Top Two algorithms with adaptive choice $\beta$ was an open problem in BAI. In this work, we prove that their choice of IDS proportions also yield asymptotically optimal algorithms for $\varepsilon$-BAI. While the proof of Theorem 1 assumes the existence of a unique best arm, it holds for instances having sub-optimal arms with the same mean. This is an improvement compared to existing asymptotic guarantees on Top Two algorithms which rely on the assumption that the means of all arms are different [32, 36, 21]. The improvement is possible thanks to the regularization induced by the slack $\varepsilon_0 > 0$.

While asymptotic optimality in the $\varepsilon$-BAI setting was already achieved for various algorithms (e.g. $\varepsilon$-Track-and-Stop (TaS) [16], Sticky TaS [10] or L$\varepsilon$BAI [19]), none of them obtained non-asymptotic guarantees. Despite their theoretical interest, asymptotic results provide no guarantee on the performance for moderate $\delta$. Furthermore, asymptotic results on Top Two algorithms require a unique best arm regardless of the considered error $\varepsilon$: reaching asymptotic ($\beta$-)optimality implicitly

means that the algorithm eventually allocates samples in an optimal way that depends on the identity of the unique best arm, and that requires the unique best arm to be identified. As our focus is $\varepsilon$-BAI, our guarantees should only require that one of the $\varepsilon$-good arms is identified and should hold for instances having multiple best arms. The upper bound should scale with $\varepsilon_0^{-2}$ instead of $\Delta_{\min}^{-2}$ when $\Delta_{\min}$ is small. Theorem 2 satisfies these requirements.

**Theorem 2.** *Let $\delta \in (0, 1)$ and $\varepsilon_0 > 0$. Combined with $GLR_{\varepsilon_0}$ stopping (3), the EB-TC$_{\varepsilon_0}$ algorithm with fixed $\beta = 1/2$ is $(\varepsilon_0, \delta)$-PAC and satisfies that, for all $\nu \in \mathcal{D}^K$ with mean $\mu$,*

$$\mathbb{E}_\nu[\tau_{\varepsilon_0,\delta}] \leq \inf_{\tilde{\varepsilon} \in [0,\varepsilon_0]} \max \left\{ T_{\mu,\varepsilon_0}(\delta, \tilde{\varepsilon}) + 1, \; S_{\mu,\varepsilon_0}(\tilde{\varepsilon}) \right\} + 2K^2 \; ,$$

*where $T_{\mu,\varepsilon_0}(\delta, \tilde{\varepsilon})$ and $S_{\mu,\varepsilon_0}(\tilde{\varepsilon})$ are defined by*

$$T_{\mu,\varepsilon_0}(\delta, \tilde{\varepsilon}) = \sup \left\{ n \mid n - 1 \leq 2(1+\gamma)^2 \sum_{i \in \mathcal{I}_{\tilde{\varepsilon}}(\mu)} T_{\varepsilon_0,1/2}(\mu, i)(\sqrt{c(n-1,\delta)} + \sqrt{4 \log n})^2 \right\} \; ,$$

$$S_{\mu,\varepsilon_0}(\tilde{\varepsilon}) = h_1 \left( \frac{16(1+\gamma^{-1})}{a_{\mu,\varepsilon_0}(\tilde{\varepsilon})} H_{\mu,\varepsilon_0}(\tilde{\varepsilon}), \; \frac{(1+\gamma^{-1})K^2}{a_{\mu,\varepsilon_0}(\tilde{\varepsilon})} + 1 \right) \; ,$$

$$a_{\mu,\varepsilon_0}(\tilde{\varepsilon}) = \frac{\min_{i \in \mathcal{I}_{\tilde{\varepsilon}}(\mu)} T_{\varepsilon_0,1/2}(\mu, i)}{\sum_{i \in \mathcal{I}_{\tilde{\varepsilon}}(\mu)} T_{\varepsilon_0,1/2}(\mu, i)} \min_{i \in \mathcal{I}_{\tilde{\varepsilon}}(\mu), j \neq i} w_{\varepsilon_0,1/2}(\mu, i)_j \; ,$$

*where $\gamma \in (0, 1/2]$ is an analysis parameter and $h_1(y, z) \approx z + y \log(z + y \log(y))$ as in Lemma 51. $T_{\varepsilon_0,1/2}(\mu, i)$ and $w_{\varepsilon_0,1/2}(\mu, i)$ are defined in (1) and*

$$H_{\mu,\varepsilon_0}(\tilde{\varepsilon}) := \frac{2|i^\star(\mu)|}{\Delta_\mu(\tilde{\varepsilon})^2} + (|\mathcal{I}_{\tilde{\varepsilon}}(\mu) \setminus i^\star(\mu)|) C_{\mu,\varepsilon_0}(\tilde{\varepsilon})^2 + \sum_{i \notin \mathcal{I}_{\tilde{\varepsilon}}(\mu)} \max\{C_{\mu,\varepsilon_0}(\tilde{\varepsilon}), \sqrt{2}\Delta_i^{-1}\}^2 \; , \quad (5)$$

*with $\Delta_\mu(\tilde{\varepsilon}) = \min_{k \notin \mathcal{I}_{\tilde{\varepsilon}}(\mu)} \Delta_k$ and $C_{\mu,\varepsilon_0}(\tilde{\varepsilon}) = \max\{2\Delta_\mu(\tilde{\varepsilon})^{-1} - \varepsilon_0^{-1}, \varepsilon_0^{-1}\}$.*

The upper bound on $\mathbb{E}_\nu[\tau_{\varepsilon_0,\delta}]$ involves a $\delta$-dependent term $T_{\mu,\varepsilon_0}(\delta, \tilde{\varepsilon})$ and a $\delta$-independent term $S_{\mu,\varepsilon_0}(\tilde{\varepsilon})$. The choice of $\tilde{\varepsilon}$ influences the compromise between those, and the infimum over $\tilde{\varepsilon}$ means that our algorithm benefits from the best possible trade-off. In the asymptotic regime, we take $\tilde{\varepsilon} = 0$ and $\gamma \to 0$ and we obtain $\lim_{\delta \to 0} \mathbb{E}_\nu[\tau_{\varepsilon_0,\delta}]/\log(1/\delta) \leq 2|i^\star(\mu)|T_{\varepsilon_0,1/2}(\mu)$. When $|i^\star(\mu)| = 1$, we recover the asymptotic result of Theorem 1 up to a multiplicative factor 2. For multiple best arms, the asymptotic sample complexity is at most a factor $2|i^\star(\mu)|$ from the $\beta$-optimal one.

Given a finite confidence, the dominant term will be $S_{\mu,\varepsilon_0}(\tilde{\varepsilon})$. For $\tilde{\varepsilon} = 0$, we show that $H_{\mu,\varepsilon_0}(0) = \mathcal{O}(K \min\{\Delta_{\min}, \varepsilon_0\}^{-2})$, hence we should consider $\tilde{\varepsilon} > 0$ to avoid the dependency in $\Delta_{\min}$. For $\tilde{\varepsilon} = \varepsilon_0$, there exists instances such that $\max_{i \in \mathcal{I}_{\varepsilon_0}(\mu)} T_{\varepsilon_0,1/2}(\mu, i)$ is arbitrarily large, hence $S_{\mu,\varepsilon_0}(\varepsilon_0)$ will be very large as well. The best trade-off is attained in the interior of the interval $(0, \varepsilon_0)$. For $\tilde{\varepsilon} = \varepsilon_0/2$, Lemma 10 shows that $T_{\varepsilon_0,1/2}(\mu, i) = \mathcal{O}(K/\varepsilon_0^2)$ for all $i \in \mathcal{I}_{\varepsilon_0/2}(\mu)$ and $H_{\mu,\varepsilon_0}(\varepsilon_0/2) = \mathcal{O}(K/\varepsilon_0^2)$. Therefore, we obtain an upper bound $\mathcal{O}(|\mathcal{I}_{\varepsilon_0/2}(\mu)|K\varepsilon_0^{-2} \log \varepsilon_0^{-1})$.

Likewise, Lemma 10 shows that $\min_{j \neq i} w_{\varepsilon_0,1/2}(\mu, i)_j \geq (16(K - 2) + 2)^{-1}$ for all $i \in \mathcal{I}_{\varepsilon_0/2}(\mu)$. While the dependency in $a_{\mu,\varepsilon_0}(\varepsilon_0/2)$ is milder in $\varepsilon$-BAI than in BAI (as it is bounded away from 0), we can improve it by using a refined analysis (see Appendix E). Introduced in [20], this method allows to clip $\min_{j \neq i} w_{\varepsilon_0,1/2}(\mu, i)_j$ by a fixed value $x \in [0, (K - 1)^{-1}]$ for all $i \in \mathcal{I}_{\tilde{\varepsilon}}(\mu)$.

**Comparison with existing upper bounds** The LUCB algorithm [24] has a structure similar to a Top Two algorithm, with the differences that LUCB samples both the leader and the challenger and that it stops when the gap between the UCB and LCB indices is smaller than $\varepsilon_0$. As LUCB satisfies $\mathbb{E}_\mu[\tau_{\varepsilon_0,\delta}] \leq 292H_{\varepsilon_0}(\mu) \log(H_{\varepsilon_0}(\mu)/\delta) + 16$ where $H_{\varepsilon_0}(\mu) = \sum_i(\max\{\Delta_i, \varepsilon_0/2\})^{-2}$, it enjoys better scaling than EB-TC$_{\varepsilon_0}$ for finite confidence. However, since the empirical allocation of LUCB is not converging towards $w_{\varepsilon_0,1/2}(\mu)$, it is not asymptotically 1/2-optimal. While LUCB has better moderate confidence guarantees, there is no hope to prove anytime performance bounds since LUCB indices depends on $\delta$. In contrast, EB-TC$_{\varepsilon_0}$ enjoys such guarantees (see Section 4).

**Key technical tool for the non-asymptotic analysis** We want to ensure that EB-TC$_{\varepsilon_0}$ eventually selects only $\varepsilon$-good arms as leader, for any error $\varepsilon \geq 0$. Our proof strategy is to show that if the leader is not an $\varepsilon$-good arm and empirical means do not deviate too much from the true means, then either the current leader or the current challenger was selected as leader or challenger less than a given quantity. We obtain a bound on the total number of times that can happen.

**Lemma 4.** *Let $\delta \in (0,1]$ and $n > K$. Let $T_n(i) := \sum_{j\neq i}(T_n(i,j) + T_n(j,i))$ be the number of times arm $i$ was selected in the leader/challenger pair. Assume there exists a sequence of events $(A_t(n,\delta))_{n\geq t>K}$ and positive reals $(D_i(n,\delta))_{i\in[K]}$ such that, for all $t \in \{K+1,\ldots,n\}$, under the event $A_t(n,\delta)$,*

$$\exists i_t \in [K], \quad T_t(i_t) \leq D_{i_t}(n,\delta) \quad and \quad T_{t+1}(i_t) = T_t(i_t) + 1 . \tag{6}$$

*Then, we have $\sum_{t=K+1}^n \mathbb{1}\left(A_t(n,\delta)\right) \leq \sum_{i\in[K]} D_i(n,\delta)$.*

To control the deviation of the empirical means and empirical gaps to their true value, we use a sequence of concentration events $(\mathcal{E}_{n,\delta})_{n>T}$ defined in Lemma 45 (Appendix G.2) such that $\mathbb{P}_\nu(\mathcal{E}_{n,\delta}^\complement) \leq K^2\delta n^{-s}$ where $s \geq 0$ and $\delta \in (0,1]$. For the EB-TC$_{\varepsilon_0}$ algorithm with fixed $\beta = 1/2$, we prove that, under $\mathcal{E}_{n,\delta}$, $\{B_t^{\text{EB}} \notin \mathcal{I}_\varepsilon(\mu)\}$ is a "bad" event satisfying the assumption of Lemma 4. This yields Lemma 5, which essentially says that the leader is an $\varepsilon$-good arm except for a logarithmic number of rounds.

**Lemma 5.** *Let $\delta \in (0,1]$, $n > K$ and $\varepsilon \geq 0$. Under the event $\mathcal{E}_{n,\delta}$, we have*

$$\sum_{i\in\mathcal{I}_\varepsilon(\mu)} \sum_j T_n(i,j) \geq n - 1 - 8H_{\mu,\varepsilon_0}(\varepsilon)f_2(n,\delta) - 3K^2 ,$$

*where $f_2(n,\delta) = \log(1/\delta) + (2+s)\log n$ and $H_{\mu,\varepsilon_0}(\varepsilon)$ is defined in (5).*

Noticeably, Lemma 5 holds for any $\varepsilon \geq 0$ even when there are multiple best arms. As expected the number of times the leader is not among the $\varepsilon_0$-good arms depends on $H_{\mu,\varepsilon_0}(\varepsilon_0) = \mathcal{O}(K/\varepsilon_0^2)$. The number of times the leader is not among the best arms depends on $H_{\mu,\varepsilon_0}(0) = \mathcal{O}(K(\min\{\Delta_{\min},\varepsilon_0\})^{-2})$.

**Time-varying slack** Theorem 1 shows the asymptotic optimality of the EB-TC$_{\varepsilon_0}$ algorithm with IDS for $\varepsilon_0$-BAI (where $\varepsilon_0 > 0$). To obtain optimality for BAI, we consider time-varying slacks $(\varepsilon_n)_n$, where $(\varepsilon_n)_n$ is decreasing, $\varepsilon_n > 0$ and $\varepsilon_n \to_{+\infty} 0$. A direct adaptation of our asymptotic analysis on $\mathbb{E}_\nu[\tau_{0,\delta}]$ (see Appendix D), regardless of the choice of $(\varepsilon_n)_n$, one can show that using GLR$_0$ stopping, the EB-TC$_{(\varepsilon_n)_n}$ algorithm with IDS is $(0,\delta)$-PAC and is asymptotically optimal in BAI. Its empirical performance is however very sensitive to the choice of $(\varepsilon_n)_n$ (Appendix J.2.3).

# 4 Beyond fixed-confidence guarantees

Could an algorithm analyzed in the fixed-confidence setting be used for the fixed-budget or even anytime setting? This question is especially natural for EB-TC$_{\varepsilon_0}$, which does not depend on the confidence parameter $\delta$. Yet its answer is not obvious, as it is known that algorithms that have *optimal* asymptotic guarantees in the fixed-confidence setting can be sub-optimal in terms of error probability. Indeed [28] prove in their Appendix C that for any asymptotically optimal (exact) BAI algorithm, there exists instances in which the error probability cannot decay exponentially with the horizon, which makes them worse than the (minimax optimal) uniform sampling strategy [5].

Their argument also applies to $\beta$-optimal algorithms, hence to EB-TC$_0$ with $\beta = 1/2$. However, whenever $\varepsilon_0$ is positive, Theorem 3 reveals that the error probability of EB-TC$\varepsilon_0$ always decays exponentially, which redeems the use of optimal fixed-confidence algorithms for a relaxed BAI problem in the anytime setting. Going further, this result provides an anytime bound on the probability to recommend an arm that is not $\varepsilon$-optimal, for any error $\varepsilon \geq 0$. This bound involves instance-dependent complexities depending solely on the gaps in $\mu$. To state it, we define $C_\mu := |\{\mu_i \mid i \in [K]\}|$ as the number of distinct arm means in $\mu$ and let $\mathcal{C}_\mu(i) := \{k \in [K] \mid \mu_\star - \mu_k = \Delta_i\}$ be the set of arms having mean gap $\Delta_i$ where the gaps are sorted by increasing order $0 = \Delta_1 < \Delta_2 < \cdots < \Delta_{C_\mu}$. For all $\varepsilon \geq 0$, let $i_\mu(\varepsilon) = i$ if $\varepsilon \in [\Delta_i, \Delta_{i+1})$ (with the convention $\Delta_{C_\mu+1} = +\infty$). Theorem 3 shows that the exponential decrease of $\mathbb{P}_\nu(\hat{\imath}_n \notin \mathcal{I}_\varepsilon(\mu))$ is linear.

**Theorem 3.** *(see Theorem 6 in Appendix F) Let $\varepsilon_0 > 0$. The EB-TC$_{\varepsilon_0}$ algorithm with fixed proportions $\beta = 1/2$ satisfies that, for all $\nu \in \mathcal{D}^K$ with mean $\mu$, for all $\varepsilon \geq 0$, for all $n > 5K^2/2$,*

$$\mathbb{P}_\nu\left(\hat{\imath}_n \notin \mathcal{I}_\varepsilon(\mu)\right) \leq K^2 e^2 (2 + \log n)^2 \exp\left(-p\left(\frac{n - 5K^2/2}{8H_{i_\mu(\varepsilon)}(\mu,\varepsilon_0)}\right)\right) .$$

*where $p(x) = x - \log x$ and $(H_i(\mu,\varepsilon_0))_{i\in[C_\mu-1]}$ are such that $H_1(\mu,\varepsilon_0) = K(2\Delta_{\min}^{-1} + 3\varepsilon_0^{-1})^2$ and $K/\Delta_{i+1}^{-2} \leq H_i(\mu,\varepsilon_0) \leq K\min_{j\in[i]}\max\{2\Delta_{j+1}^{-1}, 2\frac{\Delta_j/\varepsilon_0+1}{\Delta_{i+1}-\Delta_j} + 3\varepsilon_0^{-1}\}^2$ for all $i > 1$.*

This bound can be compared with the following uniform $\varepsilon$-error bound of the strategy using uniform sampling and recommending the empirical best arm:

$$\mathbb{P}\left(\hat{\imath}_n^{\mathrm{U}} \notin \mathcal{I}_\varepsilon(\mu)\right) \leq \sum_{i \notin \mathcal{I}_\varepsilon(\mu)} \exp\left(-\frac{\Delta_i^2 \lfloor n/K \rfloor}{4}\right) \leq K \exp\left(-\frac{n-K}{4K\Delta_{i_\mu(\varepsilon)+1}^{-2}}\right)$$

Recalling that the quantity $H_i(\mu, \varepsilon_0)$ in Theorem 3 is always bounded from below by $2K\Delta_{i+1}^{-1}$, we get that our upper bound is larger than the probability of error of the uniform strategy, but the two should be quite close. For example for $\varepsilon = 0$, we have

$$\mathbb{P}_\nu\left(\hat{\imath}_n \notin i^\star(\mu)\right) \leq \exp\left(-\Theta\left(\frac{n}{K(\Delta_{\min}^{-1} + \varepsilon_0^{-1})^2}\right)\right), \ \mathbb{P}_\nu\left(\hat{\imath}_n^{\mathrm{U}} \notin i^\star(\mu)\right) \leq \exp\left(-\Theta\left(\frac{n}{K\Delta_{\min}^{-2}}\right)\right).$$

Even if they provide a nice sanity-check for the use of a sampling rule with optimal fixed-confidence guarantees for $\varepsilon_0$-BAI in the anytime regime, we acknowledge that these guarantees are far from optimal. Indeed, the work of [41] provides tighter anytime uniform $\varepsilon$-error probability bounds for two algorithms: an anytime version of Sequential Halving [25] using a doubling trick (called DSH), and an algorithm called Brackeiting Sequential Halving, that is designed to tackle a very large number of arms. Their upper bounds are of the form $\mathbb{P}_\nu\left(\hat{\imath}_n \notin \mathcal{I}_\varepsilon(\mu)\right) \leq \exp\left(-\Theta\left(n/H(\varepsilon)\right)\right)$ with $H(\varepsilon) = \frac{1}{g(\varepsilon/2)} \max_{i \geq g(\varepsilon)+1} \frac{i}{\Delta_i^2}$ where $g(\varepsilon) = |\{i \in [K] \mid \mu_i \geq \mu^\star - \varepsilon\}|$. Therefore, they can be much smaller than $K\Delta_{i_\mu(\varepsilon)+1}^{-2}$.

The BUCB algorithm of [26] is also analyzed for any level of error $\varepsilon$, but in a different fashion. The authors provide bounds on its $(\varepsilon, \delta)$-*unverifiable sample complexity*, defined as the expectation of the smallest stopping time $\tilde{\tau}$ satisfying $\mathbb{P}(\forall t \geq \tilde{\tau}, \hat{\imath}_n \in \mathcal{I}_\varepsilon(\mu)) \geq 1 - \delta$. This notion is different from the sample complexity we use in this paper, which is sometimes called *verifiable* since it is the time at which the algorithm can guarantee that its error probability is less than $\delta$. Interestingly, to prove Theorem 3 we first prove a bound on the unverifiable sample complexity of EB-TC$_{\varepsilon_0}$ which is valid for all $(\varepsilon, \delta)$, neither of which are parameters of the algorithm. More precisely, we prove that $\mathbb{P}_\nu\left(\forall n > U_{i_\mu(\varepsilon),\delta}(\mu, \varepsilon_0), \ \hat{\imath}_n \in \mathcal{I}_\varepsilon(\mu)\right) \geq 1 - \delta$ for $U_{i,\delta}(\mu, \varepsilon_0) =_{\delta \to 0} 8H_i(\mu, \varepsilon_0) \log(1/\delta) + \mathcal{O}(\log\log(1/\delta))$. As this statement is valid for all $\delta \in (0,1)$, applying it for each $n$ to $\delta_n$ such that $U_{i_\mu(\varepsilon),\delta_n}(\mu, \varepsilon_0) = n$, we obtain Theorem 3. We remark that such a trick cannot be applied to BUCB to get uniform $\varepsilon$-error bounds for any time, as the algorithm does depend on $\delta$.

**Simple regret** As already noted by [41], uniform $\varepsilon$-error bounds easily yield simple regret bounds. We state in Corollary 1 the one obtained for EB-TC$_{\varepsilon_0}$. As a motivation to derive simple regret bounds, we observe that they readily translate to bounds on the cumulative regret for an agent observing the stream of recommendations $(\hat{\imath}_n)$ and playing arm $\hat{\imath}_n$. An exponentially decaying simple regret leads to a constant cumulative regret in this decoupled exploration/exploitation setting [2, 33].

**Corollary 1.** *Let $\varepsilon_0 > 0$. Let $p(x)$ and $(H_i(\mu, \varepsilon_0))_{i \in [C_\mu - 1]}$ be defined as in Theorem 3. The EB-TC$_{\varepsilon_0}$ algorithm with fixed $\beta = 1/2$ satisfies that, for all $\nu \in \mathcal{D}^K$ with mean $\mu$, for all $n > 5K^2/2$,*

$$\mathbb{E}_\nu[\mu_\star - \mu_{\hat{\imath}_n}] \leq K^2 e^2 (2 + \log n)^2 \sum_{i \in [C_\mu - 1]} (\Delta_{i+1} - \Delta_i) \exp\left(-p\left(\frac{n - 5K^2/2}{8H_i(\mu, \varepsilon_0)}\right)\right).$$

Following the discussion above, this bound is not expected to be state-of-the-art, it rather justifies that EB-TC$_{\varepsilon_0}$ with $\varepsilon_0 > 0$ is not too much worse than the uniform sampling strategy. Yet, as we shall see in our experiments, the practical story is different. In Section 5, we compare the simple regret of EB-TC$_{\varepsilon_0}$ to that of DSH in synthetic experiments with a moderate number of arms, revealing the superior performance of EB-TC$_{\varepsilon_0}$.

## 5 Experiments

We assess the performance of the EB-TC$_{\varepsilon_0}$ algorithm on Gaussian instances both in terms of its empirical stopping time and its empirical simple regret, and we show that it perform favorably compared to existing algorithms in both settings. For the sake of space, we only show the results for large sets of arms and for a specific instance with $|i^\star(\mu)| = 2$.

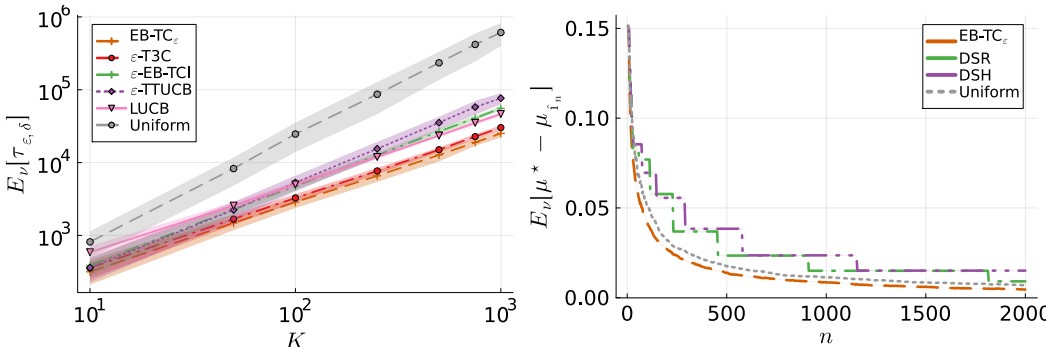

Figure 2: (a) Empirical stopping time on "$\alpha = 0.3$" instances for varying $K$ and stopping rule (3) using $(\varepsilon, \delta) = (0.1, 0.01)$. The BAI algorithms T3C, EB-TCI and TTUCB are modified to be $\varepsilon$-BAI ones. (b) Empirical simple regret on instance $\mu = (0.6, 0.6, 0.55, 0.45, 0.3, 0.2)$, in which EB-TC$_{\varepsilon_0}$ with slack $\varepsilon_0 = 0.1$ and fixed $\beta = 1/2$ is used.

**Empirical stopping time** We study the impact of large sets of arms (up to $K = 1000$) in $\varepsilon$-BAI for $(\varepsilon, \delta) = (0.1, 0.01)$ on the "$\alpha = 0.3$" scenario of [18] which sets $\mu_i = 1 - ((i-1)/(K-1))^\alpha$ for all $i \in [K]$. EB-TC$_{\varepsilon_0}$ with IDS and slack $\varepsilon_0 = \varepsilon$ is compared to existing $\varepsilon$-BAI algorithms having low computational cost. This precludes algorithms such as $\varepsilon$-Track-and-Stop (TaS) [16], Sticky TaS [10] or $\varepsilon$-BAI adaptation of FWS [39] and DKM [11]. In Appendix J.2.2, we compare EB-TC$_\varepsilon$ to those algorithms on benchmarks with smaller number of arms. We show that EB-TC$_\varepsilon$ performs on par with $\varepsilon$-TaS and $\varepsilon$-FWS, but outperforms $\varepsilon$-DKM. As Top Two benchmarks with fixed $\beta = 1/2$, we consider T3C [36], EB-TCI [21] and TTUCB [20]. To provide a fair comparison, we adapt them to tackle $\varepsilon$-BAI by using the stopping rule (3) and by adapting their sampling rule to use the TC$\varepsilon$ challenger from (2) (with a penalization $\log N_{n,i}$ for EB-TCI). We use the heuristic threshold $c(n, \delta) = \log((1 + \log n)/\delta)$. While it is too small to ensure the $(\varepsilon, \delta)$-PAC property, it still yields an empirical error which is several orders of magnitude lower than $\delta$. Finally, we compare with LUCB [24] and uniform sampling. For a fair comparison, LUCB uses $\sqrt{2c(n-1, \delta)/N_{n,i}}$ as bonus, which is also too small to yield valid confidence intervals. Our results are averaged over 100 runs, and the standard deviations are displayed. In Figure 2(a), we see that EB-TC$_\varepsilon$ performs on par with the $\varepsilon$-T3C heuristic, and significantly outperforms the other algorithms. While the scaling in $K$ of $\varepsilon$-EB-TCI and LUCB appears to be close to the one of EB-TC$_\varepsilon$, $\varepsilon$-TTUCB and uniform sampling obtain a worse one. Figure 2(a) also reveals that the regularization ensured by the TC$\varepsilon$ challenger is sufficient to ensure enough exploration, hence other exploration mechanisms are superfluous (TS/UCB leader or TCI challenger).

**Anytime empirical simple regret** The EB-TC$_{\varepsilon_0}$ algorithm with fixed $\beta = 1/2$ and $\varepsilon_0 = 0.1$ is compared to existing algorithms on the instance $\mu = (0.6, 0.6, 0.55, 0.45, 0.3, 0.2)$ from [16], which has two best arms. As benchmark, we consider Doubling Successive Reject (DSR) and Doubling Sequential Halving (DSH), which are adaptations of the elimination based algorithms SR [1] and SH [25]. SR eliminates one arm with the worst empirical mean at the end of each phase, and SH eliminated half of them but drops past observations between each phase. These doubling-based algorithms have empirical error decreasing by steps: they change their recommendation only before they restart. In Figure 2(b), we plot the average of the simple regret over 10000 runs and the standard deviation of that average (which is too small to see clearly). We observe that EB-TC$_{\varepsilon_0}$ outperforms uniform sampling, as well as DSR and DSH, which both perform worse due to the dropped observations. The favorable performance of EB-TC$_{\varepsilon_0}$ is confirmed on other instances from [16], and for "two-groups" instances with varying $|i^\star(\mu)|$ (see Figures 10 and 12).

**Supplementary experiments** Extensive experiments and implementation details are available in Appendix J. In Appendix J.2.1, we compare the performance of EB-TC$_{\varepsilon_0}$ with different slacks $\varepsilon_0$ for IDS and fixed $\beta = 1/2$. In Appendix J.2.2, we demonstrate the good empirical performance of EB-TC$_{\varepsilon_0}$ compared to state-of-the art methods in the fixed-confidence $\varepsilon$-BAI setting, compared to DSR and DSH for the empirical simple regret, and compared to SR and SH for the probability of 0-error in the fixed-budget setting (Figure 13). We consider a wide range of instances: random ones, benchmarks from the literature [18, 16] and "two-groups" instances with varying $|i^\star(\mu)|$.

## 6  Perspectives

We have proposed the EB-TC$_{\varepsilon_0}$ algorithm, which is easy to understand and implement. EB-TC$_{\varepsilon_0}$ is the first algorithm to be simultaneously asymptotically optimal in the fixed-confidence $\varepsilon_0$-BAI setting (Theorem 1), have finite-confidence guarantees (Theorem 2), and have also anytime guarantees on the probability of error at any level $\varepsilon$ (Theorem 3), hence on the simple regret (Corollary 1). Furthermore, we have demonstrated that the EB-TC$_{\varepsilon_0}$ algorithm achieves superior performance compared to other algorithms, in benchmarks where the number of arms is moderate to large. In future work, we will investigate its possible adaptation to the data-poor regime of [41] in which the number of arms is so large that any algorithm sampling each arm once is doomed to failure.

While our results hold for general sub-Gaussian distributions, the EB-TC$_{\varepsilon_0}$ algorithm with IDS and slack $\varepsilon_0 > 0$ only achieves asymptotic optimality for $\varepsilon_0$-BAI with Gaussian bandits. Even though IDS has been introduced by [40] for general single-parameter exponential families, it is still an open problem to show asymptotic optimality for distributions other than Gaussians. While our non-asymptotic guarantees on $\mathbb{E}_\nu[\tau_{\varepsilon_0,\delta}]$ and $\mathbb{E}_\nu[\mu_\star - \mu_{\hat{\imath}_n}]$ were obtained for the EB-TC$_{\varepsilon_0}$ algorithm with fixed $\beta = 1/2$, we observed empirically better performance when using IDS. Deriving similar (or better) non-asymptotic guarantees for IDS is an interesting avenue for future work.

Finally, the EB-TC$_{\varepsilon_0}$ algorithm is a promising method to tackle structured bandits. Despite the existence of heuristics for settings such as Top-k identification [40], it is still an open problem to efficiently adapt Top Two approaches to cope for other structures such as $\varepsilon$-BAI in linear bandits.

## Acknowledgments and Disclosure of Funding

Experiments presented in this paper were carried out using the Grid'5000 testbed, supported by a scientific interest group hosted by Inria and including CNRS, RENATER and several Universities as well as other organizations (see https://www.grid5000.fr). This work has been partially supported by the THIA ANR program "AI_PhD@Lille". The authors acknowledge the funding of the French National Research Agency under the project BOLD (ANR-19-CE23-0026-04) and the project FATE (ANR22-CE23-0016-01).

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

## A  Outline

The appendices are organized as follows:

- Notation are summarized in Appendix B.

- In Appendix C, we show how the characteristic times and optimal allocations for $\varepsilon$-BAI can be reduced to the ones of a BAI problem with modified instances (Lemma 9).

- The asymptotic upper bound on the expected sample complexity of the EB-TC$_{\varepsilon_0}$ algorithm when combined with the stopping rule (4) is proven in Appendix D (Theorem 1).

- The non-asymptotic upper bound on the expected sample complexity of the EB-TC$_{\varepsilon_0}$ algorithm when combined with the stopping rule (4) is proven in Appendix E (Theorem 5).

- The upper bound on the probability of error and on the simple regret of the EB-TC$_{\varepsilon_0}$ algorithm is proven in Appendix F (Theorem 6), as well as the upper bound on its unverifiable expected sample complexity (Theorem 7).

- Appendix G gathers concentration results used for the calibration of the stopping rule (Lemma 3) and to control the empirical means and gaps.

- Appendix H gathers existing and new technical results which are used for our proofs.

- In Appendix I, we discuss the multiplicative notion of $\varepsilon$-goodness, propose the EB-TC$_{\varepsilon_0}^{\mathrm{m}}$ algorithm and show asymptotic guarantees on the expected sample complexity when combined with the appropriate GLR stopping rule (Theorem 8).

- Implementation details and additional experiments are presented in Appendix J.

Table 1: Notation for the setting.

| Notation | Type | Description |
|---|---|---|
| $K$ | $\mathbb{N}$ | Number of arms |
| $\nu_i$ | $\mathcal{D}$ | Sub-Gaussian distribution of arm $i \in [K]$ |
| $\nu$ | $\mathcal{D}^K$ | Vector of sub-Gaussian distributions, $\nu := (\nu_i)_{i \in [K]}$ |
| $\mu_i$ | $\mathbb{R}$ | Mean of arm $i \in [K]$ |
| $\mu$ | $\mathbb{R}^K$ | Vector of means, $\mu := (\mu_i)_{i \in [K]}$ |
| $\mu_\star$ | $\mathbb{R}$ | Largest mean, $\mu_\star := \max_{i \in [K]} \mu_i$ |
| $\Delta_i$ | $\mathbb{R}_+$ | Gap of arm $i$, $\Delta_i := \mu_\star - \mu_i$ |
| $i^\star(\mu)$ | $[K]$ | Best arms, $i^\star(\mu) := \arg\max_{i \in [K]} \mu_i$ |
| $\varepsilon$ | $\mathbb{R}_+$ | Additive error of $\varepsilon$-good arms |
| $\mathcal{I}_\varepsilon(\mu)$ | $[K]$ | $\varepsilon$-Good arms, $\mathcal{I}_\varepsilon(\mu) := \{i \in [K] \mid \Delta_i \le \varepsilon\}$ |
| $T_\varepsilon(\mu), T_{\varepsilon,\beta}(\mu), T_{\varepsilon,\beta}(\mu, i)$ | $\mathbb{R}_+^\star$ | Asymptotic ($\beta$-)characteristic time for $\varepsilon$-BAI |
| $w_\varepsilon(\mu), w_{\varepsilon,\beta}(\mu), w_{\varepsilon,\beta}(\mu, i)$ | $\triangle_K$ | Asymptotic ($\beta$-)optimal allocation for $\varepsilon$-BAI |
| $C_\mu$ | $[K]$ | Number of equivalence classes, $C_\mu := |\{\mu_i \mid i \in [K]\}|$ |
| $\mathcal{C}_\mu(i)$ | $[K]$ | Equivalence class, $\mathcal{C}_\mu(i) := \{k \in [K] \mid \mu_\star - \mu_k = \Delta_i\}$ |

## B  Notation

We recall some commonly used notation: the set of integers $[n] := \{1, \cdots, n\}$, the complement $X^{\complement}$ and interior $\mathring{X}$ of a set $X$, the indicator function $\mathbb{1}(X)$ of an event, the probability $\mathbb{P}_\nu$ and the expectation $\mathbb{E}_\nu$ under distribution $\nu$, Landau's notation $o, \mathcal{O}, \Omega$ and $\Theta$, the $(K-1)$-dimensional probability simplex $\triangle_K := \left\{ w \in \mathbb{R}_+^K \mid w \ge 0, \sum_{i \in [K]} w_i = 1 \right\}$. The functions $\mathcal{C}_G$ and $g_G$ are defined in (23), $\overline{W}_{-1}$ in Lemma 49, $h_1$ in Lemma 51, $\zeta$ is the Riemann $\zeta$ function. While Table 1 gathers problem-specific notation, Table 2 groups notation for the algorithms.

Table 2: Notation for algorithms.

| Notation | Type | Description |
|---|---|---|
| $B_n, B_n^{\text{EB}}$ | $[K]$ | (EB) Leader at time $n$ |
| $C_n, C_n^{\text{TC}\varepsilon_0}$ | $[K]$ | (TC$_{\varepsilon_0}$) Challenger at time $n$ |
| $I_n$ | $[K]$ | Arm sampled at time $n$ |
| $X_{n,I_n}$ | $\mathbb{R}$ | Sample observed at the end of time $n$, i.e. $X_{n,I_n} \sim \nu_{I_n}$ |
| $\mathcal{F}_n$ | | History before time $n$, i.e. $\mathcal{F}_n := \sigma(I_1, X_{1,I_1}, \cdots, I_n, X_{n,I_n})$ |
| $\hat{\imath}_n$ | $[K]$ | Arm recommended before time $n$ |
| $\tau_{\varepsilon,\delta}$ | $\mathbb{N}$ | Sample complexity (stopping time) |
| $c(n, \delta)$ | $\mathbb{N} \times (0,1) \to \mathbb{R}_+^\star$ | Stopping threshold function |
| $N_{n,i}$ | $\mathbb{N}$ | Number of pulls of arm $i$ before time $n$ |
| $\mu_{n,i}$ | $\mathcal{I}$ | Empirical mean of arm $i$ before time $n$ |
| $T_n(i,j)$ | $\mathbb{N}$ | Counts of $(B_t, C_t) = (i,j)$ before time $n$ |
| $T_n(i)$ | $\mathbb{N}$ | Counts of $i \in \{B_t, C_t\}$ before time $n$ |
| $N_{n,j}^i$ | $\mathbb{N}$ | Counts of $(B_t, C_t, I_t) = (i,j,j)$ before time $n$ |
| $\beta$ | $(0,1)$ | Fixed proportion |
| $\beta_n(i,j)$ | $(0,1)$ | IDS proportions |
| $\bar{\beta}_n(i,j)$ | $(0,1)$ | Averaged IDS proportions |

## C  Characteristic Times

In Appendix C, we first recall properties satisfied by the ($\beta$-)characteristic times and ($\beta$-)optimal allocation for the BAI setting with Gaussian bandits. Then, we show how a $\varepsilon$-BAI problem can be reduced to a BAI one on a modified (and easier) instance.

**BAI setting**   Let $\nu \in \mathcal{D}^K$ with mean $\mu \in \mathbb{R}^K$. The ($\beta$-)characteristic times for the fixed-confidence BAI setting with Gaussian bandits $\mathcal{N}(\mu, 1)$ are defined as

$$T^\star(\mu) = \min_{\beta \in (0,1)} T_\beta^\star(\mu) \quad \text{with} \quad 2T_\beta^\star(\mu)^{-1} = \max_{w \in \triangle_K, w_{i^\star} = \beta} \min_{i^\star \neq i} \frac{(\mu_{i^\star} - \mu_i)^2}{1/\beta + 1/w_i} \,, \tag{7}$$

see [15, 34] for example. Let us denote by $w_\beta^\star(\mu)$ and $w^\star(\mu)$ their maximizer, which we will refer to as ($\beta$-)optimal allocations. Note that $(T^\star(\mu), T_\beta^\star(\mu)) = (T_0(\mu), T_{0,\beta}(\mu))$ and $(w^\star(\mu), w_\beta^\star(\mu)) = (w_0(\mu), w_{0,\beta}(\mu))$.

We recall in the following some fundamental results on those quantities ($\beta$-)characteristic times and ($\beta$-)optimal allocations. Lemma 6 proves that the ($\beta$-)optimal allocations are unique, have strictly positive elements, and shows a worst-case inequality $T_{1/2}^\star(\mu) \leq 2T^\star(\mu)$. Lemma 6 also holds for any single-parameter exponential family of distributions and for the non-parametric family of bounded distributions [21].

**Lemma 6** ([15, 34, 21]). *If $i^\star(\mu)$ is a singleton and $\beta \in (0,1)$, then $w^\star(\mu)$ and $w_\beta^\star(\mu)$ are singletons, i.e. the optimal allocations are unique, and $w^\star(\mu)_i > 0$ and $w_\beta^\star(\mu)_i > 0$ for all $i \in [K]$. Moreover, we have $T_{1/2}^\star(\mu) \leq 2T^\star(\mu)$ and with $\beta^\star = w_{i^\star}^\star(\mu)$. Additionally, we have $H(\mu) \leq T^\star(\mu) \leq 2H(\mu)$ where $H(\mu) = 2\sum_{i \in [K]} \Delta_i^{-2}$ where $\Delta_{i^\star} = \Delta_{\min}$.*

For Gaussian bandits, Lemma 7 shows that the ($\beta$-)characteristic times and ($\beta$-)optimal allocations can be obtained by solving a simpler optimization problem, which is amenable to faster implementation (e.g. Newton's iterates). It also provides upper and lower bound on the optimal allocation of the best arm, and gives hints that a tighter worst-case inequality $T_{1/2}^\star(\mu) \leq r_K T^\star(\mu)$ might hold.

**Lemma 7** ([3, 20]). *Assume that $i^\star(\mu) = \{i^\star\}$. Let $r(\mu)$ and $r_\beta(\mu)$ be the solution of $\psi_\mu(r) = 0$ and $\varphi_{\mu,\beta}(r) = 0$ with, for all $r \in (1/\min_{i \neq i^\star}(\mu_{i^\star} - \mu_i)^2, +\infty)$,*

$$\psi_\mu(r) = \sum_{i \neq i^\star} \frac{1}{\left(r(\mu_{i^\star} - \mu_i)^2 - 1\right)^2} - 1 \quad \text{and} \quad \varphi_{\mu,\beta}(r) = \sum_{i \neq i^\star} \frac{1}{r(\mu_{i^\star} - \mu_i)^2 - 1} - \frac{1 - \beta}{\beta} \,,$$

*Then, $\psi_\mu$ and $\varphi_{\mu,\beta}$ are convex and decreasing. Moreover,*

$$T^\star(\mu) = \frac{2r(\mu)}{1 + \sum_{i \neq i^\star} \frac{1}{r(\mu)(\mu_{i^\star} - \mu_i)^2 - 1}} \quad \text{and} \quad T^\star_\beta(\mu) = \frac{2r_\beta(\mu)}{\beta} .$$

*Additionally, for all $i \neq i^\star$, we have $\beta w^\star_\beta(\mu)_i^{-1} = \beta T^\star_\beta(\mu)(\mu_{i^\star} - \mu_i)^2/2 - 1$. For $K = 2$, $w^\star(\mu) = (0.5, 0.5)$ and $T^\star(\mu) = T^\star_{1/2}(\mu) = 8(\mu_1 - \mu_2)^2$. For $K \geq 3$, we have*

$$1/(\sqrt{K-1} + 1) \leq w^\star(\mu)_{i^\star} \leq 1/2 .$$

*Let $r_K = 2K/(1 + \sqrt{K-1})^2$, $j^\star \in \arg\min_{j \neq i^\star} \mu_{i^\star} - \mu_j$ and $x \in \mathbb{R}^{K-2}$ such that $x_j = \frac{\mu_{i^\star} - \mu_j}{\mu_{i^\star} - \mu_{j^\star}}$. Then, we have $T^\star_{1/2}(\mu)/T^\star(\mu) = \Omega(x)$ (i.e. independent of $\mu_{i^\star}$ and $\mu_{i^\star} - \mu_{j^\star}$) and*

$$\Omega(1_{K-2}) = r_K \quad \text{and} \quad \nabla_x \Omega(1_{K-2}) = 0_{K-2} .$$

Lemma 8 gathers necessary conditions on the ($\beta$-)optimal allocations at the equilibrium such as equality of the transportation costs, and a link between the squared allocation which was refered to as *overall balance* in [40].

**Lemma 8** ([3, 40]). *Assume that $i^\star(\mu) = \{i^\star\}$ and $\beta \in (0, 1)$. Then, for all $i \neq i^\star$,*

$$2T^\star(\mu)^{-1} = \frac{(\mu_{i^\star} - \mu_i)^2}{1/w^\star(\mu)_{i^\star} + 1/w^\star(\mu)_i} \quad \text{and} \quad 2T^\star_\beta(\mu)^{-1} = \frac{(\mu_{i^\star} - \mu_i)^2}{1/\beta + 1/w^\star_\beta(\mu)_i} .$$

*Moreover, we have*

$$w^\star(\mu)_{i^\star}^2 = \sum_{i \neq i^\star} w^\star(\mu)_i^2 .$$

**Reduction of an $\varepsilon$-BAI problem to a BAI problem**    Lemma 9 gives a reduction of a $\varepsilon$-BAI problem to a BAI one on a modified instance, which is easier. Thanks to Lemma 9, it is possible to leverage existing results on $T^\star(\mu), T^\star_\beta(\mu), w^\star(\mu), w^\star_\beta(\mu)$ (such as Lemmas 6, 7 and 8) in order to study $T_\varepsilon(\mu)$, $T_{\varepsilon,\beta}(\mu), T_{\varepsilon,\beta}(\mu, i), w_\varepsilon(\mu), w_{\varepsilon,\beta}(\mu)$ and $w_{\varepsilon,\beta}(\mu, i)$.

**Lemma 9.** *Let $\mu \in \mathbb{R}^K$, $\varepsilon \geq 0$ and $\tilde{\varepsilon} \in [0, \varepsilon]$ and $\beta \in (0, 1)$. For all $i \in \mathcal{I}_{\tilde{\varepsilon}}(\mu)$, we define $\mu_\varepsilon(i)$ as $\mu_\varepsilon(i)_j = \mu_j - \varepsilon$ for all $j \neq i$ and $\mu_\varepsilon(i)_i = \mu_i$. Then, for all $i \in \mathcal{I}_{\tilde{\varepsilon}}(\mu)$, $T_{\varepsilon,\beta}(\mu, i) = T^\star_\beta(\mu_\varepsilon(i))$ and $w_{\varepsilon,\beta}(\mu, i) = w^\star_\beta(\mu_\varepsilon(i))$. Moreover, for all $i^\star \in i^\star(\mu)$,*

$$T_\varepsilon(\mu) = T^\star(\mu_\varepsilon(i^\star)) \quad \text{and} \quad T_{\varepsilon,\beta}(\mu) = T^\star_\beta(\mu_\varepsilon(i^\star)) .$$

*Moreover, we have $w_\varepsilon(\mu) = \bigcup_{i^\star \in i^\star(\mu)} w^\star(\mu_\varepsilon(i^\star))$ and $w_{\varepsilon,\beta}(\mu) = \bigcup_{i^\star \in i^\star(\mu)} w^\star_\beta(\mu_\varepsilon(i^\star))$.*

*Proof.* For $\varepsilon = 0$, the result is direct by definition. Let $\varepsilon > 0$ and $\tilde{\varepsilon} \in [0, \varepsilon]$. The first part is obtained by definition of $T^\star_{\varepsilon,\beta}(\mu, i)$, $w^\star_{\varepsilon,\beta}(\mu, i)$. Let $i^\star \in i^\star(\mu)$ and $\mu_\varepsilon(i^\star)$ defined as above, hence $i^\star(\mu_\varepsilon(i^\star)) = \{i^\star\}$. For all $i \in \mathcal{I}_\varepsilon(\mu) \setminus i^\star(\mu)$, let us denote by $\lambda_\varepsilon^{(i^\star, i)}$ the instance such that $\lambda_{\varepsilon,i}^{(i^\star, i)} = \mu_i$ and $\lambda_{\varepsilon,j}^{(i^\star, i)} = \mu_j - \varepsilon$ for all $j \neq i$. If $\mu_{i^\star} - \mu_i = \varepsilon$ then $i^\star(\lambda_\varepsilon^{(i^\star, i)}) = \{i\} \cup i^\star(\mu) \setminus \{i^\star\}$, otherwise $i^\star(\lambda_\varepsilon^{(i^\star, i)}) = \{i\}$. We consider the permutation $\sigma$ that swaps arm $i$ with arm $i^\star$. By symmetry, we have $T^\star(\lambda_\varepsilon^{(i^\star, i)}) = T^\star(\sigma(\lambda_\varepsilon^{(i^\star, i)}))$. Moreover, we have that the gaps of $\sigma(\lambda_\varepsilon^{(i^\star, i)})$ are all strictly smaller than the gaps of $\mu_\varepsilon$ since $\varepsilon \geq \Delta_i > 0$. Therefore, Lemma 11 of Barrier et al (2022) yields that $T^\star(\sigma(\lambda_\varepsilon^{(i^\star, i)})) > T^\star(\mu_\varepsilon)$. We have proved that

$$\forall i^\star \in i^\star(\mu), \quad T^\star(\mu_\varepsilon(i^\star)) < \min_{i \in \mathcal{I}_\varepsilon(\mu) \setminus i^\star(\mu)} T^\star(\lambda_\varepsilon^{(i^\star, i)}) .$$

By symmetry $T^\star(\mu_\varepsilon(i^\star))$ is constant for all $i^\star \in i^\star(\mu)$, hence we have shown that

$$\forall i^\star \in i^\star(\mu), \quad T_\varepsilon(\mu) = T^\star(\mu_\varepsilon(i^\star)) .$$

It also shows that $w_\varepsilon(\mu) = \bigcup_{i^\star \in i^\star(\mu)} w^\star(\mu_\varepsilon(i^\star))$. The same reasoning yields the result for $T_{\varepsilon,\beta}(\mu)$ and $w_{\varepsilon,\beta}(\mu)$. $\qquad\square$

$\varepsilon$-**BAI problems cannot be arbitrarily difficult**  Lemma 10 shows that, for all $i \in \mathcal{I}_{\varepsilon/2}(\mu)$, $T_{\varepsilon,1/2}(\mu, i)$ cannot be arbitrarily large and $\min_{j \neq i} w_{\varepsilon_0,1/2}(\mu, i)_j$ cannot be arbitrarily small. The proof relies on Lemma 9 and existing results for BAI.

**Lemma 10.** *Let $\varepsilon > 0$ and $\mu \in \mathbb{R}^K$. Then, for all $i \in \mathcal{I}_{\varepsilon/2}(\mu)$, we have $T_{\varepsilon,1/2}(\mu, i) \leq 32K/\varepsilon^2$ and $\min_{j \neq i} w_{\varepsilon,1/2}(\mu, i)_j \geq (16(K-2)+2)^{-1}$.*

*Proof.* Let $i \in \mathcal{I}_{\varepsilon/2}(\mu)$, hence $\mu_i \geq \mu_\star - \varepsilon/2$. Let $\mu_\varepsilon(i)$ as in Lemma 9 which satisfies that $i^\star(\mu_\varepsilon(i)) = \{i\}$. Let $\Delta_{i,j} = \mu_i - \mu_j$ and $\Delta_{i,i} = \mu_i - \max_{j \neq i} \mu_j$. Then, we have

$$T_{\varepsilon,1/2}(\mu, i) = T_{1/2}^\star(\mu_\varepsilon(i)) \leq 2T^\star(\mu_\varepsilon(i)) \leq 8 \sum_{j \in [K]} (\Delta_{i,j} + \varepsilon)^{-2} \leq 8 \sum_{j \in [K]} (\Delta_j + \varepsilon/2)^{-2} \,,$$

where we used Lemma 6 for the two first inequality and $i \in \mathcal{I}_{\varepsilon/2}(\mu)$ for the last one. Since $\Delta_j \geq 0$, we conclude that $T_{\varepsilon,\beta}(\mu, i) \leq 32K/\varepsilon^2$. Moreover, using Lemma 6, we have

$$T_{\varepsilon,1/2}(\mu, i) = T_{1/2}^\star(\mu_\varepsilon(i)) \geq T^\star(\mu_\varepsilon(i)) \geq 2 \sum_{j \in [K]} (\Delta_{i,j} + \varepsilon)^{-2} \,.$$

Likewise, using Lemma 9 and Lemma 7, we obtain that, for all $j \neq i$,

$$w_{\varepsilon,1/2}(\mu)_j^{-1}/2 = w_{1/2}^\star(\mu_\varepsilon(i))_j^{-1}/2 = T_{1/2}^\star(\mu_\varepsilon(i))(\mu_i - \mu_j + \varepsilon)^2/4 - 1$$

$$\leq 2 \sum_{k \notin \{i,j\}} \left( \frac{\mu_i - \mu_j + \varepsilon}{\mu_i - \mu_k + \varepsilon} \right)^2 + 1$$

where the last inequality uses what we proved above. When $\mu_k \leq \mu_j$, the ratio is smaller than one. When $\mu_k > \mu_j$, we have $\mu_\star \geq \mu_k > \mu_j \geq \mu_\star - \varepsilon/2$, hence $\mu_k - \mu_j \leq \varepsilon/2$ and

$$\frac{\mu_i - \mu_j + \varepsilon}{\mu_i - \mu_k + \varepsilon} \leq 1 + \frac{\varepsilon/2}{\mu_\star - \mu_k + \varepsilon/2} \leq 2 \,.$$

Therefore, we obtain $w_{\varepsilon,1/2}(\mu)_j^{-1}/2 \leq 8(K-2) + 1$, which concludes the result. $\qquad\square$

Lemma 11 links the characteristic times for $\varepsilon$-BAI where $\varepsilon \in \{\varepsilon_0, \varepsilon_1\}$.

**Lemma 11.** *Let $\mu \in \mathbb{R}^K$ such that $|i^\star(\mu)| = 1$. Let $\varepsilon_0 > \varepsilon_1$. Then, for all $\beta \in (0,1)$, we have*
$T_{\varepsilon_0}(\mu)(\Delta_{\min}+\varepsilon_0)^2 \geq T_{\varepsilon_1}(\mu)(\Delta_{\min}+\varepsilon_1)^2$ *and* $T_{\varepsilon_0,\beta}(\mu)(\Delta_{\min}+\varepsilon_0)^2 \geq T_{\varepsilon_1,\beta}(\mu)(\Delta_{\min}+\varepsilon_1)^2$.
*Let $\varepsilon_0 < \varepsilon_1$. Then,*
$T_{\varepsilon_0}(\mu)(\Delta_{\max}+\varepsilon_0)^2 \geq T_{\varepsilon_1}(\mu)(\Delta_{\max}+\varepsilon_1)^2$ *and* $T_{\varepsilon_0,\beta}(\mu)(\Delta_{\max}+\varepsilon_0)^2 \geq T_{\varepsilon_1,\beta}(\mu)(\Delta_{\max}+\varepsilon_1)^2$.

*Proof.* Let $i^\star(\mu) = \{i^\star\}$. Let $\varepsilon_0 > \varepsilon_1$. Using Lemma 9, we have

$$2T_\varepsilon(\mu)^{-1}(\Delta_{\min} + \varepsilon)^{-2} = \max_{w \in \triangle_K} \min_{j \neq i^\star} \frac{\tilde{\Delta}_j(\varepsilon)^2}{1/w_{i^\star} + 1/w_j} \quad \text{with} \quad \tilde{\Delta}_j(\varepsilon) = \frac{\mu_\star - \mu_j + \varepsilon}{\Delta_{\min} + \varepsilon} \,.$$

To conclude the first part of the first result, a sufficient condition is to show that $\tilde{\Delta}_j(\varepsilon_1) \geq \tilde{\Delta}_j(\varepsilon_0)$ for all $j \neq i^\star$. Direct manipulations show that, for all $j \neq i^\star$,

$$\tilde{\Delta}_j(\varepsilon_1) \geq \tilde{\Delta}_j(\varepsilon_0) \iff 1 - \frac{\varepsilon_0 - \varepsilon_1}{\mu_\star - \mu_j + \varepsilon_0} \geq 1 - \frac{\varepsilon_0 - \varepsilon_1}{\Delta_{\min} + \varepsilon_0} \iff \mu_\star - \mu_j \geq \Delta_{\min} \,,$$

hence the result holds. The same proof can be used to obtain the second part of the first result.

Let $\varepsilon_0 < \varepsilon_1$. Using Lemma 9, we have

$$2T_\varepsilon(\mu)^{-1}(\Delta_{\max} + \varepsilon)^{-2} = \max_{w \in \triangle_K} \min_{j \neq i^\star} \frac{\bar{\Delta}_j(\varepsilon)^2}{1/w_{i^\star} + 1/w_j} \quad \text{with} \quad \bar{\Delta}_j(\varepsilon) = \frac{\mu_\star - \mu_j + \varepsilon}{\Delta_{\max} + \varepsilon} \,.$$

To conclude the first part of the second result, a sufficient condition is to show that $\bar{\Delta}_j(\varepsilon_1) \geq \bar{\Delta}_j(\varepsilon_0)$ for all $j \neq i^\star$. Direct manipulations show that, for all $j \neq i^\star$,

$$\bar{\Delta}_j(\varepsilon_1) \geq \bar{\Delta}_j(\varepsilon_0) \iff 1 + \frac{\varepsilon_1 - \varepsilon_0}{\mu_\star - \mu_j + \varepsilon_0} \geq 1 + \frac{\varepsilon_1 - \varepsilon_0}{\Delta_{\max} + \varepsilon_0} \iff \mu_\star - \mu_j \leq \Delta_{\max} \,,$$

hence the result holds. The same proof can be used to obtain the second part of the second result. $\quad\square$

# D  Asymptotic analysis

Let $\varepsilon_0 > 0$, $\beta \in (0,1)$ and $\delta \in (0,1)$. In this section, we provide an asymptotic analysis of EB-TC$_{\varepsilon_0}$ slack $\varepsilon_0 > 0$ combined with the stopping rule (3) with parameters $(\varepsilon_0, \delta)$. First, we detail IDS proportions in Appendix D.1. Then, we sketch the proof for fixed proportions $\beta$ in Appendix D.2.

In the following, we consider a sub-Gaussian bandit with distribution $\nu \in \mathcal{D}^K$ having mean parameter $\mu \in \mathbb{R}^K$ with a unique best arm, i.e. $i^\star(\mu) = \{i^\star\}$. We restate Theorem 1 below.

**Theorem 4** (Theorem 1). *Let $\varepsilon_0 > 0$, $\varepsilon_1 \geq 0$ and $\delta \in (0,1)$. Using the threshold (4) in the stopping rule (3) with slack $\varepsilon_1$, the EB-TC$_{\varepsilon_0}$ algorithm is $(\varepsilon_1, \delta)$-PAC. For IDS proportions, it satisfies that, for all $\nu \in \mathcal{D}^K$ such that $|i^\star(\mu)| = 1$,*

$$\limsup_{\delta \to 0} \frac{\mathbb{E}_\nu[\tau_{\varepsilon_1, \delta}]}{\log(1/\delta)} \leq T_{\varepsilon_0}(\mu) \left( 1 + \max_{i \neq i^\star} \frac{\varepsilon_0 - \varepsilon_1}{\mu_{i^\star} - \mu_i + \varepsilon_1} \right)^2 .$$

*Let $\beta \in (0,1)$. For fixed proportions $\beta$, it satisfies that, for all $\nu \in \mathcal{D}^K$ such that $|i^\star(\mu)| = 1$,*

$$\limsup_{\delta \to 0} \frac{\mathbb{E}_\nu[\tau_{\varepsilon_1, \delta}]}{\log(1/\delta)} \leq T_{\varepsilon_0, \beta}(\mu) \left( 1 + \max_{i \neq i^\star} \frac{\varepsilon_0 - \varepsilon_1}{\mu_{i^\star} - \mu_i + \varepsilon_1} \right)^2 .$$

Theorem 1 provides asymptotic upper bound on the expected sample complexity of the EB-TC$_{\varepsilon_0}$ algorithm when combined with the stopping rule (3). When $\varepsilon_1 = \varepsilon_0$ and for Gaussian distributions, combining Lemma 1 and Theorem 1 shows that the EB-TC$_{\varepsilon_0}$ algorithm is asymptotically optimal for $\varepsilon_0$-BAI when using IDS proportions and asymptotically $\beta$-optimal for $\varepsilon_0$-BAI when using fixed proportions $\beta$. Until recently [40], proving asymptotic optimality of Top Two algorithms with adaptive choice $\beta$ was an open problem in the BAI literature. Our work extends their analysis to tackle the $\varepsilon$-BAI setting. Compared to previous work, we removed the assumption that sub-optimal arms should have distinct means.

Considering $\varepsilon_1 \neq \varepsilon_0$ can be interesting when the practitioner decides to tackle $\varepsilon$-BAI with an error $\varepsilon$ which is different from the slack $\varepsilon_0$ used by the EB-TC$_{\varepsilon_0}$ algorithm. Those situations occur when the practitioner is observing the data collected by the EB-TC$_{\varepsilon_0}$ algorithm without having control over it. Theorem 1 show that the resulting sequential hypothesis testing has still asymptotic guarantees on the expected sample complexity. Quite naturally, those guarantees are not asymptotically optimal even for Gaussian distributions. The sub-optimality multiplicative gap of IDS proportions (that is, the ratio of the upper bound in Theorem 1 and the characteristic time for $\varepsilon_1$-BAI, $T_{\varepsilon_1}(\mu)$) is

$$\frac{T_{\varepsilon_0}(\mu)}{T_{\varepsilon_1}(\mu)} \frac{(\Delta_{\min} + \varepsilon_0)^2}{(\Delta_{\min} + \varepsilon_1)^2} \geq 1 \quad \text{when } \varepsilon_0 > \varepsilon_1 \text{ , otherwise} \quad \frac{T_{\varepsilon_0}(\mu)}{T_{\varepsilon_1}(\mu)} \frac{(\Delta_{\max} + \varepsilon_0)^2}{(\Delta_{\max} + \varepsilon_1)^2} \geq 1 \text{ ,}$$

where the inequalities comes from Lemma 11 and justify that there is no contradiction with the asymptotic lower bound on the expected sample complexity (Lemma 1). The same reasoning applies to fixed proportions $\beta$ by replacing $T_\varepsilon(\mu)$ by $T_{\varepsilon, \beta}(\mu)$.

**Concentration result**  For both proofs, we use the following concentration result of the empirical mean for sub-Gaussian observations (Lemma 12). This is a standard tool for the asymptotic analysis of Top Two algorithms (e.g. Lemma 3 in [32], Lemma 5 in [36] or Lemma 14 in [21]), hence we omit the proof.

**Lemma 12.** *There exists a sub-Gaussian random variable $W_\mu$ such that almost surely, for all $i \in [K]$ and all $n$ such that $N_{n,i} \geq 1$,*

$$|\mu_{n,i} - \mu_i| \leq W_\mu \sqrt{\frac{\log(e + N_{n,i})}{N_{n,i}}} .$$

*In particular, any random variable which is polynomial in $W_\mu$ has a finite expectation.*

## D.1  IDS proportions

Using Lemma 9, the optimal allocation for $\varepsilon_0$-BAI are defined as

$$w_{\varepsilon_0}(\mu) := \arg\max_{w \in \triangle_K} \min_{i \neq i^\star} \frac{(\mu_{i^\star} - \mu_i + \varepsilon_0)^2}{1/w_{i^\star} + 1/w_i} .$$

Since $|i^\star(\mu)| = 1$, let $\mu_{\varepsilon_0} = \mu_{\varepsilon_0}(i^\star)$ where $\mu_\varepsilon(i^\star)$ as in Lemma 9. Since we have $T_{\varepsilon_0}(\mu) = T^\star(\mu_{\varepsilon_0})$ and $w_{\varepsilon_0}(\mu) = w^\star(\mu_{\varepsilon_0})$, Lemma 6 and 8 yield that $w_{\varepsilon_0}(\mu) = \{w^\star\}$ is a singleton with unique element denotes by $w^\star$ which satisfies that $\min_{i \in [K]} w_i^\star > 0$,

$$\sum_{i \neq i^\star} \left(\frac{w_i^\star}{w_{i^\star}^\star}\right)^2 = 1 \,, \tag{8}$$

and

$$\forall i \in [K] \setminus \{i^\star\}, \quad \frac{(\mu_{i^\star} - \mu_i + \varepsilon_0)^2}{1/w_{i^\star}^\star + 1/w_i^\star} = 2 T_{\varepsilon_0}(\mu)^{-1} \,. \tag{9}$$

[40] refers to (8) as the overall balance equation. The key novelty of their analysis is to show that asymptotically the empirical proportions satisfy the same equation, referred to as the empirical overall balance, when using IDS proportions with sampling to choose among the leader and the challenger. The key novelty in our work lies in proving that the empirical overall balance equation is also satisfied when using IDS proportions with $K(K-1)$ tracking procedures to select between the leader and the challenger (see Appendix D.1.2).

**Convergence time** We follow an asymptotic analysis similar to the ones used in [32, 36, 21] with the improvement from [40]. Let $\gamma > 0$. Let us define the *convergence time* $T_{\varepsilon_0, \gamma}$, which is a random variable quantifies the number of samples required for the empirical allocations $(N_{n,i}/N_{n,i^\star})_{i \neq i^\star}$ to be $\gamma$-close to $(w_i^\star/w_{i^\star}^\star)_{i \neq i^\star}$:

$$T_{\varepsilon_0, \gamma} := \inf \left\{ T \geq 1 \mid \forall n \geq T, \max_{i \neq i^\star} \left| \frac{N_{n,i}}{N_{n,i^\star}} - \frac{w_i^\star}{w_{i^\star}^\star} \right| \leq \gamma \right\} \,. \tag{10}$$

Lemma 13 gives a sufficient condition to prove Theorem 1 for IDS proportions. The case $\varepsilon_0 = \varepsilon_1$ is a direct consequence of combining Theorem 13 and Proposition 14 in [40]. The proof is inspired by existing methods, e.g. Theorem 2 in [21] or Theorem 3 in [32].

**Lemma 13.** *Let $\varepsilon_0 > 0$, $\varepsilon_1 \geq 0$ and $\delta \in (0, 1)$. Assume that there exists $\gamma_1(\mu) > 0$ such that for all $\gamma \in (0, \gamma_1(\mu)]$, $\mathbb{E}_\nu[T_{\varepsilon_0, \gamma}] < +\infty$. Using the threshold (4) in the stopping rule (3) with slack $\varepsilon_1$ yields an algorithm such that, for all $\nu \in \mathcal{D}^K$ such that $|i^\star(\mu)| = 1$,*

$$\limsup_{\delta \to 0} \frac{\mathbb{E}_\nu[\tau_{\varepsilon_1, \delta}]}{\log(1/\delta)} \leq T_{\varepsilon_0}(\mu) \left(1 + \max_{i \neq i^\star} \frac{\varepsilon_0 - \varepsilon_1}{\mu_{i^\star} - \mu_i + \varepsilon_1}\right)^2 \,.$$

*Proof.* Let $\gamma > 0$. Let us define by

$$\tilde{T}_{\gamma, \varepsilon_0} := \inf \left\{ T \geq 1 \mid \forall n \geq T, \left\| \frac{N_n}{n-1} - w^\star \right\|_\infty \leq \gamma \right\} \,.$$

Using that

$$\sum_{i \neq i^\star} \frac{w_i^\star}{w_{i^\star}^\star} = \frac{1}{w_{i^\star}^\star} - 1 \quad \text{and} \quad \sum_{i \neq i^\star} \frac{N_{n,i}}{N_{n,i^\star}} = \frac{n-1}{N_{n,i^\star}} - 1 \,,$$

it is direct to see that for all $n \geq T_\gamma$

$$\left| \frac{N_{n,i^\star}}{n-1} - w_{i^\star}^\star \right| = \left| \frac{n-1}{N_{n,i^\star}} - \frac{1}{w_{i^\star}^\star} \right| \frac{N_{n,i^\star}}{n-1} w_{i^\star}^\star \leq \left| \frac{n-1}{N_{n,i^\star}} - \frac{1}{w_{i^\star}^\star} \right| \leq \gamma(K-1) \,,$$

$$\left| \frac{N_{n,i}}{n-1} - w_i^\star \right| = w_{i^\star}^\star \left| \frac{N_{n,i}}{n-1} \left( \frac{1}{w_{i^\star}^\star} - \frac{n-1}{N_{n,i^\star}} \right) + \frac{N_{n,i}}{N_{n,i^\star}} - \frac{w_i^\star}{w_{i^\star}^\star} \right|$$

$$\leq \left| \frac{1}{w_{i^\star}^\star} - \frac{n-1}{N_{n,i^\star}} \right| + \left| \frac{N_{n,i}}{N_{n,i^\star}} - \frac{w_i^\star}{w_{i^\star}^\star} \right| \leq \gamma K \,.$$

Therefore, for all $\gamma \in (0, \gamma_1(\mu)/K]$, $\mathbb{E}_\nu[\tilde{T}_{\gamma, \varepsilon_0}] < +\infty$.

Let $\mu_{\varepsilon_0}$ as in Lemma 9. Since we have $T_{\varepsilon_0}(\mu) = T^\star(\mu_{\varepsilon_0})$ and $w_{\varepsilon_0}(\mu) = w^\star(\mu_{\varepsilon_0})$, we can leverage existing results from the BAI literature, e.g. Theorem 2 in [21] or Theorem 3 in [32]. The sole criterion on the stopping threshold is to be asymptotically tight (Definition 1).

**Definition 1.** *A threshold $c : \mathbb{N} \times (0, 1] \to \mathbb{R}_+$ is said to be asymptotically tight if there exists $\alpha \in [0, 1)$, $\delta_0 \in (0, 1]$, functions $f, \bar{T} : (0, 1] \to \mathbb{R}_+$ and $C$ independent of $\delta$ satisfying: (1) for all $\delta \in (0, \delta_0]$ and $n \geq \bar{T}(\delta)$, then $c(n, \delta) \leq f(\delta) + Cn^{\alpha}$, (2) $\limsup_{\delta \to 0} f(\delta) / \log(1/\delta) \leq 1$ and (3) $\limsup_{\delta \to 0} \bar{T}(\delta) / \log(1/\delta) = 0$.*

Since $\mathcal{C}_G$ defined in (23) satisfies $\mathcal{C}_G \approx x + \log(x)$, it is direct to see that

$$c(n, \delta) = 2\mathcal{C}_G \left( \frac{1}{2} \log \left( \frac{K-1}{\delta} \right) \right) + 4 \log \left( 4 + \log \frac{n}{2} \right)$$

is asymptotically tight, e.g. by taking $(\alpha, \delta_0, C) = (1/2, 1, 4)$, $f(\delta) = 2\mathcal{C}_G \left( \frac{1}{2} \log \left( \frac{K-1}{\delta} \right) \right)$ and $\bar{T}(\delta) = 1$.

Based on continuity arguments and using that $\mathbb{E}_\nu[T_{\varepsilon_0, \gamma}] < +\infty$, the proof of Theorem 2 in [21] yields that

$$\limsup_{\delta \to 0} \frac{\mathbb{E}_\nu[\tau_\delta]}{\log(1/\delta)} \leq \left( \min_{i \neq i^\star} \frac{(\mu_{i^\star} - \mu_i + \varepsilon_1)^2}{2(1/w_{i^\star}^\star + 1/w_i^\star)} \right)^{-1} = T_{\varepsilon_0}(\mu) \max_{i \neq i^\star} \left( \frac{\mu_{i^\star} - \mu_i + \varepsilon_0}{\mu_{i^\star} - \mu_i + \varepsilon_1} \right)^2 ,$$

where the equality uses the condition at the equilibrium (9). This concludes the proof. □

Using Lemma 13, the proof of Theorem 1 for IDS proportions boils down to showing that $\mathbb{E}_\nu[T_{\varepsilon_0, \gamma}] < +\infty$. As in [32, 36, 21, 40], the proof is divided in several steps. First, we show that all the arms are sufficient explored for $n$ large enough (Appendix D.1.1). Second, we prove that the empirical overall balance equation approximately holds for $n$ large enough (Appendix D.1.2). Finally, we show convergence of the empirical ratio of proportions towards the ratio of optimal allocation, i.e. $\mathbb{E}_\nu[T_{\varepsilon_0, \gamma}] < +\infty$ (Appendix D.1.3).

### D.1.1 Sufficient exploration

To upper bound the expected convergence time, as prior work we first establish sufficient exploration. Given an arbitrary threshold $L \in \mathbb{R}_+^*$, we define the sampled enough set and its arms with highest mean (when not empty) as

$$S_n^L := \{i \in [K] \mid N_{n,i} \geq L\} \quad \text{and} \quad \mathcal{I}_n^\star := \arg\max_{i \in S_n^L} \mu_i . \tag{11}$$

In all generality $\mathcal{I}_n^\star$ is a set, yet we obtain $\mathcal{I}_n^\star = \{i^\star\}$ as soon as $i^\star \in S_n^L$. We define the highly and the mildly under-sampled sets

$$U_n^L := \{i \in [K] \mid N_{n,i} < \sqrt{L}\} \quad \text{and} \quad V_n^L := \{i \in [K] \mid N_{n,i} < L^{3/4}\} . \tag{12}$$

[21] identifies the properties that the leader and the challenger should satisfy to ensure sufficient exploration. We improve on their analysis by removing the need for the distinct means assumption, and supporting IDS instead of a fixed allocation $\beta$.

Lemma 14 shows that the transportation cost is strictly positive and increases linearly. Compared to previous results, the key improvement lies in the fact that the lower bound holds for $(i, j) \in \mathcal{I}_n^\star \times S_n^L$ instead of $(i, j) \in \mathcal{I}_n^\star \times (S_n^L \setminus \mathcal{I}_n^\star)$. This is due to the $\varepsilon_0$ regularization, which removes the need to assume that the means are distinct.

**Lemma 14.** *Let $S_n^L$ and $\mathcal{I}_n^\star$ as in (11). There exists $L_0$ with $\mathbb{E}_\mu[(L_0)^\alpha] < +\infty$ for all $\alpha > 0$ such that if $L \geq L_0$, for all $n$ such that $S_n^L \neq \emptyset$, for all $(i, j) \in \mathcal{I}_n^\star \times S_n^L$ such that $i \neq j$, we have*

$$\frac{\mu_{n,i} - \mu_{n,j} + \varepsilon_0}{\sqrt{1/N_{n,i} + 1/N_{n,j}}} \geq \sqrt{L} \frac{\varepsilon_0}{2\sqrt{2}} .$$

*Proof.* Using Lemma 12 and $\min\{N_{n,i}, N_{n,j}\} \geq L$, we obtain

$$\mu_{n,i} - \mu_{n,j} \geq \mu_i - \mu_j - W_\mu \left( \sqrt{\frac{\log(e + N_{n,i})}{N_{n,i}}} + \sqrt{\frac{\log(e + N_{n,j})}{N_{n,j}}} \right)$$

$$\geq -2W_\mu \sqrt{\frac{\log(e + L)}{L}} \geq -\frac{\varepsilon_0}{2} ,$$

where the last inequality is obtained for $L \geq L_0 = L_1 - e$ which is defined as

$$L_1 = \sup\left\{x \mid x < 16W_\mu^2 \log(x)/\varepsilon_0^2 + e\right\} \leq h_1\left(16W_\mu/\varepsilon_0^2, e\right) .$$

The last inequality is obtained by using Lemma 51, and we recall that $h_1$ is defined in Lemma 51. Since $h_1(x, e) \sim_{x \to +\infty} x \log x$, we have $\mathbb{E}_\mu[(L_0)^\alpha] < +\infty$ for all $\alpha > 0$ by using Lemma 12 (polynomial in $W_\mu$). Then, we have

$$\frac{\mu_{n,i} - \mu_{n,j} + \varepsilon_0}{\sqrt{1/N_{n,i} + 1/N_{n,j}}} \geq \frac{\varepsilon_0/2}{\sqrt{1/N_{n,i} + 1/N_{n,j}}} \geq \sqrt{L}\frac{\varepsilon_0}{2\sqrt{2}} .$$

$\square$

Lemma 15 gives an upper bound on the transportation costs between a sampled enough arm and an under-sampled one.

**Lemma 15.** *Let $S_n^L$ as in (11). There exists $L_1$ with $\mathbb{E}_\mu[(L_1)^\alpha] < +\infty$ for all $\alpha > 0$ such that for all $L \geq L_1$ and all $n \in \mathbb{N}$,*

$$\forall (i,j) \in S_n^L \times \overline{S_n^L}, \quad \frac{\mu_{n,i} - \mu_{n,j} + \varepsilon_0}{\sqrt{1/N_{n,i} + 1/N_{n,j}}} \leq \sqrt{L}(D_1 + 4W_\mu) ,$$

*where $D_1 > 0$ is a problem dependent constant and $W_\mu$ is the random variables defined in Lemma 12.*

*Proof.* Using Lemma 12 and $N_{n,i} \geq L \geq N_{n,j} \geq 1$, we obtain

$$\frac{\mu_{n,i} - \mu_{n,j} + \varepsilon_0}{\sqrt{1/N_{n,i} + 1/N_{n,j}}} \leq \sqrt{N_{n,j}}|\mu_{n,i} - \mu_{n,j} + \varepsilon_0|$$

$$\leq \sqrt{L}\left(|\mu_i - \mu_j + \varepsilon_0| + W_\mu\left(\sqrt{\frac{\log(e + N_{n,i})}{N_{n,i}}} + \sqrt{\frac{\log(e + N_{n,j})}{N_{n,j}}}\right)\right)$$

$$\leq \sqrt{L}\left(|\mu_i - \mu_j + \varepsilon_0| + 2W_\mu\sqrt{\log(e + 1)}\right)$$

$$\leq \sqrt{L}(D_1 + 4W_\mu) ,$$

for $D_1 = \max_{i \neq j} |\mu_i - \mu_j + \varepsilon_0|$. $\square$

Lemma 16 shows the desired property for the EB leader. Since it corresponds to Lemma 17 in [21], we omit the proof.

**Lemma 16** (Lemma 17 in [21]). *There exists $L_2$ with $\mathbb{E}_\nu[(L_2)^\alpha] < +\infty$ for all $\alpha > 0$ such that if $L \geq L_2$, for all $n$ (at most polynomial in $L$) such that $S_n^L \neq \emptyset$, $B_n^{EB} \in S_n^L$ implies $B_n^{EB} \in \mathcal{I}_n^\star$.*

Lemma 17 show that the desired property for the TC challenger. Compared to existing result, we improve by remove the assumption that all arms have distinct means thanks to the regularization $\varepsilon_0 > 0$.

**Lemma 17.** *There exists $L_3$ with $\mathbb{E}_\nu[L_3] < +\infty$ such that if $L \geq L_3$, for all $n$ (at most polynomial in $L$) such that $U_n^L \neq \emptyset$, $B_n^{EB} \notin V_n^L$ implies $C_n^{TC\varepsilon_0} \in V_n^L$.*

*Proof.* In the following, we consider $U_n^L \neq \emptyset$ (hence $V_n^L \neq \emptyset$) and $B_n^{EB} \notin V_n^L$. Let $\mathcal{J}_n^\star = \arg\max_{i \in \overline{V_n^L}} \mu_i$ and $L_2$ as in Lemma 16. Then, for $L \geq L_2^{4/3}$, we have $B_n^{EB} \in \mathcal{J}_n^\star$.

Let $L_0$ and $(L_1, D_1)$ as in Lemmas 14 and 15. Then, for all $L \geq \max\{L_0^{4/3}, L_2^{4/3}, L_1^2\}$,

$$B_n^{EB} \in \mathcal{J}_n^\star ,$$

$$\forall (i,j) \in \mathcal{J}_n^\star \times \overline{V_n^L}, \text{ s.t. } i \neq j, \quad \frac{\mu_{n,i} - \mu_{n,j} + \varepsilon_0}{\sqrt{1/N_{n,i} + 1/N_{n,j}}} \geq L^{3/8}\frac{\varepsilon_0}{2\sqrt{2}} ,$$

$$\forall (i,j) \in \overline{U_n^L} \times U_n^L, \quad \frac{\mu_{n,i} - \mu_{n,j} + \varepsilon_0}{\sqrt{1/N_{n,i} + 1/N_{n,j}}} \leq L^{1/4}(D_1 + 4W_\mu) .$$

Since $\mathcal{J}_n^\star \subseteq \overline{V_n^L} \subseteq \overline{U_n^L}$, for all $L \geq L_3 := \max\{L_0^{4/3}, L_2^{4/3}, L_1^2, \left(\frac{2\sqrt{2}}{\varepsilon_0}(D_1 + 4W_\mu)\right)^8 + 1\}$ we have

$$\forall (i, k, j) \in \mathcal{J}_n^\star \times U_n^L \times \overline{V_n^L}, \text{ s.t. } i \neq j, \quad \frac{\mu_{n,i} - \mu_{n,j} + \varepsilon_0}{\sqrt{1/N_{n,i} + 1/N_{n,j}}} > \frac{\mu_{n,i} - \mu_{n,k} + \varepsilon_0}{\sqrt{1/N_{n,i} + 1/N_{n,k}}} \, .$$

As $B_n^{\text{EB}} \in \mathcal{J}_n^\star$, the definition $C_n^{\text{TC}\varepsilon_0}$ yields that $C_n^{\text{TC}\varepsilon_0} \in V_n^L$. Otherwise the above strict inequality would wield a contradiction. Using Lemma 12, we show

$$\mathbb{E}_{\boldsymbol{F}}[L_3] \leq \mathbb{E}_{\boldsymbol{F}}\left[\left(\frac{2\sqrt{2}}{\varepsilon_0}(D_1 + 4W_\mu)\right)^8 + 1\right] + \mathbb{E}_{\boldsymbol{F}}[(L_0)^{4/3}] + \mathbb{E}_{\boldsymbol{F}}[(L_2)^{4/3}] + \mathbb{E}_{\boldsymbol{F}}[(L_1)^2] < +\infty \, ,$$

hence this concludes the proof. $\qquad\square$

Lemma 18 shows that all arms are sufficiently explored and that the leader is the unique best arm for $n$ large enough.

**Lemma 18.** *There exist $N_1$ with $\mathbb{E}_\nu[N_1] < +\infty$ such that for all $n \geq N_1$ and all $i \in [K]$, $N_{n,i} \geq \sqrt{n/K}$ and $B_n^{EB} = i^\star$.*

*Proof.* Let $L_2$ and $L_3$ as in Lemmas 16 and 17. Therefore, for $L \geq L_4 := \max\{L_3, L_2^{4/3}\}$, for all $n$ such that $U_n^L \neq \emptyset$, $B_n^{\text{EB}} \in V_n^L$ or $C_n^{\text{TC}\varepsilon_0} \in V_n^L$. We have $\mathbb{E}_\nu[L_4] < +\infty$. There exists a deterministic $L_5$ such that for all $L \geq L_5$, $\lfloor L \rfloor \geq KL^{3/4}$. Let $L \geq \max\{L_5, L_4\}$.

Suppose towards contradiction that $U_{\lfloor KL \rfloor}^L$ is not empty. Then, for any $1 \leq t \leq \lfloor KL \rfloor$, $U_t^L$ and $V_t^L$ are non empty as well. Using the pigeonhole principle, there exists some $i \in [K]$ such that $N_{\lfloor L \rfloor, i} \geq L^{3/4}$. Thus, we have $\left|V_{\lfloor L \rfloor}^L\right| \leq K - 1$. Our goal is to show that $\left|V_{\lfloor 2L \rfloor}^L\right| \leq K - 2$. A sufficient condition is that one arm in $V_{\lfloor L \rfloor}^L$ is pulled at least $L^{3/4}$ times between $\lfloor L \rfloor$ and $\lfloor 2L \rfloor - 1$.

If we have

$$\sum_{t=\lfloor L \rfloor}^{\lfloor 2L \rfloor - 1} \mathbb{1}\left(\{B_t^{\text{EB}}, C_t^{\text{TC}\varepsilon_0}\} \subseteq V_t^L\right) \geq KL^{3/4} \, ,$$

then, using that $V_t^L \subseteq V_{\lfloor L \rfloor}^L$, we obtain

$$\sum_{t=\lfloor L \rfloor}^{\lfloor 2L \rfloor - 1} \mathbb{1}\left(I_t \in V_{\lfloor L \rfloor}^L\right) \geq \sum_{t=\lfloor L \rfloor}^{\lfloor 2L \rfloor - 1} \mathbb{1}\left(I_t \in V_t^L\right) \geq \sum_{t=\lfloor L \rfloor}^{\lfloor 2L \rfloor - 1} \mathbb{1}\left(\{B_t^{\text{EB}}, C_t^{\text{TC}\varepsilon_0}\} \subseteq V_t^L\right) \geq KL^{3/4} \, ,$$

Therefore, there exists $i \in V_{\lfloor L \rfloor}^L$ which is sampled $L^{3/4}$ times between $\lfloor L \rfloor$ and $\lfloor 2L \rfloor - 1$.

In the following, we assume that

$$\sum_{t=\lfloor L \rfloor}^{\lfloor 2L \rfloor - 1} \mathbb{1}\left(\{B_t^{\text{EB}}, C_t^{\text{TC}\varepsilon_0}\} \subseteq V_t^L\right) < KL^{3/4} \, ,$$

Since $B_t^{\text{EB}} \in V_t^L$ or $C_t^{\text{TC}\varepsilon_0} \in V_t^L$, we have

$$\sum_{t=\lfloor L \rfloor}^{\lfloor 2L \rfloor - 1} \left(\mathbb{1}\left(B_t^{\text{EB}} \in V_t^L, C_t^{\text{TC}\varepsilon_0} \notin V_t^L\right) + \mathbb{1}\left(B_t^{\text{EB}} \notin V_t^L, C_t^{\text{TC}\varepsilon_0} \in V_t^L\right)\right) > \lfloor 2L \rfloor - \lfloor L \rfloor - KL^{3/4} \, .$$

Therefore, we have

**Case 1:** $\quad \displaystyle\sum_{t=\lfloor L \rfloor}^{\lfloor 2L \rfloor - 1} \mathbb{1}\left(B_t^{\text{EB}} \in V_t^L, C_t^{\text{TC}\varepsilon_0} \notin V_t^L\right) > \left(\lfloor 2L \rfloor - \lfloor L \rfloor - KL^{3/4}\right)/2 \, ,$

or **Case 2:** $\quad \displaystyle\sum_{t=\lfloor L \rfloor}^{\lfloor 2L \rfloor - 1} \mathbb{1}\left(B_t^{\text{EB}} \notin V_t^L, C_t^{\text{TC}\varepsilon_0} \in V_t^L\right) > \left(\lfloor 2L \rfloor - \lfloor L \rfloor - KL^{3/4}\right)/2 \, .$

Since $\beta_t(i,j) = N_{t,j}/(N_{t,j} + N_{t,i})$, we obtain

$$1/2 \leq \begin{cases} \beta_t(B_t^{\text{EB}}, C_t^{\text{TC}\varepsilon_0}) & \text{if} \quad B_t^{\text{EB}} \in V_t^L, C_t^{\text{TC}\varepsilon_0} \notin V_t^L \\ 1 - \beta_t(B_t^{\text{EB}}, C_t^{\text{TC}\varepsilon_0}) & \text{if} \quad B_t^{\text{EB}} \notin V_t^L, C_t^{\text{TC}\varepsilon_0} \in V_t^L \end{cases}.$$

This lower bound will be crucial to argue that the challenger is sampled enough in case 2 and the leader is sampled enough in case 1.

**Case 1.** Using Lemma 46 and the above, we obtain

$$\sum_{t=\lfloor L \rfloor}^{\lfloor 2L \rfloor - 1} \mathbb{1}\left(I_t \in V_{\lfloor L \rfloor}^L\right) \geq \sum_{i \in V_{\lfloor L \rfloor}^L} \sum_{j \neq i} \sum_{t=\lfloor L \rfloor}^{\lfloor 2L \rfloor - 1} \mathbb{1}\left(I_t = i, (B_t^{\text{EB}}, C_t^{\text{TC}\varepsilon_0}) = (i,j)\right)$$

$$\geq \sum_{i \in V_{\lfloor L \rfloor}^L} \sum_{j \neq i} \sum_{t=\lfloor L \rfloor}^{\lfloor 2L \rfloor - 1} \beta_t(i,j) \mathbb{1}\left((B_t^{\text{EB}}, C_t^{\text{TC}\varepsilon_0}) = (i,j)\right) - K^2$$

$$\geq \sum_{t=\lfloor L \rfloor}^{\lfloor 2L \rfloor - 1} \beta_t(B_t^{\text{EB}}, C_t^{\text{TC}\varepsilon_0}) \mathbb{1}\left(B_t^{\text{EB}} \in V_t^L, C_t^{\text{TC}\varepsilon_0} \notin V_t^L\right) - K^2$$

$$\geq \left(\lfloor 2L \rfloor - \lfloor L \rfloor - KL^{3/4}\right)/4 - K^2 \geq KL^{3/4},$$

where the last inequality is obtained for $L \geq L_6 + 1$ with

$$L_6 = \sup\left\{L \in \mathbb{N} \mid \left(\lfloor 2L \rfloor - \lfloor L \rfloor - KL^{3/4}\right)/4 - K^2 < KL^{3/4}\right\}.$$

Therefore, there exists $i \in V_{\lfloor L \rfloor}^L$ which is sampled $L^{3/4}$ times between $\lfloor L \rfloor$ and $\lfloor 2L \rfloor - 1$.

**Case 2.** Using Lemma 46 and the above, we obtain

$$\sum_{t=\lfloor L \rfloor}^{\lfloor 2L \rfloor - 1} \mathbb{1}\left(I_t \in V_{\lfloor L \rfloor}^L\right) \geq \sum_{j \in V_{\lfloor L \rfloor}^L} \sum_{i \neq j} \sum_{t=\lfloor L \rfloor}^{\lfloor 2L \rfloor - 1} \mathbb{1}\left(I_t = j, (B_t^{\text{EB}}, C_t^{\text{TC}\varepsilon_0}) = (i,j)\right)$$

$$\geq \sum_{j \in V_{\lfloor L \rfloor}^L} \sum_{i \neq j} \sum_{t=\lfloor L \rfloor}^{\lfloor 2L \rfloor - 1} (1 - \beta_t(i,j)) \mathbb{1}\left((B_t^{\text{EB}}, C_t^{\text{TC}\varepsilon_0}) = (i,j)\right) - K^2$$

$$\geq \sum_{t=\lfloor L \rfloor}^{\lfloor 2L \rfloor - 1} (1 - \beta_t(B_t^{\text{EB}}, C_t^{\text{TC}\varepsilon_0})) \mathbb{1}\left(B_t^{\text{EB}} \notin V_t^L, C_t^{\text{TC}\varepsilon_0} \in V_t^L\right) - K^2$$

$$\geq \left(\lfloor 2L \rfloor - \lfloor L \rfloor - KL^{3/4}\right)/4 - K^2 \geq KL^{3/4},$$

where the last inequality is obtained for $L \geq L_6 + 1$. Therefore, there exists $i \in V_{\lfloor L \rfloor}^L$ which is sampled $L^{3/4}$ times between $\lfloor L \rfloor$ and $\lfloor 2L \rfloor - 1$.

**Summary.** We have shown $\left|V_{\lfloor 2L \rfloor}^L\right| \leq K - 2$. By induction, for any $1 \leq k \leq K$, we have $\left|V_{\lfloor kL \rfloor}^L\right| \leq K - k$, and finally $U_{\lfloor KL \rfloor}^L = \emptyset$ for all $L \geq L_7 := \max\{L_5, L_4, L_6 + 1\}$. Defining $N_0 = KL_7$, we have $\mathbb{E}_\nu[N_0] < +\infty$. For all $n \geq N_0$, we let $L = n/K$, hence we have $U_n^{n/K} = U_{\lfloor KL \rfloor}^L = \emptyset$. This concludes the proof of sufficient exploration.

**Leader is best arm.** Using Lemma 12, we obtain that for all $n \geq N_0$,

$$\forall i \neq i^\star, \quad \mu_{n,i} \leq \mu_i + W_\mu \sqrt{\frac{\log(e + N_{n,i})}{N_{n,i}}} \leq \mu_i + W_\mu \sqrt{\frac{\log(e + \sqrt{n/K})}{\sqrt{n/K}}},$$

$$\mu_{n,i^\star} \leq \mu_{i^\star} - W_\mu \sqrt{\frac{\log(e + N_{n,i^\star})}{N_{n,i^\star}}} \leq \mu_{i^\star} - W_\mu \sqrt{\frac{\log(e + \sqrt{n/K})}{\sqrt{n/K}}}.$$

Therefore, we have $B_n^{\text{EB}} = i^\star$ for all $n \geq N_1 := \max\{N_0, K(X_0 - e)^2 + 1\}$ where

$$X_0 = \sup\left\{ x > 1 \mid x < \frac{16W_\mu^2}{\min_{i \neq i^\star}(\mu_{i^\star} - \mu_i)^2} \log(x) + e \right\} \leq h_1\left( \frac{16W_\mu^2}{\min_{i \neq i^\star}(\mu_{i^\star} - \mu_i)^2}, e \right),$$

where the last inequality is obtained by using Lemma 51 with $h_1$ defined therein. Since $h_1(x, e) \sim_{x \to +\infty} x \log x$, we have

$$\mathbb{E}_\nu[N_1] < \mathbb{E}_\nu[N_0] + \mathbb{E}_\nu[K(X_0 - e)^2 + 1] < +\infty.$$

This concludes the proof. $\qquad\square$

### D.1.2 Empirical overall balance

As in [40], the key to obtain asymptotic optimality is to show that the empirical proportion satisfy the empirical overall balance equation. Compared to them, the novelty of Lemma 19 is that we use IDS proportions with $K(K-1)$ tracking procedures to select between the leader and the challenger instead of sampling.

**Lemma 19.** *Let $\gamma > 0$. There exists $N_2$ with $\mathbb{E}_\nu[N_2] < +\infty$ such that for all $n \geq N_2$*

$$\left| \left( \frac{N_{n,i^\star}}{n-1} \right)^2 - \sum_{i \neq i^\star} \left( \frac{N_{n,i}}{n-1} \right)^2 \right| \leq \gamma.$$

*Proof.* Let $N_1$ as in Lemma 18. Then, we have for all $n > N_1$

$$\frac{N_{n,i^\star}}{n-1} = \frac{N_{n,i^\star}^{i^\star}}{n-1} + \frac{\sum_{i \neq i^\star} N_{N_1,i^\star}^i}{n-1} \quad , \quad \frac{N_{n,i}}{n-1} = \frac{N_{n,i}^{i^\star}}{n-1} + \frac{\sum_{j \neq i^\star} N_{N_1,i}^j}{n-1},$$

$$(1 - \bar{\beta}_n(i^\star, j))\frac{T_n(i^\star, j)}{n-1} = \frac{1}{n-1} \sum_{t=N_1}^{n-1} \mathbb{1}(C_t = j)(1 - \beta_t(i^\star, j)) + (1 - \bar{\beta}_{N_1}(i^\star, j))\frac{T_{N_1}(i^\star, j)}{n-1},$$

$$\sum_{j \neq i^\star} \bar{\beta}_n(i^\star, j)\frac{T_n(i^\star, j)}{n-1} = \frac{1}{n-1} \sum_{t=N_1}^{n-1} \beta_t(i^\star, C_t) + \sum_{j \neq i^\star} \bar{\beta}_{N_1}(i^\star, j)\frac{T_{N_1}(i^\star, j)}{n-1}$$

Chaining the inequalities with Lemma 46, we obtain that

$$-\frac{K}{n-1} \leq \frac{N_{n,i^\star}}{n-1} - \frac{1}{n-1} \sum_{t=N_1}^{n-1} \beta_t(i^\star, C_t) \leq \frac{2N_1 + K}{n-1},$$

$$-\frac{1}{n-1} \leq \frac{N_{n,i}}{n-1} - \frac{1}{n-1} \sum_{t=N_1}^{n-1} \mathbb{1}(C_t = j)(1 - \beta_t(i^\star, j)) \leq \frac{2N_1 + 1}{n-1}.$$

Using that $a^2 - b^2 = (a-b)(a+b) \leq 2(a-b)$ for $(a, b) \in (0, 1)^2$, we obtain

$$-2\frac{K}{n-1} \leq \left( \frac{N_{n,i^\star}}{n-1} \right)^2 - \left( \frac{1}{n-1} \sum_{t=N_1}^{n-1} \beta_t(i^\star, C_t) \right)^2 \leq 2\frac{2N_1 + K}{n-1},$$

$$-2\frac{1}{n-1} \leq \left( \frac{N_{n,i}}{n-1} \right)^2 - \left( \frac{1}{n-1} \sum_{t=N_1}^{n-1} \mathbb{1}(C_t = j)(1 - \beta_t(i^\star, j)) \right)^2 \leq 2\frac{2N_1 + 1}{n-1}.$$

For all $n > N_1$, let us denote by

$$G_n = \left( \sum_{t=N_1}^{n-1} \beta_t(i^\star, C_t) \right)^2 - \sum_{j \neq i^\star} \left( \sum_{t=N_1}^{n-1} \mathbb{1}(C_t = j)(1 - \beta_t(i^\star, j)) \right)^2.$$

Therefore, we have

$$-4K\frac{N_1 + 1}{n-1} \leq \left( \frac{N_{n,i^\star}}{n-1} \right)^2 - \sum_{i \neq i^\star} \left( \frac{N_{n,i}}{n-1} \right)^2 - \frac{G_n}{(n-1)^2} \leq 4\frac{N_1 + K}{n-1}.$$

Direct manipulations yield that

$$\frac{1}{2}(G_{n+1} - G_n) = \beta_n(i^\star, C_n) \sum_{t=N_1}^{n-1} \beta_t(i^\star, C_t) - (1 - \beta_n(i^\star, C_n)) \sum_{t=N_1}^{n-1} \mathbb{1}\left(C_t = C_n\right)\left(1 - \beta_t(i^\star, C_n)\right)$$
$$+ \beta_n(i^\star, C_n) - 1/2 \,.$$

Therefore, we obtain by using the above inequalities that

$$\frac{1}{2}(G_{n+1} - G_n) \leq \beta_n(i^\star, C_n) N_{n,i^\star} - (1 - \beta_n(i^\star, C_n)) N_{n,C_n} + 2N_1 + K + 3/2$$
$$= 2N_1 + K + 3/2 \,,$$

$$\frac{1}{2}(G_{n+1} - G_n) \geq \beta_n(i^\star, C_n) N_{n,i^\star} - (1 - \beta_n(i^\star, C_n)) N_{n,C_n} - 2N_1 - K - 3/2$$
$$= -2N_1 - K - 3/2 \,,$$

where we used that $\beta_n(i^\star, C_n) N_{n,i^\star} = (1 - \beta_n(i^\star, C_n)) N_{n,C_n}$ since $\beta_n(i^\star, C_n) = \frac{N_{n,C_n}}{N_{n,i^\star} + N_{n,C_n}}$.
Summing those inequalities, we obtain

$$G_n = \sum_{t=N_1+1}^{n-1} (G_{t+1} - G_t) + G_{N_1+1} \leq (4N_1 + 2K + 3)(n - 1 - N_1) + 1$$
$$\geq -(4N_1 + 2K + 3)(n - 1 - N_1) - 1 \,,$$

where we used that $G_{N_1+1} = 2\beta_{N_1}(i^\star, C_{N_1}) - 1 \in [-1, 1]$. Combining everything together, we have shown that

$$\left(\frac{N_{n,i^\star}}{n-1}\right)^2 - \sum_{i \neq i^\star} \left(\frac{N_{n,i}}{n-1}\right)^2 \leq 4\frac{N_1 + K}{n-1} + \frac{(4N_1 + 2K + 3)(n - 1 - N_1) + 1}{(n-1)^2}$$
$$\geq -4K\frac{N_1 + 1}{n-1} - \frac{(4N_1 + 2K + 3)(n - 1 - N_1) + 1}{(n-1)^2} \,.$$

Therefore, we have shown the desired results for $n \geq N_2$ defined as

$$N_2 = \inf\left\{n > 2 \mid 4K\frac{N_1 + 1}{n-1} + \frac{(4N_1 + 2K + 3)(n - 1 - N_1) + 1}{(n-1)^2} \leq \gamma\right\},$$

which satisfies $\mathbb{E}_\nu[N_2] < +\infty$ since it is a linear function of $N_1$. □

As in [40], using Lemma 19 allows to bound the empirical proportion allocated to the unique best arm $N_{n,i^\star}/(n-1)$ away from 0 (Lemma 20).

**Lemma 20.** *There exists $N_3$ with $\mathbb{E}_\nu[N_3] < +\infty$ such that for all $n \geq N_3$*

$$\frac{1}{4\sqrt{2(K-1)}} \leq \frac{N_{n,i^\star}}{n-1} \leq \frac{3}{4} \,.$$

*Proof.* Let $N_2$ as in Lemma 19 for $\gamma = 1/2$. Using Lemma 19, we obtain $\frac{N_{n,i^\star}}{n-1} \leq \frac{3}{4}$ since

$$1/2 \geq \left(\frac{N_{n,i^\star}}{n-1}\right)^2 - \sum_{i \neq i^\star} \left(\frac{N_{n,i}}{n-1}\right)^2 \geq \left(\frac{N_{n,i^\star}}{n-1}\right)^2 - \left(1 - \frac{N_{n,i^\star}}{n-1}\right)^2 = 2\frac{N_{n,i^\star}}{n-1} - 1 \,.$$

Let $\tilde{N}_2$ as in Lemma 19 for $\gamma = \frac{1}{32(K-1)}$. Similarly by using Lemma 19, we obtain $\frac{N_{n,i^\star}}{n-1} \geq \frac{1}{4\sqrt{2(K-1)}}$ since

$$\left(\frac{N_{n,i^\star}}{n-1}\right)^2 \geq -\frac{1}{32(K-1)} + \sum_{i \neq i^\star} \left(\frac{N_{n,i}}{n-1}\right)^2 \geq -\frac{1}{32(K-1)} + \frac{1}{K-1}\left(1 - \frac{N_{n,i^\star}}{n-1}\right)^2$$
$$\geq \frac{1}{32(K-1)} \,.$$

Taking $N_3 = \max\{N_2, \tilde{N}_2\}$ yields the result. □

Lemma 21 is a rescaled version of the empirical overall balance equation which is proven by simply combining Lemma 19 and Lemma 20.

**Lemma 21.** *Let $\gamma > 0$. There exists $N_4$ with $\mathbb{E}_\nu[N_4] < +\infty$ such that for all $n \geq N_4$*

$$\left| 1 - \sum_{i \neq i^\star} \left( \frac{N_{n,i}}{N_{n,i^\star}} \right)^2 \right| \leq \gamma \,.$$

*Proof.* Let $N_2$ as in Lemma 19 for $\frac{\gamma}{32(K-1)}$. Let $N_3$ as in Lemma 20. Direct manipulation shows that, for all $n \geq N_4 = \max\{N_2, N_3\}$,

$$\left| 1 - \sum_{i \neq i^\star} \left( \frac{N_{n,i}}{N_{n,i^\star}} \right)^2 \right| = \left( \frac{n-1}{N_{n,i^\star}} \right)^2 \left| \left( \frac{N_{n,i^\star}}{n-1} \right)^2 - \sum_{i \neq i^\star} \left( \frac{N_{n,i}}{n-1} \right)^2 \right| \leq 32(K-1) \frac{\gamma}{32(K-1)} = \gamma \,.$$

This concludes the result. $\qquad\square$

### D.1.3 Convergence towards optimal ratio of allocation

As in [40], we show that a challenger will not be sampled next when the ratio of its empirical proportion compared to the one of $i^\star$ overshoots the ratio of the optimal allocations (Lemma 22).

**Lemma 22.** *Let $\gamma > 0$. There exists $N_5$ with $\mathbb{E}_\nu[N_5] < +\infty$ such that for all $n \geq N_5$ and all $i \neq i^\star$,*

$$\frac{N_{n,i}}{N_{n,i^\star}} \geq \frac{w_i^\star}{w_{i^\star}^\star} + \gamma \quad \implies \quad C_n^{TC\varepsilon_0} \neq i \,.$$

*Proof.* Let $\gamma > 0$. Let $i \neq i^\star$ such that

$$\frac{N_{n,i}}{N_{n,i^\star}} \geq \frac{w_i^\star}{w_{i^\star}^\star} + \gamma \,.$$

Suppose towards contradiction that

$$\forall j \neq i^\star, \quad \frac{N_{n,j}}{N_{n,i^\star}} > \frac{w_j^\star}{w_{i^\star}^\star} \,.$$

Let $\tilde{\gamma} > 0$. Let $N_4$ as in Lemma 21 for $\tilde{\gamma}$. then using Lemma 21 and 8 yields that, for all $n \geq N_4$,

$$\tilde{\gamma} \geq \sum_{j \neq i^\star} \left( \frac{N_{n,j}}{N_{n,i^\star}} \right)^2 - 1 \geq \left( \frac{w_i^\star}{w_{i^\star}^\star} + \gamma \right)^2 - \left( \frac{w_i^\star}{w_{i^\star}^\star} \right)^2 = \gamma \left( \gamma + 2 \frac{w_i^\star}{w_{i^\star}^\star} \right) \,.$$

Therefore, we have a contradiction if we take $\tilde{\gamma}$ small enough (e.g. $\tilde{\gamma} < \gamma^2$), hence we have shown that, for all $n \geq N_4$,

$$\frac{N_{n,i}}{N_{n,i^\star}} \geq \frac{w_i^\star}{w_{i^\star}^\star} + \gamma \quad \implies \quad \exists j \notin \{i^\star, i\}, \quad \frac{N_{n,j}}{N_{n,i^\star}} \leq \frac{w_j^\star}{w_{i^\star}^\star} \,.$$

Let $N_1$ as in Lemma 18. Using that $B_n^{\mathrm{EB}} = i^\star$ for all $n \geq N_1$ and the definition of $C_n^{\mathrm{TC}\varepsilon_0}$, we known that

$$\frac{\mu_{n,i^\star} - \mu_{n,i} + \varepsilon_0}{\sqrt{1/N_{n,i^\star} + 1/N_{n,i}}} > \frac{\mu_{n,i^\star} - \mu_{n,j} + \varepsilon_0}{\sqrt{1/N_{n,i^\star} + 1/N_{n,j}}} \quad \implies \quad C_n^{\mathrm{TC}\varepsilon_0} \neq i \,.$$

To conclude the proof, it is sufficient to show that the ratio of the two quantities is strictly larger than one. For all $n \geq \max\{N_1, N_4\}$, we obtain

$$\frac{\mu_{n,i^\star} - \mu_{n,i} + \varepsilon_0}{\mu_{n,i^\star} - \mu_{n,j} + \varepsilon_0} \sqrt{\frac{1 + N_{n,i^\star}/N_{n,j}}{1 + N_{n,i^\star}/N_{n,i}}} \geq \frac{\mu_{n,i^\star} - \mu_{n,i} + \varepsilon_0}{\mu_{n,i^\star} - \mu_{n,j} + \varepsilon_0} \sqrt{\frac{1 + w_{i^\star}^\star/w_j^\star}{1 + (w_i^\star/w_{i^\star}^\star + \gamma)^{-1}}} \,.$$

Let $\tilde{\gamma} > 0$. Using Lemmas 12 and 18, we have, for all $n \geq N_1$ and all $k \neq i^\star$,

$$\left| \frac{\mu_{n,i^\star} - \mu_{n,k} + \varepsilon_0}{\mu_{i^\star} - \mu_k + \varepsilon_0} - 1 \right| \leq \frac{W_\mu}{\mu_{i^\star} - \mu_k + \varepsilon_0} \left( \sqrt{\frac{\log(e + N_{n,k})}{N_{n,k}}} + \sqrt{\frac{\log(e + N_{n,i^\star})}{N_{n,i^\star}}} \right)$$

$$\leq \frac{2W_\mu}{\min_{k \neq i^\star}(\mu_{i^\star} - \mu_k + \varepsilon_0)} \sqrt{\frac{\log(e + \sqrt{n/K})}{\sqrt{n/K}}} \leq \tilde{\gamma}$$

for all $n \geq N_6 = K(X_0 - e)^2 + 1$ which is defined as

$$X_0 = \sup \left\{ x \geq 1 \mid x < \frac{4W_\mu^2}{\tilde{\gamma}^2 \min_{k \neq i^\star}(\mu_{i^\star} - \mu_k + \varepsilon_0)^2} \log x + e \right\}$$

$$\leq h_1 \left( \frac{4W_\mu^2}{\tilde{\gamma}^2 \min_{k \neq i^\star}(\mu_{i^\star} - \mu_k + \varepsilon_0)^2}, e \right) .$$

where the last inequality is obtained by using Lemma 51 with $h_1$ defined therein. Since $h_1(x,y) \sim_{x \to +\infty} x \log x$, we have $\mathbb{E}_\nu[N_6] < +\infty$ since it polynomial in $W_\mu$ (by using Lemma 12). Let $\kappa > 0$. We have shown that, for all $n \geq \max\{N_1, N_4, N_6\}$,

$$\frac{\mu_{n,i^\star} - \mu_{n,i} + \varepsilon_0}{\mu_{n,i^\star} - \mu_{n,j} + \varepsilon_0} \sqrt{\frac{1 + N_{n,i^\star}/N_{n,j}}{1 + N_{n,i^\star}/N_{n,i}}} \geq \frac{\mu_{i^\star} - \mu_i + \varepsilon_0}{\mu_{i^\star} - \mu_j + \varepsilon_0} \sqrt{\frac{1 + w_{i^\star}^\star/w_j^\star}{1 + (w_i^\star/w_{i^\star}^\star + \gamma)^{-1}}} \frac{1 - \tilde{\gamma}}{1 + \tilde{\gamma}}$$

$$= \sqrt{\frac{1 + w_{i^\star}^\star/w_i^\star}{1 + (w_i^\star/w_{i^\star}^\star + \gamma)^{-1}}} \frac{1 - \tilde{\gamma}}{1 + \tilde{\gamma}} .$$

where the equality uses that the transportation costs are equal at the equilibrium, i.e. (9). Taking $\tilde{\gamma}$ small enough, we have that shown that, for all $n \geq \max\{N_1, N_4, N_6\}$,

$$\frac{N_{n,i}}{N_{n,i^\star}} \geq \frac{w_i^\star}{w_{i^\star}^\star} + \gamma \quad \implies \quad \frac{\mu_{n,i^\star} - \mu_{n,i} + \varepsilon_0}{\mu_{n,i^\star} - \mu_{n,j} + \varepsilon_0} \sqrt{\frac{1 + N_{n,i^\star}/N_{n,j}}{1 + N_{n,i^\star}/N_{n,i}}} > 1 \quad \implies \quad C_n^{\mathrm{TC}\varepsilon_0} \neq i .$$

This concludes the proof. $\qquad\square$

Lemma 23 shows no sub-optimal arm is overshooting the ratio of its optimal allocation for $n$ large enough.

**Lemma 23.** *Let $\gamma > 0$. There exists $N_6$ with $\mathbb{E}_\nu[N_6] < +\infty$ such that for all $n \geq N_6$ and all $i \neq i^\star$,*

$$\frac{N_{n,i}}{N_{n,i^\star}} \leq \frac{w_i^\star}{w_{i^\star}^\star} + \gamma .$$

*Proof.* Let $\gamma > 0$. Let $N_1$ as in Lemma 18. Let $N_3$ and $N_5$ as in Lemmas 20 and 22. Let $M \geq \max\{N_3, N_5, N_1\}$ and $n > M$. Let $t_{n,i}(\gamma) = \max\left\{ M, \max\left\{ t \in \{M, \cdots, n-1\} \mid \frac{N_{t,i}}{N_{t,i^\star}} < \frac{w_i^\star}{w_{i^\star}^\star} + \gamma/2 \right\} \right\}$ for all $i \neq i^\star$. In particular, using Lemma 22, for all $n > M$

$$\forall t > t_{n,i}(\gamma), \quad \frac{N_{t,i}}{N_{t,i^\star}} \geq \frac{w_i^\star}{w_{i^\star}^\star} + \gamma/2 \quad \text{hence} \quad i \neq C_t^{\mathrm{TC}\varepsilon_0} ,$$

and

$$N_{t_{n,i}(\gamma),i} \leq \max\left\{ M, \left( \frac{w_i^\star}{w_{i^\star}^\star} + \gamma/2 \right) N_{t_{n,i}(\gamma),i^\star} \right\}$$

Then, we have for all $n \geq M$,

$$N_{n,i} = N_{t_{n,i}(\gamma),i} + \mathbb{1}\left( i = I_{t_{n,i}(\gamma)} \right) + \sum_{t=t_{n,i}(\gamma)+1}^{n-1} \mathbb{1}\left( i = I_t = C_t^{\mathrm{TC}\varepsilon_0} \right)$$

$$\leq \max\left\{ M, \left( \frac{w_i^\star}{w_{i^\star}^\star} + \gamma/2 \right) N_{t_{n,i}(\gamma),i^\star} \right\} + 1 .$$

Using that $N_{t_{n,i}(\gamma),i^\star} \leq N_{n,i^\star}$, $N_{n,i^\star} \geq \frac{n-1}{4\sqrt{2(K-1)}}$ (Lemma 20) and $\max\{a, b\} \leq a + b$, we obtain that

$$\frac{N_{n,i}}{N_{n,i^\star}} \leq \frac{4(M+1)\sqrt{2(K-1)}}{n-1} + \frac{w_i^\star}{w_{i^\star}^\star} + \gamma/2 \leq \frac{w_i^\star}{w_{i^\star}^\star} + \gamma \,,$$

where the last inequality holds for $n \geq N_7 = 8(M+1)\sqrt{2(K-1)}/\gamma + 1$. Taking $N_6 = \max\{N_3, N_5, N_1, N_7\}$ yields the result since $\mathbb{E}_\nu[N_6] < +\infty$. $\qquad\square$

Finally, Lemma 24 shows that the sufficient condition (see Lemma 13) to obtain the asymptotic upper bound on the expected sample complexity is satisfied.

**Lemma 24.** *Let $\varepsilon_0 > 0$, $\gamma > 0$ and $T_{\varepsilon_0,\gamma}$ as in (10). Under the EB-TC$_{\varepsilon_0}$ sampling rule with IDS proportions, we have $\mathbb{E}_\nu[T_{\varepsilon_0,\gamma}] < +\infty$.*

*Proof.* Let $\gamma > 0$ and $\tilde{\gamma}$. Let $N_4$ and $N_6$ as in Lemmas 21 and 23 for $\tilde{\gamma}$. Then, using Lemmas 21 and 23, we have for all $n \geq \max\{N_4, N_6\}$,

$$\left(\frac{N_{n,i}}{N_{n,i^\star}}\right)^2 \geq 1 - \sum_{j \notin \{i,i^\star\}} \left(\frac{N_{n,j}}{N_{n,i^\star}}\right)^2 - \tilde{\gamma}$$

$$\geq 1 - \sum_{j \notin \{i,i^\star\}} \left(\frac{w_j^\star}{w_{i^\star}^\star} + \tilde{\gamma}\right)^2 - \tilde{\gamma}$$

$$= \left(\frac{w_i^\star}{w_{i^\star}^\star}\right)^2 - \left((K-2)\tilde{\gamma} + 2\sum_{j \notin \{i,i^\star\}} \frac{w_j^\star}{w_{i^\star}^\star} + 1\right)\tilde{\gamma}$$

where the equality uses (9). Therefore, we have shown that for all $n \geq \max\{N_4, N_6\}$,

$$\frac{N_{n,i}}{N_{n,i^\star}} \geq \frac{w_i^\star}{w_{i^\star}^\star} \sqrt{1 - \left(\frac{w_{i^\star}^\star}{w_i^\star}\right)^2 \left((K-2)\tilde{\gamma} + 2\sum_{j \notin \{i,i^\star\}} \frac{w_j^\star}{w_{i^\star}^\star} + 1\right)\tilde{\gamma}} \geq \frac{w_i^\star}{w_{i^\star}^\star} - \gamma \,,$$

where the last inequality holds for $\tilde{\gamma}$ small enough as a function of $\gamma$ and $w^\star$. Therefore, we obtained

$$\max_{i \neq i^\star} \left|\frac{N_{n,i}}{N_{n,i^\star}} - \frac{w_i^\star}{w_{i^\star}^\star}\right| \leq \gamma \,,$$

which concludes the proof, i.e. $\mathbb{E}_\nu[T_\gamma] < +\infty$. $\qquad\square$

Combining Lemmas 13 and 24 concludes the first part of Theorem 1, i.e.

$$\limsup_{\delta \to 0} \frac{\mathbb{E}_\nu[\tau_{\varepsilon_1,\delta}]}{\log(1/\delta)} \leq T_{\varepsilon_0}(\mu)\left(1 + \max_{i \neq i^\star} \frac{\varepsilon_0 - \varepsilon_1}{\mu_{i^\star} - \mu_i + \varepsilon_1}\right)^2 \,.$$

### D.2 Fixed proportions

Using Lemma 9, the $\beta$-optimal allocation for $\varepsilon_0$-BAI are defined as

$$w_{\varepsilon_0,\beta}(\mu) := \underset{w \in \triangle_K, w_{i^\star} = \beta}{\arg\max} \min_{i \neq i^\star} \frac{(\mu_{i^\star} - \mu_i + \varepsilon_0)^2}{1/\beta + 1/w_i} \,.$$

Since $|i^\star(\mu)| = 1$, let $\mu_{\varepsilon_0} = \mu_{\varepsilon_0}(i^\star)$ where $\mu_\varepsilon(i^\star)$ as in Lemma 9. Since we have $T_{\varepsilon_0,\beta}(\mu) = T_\beta^\star(\mu_{\varepsilon_0})$ and $w_{\varepsilon_0,\beta}(\mu) = w_\beta^\star(\mu_{\varepsilon_0})$, Lemma 6 and 8 yield that $w_{\varepsilon_0,\beta}(\mu) = \{w_\beta^\star\}$ is a singleton with unique element denotes by $w_\beta^\star$ which satisfies that $\min_{i \in [K]} w_{\beta,i}^\star > 0$ and

$$\forall i \in [K] \setminus \{i^\star\}, \quad \frac{(\mu_{i^\star} - \mu_i + \varepsilon_0)^2}{1/\beta + 1/w_{\beta,i}^\star} = 2T_{\varepsilon_0,\beta}(\mu)^{-1} \,. \tag{13}$$

When considering fixed proportions $\beta$, the proof strategy is the same as in Appendix D.1. The sole difference lies in the fact that the empirical allocation of the unique best arm is converging towards a fixed $\beta$.

**Convergence time**   Let $\gamma > 0$. Let us define the *convergence time* $T_{\varepsilon_0,\beta,\gamma}$, which is a random variable quantifies the number of samples required for the empirical allocations $N_n$ to be $\gamma$-close to $w_\beta^\star$:

$$T_{\varepsilon_0,\beta,\gamma} := \inf \left\{ T \geq 1 \mid \forall n \geq T, \; \left\| \frac{N_n}{n-1} - w_\beta^\star \right\|_\infty \leq \gamma \right\} . \tag{14}$$

Lemma 25 gives a sufficient condition for asymptotic $\beta$-optimality. The case $\varepsilon_0 = \varepsilon_1$ is a direct consequence of existing methods, e.g. Theorem 2 in [21] or Theorem 3 in [32].

**Lemma 25.** *Let $\varepsilon_0 > 0$, $\varepsilon_1 \geq 0$ and $\delta \in (0,1)$. Assume that there exists $\gamma_1(\mu) > 0$ such that for all $\gamma \in (0, \gamma_1(\mu)]$, $\mathbb{E}_\nu[T_{\varepsilon_0,\beta,\gamma}] < +\infty$. Using the threshold* (4) *in the stopping rule* (3) *with slack $\varepsilon_1$ yields an algorithm such that, for all $\nu \in \mathcal{D}^K$ such that $|i^\star(\mu)| = 1$,*

$$\limsup_{\delta \to 0} \frac{\mathbb{E}_\nu[\tau_{\varepsilon_1,\delta}]}{\log(1/\delta)} \leq T_{\varepsilon_0,\beta}(\mu) \left( 1 + \max_{i \neq i^\star} \frac{\varepsilon_0 - \varepsilon_1}{\mu_{i^\star} - \mu_i + \varepsilon_1} \right)^2 .$$

*Proof.*   The proof is the same of the ones of Lemma 13. The sole difference is that we have $w_{\beta,i^\star}^\star = \beta$ by definition.   □

**Sufficient exploration**   Lemma 26 shows that all arms are sufficiently explored and that the leader is the unique best arm for $n$ large enough.

**Lemma 26.** *There exist $N_1$ with $\mathbb{E}_\nu[N_1] < +\infty$ such that for all $n \geq N_1$ and all $i \in [K]$, $N_{n,i} \geq \sqrt{n/K}$ and $B_n^{EB} = i^\star$.*

*Proof.*   Since Lemmas 16 and 17 holds, we can use the intermediate results of Lemma 18. It is direct to see that the proof can be conducted similarly by noting that, for all $i \neq j$,

$$\min\{\beta_t(i,j), 1 - \beta_t(i,j)\} = \min\{\beta, 1 - \beta\} > 0 .$$

□

**Convergence towards optimal allocation**   Combining Lemma 26 with the proof of Lemma F.8 in [20] yields Lemma 27.

**Lemma 27.** *Let $\gamma > 0$. There exists $N_2$ with $\mathbb{E}_\mu[N_2] < +\infty$ such that for all $n \geq N_2$,*

$$\left| \frac{N_{n,i^\star}}{n-1} - \beta \right| \leq \gamma .$$

Combining Lemmas 26 and 27 with the proof of Lemma F.9 in [20] yields Lemma 28.

**Lemma 28.** *Let $\gamma > 0$. There exists $N_5$ with $\mathbb{E}_\nu[N_5] < +\infty$ such that for all $n \geq N_5$ and all $i \neq i^\star$,*

$$\frac{N_{n,i}}{n-1} \geq w_{\beta,i}^\star + \gamma \quad \Longrightarrow \quad C_n^{TC\varepsilon_0} \neq i .$$

Combining Lemmas 27 and 28 with the proof of Lemma F.10 in [20] yields Lemma 29.

**Lemma 29.** *Let $\gamma > 0$. There exists $N_6$ with $\mathbb{E}_\nu[N_6] < +\infty$ such that for all $n \geq N_6$,*

$$\left\| \frac{N_n}{n-1} - w_\beta^\star \right\|_\infty \leq \gamma .$$

Lemma 30 is a direct corollary of Lemma 29.

**Lemma 30.** *Let $\varepsilon_0 > 0$, $\beta \in (0,1)$, $\gamma > 0$ and $T_{\varepsilon_0,\beta,\gamma}$ as in* (14). *Under the EB-TC$_{\varepsilon_0}$ sampling rule with fixed proportions $\beta$, we have $\mathbb{E}_\nu[T_{\varepsilon_0,\beta,\gamma}] < +\infty$.*

Combining Lemmas 25 and 30 concludes the second part of Theorem 1, i.e.

$$\limsup_{\delta \to 0} \frac{\mathbb{E}_\nu[\tau_{\varepsilon_1,\delta}]}{\log(1/\delta)} \leq T_{\varepsilon_0,\beta}(\mu) \left( 1 + \max_{i \neq i^\star} \frac{\varepsilon_0 - \varepsilon_1}{\mu_{i^\star} - \mu_i + \varepsilon_1} \right)^2 .$$

# E  Non-asymptotic analysis

Let $\varepsilon_0 > 0$, $\beta \in (0,1)$ and $\delta \in (0,1)$. In this section, we provide a non-asymptotic analysis of EB-TC$_{\varepsilon_0}$ with fixed proportions $\beta$ and slack $\varepsilon_0 > 0$ when combined with the stopping rule (3) with parameters $(\varepsilon_0, \delta)$. In practice, we will mostly be interested in the case $\beta = 1/2$. First, we prove Lemma 5 in Appendix E.1. Then, we detail the proof of Theorem 2 in Appendix E.2.

In the following, we consider a sub-Gaussian bandit with distribution $\nu \in \mathcal{D}^K$ having mean parameter $\mu \in \mathbb{R}^K$. In particular, our analysis holds when several arms have the largest mean $\mu_\star$.

## E.1  Proof of Lemma 5

Let $s \geq 0$. For all $n > K$ and $\delta \in (0,1]$, let $\mathcal{E}_{n,\delta} = \mathcal{E}_{n,\delta}^1 \cap \mathcal{E}_{n,\delta}^2$ with $(\mathcal{E}_{n,\delta}^1)_{n>K}$ and $(\mathcal{E}_{n,\delta}^2)_{n>K}$ as in (24) and (25), i.e.

$$
\mathcal{E}_{n,\delta}^1 = \left\{ \forall k \in [K], \forall t \leq n, \; |\mu_{t,k} - \mu_k| < \sqrt{\frac{2f_1(n,\delta)}{N_{t,k}}} \right\} \, ,
$$

$$
\mathcal{E}_{n,\delta}^2 = \left\{ \forall (i,k) \in [K]^2 \text{ s.t. } i \neq k, \; \forall t \leq n, \; \frac{|(\mu_{t,i} - \mu_{t,k}) - (\mu_i - \mu_k)|}{\sqrt{1/N_{t,i} + 1/N_{t,k}}} < \sqrt{2f_2(n,\delta)} \right\} \, ,
$$

where $f_2(x,\delta) = \log(1/\delta) + (1+s)\log(x)$ and $f_2(x,\delta) = \log(1/\delta) + (2+s)\log(x)$.

Lemma 31 shows that when there are arms with strictly higher true mean than the one of the leader, then at least one of those arms is undersampled.

**Lemma 31.** *Under $\mathcal{E}_{n,\delta}^1$, for all $t \in [n] \setminus [K]$ let $B_t^{EB} = k$. Then,*

$$
\forall i \neq k, \quad \mathbb{1}\left(\mu_i > \mu_k\right) \min\{N_{t,k}, N_{t,i}\} \leq \frac{8f_1(n,\delta)}{(\mu_i - \mu_k)^2} \, .
$$

*Proof.* Under $\mathcal{E}_{n,\delta}^1$, for all $t \in [n] \setminus [K]$, let $B_t^{\text{EB}} = k$. Then, for all $i \neq k$, we have

$$
\mu_i - \sqrt{\frac{2f_1(n,\delta)}{\min\{N_{t,k}, N_{t,i}\}}} \leq \mu_i - \sqrt{\frac{2f_1(n,\delta)}{N_{t,i}}} \leq \mu_{t,i} \leq \mu_{t,k} \leq \mu_k + \sqrt{\frac{2f_1(n,\delta)}{N_{t,k}}}
$$

$$
\leq \mu_k + \sqrt{\frac{2f_1(n,\delta)}{\min\{N_{t,k}, N_{t,i}\}}} \, .
$$

Re-ordering the above equations for $i$ such that $\mu_i > \mu_k$ yields the result. $\qquad\square$

**Lemma 32.** *Let $\varepsilon \geq 0$, $\Delta_\mu(\varepsilon) = \min_{k \notin \mathcal{I}_\varepsilon(\mu)} \Delta_k$ and $C_{\mu,\varepsilon_0}(\varepsilon) = \max\{2\Delta_\mu(\varepsilon)^{-1} - \varepsilon_0^{-1}, \varepsilon_0^{-1}\}^2$. Let $A_{\varepsilon_0,\varepsilon,i} = 2/\Delta_\mu(\varepsilon)^2$ for all $i \in i^\star(\mu)$, $A_{\varepsilon_0,\varepsilon,i} = C_{\mu,\varepsilon_0}(\varepsilon)$ for all $i \in \mathcal{I}_\varepsilon(\mu) \setminus i^\star(\mu)$, otherwise $A_{\varepsilon_0,\varepsilon,i} = \max\{C_{\mu,\varepsilon_0}(\varepsilon), 2/\Delta_i^2\}$. For all $n > K$, under event $\mathcal{E}_{n,\delta}$, for all $t \in [n] \setminus [K]$ such that $B_t^{EB} \notin \mathcal{I}_\varepsilon(\mu)$, there exists $i_t \in [K]$ such that*

$$
T_t(i_t) \leq \frac{4f_2(n,\delta)}{\min\{\beta, 1-\beta\}} A_{\varepsilon_0,\varepsilon,i_t} + 3(K-1)/2 \quad \text{and} \quad T_{t+1}(i_t) = T_t(i_t) + 1 \, .
$$

*Proof.* Let $\Delta_i = \mu_\star - \mu_i$ and $\Delta_{\max} = \max_{i \in [K]} \Delta_i$. When $\varepsilon \geq \Delta_{\max}$, we have $\mathcal{I}_\varepsilon(\mu)^{\complement} = \emptyset$, hence the above result holds trivially since the event $B_t^{\text{EB}} \notin \mathcal{I}_\varepsilon(\mu)$ cannot happen. Let $\varepsilon \in [0, \Delta_{\max})$, i.e. $\mathcal{I}_\varepsilon(\mu)^{\complement} \neq \emptyset$. We will consider in two distinct cases since

$$
\{B_t \notin \mathcal{I}_\varepsilon(\mu)\} = \{B_t \notin \mathcal{I}_\varepsilon(\mu), C_t \in i^\star(\mu)\} \cup \{B_t \notin \mathcal{I}_\varepsilon(\mu), C_t \notin i^\star(\mu)\} \, .
$$

**Case 1.** Let $t \in [n] \setminus [K]$ such that $(B_t^{\text{EB}}, C_t^{\text{TC}\varepsilon_0}) = (i,j)$ with $i \notin \mathcal{I}_\varepsilon(\mu)$ and $j \in i^\star(\mu)$. Using Lemmas 46 and 31, we obtain

$$
\min\{\beta, 1-\beta\} \left(\min\{T_t(i), T_t(j)\} - 3(K-1)/2\right) \leq \min\{N_{t,i}, N_{t,j}\} \leq \frac{8f_1(n,\delta)}{\Delta_i^2} \leq \frac{8f_2(n,\delta)}{\Delta_i^2} \, ,
$$

which can be rewritten as

$$\min\{T_t(i), T_t(j)\} \leq \frac{8f_2(n,\delta)}{\min\{\beta, 1-\beta\}\Delta_i^2} + 3(K-1)/2 \, .$$

Let us define $\Delta_\mu(\varepsilon) = \min_{k \notin \mathcal{I}_\varepsilon(\mu)} \Delta_k$, and

$$\forall i \notin \mathcal{I}_\varepsilon(\mu), \quad D_{\varepsilon,i} = 2/\Delta_i^2 \quad \text{and} \quad \forall i \in i^\star(\mu), \quad D_{\varepsilon,i} = 2/\Delta_\mu(\varepsilon)^2 \, .$$

The above shows that there exists $k_t \in \mathcal{I}_\varepsilon(\mu)^\complement \cup i^\star(\mu)$ such that

$$T_t(k_t) \leq \frac{4f_2(n,\delta)}{\min\{\beta, 1-\beta\}} D_{\varepsilon,k_t} + 3(K-1)/2 \quad \text{and} \quad T_{t+1}(k_t) = T_t(k_t) + 1 \, .$$

**Case 2.** Let $t \in [n] \setminus [K]$ such that $(B_t^{\mathrm{EB}}, C_t^{\mathrm{TC}\varepsilon_0}) = (i,j)$ with $i \notin \mathcal{I}_\varepsilon(\mu)$ and $j \notin i^\star(\mu)$. Let $i_0 \in i^\star(\mu)$. Using the TC challenger, we obtain

$$\frac{\varepsilon_0 - \Delta_i}{\sqrt{1/N_{t,i} + 1/N_{t,i_0}}} + \sqrt{2f_2(n,\delta)} \geq \frac{\mu_{t,i} - \mu_{t,i_0} + \varepsilon_0}{\sqrt{1/N_{t,i} + 1/N_{t,i_0}}} \geq \frac{\mu_{t,i} - \mu_{t,j} + \varepsilon_0}{\sqrt{1/N_{t,i} + 1/N_{t,j}}}$$

$$\geq \varepsilon_0\sqrt{\min\{N_{t,i}, N_{t,j}\}/2} \, .$$

Using Lemma 31, we obtain

$$\frac{1}{1/N_{t,i} + 1/N_{t,i_0}} \leq \min\{N_{t,i}, N_{t,i_0}\} \leq \frac{8f_1(n,\delta)}{\Delta_i^2} \leq \frac{8f_2(n,\delta)}{\Delta_i^2} \, .$$

By distinguishing between $\varepsilon_0 > \Delta_i$ and $\varepsilon_0 \leq \Delta_i$ and using that $\Delta_i > 0$, we have

$$\frac{\varepsilon_0 - \Delta_i}{\sqrt{1/N_{t,i} + 1/N_{t,i_0}}} + \sqrt{2f_2(n,\delta)} \leq \max\{2\varepsilon_0/\Delta_i - 1, 1\}\sqrt{2f_2(n,\delta)} \, .$$

Using Lemma 46 to lower bound $\min\{N_{t,i}, N_{t,j}\}$ and reordering, we have shown that

$$\min\{T_t(i), T_t(j)\} \leq \max\left\{\left(\frac{2}{\Delta_i} - \frac{1}{\varepsilon_0}\right)^2, \frac{1}{\varepsilon_0^2}\right\} \frac{4f_2(n,\delta)}{\min\{\beta, 1-\beta\}} + 3(K-1)/2 \, .$$

Let us define $C_{\mu,\varepsilon_0}(\varepsilon) = \max\{2\Delta_\mu(\varepsilon)^{-1} - \varepsilon_0^{-1}, \varepsilon_0^{-1}\}^2$. The above shows that, there exists $k_t \notin i^\star(\mu)$ such that

$$T_t(k_t) \leq \frac{4f_2(n,\delta)}{\min\{\beta, 1-\beta\}} C_{\mu,\varepsilon_0}(\varepsilon) + 3(K-1)/2 \quad \text{and} \quad T_{t+1}(k_t) = T_t(k_t) + 1 \, .$$

**Summary.** Let us define $(A_{\varepsilon_0,\varepsilon,i})_{i \in [K]}$ as in the statement of Lemma 32. Under $\mathcal{E}_{n,\delta}$, we have show that, when $B_t \notin \mathcal{I}_\varepsilon(\mu)$, there exists $i_t \in [K]$ such that

$$T_t(k_t) \leq \frac{4f_2(n,\delta)}{\min\{\beta, 1-\beta\}} A_{\varepsilon,\varepsilon_0,i_t} + 3(K-1)/2 \quad \text{and} \quad T_{t+1}(k_t) = T_t(k_t) + 1\} \, .$$

$\qquad\qquad\qquad\qquad\qquad\qquad\qquad\qquad\qquad\qquad\qquad\qquad\qquad\qquad\qquad\qquad\qquad\square$

For all $n > K$, under event $\mathcal{E}_{n,\delta}$, combining Lemma 4 and 32 for $A_t(n,\delta) = \{B_t^{\mathrm{EB}} \notin \mathcal{I}_\varepsilon(\mu)\}$ and $D_i(n,\delta) = \frac{4f_2(n,\delta)}{\min\{\beta, 1-\beta\}} A_{\varepsilon_0,\varepsilon,i} + 3(K-1)/2$ yields that

$$\sum_{t=K+1}^{n} \mathbb{1}\left(B_t^{\mathrm{EB}} \notin \mathcal{I}_\varepsilon(\mu)\right) \leq \frac{4f_2(n,\delta)}{\min\{\beta, 1-\beta\}} H_{\mu,\varepsilon_0}(\varepsilon) + 3K(K-1)/2 \, .$$

where we used that $\sum_{i \in [K]} A_{\varepsilon_0,\varepsilon,i} = H_{\mu,\varepsilon_0}(\varepsilon)$ where $H_{\mu,\varepsilon_0}(\varepsilon)$ is defined in (5). To conclude the proof of Lemma 5, we use that

$$\sum_{t=K+1}^{n} \mathbb{1}\left(B_t^{\mathrm{EB}} \notin \mathcal{I}_\varepsilon(\mu)\right) = n - 1 - \sum_{i \in \mathcal{I}_\varepsilon(\mu)} \sum_j T_n(i,j) \, .$$

Let $\tilde{f}_1(n,\delta)$ and $\tilde{f}_2(n,\delta)$ defined as in (26) and (28). Using Lemma 50, we have $\tilde{f}_1(n,\delta) \leq \tilde{f}_2(n,\delta)$. Therefore, it is direct to see that Lemma 33 can be proven with the same proof as for Lemma 5 based on the concentration event $\tilde{\mathcal{E}}_{n,\delta} = \tilde{\mathcal{E}}_{n,\delta}^1 \cap \tilde{\mathcal{E}}_{n,\delta}^2$ with $(\tilde{\mathcal{E}}_{n,\delta}^1)_{n>K}$ and $(\tilde{\mathcal{E}}_{n,\delta}^2)_{n>K}$ as in (27) and (29).

**Lemma 33.** *Let $\delta \in (0,1]$, $\varepsilon \geq 0$ and $H_{\mu,\varepsilon_0}(\varepsilon)$ as in (5). For all $n > K$, under the event $\tilde{\mathcal{E}}_{n,\delta}$,*

$$\sum_{i \in \mathcal{I}_\varepsilon(\mu)} \sum_j T_n(i,j) \geq n - 1 - \left( \frac{4 H_{\mu,\varepsilon_0}(\varepsilon)}{\min\{\beta, 1-\beta\}} \tilde{f}_2(n,\delta) + 3K(K-1)/2 \right) .$$

### E.2 Proof of Theorem 2

Let $s > 1$. For all $n > K$, let $\mathcal{E}_n = \mathcal{E}_{n,1}^1 \cap \mathcal{E}_{n,1}^2$ with $(\mathcal{E}_{n,\delta}^1)_{n>K}$ and $(\mathcal{E}_{n,\delta}^2)_{n>K}$ as in (24) and (25). Using Lemma 45, we know that $\sum_{n>K} \mathbb{P}_\nu(\mathcal{E}_n^\complement) \leq \frac{K(K+1)}{2} \zeta(s)$.

Let $\varepsilon_0 > 0$, $\varepsilon_1 \geq \varepsilon_0$ and $\delta \in (0,1)$. Since $\varepsilon_1 \geq \varepsilon_0$, and using the definition of the stopping rule (3), it is direct to see that $\mathbb{E}_\nu[\tau_{\varepsilon_1,\delta}] \leq \mathbb{E}_\nu[\tau_{\varepsilon_0,\delta}]$.

Suppose that we have constructed a time $T(\delta) > K$ be such that for $n \geq T(\delta)$, $\mathcal{E}_n \subset \{\tau_{\varepsilon_0,\delta} \leq n\}$. Then, Lemma 47 yields

$$\mathbb{E}_\nu[\tau_{\varepsilon_0,\delta}] \leq T(\delta) + \frac{K(K+1)}{2} \zeta(s) .$$

We will construct an infinite number of times $\{T(\delta,u)\}_{u \in \mathcal{U}}$ such that the above property holds, hence taking the infimum yields

$$\mathbb{E}_\nu[\tau_{\varepsilon_0,\delta}] \leq \inf_{u \in \mathcal{U}} T(\delta,u) + \frac{K(K+1)}{2} \zeta(s) .$$

This proof strategy is the one used to prove Theorem 5. Theorem 2 is a corollary of Theorem 5. Note that the notation of both theorems differ slightly since we provide a shorter statement in the main content. The key difference lies in the refined analysis used to clip $\min_{j \neq i} w_{\varepsilon_0,1/2}(\mu,i)_j$ by a fixed value $x \in [0, (K-1)^{-1}]$ for all $i \in \mathcal{I}_\varepsilon(\mu)$.

**Theorem 5.** *Let $\varepsilon_0 > 0$, $\varepsilon_1 \geq \varepsilon_0$ and $\delta \in (0,1)$. Using the threshold (4) in the stopping rule (3) with error $\varepsilon_1$, the EB-TC$_{\varepsilon_0}$ algorithm with fixed proportions $\beta = 1/2$ is $(\varepsilon_1,\delta)$-PAC and satisfies that, for all $\nu \in \mathcal{D}^K$, $\mathbb{E}_\nu[\tau_{\varepsilon_1,\delta}] \leq \mathbb{E}_\nu[\tau_{\varepsilon_0,\delta}]$ and*

$$\mathbb{E}_\nu[\tau_{\varepsilon_0,\delta}] \leq \inf_{(\varepsilon,x) \in [0,\varepsilon_0] \times [0,(K-1)^{-1}]} \max \left\{ T_{\mu,\varepsilon_0}(\delta,\varepsilon,x) + 1, \, S_{\mu,\varepsilon_0}(\varepsilon,x) \right\} + \zeta(s) \frac{K(K+1)}{2} ,$$

*where*

$$T_{\mu,\varepsilon_0}(\delta,\varepsilon,x) = \sup \left\{ n \mid n - 1 \leq \frac{2(1+\gamma)^2 \sum_{i \in \mathcal{I}_\varepsilon(\mu)} T_{\varepsilon_0,1/2}(\mu,i)}{(1-x)^{d_{\mu,\varepsilon_0}(\varepsilon,x)}} \right.$$

$$\left. \left( \sqrt{c(n-1,\delta)} + \sqrt{(2+s)\log n} \right)^2 \right\} ,$$

$$S_{\mu,\varepsilon_0}(\varepsilon,x) = h_1 \left( \frac{4(2+s)(1+\gamma^{-1})}{a_{\mu,\varepsilon_0}(\varepsilon,x) v_{\mu,\varepsilon_0}(\varepsilon)} H_{\mu,\varepsilon_0}(\varepsilon), \frac{(1+\gamma^{-1})(3K^2/4+1)}{a_{\mu,\varepsilon_0}(\varepsilon,x) v_{\mu,\varepsilon_0}(\varepsilon)} + 1 \right) ,$$

$$v_{\mu,\varepsilon_0}(\varepsilon) = \frac{\min_{i \in \mathcal{I}_\varepsilon(\mu)} T_{\varepsilon_0,1/2}(\mu,i)}{\sum_{i \in \mathcal{I}_\varepsilon(\mu)} T_{\varepsilon_0,1/2}(\mu,i)}$$

$$a_{\mu,\varepsilon_0}(\varepsilon,x) = \min_{i \in \mathcal{I}_\varepsilon(\mu)} (1-x)^{d_{\mu,\varepsilon_0}(x,i)} \max\{\min_{j \neq i} w_{\varepsilon_0,1/2}(\mu,i)_j, x/2\} ,$$

$$d_{\mu,\varepsilon_0}(\varepsilon,x) = \max_{i \in \mathcal{I}_\varepsilon(\mu)} d_{\mu,\varepsilon_0}(x,i) \quad \text{with} \quad d_{\mu,\varepsilon_0}(x,i) = |\{j \neq i \mid w_{\varepsilon_0,1/2}(\mu,i)_j < x/2\}| ,$$

*where $\gamma \in (0,1/2]$ and $s > 1$ are analysis parameters. $T_{\varepsilon_0,1/2}(\mu,i)$ and $w_{\varepsilon_0,1/2}(\mu,i)$ are defined in (1), $H_{\mu,\varepsilon_0}(\varepsilon)$ in (5), $\zeta$ is the Riemann $\zeta$ function and $h_1(z,y) = z\overline{W}_{-1}(\log(z) + y/z)$ is defined in Lemma 51 and satisfies that $h_1(z,y) \approx x\log(z) + y + \log(\log(z) + y/z)$.*

*Proof.* Let $\varepsilon \in [0,\varepsilon_0]$ be an analysis parameter, and $I_\varepsilon = |\mathcal{I}_\varepsilon(\mu)|$. Let $v \in \mathring{\triangle}_{I_\varepsilon}$ defined as

$$\forall i \in \mathcal{I}_\varepsilon(\mu), \quad v_i = \frac{T_{\varepsilon_0,1/2}(\mu,i)}{\sum_{j \in \mathcal{I}_\varepsilon(\mu)} T_{\varepsilon_0,1/2}(\mu,j)} , \quad \text{hence} \quad \min_{i \in \mathcal{I}_\varepsilon(\mu)} v_i = v_{\mu,\varepsilon_0}(\varepsilon) .$$

The main technical result on which our proof relies on is Lemma 34, which we will prove afterwards.

**Lemma 34.** *There exist $D_0 > 0$ and $T_\mu > 0$ such that for all $n \geq T_\mu$ such that $\mathcal{E}_n \cap \{n < \tau_{\varepsilon_1,\delta}\}$ holds true, there exists $i_0 \in \mathcal{I}_\varepsilon(\mu)$ and $t \leq n$ with $B_t^{EB} = i_0$, which satisfies*

$$1/N_{t,i_0}^{i_0} + 1/N_{t,C_t^{TC\varepsilon_0}}^{i_0} \leq \frac{D_0}{v_{i_0}(n-1)}\left(1/\beta + 1/w_\beta^\star(i_0)_{C_t^{TC\varepsilon_0}}\right) . \tag{15}$$

Before proving Lemma 34, we show how it can be used to conclude. Based on Lemma 34, let $D_0 > 0$, $T_\mu > 0$, $n > T_\mu$ such that $\mathcal{E}_n \cap \{n < \tau_{\varepsilon_1,\delta}\}$ holds true, $i_0$ and $t \leq n$ such that as $B_t^{EB} = i_0$ and

$$1/N_{t,i_0}^{i_0} + 1/N_{t,C_t^{TC\varepsilon_0}}^{i_0} \leq \frac{D_0}{v_{i_0}(n-1)}\left(1/\beta + 1/w_\beta^\star(i_0)_{C_t^{TC\varepsilon_0}}\right) .$$

Using the stopping rule (3) with error $\varepsilon_0$, $t \leq n < \tau_{\varepsilon_1,\delta}$, $B_t^{EB} = \hat{\imath}_t = i_0$ and the definition of the $TC_{\varepsilon_0}$challenger in (2) with slack $\varepsilon_0$, we obtain

$$
\begin{aligned}
\sqrt{2c(n-1,\delta)} &\geq \min_{i\neq i_0} \frac{\mu_{t,i_0} - \mu_{t,i} + \varepsilon_0}{\sqrt{1/N_{t,i_0} + 1/N_{t,i}}} \\
&\geq \frac{\mu_{i_0} - \mu_{C_t^{TC\varepsilon_0}} + \varepsilon_0}{\sqrt{1/N_{t,i_0} + 1/N_{t,C_t^{TC\varepsilon_0}}}} - \sqrt{2(2+s)\log n} \\
&\geq \sqrt{\frac{1/\beta + w_\beta^\star(i_0)_{C_t^{TC\varepsilon_0}}^{-1}}{1/N_{t,i_0}^{i_0} + 1/N_{t,C_t^{TC\varepsilon_0}}^{i_0}}}\sqrt{2T_{\varepsilon_0,\beta}(\mu,i_0)^{-1}} - \sqrt{2(2+s)\log n} \\
&\geq \sqrt{\frac{2(n-1)}{D_0 \sum_{i\in\mathcal{I}_\varepsilon(\mu)} T_{\varepsilon_0,\beta}(\mu,i)}} - \sqrt{2(2+s)\log n} .
\end{aligned}
$$

The third inequality is obtained since $\mathcal{E}_n$ holds. The fourth inequality is obtained since $N_{t,i}^{i_0} \geq N_{t,i}^{i_0}$ for all $i \in [K]$ and by using Lemmas 8 and 9. The last inequality is obtained by Lemma 34 and using the definition of $v_{i_0}$.

Let's define $T(\delta, D_0) := \sup\{n > K \mid n - 1 < D_0 \sum_{i\in\mathcal{I}_\varepsilon(\mu)} T_{\varepsilon_0,\beta}(\mu,i)(\sqrt{c(n-1,\delta)} + \sqrt{(2+s)\log n})^2\}$. Therefore, we have $\mathcal{E}_n \cap \{n < \tau_{\varepsilon_0,\delta}\} = \emptyset$ (i.e. $\mathcal{E}_n \subset \{\tau_{\varepsilon_0,\delta} \leq n\}$) for all $n \geq \max\{T(\delta, D_0), T_\mu + 1\}$. This concludes the construction.

To conclude the proof, we will show that Lemma 34 for many choices of $D_0$ and $T_\mu$.

**Vanilla Proof of Lemma 34.** Let $g_\varepsilon(n) = \frac{4H_{\mu,\varepsilon_0}(\varepsilon)}{\min\{\beta, 1-\beta\}}(2+s)\log n + 3K(K-1)/2$. Using Lemma 5, under the event $\mathcal{E}_n$,

$$\sum_{i\in\mathcal{I}_\varepsilon(\mu)}\sum_j T_n(i,j) \geq n - 1 - g_\varepsilon(n) .$$

Therefore, the pigeonhole principle yields that there exists $i_0 \in \mathcal{I}_\varepsilon(\mu)$ such that

$$\sum_j T_n(i_0,j) \geq v_{i_0}(n - 1 - g_\varepsilon(n)) .$$

The crux of the problem is to relate $N_{t,C_t}^{i_0}/(t-1)$ and $w_\beta^\star(i_0)_{C_t^{TC\varepsilon_0}}$. To do so, we use the approach of [20] that builds on the idea behind the proof for APT from [30]: consider an arm being over-sampled and study the last time this arm was pulled.

By the pigeonhole principle, at time $n$, there is an index $k_1 \neq i_0$ such that

$$N_{n,k_1}^{i_0} \geq \frac{w_\beta^\star(i_0)_{k_1}}{1-\beta}\sum_{j\neq i_0} N_{n,j}^{i_0} , \tag{16}$$

and we take such $k_1$. Let $t_1 := \sup\{t < n \mid (B_t, C_t) = (i_0, k_1)\}$ be the last time at which $i_0$ was the leader and $k_1$ was the challenger. If $I_{t_1} = B_{t_1}$ then $N_{t_1,k_1}^{i_0} = N_{n,k_1}^{i_0}$, otherwise $N_{t_1,k_1}^{i_0} = N_{n,k_1}^{i_0} - 1$.

In both cases, we have $N^{i_0}_{t_1,k_1} \geq N^{i_0}_{n,k_1} - 1$. Combined with the above and using that $w^\star_\beta(i_0)_{k_1} \leq 1 - \beta$ and $v_{i_0} \leq 1$, we obtain

$$N^{i_0}_{t_1,k_1} \geq N^{i_0}_{n,k_1} - 1 \geq \frac{w^\star_\beta(i_0)_{k_1}}{1 - \beta} \sum_{j \neq i_0} N^{i_0}_{n,j} - 1$$

$$\geq w^\star_\beta(i_0)_{k_1} \left( \sum_{j \neq i_0} T_n(i_0, j) - \frac{K-1}{2(1-\beta)} \right) - 1$$

$$\geq w^\star_\beta(i_0)_{k_1} v_{i_0}(n-1) - (1-\beta) v_{i_0} g_\varepsilon(n) - \frac{K+1}{2}$$

$$\geq w^\star_\beta(i_0)_{k_1} v_{i_0}(n-1) - h_\varepsilon(n) \,,$$

$$\text{with} \quad h_\varepsilon(n) = \frac{4(1-\beta)H_{\mu,\varepsilon_0}(\varepsilon)}{\min\{\beta, 1-\beta\}} f_2(n) + (1-\beta)3K(K-1)/2 + \frac{K+1}{2} \,.$$

Let $w_{i_0,-} > 0$ be a lower bound on $w^\star_\beta(i_0)_{k_1}$, e.g. $w_{i_0,-} = \min_{i \neq i_0} w^\star_\beta(i_0)_i$. For instances $\mu$ such that $w^\star_\beta(i_0)_{k_1}$ is small, the equation (16) can be satisfied at the very beginning, hence $t_1$ might be sublinear in $n$. Due to the missing link between $t_1$ and $n$, we use the following inequality

$$1/N^{i_0}_{t_1,i_0} + 1/N^{i_0}_{t_1,k_1} \leq \frac{1}{v_{i_0}(n-1)} \left( 1/\beta + \frac{v_{i_0}(n-1)}{N^{i_0}_{t_1,k_1}} \right) \left( \frac{N^{i_0}_{t_1,k_1}}{N^{i_0}_{t_1,i_0}} + 1 \right) \,,$$

which is a suboptimal step which artificially introduces $1/\beta$.

Let $\gamma \in (0, 1/2]$. It remains to control both terms. First, we obtain

$$1/\beta + \frac{v_{i_0}(n-1)}{N^{i_0}_{t_1,k_1}} \leq 1/\beta + \frac{1}{w^\star_\beta(i_0)_{k_1} - \frac{h_\varepsilon(n)}{v_{i_0}(n-1)}} \leq (1+\gamma)\left(1/\beta + 1/w^\star_\beta(i_0)_{k_1}\right) \,,$$

for all $n > C_1(w_{i_0,-}, v_{\mu,\varepsilon_0}(\varepsilon))$. The last inequality is obtained by definition of

$$C_1(w, v) := \sup \left\{ n \geq 1 \mid n - 1 < \frac{h_\varepsilon(n)}{wv} \left(1 + \gamma^{-1}\right) \right\} \,, \tag{17}$$

which ensures that, for all $n > C_1(w_{i_0,-}, v_{\mu,\varepsilon_0}(\varepsilon))$, the last condition of the equivalence

$$w^\star_\beta(i_0)_{k_1} - \frac{h_\varepsilon(n)}{v_{i_0}(n-1)} \geq (1+\gamma)^{-1} w^\star_\beta(i_0)_{k_1} \quad \Longleftrightarrow \quad n - 1 \geq \frac{h_\varepsilon(n)}{v_{i_0} w^\star_\beta(i_0)_{k_1}} \left(1 + \gamma^{-1}\right)$$

is satisfied since $w^\star_\beta(i_0)_{k_1} \geq w_{i_0,-}$, $v_{i_0} \geq v_{\mu,\varepsilon_0}(\varepsilon)$ and $n > C_1(w_{i_0,-}, v_{\mu,\varepsilon_0}(\varepsilon))$.

Second, using Lemma 46, we can show that

$$N^{i_0}_{t_1,i_0} \geq T_{t_1}(i_0, k_1) - N^{i_0}_{t_1,k_1} \geq \beta T_{t_1}(i_0, k_1) - 1 \geq \frac{\beta}{1-\beta} N^{i_0}_{t_1,k_1} - \frac{1}{1-\beta} \,.$$

Then, re-ordering the above and using that $N^{i_0}_{t_1,k_1} \geq w^\star_\beta(i_0)_{k_1} v_{i_0}(n-1) - h_\varepsilon(n)$, we obtain

$$\frac{N^{i_0}_{t_1,k_1}}{N^{i_0}_{t_1,i_0}} + 1 \leq \left( \frac{\beta}{1-\beta} - \frac{1}{(1-\beta)\left(w^\star_\beta(i_0)_{k_1} v_{i_0}(n-1) - h_\varepsilon(n)\right)} \right)^{-1} + 1 \leq (1+\gamma)/\beta \,,$$

for all $n > C_2(w_{i_0,-}, v_{\mu,\varepsilon_0}(\varepsilon))$. The last inequality is obtained by definition of

$$C_2(w, v) := \sup \left\{ n \in \mathbb{N}^\star \mid n - 1 < \frac{1}{wv} \left( h_\varepsilon(n) + \frac{\gamma+1-\beta}{\beta\gamma} \right) \right\} \,, \tag{18}$$

which ensures that, for all $n > C_2(w_{i_0,-}, v_{\mu,\varepsilon_0}(\varepsilon))$, the last condition of the equivalence

$$\left( \frac{\beta}{1-\beta} - \frac{1}{(1-\beta)\left(w^\star_\beta(i_0)_{k_1} v_{i_0}(n-1) - h_\varepsilon(n)\right)} \right)^{-1} + 1 \leq (1+\gamma)/\beta$$

$$\Longleftrightarrow \quad \frac{1}{w^\star_\beta(i_0)_{k_1} v_{i_0}(n-1) - h_\varepsilon(n)} \leq \frac{\beta\gamma}{\gamma+1-\beta}$$

$$\Longleftrightarrow \quad n - 1 \geq \frac{1}{w^\star_\beta(i_0)_{k_1} v_{i_0}} \left( h_\varepsilon(n) + \frac{\gamma+1-\beta}{\beta\gamma} \right)$$

is satisfied since $w_\beta^\star(i_0)_{k_1} \geq w_{i_0,-}$, $v_{i_0} \geq v_{\mu,\varepsilon_0}(\varepsilon)$ and $n > C_2(w_{i_0,-}, v_{\mu,\varepsilon_0}(\varepsilon))$.

Let $T_\mu \geq \max_{k\in[2]} C_k(\min_{i\in\mathcal{I}_\varepsilon(\mu)} w_{i,-}, v_{\mu,\varepsilon_0}(\varepsilon))$ and $D_0 = (1+\gamma)^2/\beta$. In summary, we have shown that for all $n > T_\mu$, there exists $i_0 \in \mathcal{I}_\varepsilon(\mu)$ and $t_1 \leq n$ with $B_{t_1} = i_0$ and such that (15) holds.

For $\beta = 1/2$, using that $K \geq 2$, $1 < 1 + \gamma^{-1}$ and $5/2 + \gamma^{-1} \leq 3(1 + \gamma^{-1})/2$ (since $\gamma \in (0, 1/2]$), we obtain that

$$\max_{i\in[2]} C_i(w,v) \leq \sup\left\{ n \geq 1 \mid n - 1 < \frac{1+\gamma^{-1}}{wv}\left(4(2+s)H_{\mu,\varepsilon_0}(\varepsilon)\log n + 3K^2/4 + 1\right)\right\}$$

$$< h_1\left(\frac{4(2+s)(1+\gamma^{-1})}{wv}H_{\mu,\varepsilon_0}(\varepsilon), \frac{(1+\gamma^{-1})(3K^2/4+1)}{wv} + 1\right),$$

where the last inequality uses Lemma 51 and $h_1(x,y)$ defined therein.

Using $w_{i_0,-} = \min_{i\neq i_0} w_\beta^\star(i_0)_i$ and $\beta = 1/2$, the first part of Theorem 5 (i.e. $x = 0$) is obtained by taking the infimum over $(\varepsilon, x) \in [0, \varepsilon_0] \times \{0\}$.

**Refined Proof of Lemma 34.** When considering large $K$ or instances with unbalanced $\beta$-optimal allocation, $\min_{i_0\in\mathcal{I}_\varepsilon(\mu)} \min_{i\neq i_0} w_\beta^\star(i_0)_i$ can become arbitrarily small, hence a dependence in its inverse is undesired. Thanks to the refined analysis from [20], we can clip it with a value of our choosing which is away from zero.

Let $x \in (0, 1/(K-1)]$ be an allocation threshold and $d_\mu(x,i) := |\{j \in [K] \setminus \{i\} \mid w_\beta^\star(i)_j < (1-\beta)x\}|$ for all $i \in \mathcal{I}_\varepsilon(\mu)$. The equation (16) is only informative when (1) $w_\beta^\star(i_0)_{k_1} \geq (1-\beta)x$ or (2) $w_\beta^\star(i_0)_{k_1} < (1-\beta)x$ and $N_{n,k_1}^{i_0} \geq x\sum_{j\neq i_0} N_{n,j}^{i_0}$. The problematic case occurs when (3) $w_\beta^\star(i_0)_{k_1} < (1-\beta)x$ and $N_{n,k_1}^{i_0} < x\sum_{j\neq i_0} N_{n,j}^{i_0}$.

When $w_\beta^\star(i_0)_{k_1} \geq (1-\beta)x$, we can use the above computations by considering $w_{i_0,-} = \max\{(1-\beta)x, \min_{i\neq i_0} w_\beta^\star(i_0)_i\}$.

When $w_\beta^\star(i_0)_{k_1} < (1-\beta)x$ and $N_{n,k_1}^{i_0} \geq x\sum_{j\neq i_0} N_{n,j}^{i_0}$, the above proof can be conducted by using $(1-\beta)x$ instead of $w_\beta^\star(i_0)_{k_1}$ since we will obtain that

$$N_{t_1,k_1}^{i_0} \geq (1-\beta)xv_{i_0}(n-1) - h_\varepsilon(n) \quad \text{and} \quad 1/\beta + \frac{1}{(1-\beta)x} \leq 1/\beta + 1/w_\beta^\star(i_0)_{k_1}.$$

Then, we can likewise consider $w_{i_0,-} = \max\{(1-\beta)x, \min_{i\neq i_0} w_\beta^\star(i_0)_i\}$ to conclude.

When $w_\beta^\star(i_0)_{k_1} < (1-\beta)x$ and $N_{n,k_1}^{i_0} < x\sum_{j\neq i_0} N_{n,j}^{i_0}$, as in (16), the pigeonhole principle yields that there exists $k_2 \in [K] \setminus \{i_0, k_1\}$ such that

$$N_{n,k_2}^{i_0} \geq \frac{w_\beta^\star(i_0)_{k_2}}{1 - \beta - w_\beta^\star(i_0)_{k_1}} \sum_{j\notin\{i_0,k_1\}} N_{n,j}^{i_0} \geq \frac{(1-x)w_\beta^\star(i_0)_{k_2}}{1-\beta} \sum_{j\neq i_0} N_{n,j}^{i_0}.$$

Based on $k_2$ the same dichotomy as for $k_1$ happens: either we can conclude the proof when we are in case 1 or 2 or we cannot since we are in case 3. If case 3 occurs also for $k_2$, i.e. $w_\beta^\star(i_0)_{k_2} < (1-\beta)x$ and $N_{n,k_2}^{i_0} < x\sum_{j\notin\{i_0,k_1\}} N_{n,j}^{i_0}$, we should also ignore it since it is non informative and construct a third arm $k_3$.

The main idea is then to peel off arms that are not informative, till we find an informative one. By induction, we construct a sequence $(k_a)_{a\in[d]}$ of such arms, where $k_d$ is the first arm for which either (1) $w_\beta^\star(i_0)_{k_d} \geq (1-\beta)x$ or (2) $w_\beta^\star(i_0)_{k_d} < (1-\beta)x$ and $N_{n,k_d}^{i_0} \geq x\sum_{j\notin\{i_0\}\cup\{k_a\}_{a\in[d-1]}} N_{n,j}^{i_0}$ holds. This means that for all $a \in [d-1]$, we have $w_\beta^\star(i_0)_{k_a} < (1-\beta)x$ and

$$N_{n,k_a}^{i_0} < x\sum_{j\notin\{i_0\}\cup\{k_b\}_{b\in[a-1]}} N_{n,j}^{i_0}.$$

Using the pigeonhole principle for $i \in [K] \setminus (\{i_0\} \cup \{k_a\}_{a\in[d-1]})$ yields that $k_d$ satisfies

$$N_{n,k_d}^{i_0} \geq \frac{w_\beta^\star(i_0)_{k_d}}{1 - \beta - \sum_{a\in[d-1]} w_\beta^\star(i_0)_{k_a}} \sum_{j\notin\{i_0\}\cup\{k_b\}_{b\in[a-1]}} N_{n,j}^{i_0},$$

Using a simple recurrence on the arms $\{k_a\}_{a\in[d-1]}$, we obtain that

$$\sum_{j\notin\{i_0\}\cup\{k_b\}_{b\in[a-1]}} N_{n,j}^{i_0} \geq (1-x)^{d-1}\sum_{j\neq i_0} N_{n,j}^{i_0}\,.$$

Let $t_d := \sup\{t < n \mid (B_t, C_t) = (i_0, k_d)\}$. Since the arm $k_d$ satisfies case 1 or case 2, we can conclude similarly as above with an extra multiplicative factor $(1-x)^{d-1}$.

To remove the dependency in the random variable $d$, we consider the worst case scenario where $\{k_a\}_{a\in[d-1]} = \{i \in [K] \setminus \{i_0\} \mid w_\beta^\star(i_0)_i < (1-\beta)x\}$, i.e. $d-1 \leq d_\mu(x, i_0)$. In words, it means that we had to enumerate over all arms with small allocation, such that case 2 didn't hold, before finding an arm with large allocation, i.e. satisfying case 1. Let $d_{\mu,\varepsilon}(x) = \max_{i\in\mathcal{I}_\varepsilon(\mu)} d_\mu(x, i)$. Using that

$$(1-x)^{d-1} \geq (1-x)^{d_\mu(x, i_0)} \geq (1-x)^{d_{\mu,\varepsilon}(x)}\,,$$

$$w_{i_0,-} = (1-x)^{d_\mu(x, i_0)}\max\{(1-\beta)x,\ \min_{i\neq i_0} w_\beta^\star(i_0)_i\}\,,$$

we can conclude the proof by taking $T_\mu \geq \max_{k\in[2]} C_k(\min_{i\in\mathcal{I}_\varepsilon(\mu)} w_{i,-}, v_{\mu,\varepsilon_0}(\varepsilon))$ and $D_0 = \frac{(1+\gamma)^2}{\beta(1-x)^{d_{\mu,\varepsilon}(x)}}$. In other words, we have shown that $n > T_\mu$, there exists $i_0 \in \mathcal{I}_\varepsilon(\mu)$ and $t_d \leq n$ with $B_{t_d} = i_0$ and such that (15) holds. Using $\beta = 1/2$, the second part of Theorem 5 is obtained by taking the infimum over $(\varepsilon, x) \in [0, \varepsilon_0] \times (0, 1/(K-1)]$. $\qquad\square$

### E.2.1 Consequence of tighter concentration

It is direct to see that the proof of Theorem 5 will also work when using the concentration event $\tilde{\mathcal{E}}_n = \tilde{\mathcal{E}}_{n,1}^1 \cap \tilde{\mathcal{E}}_{n,1}^1$ with $(\tilde{\mathcal{E}}_{n,\delta}^1)_{n>K}$ and $(\tilde{\mathcal{E}}_{n,\delta}^2)_{n>K}$ as in (27) and (29). Let us define

$$\tilde{f}_1(n) = \frac{1}{2}\overline{W}_{-1}(2s\log n + 2\log(2 + \log n) + 2)\,,$$

$$\tilde{f}_2(n) = \overline{W}_{-1}(s\log n + 2\log(2 + \log n) + 2)\,.$$

Using Lemma 50, we know that $\tilde{f}_1(n) \leq \tilde{f}_2(n)$. Then, we will be able to show

$$\mathbb{E}_\nu[\tau_\delta] \leq \inf_{(\varepsilon,x)\in[0,\varepsilon_0]\times[0,1/(K-1)]} \max\left\{\tilde{T}_{\mu,\varepsilon_0}(\delta, \varepsilon, x) + 1,\ \tilde{S}_{\mu,\varepsilon_0}(\varepsilon, x)\right\} + \zeta(s)K(K+1)/2\,,$$

where

$$\tilde{T}_{\mu,\varepsilon_0}(\delta, \varepsilon, x) = \sup\left\{n \mid n - 1 \leq \frac{2(1+\gamma)^2\sum_{i\in\mathcal{I}_\varepsilon(\mu)} T_{\varepsilon_0,1/2}(\mu, i)}{(1-x)^{d_{\mu,\varepsilon_0}(\varepsilon, x)}}\right.$$

$$\left.\left(\sqrt{c(n-1, \delta)} + \sqrt{\tilde{f}_2(n)}\right)^2\right\}\,,$$

$$\tilde{S}_{\mu,\varepsilon_0}(\varepsilon, x) = \tilde{h}_1\left(\frac{4(1+\gamma^{-1})}{a_{\mu,\varepsilon_0}(\varepsilon, x)v_{\mu,\varepsilon_0}(\varepsilon)}H_{\mu,\varepsilon_0}(\varepsilon),\ \frac{(1+\gamma^{-1})(3K^2/4+1)}{a_{\mu,\varepsilon_0}(\varepsilon, x)v_{\mu,\varepsilon_0}(\varepsilon)} + 1\right)\,,$$

with $\tilde{h}_1(z, y) > \sup\left\{n \geq 1 \mid n < z\tilde{f}_2(n) + y\right\}$. Using Lemma 49, we obtain that

$$n \geq z\tilde{f}_2(n) + y \iff \frac{n-y}{z} - \log(n-y) \geq s\log n + 2\log(2 + \log n) + 2 - \log z$$

$$\Longleftarrow \frac{n}{z(1+s)} - \log\frac{n}{z(1+s)} \geq \log(1+s) + \frac{s}{s+1}\log z + \frac{1}{1+s}(2\log(2 + \log n) + 2 + \frac{y}{z})\,.$$

Therefore, we can consider $\tilde{h}_1(z, y) \geq z(1+s)h_2(z, y)$ with

$$h_2(z, y) = \sup\left\{x \mid x - \log x - \frac{2}{1+s}\log(\log x + 2 + \log(z(1+s))) < C(z, y)\right\}$$

where $C(z, y) = \frac{y}{(1+s)z} + \frac{s}{s+1}\log z + \log(1+s) + \frac{2}{1+s}$. Therefore, we will obtain a mulitplicative factor $(1+s)$ instead of $(2+s)$ in front of $H_{\mu,\varepsilon_0}(\varepsilon)$. This improvement is rather mild, hence we don't elaborate on it further.

# F Probability of error and simple regret

Let $\varepsilon_0 > 0$ and $\beta \in (0,1)$. In Appendix F, we provide an analysis on the probability of error and the simple regret of EB-TC$_{\varepsilon_0}$ with fixed proportions $\beta$ and slack $\varepsilon_0 > 0$ (Theorem 6 proved in Appendix F.1). In Appendix F.2, we provide an upper bound on the unverifiable expected sample complexity of EB-TC$_{\varepsilon_0}$ (Theorem 7). Then, by using Theorem 6, we prove that the policy pulling arm $\hat{\imath}_n$ at time $n$ has a constant induced regret (Corollary 2 proved in Appendix F.3). In practice, we will mostly be interested in the case $\beta = 1/2$.

Theorem 3 and Corollary 1 is a corollary of Theorem 6 up to some additional computations on the complexities $(H_i(\mu, \varepsilon_0))_{i \in [C_\mu - 1]}$, which we will detail below. Theorem 6 provides upper bound on $\mathbb{P}_\nu\left(\hat{\imath}_n \notin \mathcal{I}_\varepsilon(\mu)\right)$ holding for any time $n$ and any error $\varepsilon \geq 0$, and upper bound on $\mathbb{E}_\nu[\mu_\star - \mu_{\hat{\imath}_n}]$ holding for any time $n$.

**Theorem 6.** *Let $\varepsilon_0 > 0$. For all $\varepsilon \geq 0$, let $i_\mu(\varepsilon) = i$ if $\varepsilon \in [\Delta_i, \Delta_{i+1})$ and $i \in [C_\mu - 1]$, otherwise $i_\mu(\varepsilon) = C_\mu$. For all $i \in [C_\mu - 1]$, let $C_i(\varepsilon_0) = 2\Delta_i^{-1} - \varepsilon_0^{-1}$ and $C_{i,j}(\varepsilon_0) = 2\frac{\Delta_j/\varepsilon_0 + 1}{\Delta_i - \Delta_j} + 3\varepsilon_0^{-1}$. For all $i \in [C_\mu - 1]$, let $H_i(\mu, \varepsilon_0) := \min_{j \in [i]} \max\{\bar{H}_{i,j}(\mu, \varepsilon_0), \tilde{H}_{i,j}(\mu, \varepsilon_0)\}$ where*

$$
\bar{H}_{i,j}(\mu, \varepsilon_0) := |i^\star(\mu)| \max\left\{\sqrt{2}\Delta_{j+1}^{-1}, C_{i+1,j}(\varepsilon_0)\right\}^2
$$
$$
+ \max\{C_{j+1}(\varepsilon_0), C_{i+1,j}(\varepsilon_0)\}^2 \left(\sum_{k=2}^{j} |\mathcal{C}_\mu(k)| + \sum_{k=i+1}^{C_\mu} |\mathcal{C}_\mu(k)|\right)
$$
$$
+ \sum_{k=j+1}^{i} |\mathcal{C}_\mu(k)| \max\left\{C_{j+1}(\varepsilon_0), C_{i+1,j}(\varepsilon_0), \sqrt{2}\Delta_k^{-1}\right\}^2 ,
$$

$$
\tilde{H}_{i,j}(\mu, \varepsilon_0) := \frac{2|i^\star(\mu)|}{\Delta_{j+1}^2} + \max\left\{C_{j+1}(\varepsilon_0), \varepsilon_0^{-1}\right\}^2 \sum_{k=2}^{j} |\mathcal{C}_\mu(k)| + \frac{2\sum_{k=1}^{j} |\mathcal{C}_\mu(k)|}{(\Delta_{i+1} - \Delta_j)^2}
$$
$$
+ \sum_{k=j+1}^{C_\mu} |\mathcal{C}_\mu(k)| \max\left\{C_{j+1}(\varepsilon_0), \varepsilon_0^{-1}, \sqrt{2}\Delta_k^{-1}\right\}^2 .
$$

*The EB-TC$_{\varepsilon_0}$ algorithm with fixed proportions $\beta = 1/2$ satisfies that, for all $\nu \in \mathcal{D}^K$ and all $n \geq D_\mu$, if the algorithm has not stopped yet, then*

$$
\forall \varepsilon \geq 0, \quad \mathbb{P}_\nu\left(\hat{\imath}_n \notin \mathcal{I}_\varepsilon(\mu)\right) \leq \mathbb{1}\left(\varepsilon < \Delta_{\max}\right) \frac{K(K+1)}{2} e^2 (2 + \log n)^2 p\left(\frac{n - 5K^2/2}{8H_{i_\mu(\varepsilon)}(\mu, \varepsilon_0)}\right) ,
$$
$$
\mathbb{E}_\nu[\mu_\star - \mu_{\hat{\imath}_n}] \leq \frac{K(K+1)}{2} e^2 (2 + \log n)^2 \sum_{i \in [C_\mu - 1]} (\Delta_{i+1} - \Delta_i) p\left(\frac{n - 5K^2/2}{8H_i(\mu, \varepsilon_0)}\right) .
$$

*where $p(x) = x \exp(-x)$ and $D_\mu = 8H_1(\mu, \varepsilon_0) h_2\left(8H_1(\mu, \varepsilon_0), 5K^2/2, 2 + \log(K(K+1)/2)\right) + 5K^2/2$ with $h_2$ defined in Lemma 51.*

**Hardness of BAI** We have $C_2(\varepsilon_0) = 2\Delta_{\min}^{-1} - \varepsilon_0^{-1}$ and $C_{2,1}(\varepsilon_0) = 2\Delta_{\min}^{-1} + 3\varepsilon_0^{-1}$. Then, we obtain $H_1(\mu, \varepsilon_0) = \bar{H}_{1,1}(\mu, \varepsilon_0) = K(2\Delta_{\min}^{-1} + 3\varepsilon_0^{-1})^2$ since

$$
\tilde{H}_{1,1}(\mu, \varepsilon_0) = 4|i^\star(\mu)|\Delta_{\min}^{-2} + \sum_{k=2}^{C_\mu} |\mathcal{C}_\mu(k)| \max\left\{C_2(\varepsilon_0), \varepsilon_0^{-1}, \sqrt{2}\Delta_k^{-1}\right\}^2 .
$$

This corresponds to the complexity of the *two-groups* instances which are defined as $\mathcal{D}_\Delta := \left\{\nu \in \mathcal{D}^K \mid C_\mu = 2, \Delta_2 = \Delta\right\}$ where $\Delta > 0$. More importantly, it corresponds to the complexity of identifying one of the best arms, i.e. taking $\varepsilon = 0$ yields

$$
\mathbb{P}_\nu\left(\hat{\imath}_n \notin i^\star(\mu)\right) \leq \frac{K(K+1)}{2} e^2 (2 + \log n)^2 p\left(\frac{n - 5K^2/2}{8K(2\Delta_{\min}^{-1} + 3\varepsilon_0^{-1})^2}\right) .
$$

**Hardness of $\varepsilon$-BAI**  Let $i > 1$ and $j \in [i]$. We have $C_{i+1,j}(\varepsilon_0) = 2\frac{\Delta_j/\varepsilon_0 + 1}{\Delta_{i+1} - \Delta_j} + 3\varepsilon_0^{-1}$, $C_{j+1}(\varepsilon_0) = 2\Delta_{j+1}^{-1} - \varepsilon_0^{-1} \leq 2\Delta_{j+1}^{-1}$ and $\sqrt{2}\Delta_k^{-1} \leq 2\Delta_{j+1}^{-1}$ for all $k \geq j+1$. Direct manipulations show that, for all $i > 1$ and all $j \in [i]$, we have

$$\max\{\bar{H}_{i,j}(\mu, \varepsilon_0), \tilde{H}_{i,j}(\mu, \varepsilon_0)\} \leq K \max\left\{2\Delta_{j+1}^{-1},\ 2\frac{\Delta_j/\varepsilon_0 + 1}{\Delta_{i+1} - \Delta_j} + 3\varepsilon_0^{-1}\right\}^2.$$

Therefore, we have

$$H_i(\mu, \varepsilon_0) \leq K \min_{j \in [i]} \max\left\{2\Delta_{j+1}^{-1},\ 2\frac{\Delta_j/\varepsilon_0 + 1}{\Delta_{i+1} - \Delta_j} + 3\varepsilon_0^{-1}\right\}^2.$$

Taking $j = 1$ and then $j = 2$, we see that, for all $i > 1$,

$$H_i(\mu, \varepsilon_0) \leq K \max\left\{2\Delta_{\min}^{-1},\ 3\varepsilon_0^{-1} + 2\Delta_{i+1}^{-1}\right\}^2 < H_1(\mu, \varepsilon_0),$$

$$H_i(\mu, \varepsilon_0) \leq K \max\left\{2\Delta_3^{-1},\ 2\frac{\Delta_{\min}/\varepsilon_0 + 1}{\Delta_{i+1} - \Delta_{\min}} + 3\varepsilon_0^{-1}\right\}^2 \leq K \max\left\{2\frac{\Delta_{\min}/\varepsilon_0 + 1}{\Delta_3 - \Delta_{\min}} + 3\varepsilon_0^{-1}\right\}^2,$$

where the last inequality uses that $\Delta_3 \leq \Delta_{i+1}$ for all $i > 1$.

Similarly, we can derive a lower bound on $H_i(\mu, \varepsilon_0)$ by noting that $H_i(\mu, \varepsilon_0) \geq \min_{j \in [i]} \bar{H}_{i,j}(\mu, \varepsilon_0)$. For all $j \in [i]$, we have that $\sqrt{2}\Delta_{j+1}^{-1} \geq \sqrt{2}\Delta_{i+1}^{-1}$, $C_{j+1}(\varepsilon_0) \geq 2\Delta_{i+1}^{-1} - \varepsilon_0^{-1}$ and $C_{i+1,j}(\varepsilon_0) \geq 2\Delta_{i+1}^{-1} + 3\varepsilon_0^{-1}$, hence we obtain

$$H_i(\mu, \varepsilon_0) \geq \min_{j \in [i]} \bar{H}_{i,j}(\mu, \varepsilon_0) \geq 2K\Delta_{i+1}^{-2}.$$

**Three-groups instances**  The *three-groups* instances are defined as $\mathcal{D}_{\Delta,\Lambda} := \left\{\nu \in \mathcal{D}^K \mid C_\mu = 3, \Delta_2 = \Delta, \Delta_3 = \Lambda\right\}$ where $\Delta > 0$ and $\Lambda > \Delta$. Above results yield

$$H_2(\mu, \varepsilon_0) \leq K \left(2\frac{\Delta/\varepsilon_0 + 1}{\Lambda - \Delta} + 3\varepsilon_0^{-1}\right)^2 = \mathcal{O}\left(\frac{K}{\min\{\Lambda - \Delta, \varepsilon_0\}^2}\right).$$

### F.1   Proof of Theorem 6

In the following, we consider a sub-Gaussian bandit with distribution $\nu \in \mathbb{R}^K$ having mean parameter $\mu \in \mathbb{R}^K$. In particular, our analysis holds when several arms have the largest mean $\mu_\star$.

Let $\Delta_{\max} = \max_{i \in [K]} \mu_\star - \mu_i$. Let $\varepsilon_1$ be an analysis parameter (i.e. not the error parameter from the stopping rule (3)). When $\varepsilon_1 \geq \Delta_{\max}$, we have $\mathcal{I}_{\varepsilon_1}(\mu)^{\complement} = \emptyset$, hence $\mathbb{P}_\nu(\hat{\imath}_n \notin \mathcal{I}_{\varepsilon_1}(\mu)) = 0$. In the following, we consider $\varepsilon_1 \in [0, \Delta_{\max})$.

For all $n > K$ and $\delta \in (0, 1)$, let $\tilde{\mathcal{E}}_{n,\delta} = \tilde{\mathcal{E}}_{n,\delta}^1 \cap \tilde{\mathcal{E}}_{n,\delta}^2$ with $(\tilde{\mathcal{E}}_{n,\delta}^1)_{n>K}$ and $(\tilde{\mathcal{E}}_{n,\delta}^2)_{n>K}$ as in (27) and (29) for $s = 0$, i.e.

$$\tilde{\mathcal{E}}_{n,\delta}^1 = \left\{\forall k \in [K], \forall t \leq n,\ |\mu_{t,k} - \mu_k| < \sqrt{\frac{2\tilde{f}_1(n,\delta)}{N_{t,k}}}\right\},$$

$$\tilde{\mathcal{E}}_{n,\delta}^2 = \left\{\forall (i,k) \in [K]^2 \text{ s.t. } i \neq k,\ \forall t \leq n,\ \frac{|(\mu_{t,i} - \mu_{t,k}) - (\mu_i - \mu_k)|}{\sqrt{1/N_{t,i} + 1/N_{t,k}}} < \sqrt{2\tilde{f}_2(n,\delta)}\right\},$$

where

$$\tilde{f}_1(n,\delta) = \frac{1}{2}\overline{W}_{-1}(2\log(1/\delta) + 2\log(2 + \log n) + 2),$$

$$\tilde{f}_2(n,\delta) = \overline{W}_{-1}(\log(1/\delta) + 2\log(2 + \log n) + 2).$$

Using Lemma 50, we know that $\tilde{f}_1(n,\delta) \leq \tilde{f}_2(n,\delta)$. Using Lemmas 42 and 44, we obtain that $\mathbb{P}_\nu(\tilde{\mathcal{E}}_{n,\delta}^{\complement}) \leq \frac{K(K+1)}{2}\delta$.

Suppose that we have constructed a time $T_{\varepsilon_1}(\delta) > K$ such that $\tilde{\mathcal{E}}_{n,\delta} \subseteq \{\hat{\imath}_n \in \mathcal{I}_{\varepsilon_1}(\mu)\}$ for $n > T_{\varepsilon_1}(\delta)$. Then, using Lemma 48, we obtain

$$\mathbb{P}_\nu(\hat{\imath}_n \notin \mathcal{I}_{\varepsilon_1}(\mu)) \leq \frac{K(K+1)}{2} \inf\{\delta \mid n > T_{\varepsilon_1}(\delta)\} .$$

We will construct an infinite number of times $\{T_{\varepsilon_1}(\delta, \varepsilon)\}_{\varepsilon \in [0, \varepsilon_1]}$ such that the above property holds, hence taking the infimum yields

$$\mathbb{P}_\nu(\hat{\imath}_n \notin \mathcal{I}_{\varepsilon_1}(\mu)) \leq \frac{K(K+1)}{2} \inf\{\delta \mid n > \inf_{\varepsilon \in [0, \varepsilon_1]} T_{\varepsilon_1}(\delta, \varepsilon)\} .$$

This proof strategy is the one used to prove Theorem 6.

Let $\varepsilon \in [0, \varepsilon_1]$. Since $\hat{\imath}_n = B_n^{\mathrm{EB}}$, our goal is to construct a time $T_{\varepsilon_1}(\delta, \varepsilon)$ such that

$$\forall n > T_{\varepsilon_1}(\delta, \varepsilon), \quad \tilde{\mathcal{E}}_{n,\delta} \cap \{B_n^{\mathrm{EB}} \notin \mathcal{I}_{\varepsilon_1}(\mu)\} = \emptyset .$$

**Error due to undersampled arms** Let us denote by $U_{\varepsilon, \varepsilon_1, t}(n, \delta)$ the set of undersampled arms which are not $\varepsilon_1$-good, i.e.

$$U_{\varepsilon, \varepsilon_1, t}(n, \delta) := \mathcal{I}_{\varepsilon_1}(\mu)^{\complement} \cap \left\{i \mid N_{t,i} \leq 4C_{\varepsilon,i}\tilde{f}_2(n, \delta)\right\} \quad \text{with} \quad C_{\varepsilon,i} := \frac{2}{\min_{k \in \mathcal{I}_\varepsilon(\mu)}(\Delta_i - \Delta_k)^2} .$$

Lemma 35 shows that, for $n$ large enough, a necessary condition for an error to occur is to have an undersampled leader.

**Lemma 35.** *Let $\varepsilon_1 > 0$ and $\varepsilon \in [0, \varepsilon_1]$. Let $H_{\mu, \varepsilon_0}(\varepsilon)$ as in Lemma 5 and define*

$$\tilde{H}_{\varepsilon, \varepsilon_1}(\mu, \varepsilon_0) := H_{\mu, \varepsilon_0}(\varepsilon) + \frac{2|\mathcal{I}_\varepsilon(\mu)|}{\min_{(i,j) \in \mathcal{I}_\varepsilon(\mu) \times \mathcal{I}_{\varepsilon_1}(\mu)^{\complement}}(\Delta_j - \Delta_i)^2} . \tag{19}$$

*Let us define*

$$S_{\varepsilon, \varepsilon_1, \varepsilon_0, \mu}(\delta) = \sup\left\{n \mid n \leq \frac{4\tilde{H}_{\varepsilon, \varepsilon_1}(\mu, \varepsilon_0)}{\min\{\beta, 1 - \beta\}} \tilde{f}_2(n, \delta) + (3/2 + 1/\beta)K^2\right\} .$$

*For all $n > S_{\varepsilon, \varepsilon_1, \varepsilon_0, \mu}(\delta)$, under the event $\tilde{\mathcal{E}}_{n,\delta}$, we have $B_n^{EB} \notin \mathcal{I}_{\varepsilon_1}(\mu)$ implies that $B_n^{EB} \in U_{\varepsilon, \varepsilon_1, n}(n, \delta)$.*

*Proof.* Let $|\mathcal{I}_\varepsilon(\mu)| = I_\varepsilon$ and $g_{\varepsilon, \varepsilon_0}(n, \delta) = \frac{4H_{\mu, \varepsilon_0}(\varepsilon)}{\min\{\beta, 1-\beta\}} \tilde{f}_2(n, \delta) + 3K(K-1)/2$. Using Lemma 5 and the pigeonhole principle, there exists $i_0 \in \mathcal{I}_\varepsilon(\mu)$ such that

$$\sum_j T_n(i_0, j) \geq (n - 1 - g_{\varepsilon, \varepsilon_0}(n, \delta))/I_\varepsilon .$$

Therefore, we have

$$N_{n, i_0} \geq \beta \sum_j T_n(i_0, j) - (K - 1) \geq \beta (n - 1 - g_{\varepsilon, \varepsilon_0}(n, \delta))/I_\varepsilon - (K - 1) .$$

Let $S_{\varepsilon, \varepsilon_1, \varepsilon_0, \mu}(\delta)$ defined as in the statement of Lemma 35. Direct manipulations shows that

$$S_{\varepsilon, \varepsilon_1, \varepsilon_0, \mu}(\delta) \geq \sup\left\{n \mid n - 1 \leq g_{\varepsilon, \varepsilon_0}(n, \delta) + \frac{I_\varepsilon}{\beta}\left(4\tilde{f}_2(n, \delta) \max_{i \notin \mathcal{I}_{\varepsilon_1}(\mu)} C_{\varepsilon,i} + K - 1\right)\right\} .$$

Therefore, we have $N_{n, i_0} > 4\tilde{f}_2(n, \delta) \max_{i \notin \mathcal{I}_{\varepsilon_1}(\mu)} C_{\varepsilon,i}$ for all $n > S_{\varepsilon, \varepsilon_1, \varepsilon_0, \mu}(\delta)$.

Let $n > S_{\varepsilon, \varepsilon_1, \varepsilon_0, \mu}(\delta)$. Suppose that $B_n^{\mathrm{EB}} = i \notin \mathcal{I}_{\varepsilon_1}(\mu)$. Using Lemma 31, under the event $\tilde{\mathcal{E}}_{n,\delta}$, we obtain that

$$\min\{N_{n,i}, N_{n,i_0}\} \leq \frac{8\tilde{f}_1(n, \delta)}{(\mu_{i_0} - \mu_i)^2} \leq 4C_{\varepsilon,i}\tilde{f}_2(n, \delta) .$$

Suppose towards contradiction that $\min\{N_{n,i}, N_{n,i_0}\} = N_{n,i_0}$, then we have $N_{n,i_0} \leq 4C_{\varepsilon,i}\tilde{f}_2(n, \delta)$. This is a direct contradiction with $N_{n,i_0} > 4\tilde{f}_2(n, \delta) \max_{i \notin \mathcal{I}_{\varepsilon_1}(\mu)} C_{\varepsilon,i}$ since $n > S_{\varepsilon, \varepsilon_1, \varepsilon_0, \mu}(\delta)$. Therefore, we have shown that $\min\{N_{n,i}, N_{n,i_0}\} = N_{n,i}$, hence $i \in U_{\varepsilon, \varepsilon_1, n}(n, \delta)$. This concludes the proof. $\qquad\square$

**No remaining undersampled arms** Lemma 36 shows that, if there are still undersampled arms which are not $\varepsilon_1$-good, then either the leader or the challenger was not often selected as leader or challenger.

**Lemma 36.** *Let $\varepsilon_1 \geq 0$ and $\varepsilon \in [0,\varepsilon_1]$. Let $\Delta_\mu(\varepsilon) = \min_{k \notin \mathcal{I}_\varepsilon(\mu)} \Delta_k$, $C_{\mu,\varepsilon_0}(\varepsilon,\varepsilon_1) = 2\max_{j \notin \mathcal{I}_{\varepsilon_1}(\mu)} \frac{\Delta_j/\varepsilon_0 + 1}{\min_{k \in \mathcal{I}_\varepsilon(\mu)}(\Delta_j - \Delta_k)} + \varepsilon_0^{-1}$ and $C_{\mu,\varepsilon_0}(\varepsilon) = 2/\Delta_\mu(\varepsilon) - \varepsilon_0^{-1}$. Let $A_{\varepsilon,\varepsilon_1,\varepsilon_0,i} = \max\{2/\Delta_\mu(\varepsilon)^2, C_{\mu,\varepsilon_0}(\varepsilon,\varepsilon_1)^2\}$ for all $i \in i^\star(\mu)$, $A_{\varepsilon,\varepsilon_1,\varepsilon_0,i} = \max\{C_{\mu,\varepsilon_0}(\varepsilon)^2, C_{\mu,\varepsilon_0}(\varepsilon,\varepsilon_1)^2\}$ for all $i \in (\mathcal{I}_\varepsilon(\mu) \setminus i^\star(\mu)) \cup ([K] \setminus \mathcal{I}_{\varepsilon_1}(\mu))$, and otherwise $A_{\varepsilon,\varepsilon_1,\varepsilon_0,i} = \max\{C_{\mu,\varepsilon_0}(\varepsilon)^2, C_{\mu,\varepsilon_0}(\varepsilon,\varepsilon_1)^2, 2/\Delta_i^2\}$ for all $i \in \mathcal{I}_{\varepsilon_1}(\mu) \setminus \mathcal{I}_\varepsilon(\mu)$. For all $n > K$, under event $\tilde{\mathcal{E}}_{n,\delta}$, for all $t \in [n] \setminus [K]$ such that $U_{\varepsilon,\varepsilon_1,t}(n,\delta) \neq \emptyset$, there exists $i_t \in [K]$ such that*

$$T_t(i_t) \leq \frac{4\tilde{f}_2(n,\delta)}{\min\{\beta, 1-\beta\}} A_{\varepsilon,\varepsilon_1,\varepsilon_0,i_t} + 3(K-1)/2 \quad and \quad T_{t+1}(i_t) = T_t(i_t) + 1 \,.$$

*Proof.* We will be interested in three distinct cases since

$$\begin{aligned}
\{U_{\varepsilon,\varepsilon_1,t}(n,\delta) \neq \emptyset\} &= \{U_{\varepsilon,\varepsilon_1,t}(n,\delta) \cap \{B_t^{\mathrm{EB}}, C_t^{\mathrm{TC}}\} \neq \emptyset\} \\
&\cup \{U_{\varepsilon,\varepsilon_1,t}(n,\delta) \neq \emptyset, \, U_{\varepsilon,\varepsilon_1,t}(n,\delta) \cap \{B_t^{\mathrm{EB}}, C_t^{\mathrm{TC}}\} = \emptyset, \, B_t^{\mathrm{EB}} \notin \mathcal{I}_\varepsilon(\mu)\} \\
&\cup \{U_{\varepsilon,\varepsilon_1,t}(n,\delta) \neq \emptyset, \, U_{\varepsilon,\varepsilon_1,t}(n,\delta) \cap \{B_t^{\mathrm{EB}}, C_t^{\mathrm{TC}}\} = \emptyset, \, B_t^{\mathrm{EB}} \in \mathcal{I}_\varepsilon(\mu)\} \,,
\end{aligned}$$

**Case 1.** Let $t \in [n] \setminus [K]$ such that $\{B_t^{\mathrm{EB}}, C_t^{\mathrm{TC}}\} \cap U_{\varepsilon,\varepsilon_1,t}(n,\delta) \neq \emptyset$. Let $k_t \in \{B_t^{\mathrm{EB}}, C_t^{\mathrm{TC}}\} \cap U_{\varepsilon,\varepsilon_1,t}(n,\delta)$. For this $k_t \notin \mathcal{I}_{\varepsilon_1}(\mu)$, we have $T_{t+1}(k_t) = T_t(k_t) + 1$ and, by combining Lemma 31 and $N_{t,k_t} \leq 4C_{\varepsilon,k_t}\tilde{f}_2(n,\delta)$, we obtain that

$$T_t(k_t) \leq \frac{N_{t,k_t}}{\min\{\beta, 1-\beta\}} + \frac{3(K-1)}{2} \leq \frac{4\tilde{f}_2(n,\delta)}{\min\{\beta, 1-\beta\}} C_{\varepsilon,k_t} + 3(K-1)/2 \,.$$

**Case 2.** Let $t \in [n] \setminus [K]$ such that $U_{\varepsilon,\varepsilon_1,t}(n,\delta) \neq \emptyset$, $U_{\varepsilon,\varepsilon_1,t}(n,\delta) \cap \{B_t^{\mathrm{EB}}, C_t^{\mathrm{TC}}\} = \emptyset$ and $B_t^{\mathrm{EB}} \notin \mathcal{I}_\varepsilon(\mu)$. Let $\Delta_\mu(\varepsilon)$ and $C_{\mu,\varepsilon_0}(\varepsilon)$ defined as in the statement of Lemma 36. Let $D_{\varepsilon_0,\varepsilon,i} = 2/\Delta_\mu(\varepsilon)^2$ for all $i \in i^\star(\mu)$, $D_{\varepsilon_0,\varepsilon,i} = C_{\mu,\varepsilon_0}(\varepsilon)$ for all $i \in \mathcal{I}_\varepsilon(\mu) \setminus i^\star(\mu)$, otherwise $D_{\varepsilon_0,\varepsilon,i} = \max\{C_{\mu,\varepsilon_0}(\varepsilon), 2/\Delta_i^2\}$. Using Lemma 32, there exists $k_t \in [K]$ such that

$$T_t(k_t) \leq \frac{4\tilde{f}_2(n,\delta)}{\min\{\beta, 1-\beta\}} D_{\varepsilon,\varepsilon_0,k_t} + 3(K-1)/2 \quad and \quad T_{t+1}(k_t) = T_t(k_t) + 1 \,.$$

**Case 3.** Let $t \in [n] \setminus [K]$ such that $U_{\varepsilon,\varepsilon_1,t}(n,\delta) \neq \emptyset$, $U_{\varepsilon,\varepsilon_1,t}(n,\delta) \cap \{B_t^{\mathrm{EB}}, C_t^{\mathrm{TC}}\} = \emptyset$ and $B_t^{\mathrm{EB}} \in \mathcal{I}_\varepsilon(\mu)$. Let $j_0 \in U_{\varepsilon,\varepsilon_1,t}(n,\delta) \setminus \{B_t^{\mathrm{EB}}, C_t^{\mathrm{TC}}\}$, which is possible since $U_{\varepsilon,\varepsilon_1,t}(n,\delta) \cap \{B_t^{\mathrm{EB}}, C_t^{\mathrm{TC}}\} = \emptyset$ and $U_{\varepsilon,\varepsilon_1,t}(n,\delta) \neq \emptyset$. Let us denote by $(B_t^{\mathrm{EB}}, C_t^{\mathrm{TC}}) = (i,j)$ with $i \in \mathcal{I}_\varepsilon(\mu)$ and $j \neq j_0$. Using the TC challenger, under the event $\mathcal{E}_{n,\delta}$, we obtain

$$\frac{\mu_i - \mu_{j_0} + \varepsilon_0}{\sqrt{1/N_{t,i} + 1/N_{t,j_0}}} + \sqrt{2\tilde{f}_2(n,\delta)} \geq \frac{\mu_{t,i} - \mu_{t,j_0} + \varepsilon_0}{\sqrt{1/N_{t,i} + 1/N_{t,j_0}}} \geq \frac{\mu_{t,i} - \mu_{t,j} + \varepsilon_0}{\sqrt{1/N_{t,i} + 1/N_{t,j}}}$$

$$\geq \varepsilon_0 \sqrt{\min\{N_{t,i}, N_{t,j}\}/2} \,.$$

Since $\mu_i - \mu_{j_0} + \varepsilon_0 \geq \varepsilon_0 > 0$, we have

$$\frac{\mu_i - \mu_{j_0} + \varepsilon_0}{\sqrt{1/N_{t,i} + 1/N_{t,j_0}}} \leq \sqrt{N_{t,j_0}}(\mu_i - \mu_{j_0} + \varepsilon_0) + \sqrt{2\tilde{f}_2(n,\delta)}$$

$$\leq \left(2\frac{\mu_i - \mu_{j_0} + \varepsilon_0}{\min_{k \in \mathcal{I}_\varepsilon(\mu)}(\mu_k - \mu_{j_0})} + 1\right)\sqrt{2\tilde{f}_2(n,\delta)} \,.$$

Using Lemma 31 to lower bound $\min\{N_{t,i}, N_{t,j}\}$ and $\mu_i - \mu_{j_0} \leq \Delta_{j_0}$ for all $i \in \mathcal{I}_\varepsilon(\mu)$, direct manipulation yields that

$$\min\{T_t(i), T_t(j)\} \leq \frac{4\tilde{f}_2(n,\delta)}{\min\{\beta, 1-\beta\}} C_{\mu,\varepsilon_0}(\varepsilon,\varepsilon_1)^2 + 3(K-1)/2 \,.$$

where $C_{\mu,\varepsilon_0}(\varepsilon,\varepsilon_1)$ is defined as in the statement of Lemma 36. It is direct to see that $C_{\mu,\varepsilon_0}(\varepsilon,\varepsilon_1)^2 \geq \max\{1/\varepsilon_0^2, \max_{i\notin\mathcal{I}_{\varepsilon_1}(\mu)} C_{\varepsilon,i}\}$ and $C_{\varepsilon,i} \geq 2/\Delta_i^2$ for all $i \notin \mathcal{I}_{\varepsilon_1}(\mu)$. The above shows that there exists $k_t \in [K]$ such that

$$T_t(k_t) \leq \frac{4\tilde{f}_2(n,\delta)}{\min\{\beta,1-\beta\}} C_{\mu,\varepsilon_0}(\varepsilon,\varepsilon_1)^2 + 3(K-1)/2 \quad \text{and} \quad T_{t+1}(k_t) = T_t(k_t) + 1 \ .$$

**Summary.** Let us define $(A_{\varepsilon,\varepsilon_1,\varepsilon_0,i})_{i\in[K]}$ as in the statement of Lemma 36. Under $\tilde{\mathcal{E}}_{n,\delta}$, we have show that, when $U_{\varepsilon,\varepsilon_1,t}(n,\delta) \neq \emptyset$, there exists $i_t \in [K]$ such that

$$T_t(i_t) \leq \frac{4\tilde{f}_2(n,\delta)}{\min\{\beta,1-\beta\}} A_{\varepsilon,\varepsilon_1,\varepsilon_0,i_t} + 3(K-1)/2 \quad \text{and} \quad T_{t+1}(i_t) = T_t(i_t) + 1 \ .$$

$\square$

Lemma 37 shows that, for $n$ large enough, there is no undersampled arms which are not $\varepsilon_1$-good.

**Lemma 37.** *Let $\varepsilon_1 \geq 0$ and $\varepsilon \in [0, \varepsilon_1]$. Let $\Delta_\mu(\varepsilon)$, $C_{\mu,\varepsilon_0}(\varepsilon,\varepsilon_1)$ and $C_{\mu,\varepsilon_0}(\varepsilon)$ as in Lemma 36 and*

$$\begin{aligned}
\bar{H}_{\varepsilon,\varepsilon_1}(\mu,\varepsilon_0) := &\ |i^\star(\mu)| \max\{\sqrt{2}\Delta_\mu(\varepsilon)^{-1}, C_{\mu,\varepsilon_0}(\varepsilon,\varepsilon_1)\}^2 \\
&+ |(\mathcal{I}_\varepsilon(\mu) \setminus i^\star(\mu)) \cup ([K] \setminus \mathcal{I}_{\varepsilon_1}(\mu))| \max\{C_{\mu,\varepsilon_0}(\varepsilon), C_{\mu,\varepsilon_0}(\varepsilon,\varepsilon_1)\}^2 \\
&+ \sum_{i\in\mathcal{I}_{\varepsilon_1}(\mu)\setminus\mathcal{I}_\varepsilon(\mu)} \max\{C_{\mu,\varepsilon_0}(\varepsilon), C_{\mu,\varepsilon_0}(\varepsilon,\varepsilon_1), \sqrt{2}\Delta_i^{-1}\}^2 \ .
\end{aligned} \tag{20}$$

*Let us define*

$$T_{\varepsilon,\varepsilon_1,\varepsilon_0,\mu}(\delta) = \sup\left\{ n \mid n \leq \frac{4\bar{H}_{\varepsilon,\varepsilon_1}(\mu,\varepsilon_0)}{\min\{\beta,1-\beta\}} \tilde{f}_2(n,\delta) + 3K^2/2 \right\} \ .$$

*For all $n > T_{\varepsilon,\varepsilon_1,\varepsilon_0,\mu}(\delta)$, under the event $\tilde{\mathcal{E}}_{n,\delta}$, we have $U_{\varepsilon,\varepsilon_1,n}(n,\delta) = \emptyset$.*

*Proof.* For all $n > K$, under event $\tilde{\mathcal{E}}_{n,\delta}$, combining Lemma 4 and 36 for $A_t(n,\delta) = \{U_{\varepsilon,\varepsilon_1,t}(n,\delta) \neq \emptyset\}$ and $D_i(n,\delta) = \frac{4\tilde{f}_2(n,\delta)}{\min\{\beta,1-\beta\}} A_{\varepsilon,\varepsilon_1,\varepsilon_0,i} + 3(K-1)/2$ yields that

$$\sum_{t=K+1}^{n} \mathbb{1}\left(U_{\varepsilon,\varepsilon_1,t}(n,\delta) \neq \emptyset\right) \leq \frac{4\tilde{f}_2(n,\delta)}{\min\{\beta,1-\beta\}} \bar{H}_{\varepsilon,\varepsilon_1}(\mu,\varepsilon_0) + 3K(K-1)/2 \ .$$

where we used that $\sum_{i\in[K]} A_{\varepsilon,\varepsilon_1,\varepsilon_0,i} = \bar{H}_{\varepsilon,\varepsilon_1}(\mu,\varepsilon_0)$ where $\bar{H}_{\varepsilon,\varepsilon_1}(\mu,\varepsilon_0)$ is defined in (20).

For all $i \notin \mathcal{I}_{\varepsilon_1}(\mu)$, let us define

$$t_i(n,\delta) = \max\left\{t \in [n] \setminus [K] \mid i \in U_{\varepsilon,\varepsilon_1,t}(n,\delta)\right\} \ .$$

By definition, for all $i \notin \mathcal{I}_{\varepsilon_1}(\mu)$, we have $i \in U_{\varepsilon,\varepsilon_1,t}(n,\delta)$ for all $t \in (K, t_i(n,\delta)]$ and $i \notin U_{\varepsilon,\varepsilon_1,t}(n,\delta)$ for all $t \in (t_i(n,\delta), n]$. Therefore, for all $t \in (K, \max_{i\notin\mathcal{I}_{\varepsilon_1}(\mu)} t_i(n,\delta)]$, we have $U_{\varepsilon,\varepsilon_1,t}(n,\delta) \neq \emptyset$ and $U_{\varepsilon,\varepsilon_1,t}(n,\delta) = \emptyset$ for all $t > \max_{i\notin\mathcal{I}_{\varepsilon_1}(\mu)} t_i(n,\delta)$, hence

$$\begin{aligned}
\max_{i\notin\mathcal{I}_{\varepsilon_1}(\mu)} (t_i(n,\delta) - K) &= \sum_{t=K+1}^{n} \mathbb{1}\left(U_{\varepsilon,\varepsilon_1,t}(n,\delta) \neq \emptyset\right) \\
&\leq \frac{4\tilde{f}_2(n,\delta)}{\min\{\beta,1-\beta\}} \bar{H}_{\varepsilon,\varepsilon_1}(\mu,\varepsilon_0) + 3K(K-1)/2 \ .
\end{aligned}$$

Let $T_{\varepsilon,\varepsilon_1,\varepsilon_0,\mu}(\delta)$ defined as in the statement of Lemma 37. Direct manipulations show that

$$T_{\varepsilon,\varepsilon_1,\varepsilon_0,\mu}(\delta) \geq \sup\left\{ n \mid n - K \leq \frac{4\tilde{f}_2(n,\delta)}{\min\{\beta,1-\beta\}} \bar{H}_{\varepsilon,\varepsilon_1}(\mu,\varepsilon_0) + 3K(K-1)/2 \right\} \ .$$

Let $n > T_{\varepsilon,\varepsilon_1,\varepsilon_0,\mu}(\delta)$. Then, we have

$$n - K > \frac{4\tilde{f}_2(n,\delta)}{\min\{\beta,1-\beta\}} \bar{H}_{\varepsilon,\varepsilon_1}(\mu,\varepsilon_0) + 3K(K-1)/2 \geq \max_{i\notin\mathcal{I}_{\varepsilon_1}(\mu)} (t_i(n,\delta) - K) \ ,$$

hence $n > \max_{i\notin\mathcal{I}_{\varepsilon_1}(\mu)} t_i(n,\delta)$. This conclude the proof that $U_{\varepsilon,\varepsilon_1,n}(n,\delta) = \emptyset$. $\square$

**Upper bound probability of error** Let $S_{\varepsilon,\varepsilon_1,\varepsilon_0,\mu}(\delta)$ and $T_{\varepsilon,\varepsilon_1,\varepsilon_0,\mu}(\delta)$ as in Lemmas 35 and 37. For $\beta = 1/2$, direct manipulations show that $\max\{S_{\varepsilon,\varepsilon_1,\varepsilon_0,\mu}(\delta), T_{\varepsilon,\varepsilon_1,\varepsilon_0,\mu}(\delta)\} \leq E_{\varepsilon,\varepsilon_1,\varepsilon_0,\mu}(\delta)$ with

$$E_{\varepsilon,\varepsilon_1,\varepsilon_0,\mu}(\delta) := \sup\left\{n \mid n \leq 8\max\{\bar{H}_{\varepsilon,\varepsilon_1}(\mu,\varepsilon_0), \tilde{H}_{\varepsilon,\varepsilon_1}(\mu,\varepsilon_0)\}\tilde{f}_2(n,\delta) + 5K^2/2\right\}$$

Combining Lemmas 35 and 37, for all $n > E_{\varepsilon,\varepsilon_1,\varepsilon_0,\mu}(\delta)$, we proved that

$$\tilde{\mathcal{E}}_{n,\delta} \cap \{\hat{\imath}_n \notin \mathcal{I}_{\varepsilon_1}(\mu)\} = \tilde{\mathcal{E}}_{n,\delta} \cap \{B_n^{\mathrm{EB}} \notin \mathcal{I}_{\varepsilon_1}(\mu)\} \subseteq \tilde{\mathcal{E}}_{n,\delta} \cap \{U_{\varepsilon,\varepsilon_1,n}(n,\delta) \neq \emptyset\} = \emptyset \ ,$$

hence $\{\hat{\imath}_n \notin \mathcal{I}_{\varepsilon_1}(\mu)\} \subseteq \tilde{\mathcal{E}}_{n,\delta}^{\complement}$.

Using Lemma 48, $\mathbb{P}_\nu(\tilde{\mathcal{E}}_{n,\delta}^{\complement}) \leq K(K+1)\delta/2$ and taking the infimum over $\varepsilon \in [0,\varepsilon_1]$, yields

$$\mathbb{P}_\nu(\hat{\imath}_n \notin \mathcal{I}_{\varepsilon_1}(\mu)) \leq \frac{K(K+1)}{2} \inf\{\delta \mid n > E_{\varepsilon_1,\varepsilon_0,\mu}(\delta)\} \ .$$

where we used Lemma 52 and $\inf_{\varepsilon\in[0,\varepsilon_1]} E_{\varepsilon,\varepsilon_1,\varepsilon_0,\mu}(\delta) = E_{\varepsilon_1,\varepsilon_0,\mu}(\delta)$ with

$$E_{\varepsilon_1,\varepsilon_0,\mu}(\delta) := \sup\left\{n \mid n \leq 8\inf_{\varepsilon\in[0,\varepsilon_1]}\max\{\bar{H}_{\varepsilon,\varepsilon_1}(\mu,\varepsilon_0), \tilde{H}_{\varepsilon,\varepsilon_1}(\mu,\varepsilon_0)\}\tilde{f}_2(n,\delta) + 5K^2/2\right\} \ .$$

Using Lemma 52, we obtain that

$$\mathbb{P}_\nu\left(\hat{\imath}_n \notin \mathcal{I}_{\varepsilon_1}(\mu)\right) \leq \mathbb{1}\left(\varepsilon_1 < \Delta_{C_\mu}\right) \frac{K(K+1)}{2}e^2(2+\log n)^2$$
$$p\left(\frac{n - 5K^2/2}{8\inf_{\varepsilon\in[0,\varepsilon_1]}\max\{\bar{H}_{\varepsilon,\varepsilon_1}(\mu,\varepsilon_0), \tilde{H}_{\varepsilon,\varepsilon_1}(\mu,\varepsilon_0)\}}\right) \ ,$$

where $p(x) = x\exp(-x)$.

**More explicit complexity** In order to obtain a more interpretable result, we notice that the dependence in $(\varepsilon,\varepsilon_1)$ of $\bar{H}_{\varepsilon,\varepsilon_1}(\mu,\varepsilon_0)$ and $\tilde{H}_{\varepsilon,\varepsilon_1}(\mu,\varepsilon_0)$ is only with respect to its position in the ordered means. In the following, we will replace the dependency in $(\varepsilon,\varepsilon_1)$ by a dependency in such indices.

Let $\mu \in \mathbb{R}^K$. Let us denote by $C_\mu := |\{\mu_i \mid i \in [K]\}|$ the number of different arm means in $\mu$. For all $i \in [C_\mu]$, we denote the set of arms having mean gap $\Delta_i$ by $\mathcal{C}_\mu(i) := \{k \in [K] \mid \mu_\star - \mu_k = \Delta_i\}$ where the gaps are sorted by increasing order

$$0 = \Delta_1 < \Delta_2 < \cdots < \Delta_{C_\mu} \ .$$

In particular, we have $\mathcal{C}_\mu(1) = i^\star(\mu)$ and $\Delta_{C_\mu} = \Delta_{\max}$. Let us denote by

$$\forall \varepsilon > 0, \quad i_\mu(\varepsilon) := \begin{cases} i & \text{if } \varepsilon \in [\Delta_i, \Delta_{i+1}) \quad \text{with} \quad i \in [1, C_\mu - 1] \\ C_\mu & \text{if } \varepsilon \geq \Delta_{C_\mu} \end{cases} \ . \tag{21}$$

It is direct to see that $i_\mu(\varepsilon) = \max\{i \in [C_\mu], \mathcal{C}_\mu(i) \subset \mathcal{I}_\varepsilon(\mu)\}$ for all $\varepsilon > 0$. For all $i \in [C_\mu - 1]$, let us define $H_i(\mu,\varepsilon_0) = \min_{j\in[i]}\max\{\bar{H}_{i,j}(\mu,\varepsilon_0), \tilde{H}_{i,j}(\mu,\varepsilon_0)\}$ with $\bar{H}_{i,j}(\mu,\varepsilon_0)$ and $\tilde{H}_{i,j}(\mu,\varepsilon_0)$ as in Theorem 6. Let $\varepsilon_1 \geq 0$ and $\varepsilon \in [0,\varepsilon_1]$. Then, we have by direct manipulations that

$$\inf_{\varepsilon\in[0,\varepsilon_1]}\max\{\bar{H}_{\varepsilon,\varepsilon_1}(\mu,\varepsilon_0), \tilde{H}_{\varepsilon,\varepsilon_1}(\mu,\varepsilon_0)\} = H_{i_\mu(\varepsilon_1)}(\mu,\varepsilon_0) \ ,$$

since $\bar{H}_{\varepsilon,\varepsilon_1}(\mu,\varepsilon_0) = \bar{H}_{i_\mu(\varepsilon_1),i_\mu(\varepsilon)}(\mu,\varepsilon_0)$ and $\tilde{H}_{\varepsilon,\varepsilon_1}(\mu,\varepsilon_0) = \tilde{H}_{i_\mu(\varepsilon_1),i_\mu(\varepsilon)}(\mu,\varepsilon_0)$. This concludes the first part of Theorem 6, i.e.

$$\mathbb{P}_\nu(\hat{\imath}_n \notin \mathcal{I}_{\varepsilon_1}(\mu)) \leq \mathbb{1}\left(\varepsilon_1 < \Delta_{\max}\right) \frac{K(K+1)}{2}e^2(2+\log n)^2 p\left(\frac{n - 5K^2/2}{8H_{i_\mu(\varepsilon_1)}(\mu,\varepsilon_0)}\right) \ .$$

**Simple regret** Since $\mu_\star - \mu_{\hat{i}_n}$ is a positive random variable, we obtain that

$$\mathbb{E}_\nu[\mu_\star - \mu_{\hat{i}_n}] = \int_{\varepsilon_1=0}^{+\infty} \mathbb{P}_\nu(\mu_\star - \mu_{\hat{i}_n} > \varepsilon_1) \mathrm{d}\,\varepsilon_1 = \int_{\varepsilon_1=0}^{+\infty} \mathbb{P}_\nu(\hat{i}_n \notin \mathcal{I}_{\varepsilon_1}(\mu)) \mathrm{d}\,\varepsilon_1 \,.$$

As a function of $\varepsilon$, our upper bound on $\mathbb{P}_\nu(\hat{i}_n \notin \mathcal{I}_{\varepsilon_1}(\mu))$ is piecewise constant on $[\Delta_i, \Delta_{i+1})$ for all $i \in [C_\mu - 1]$ and has null value for $\varepsilon_1 \geq \Delta_{\max}$. Our upper is informative, i.e. smaller than 1, for all $n \geq D_\mu$ where

$$D_\mu = 8H_1(\mu, \varepsilon_0)h_2\left(8H_1(\mu, \varepsilon_0), 5K^2/2, 2 + \log\left(K(K+1)/2\right)\right) + 5K^2/2$$
$$> \sup\left\{x \mid x < 8H_1(\mu, \varepsilon_0)\overline{W}_{-1}\left(2\log\left(2 + \log x\right) + 2 + \log\left(K(K+1)/2\right)\right) + 5K^2/2\right\} \,,$$

where the inequality is obtained by using Lemma 51, and $h_2$ is defined therein. Therefore, we obtain that, for all $n \geq D_\mu$,

$$\mathbb{E}_\nu[\mu_\star - \mu_{\hat{i}_n}] \leq \frac{K(K+1)}{2}e^2(2 + \log n)^2 \sum_{i \in [C_\mu - 1]} (\Delta_{i+1} - \Delta_i)p\left(\frac{n - 5K^2/2}{8H_i(\mu, \varepsilon_0)}\right) \,,$$

which concludes the second part of Theorem 6.

### F.2 Unverifiable sample complexity

The $(\varepsilon, \delta)$-*unverifiable sample complexity* is defined as the expectation of the smallest stopping time $\tilde{\tau}$ satisfying $\mathbb{P}(\forall t \geq \tilde{\tau}, \hat{i}_n \in \mathcal{I}_\varepsilon(\mu)) \geq 1 - \delta$. Theorem 7 gives an upper bound on the unverifiable expected sample complexity of the EB-TC$_{\varepsilon_0}$ algorithm with fixed proportions $\beta = 1/2$.

**Theorem 7.** *Let $\varepsilon_0 > 0$ and $\delta \in (0, 1)$. The EB-TC$_{\varepsilon_0}$ algorithm with fixed proportions $\beta = 1/2$ satisfies that, for all $\nu \in \mathcal{D}^K$ with mean $\mu$, for all $\varepsilon \geq 0$,*

$$\mathbb{P}_\nu\left(\forall n > U_{i_\mu(\varepsilon),\delta}(\mu, \varepsilon_0), \ \hat{i}_n \in \mathcal{I}_\varepsilon(\mu)\right) \geq 1 - \delta \,.$$

*The times $U_{i_\mu(\varepsilon),\delta}(\mu, \varepsilon_0)$ are the $(\varepsilon, \delta)$-unverifiable sample complexity of EB-TC$_{\varepsilon_0}$ defined as*

$$\forall i \in [C_\mu - 1], \quad U_{i,\delta}(\mu, \varepsilon_0) = h_2\left(\log(1/\delta), 8H_i(\mu, \varepsilon_0), 8H_i(\mu, \varepsilon_0)\log\left(K(K+1)/2\right) + 5K^2/2\right), \tag{22}$$

*with $(H_i(\mu, \varepsilon_0))_{i \in [C_\mu - 1]}$ defined in Theorem 6. The function $h_2(\log(1/\delta), A, B) = 2A\overline{W}_{-1}(\frac{1}{2}\log(1/\delta) + \frac{B}{2A} + \log(2A))$ satisfies that $h_2(\log(1/\delta), A, B) =_{\delta \to 0} A\log(1/\delta) + \mathcal{O}(\log\log(1/\delta))$.*

*Proof.* In Appendix F.1, we consider the concentration event $\tilde{\mathcal{E}}_{n,\delta}$ that involved tighter concentration results with thresholds $\tilde{f}_1(n, \delta)$ and $\tilde{f}_2(n, \delta)$. Let $n > K$ and $\delta \in (0, 1)$. It is direct to see that the same argument holds for the concentration events $\mathcal{E}_{n,\delta} = \mathcal{E}_{n,\delta}^1 \cap \mathcal{E}_{n,\delta}^2$ with $(\mathcal{E}_{n,\delta}^1)_{n>K}$ and $(\mathcal{E}_{n,\delta}^2)_{n>K}$ as in (24) and (25) for $s = 0$, i.e.

$$\mathcal{E}_{n,\delta}^1 = \left\{\forall k \in [K], \forall t \leq n, \ |\mu_{t,k} - \mu_k| < \sqrt{\frac{2f_1(n, \delta)}{N_{t,k}}}\right\} \,,$$

$$\mathcal{E}_{n,\delta}^2 = \left\{\forall (i, k) \in [K]^2 \text{ s.t. } i \neq k, \ \forall t \leq n, \ \frac{|(\mu_{t,i} - \mu_{t,k}) - (\mu_i - \mu_k)|}{\sqrt{1/N_{t,i} + 1/N_{t,k}}} < \sqrt{2f_2(n, \delta)}\right\} \,,$$

where $f_1(n, \delta) = \log(1/\delta) + \log n + \log\left(K(K+1)/2\right)$ and $f_2(n, \delta) = \log(1/\delta) + 2\log n + \log\left(K(K+1)/2\right)$, which satisfy $f_1(n, \delta) \leq f_2(n, \delta)$. Using Lemmas 40 and 41, we obtain that $\mathbb{P}_\nu(\mathcal{E}_{n,\delta}^\complement) \leq \delta$.

In the following, we use the notation on the gaps introduced in Appendix F.1, the complexities $(H_i(\mu, \varepsilon_0))_{i \in [C_\mu - 1]}$ defined in Theorem 6 and the function $i_\mu(\varepsilon)$ defined in (21). To prove Theorem 6, we obtain as an intermediary step that: for all $n > E_{\varepsilon_1, \varepsilon_0, \mu}(\delta)$, $\{\hat{i}_n \notin \mathcal{I}_{\varepsilon_1}(\mu)\} \subseteq \mathcal{E}_{n,\delta}^\complement$, where

$$E_{\varepsilon_1, \varepsilon_0, \mu}(\delta) := \sup\left\{n \mid n \leq 8\inf_{\varepsilon \in [0, \varepsilon_1]}\max\{\bar{H}_{\varepsilon, \varepsilon_1}(\mu, \varepsilon_0), \tilde{H}_{\varepsilon, \varepsilon_1}(\mu, \varepsilon_0)\}f_2(n, \delta) + 5K^2/2\right\}$$
$$= \sup\left\{n \mid n \leq 8H_{i_\mu(\varepsilon_1)}f_2(n, \delta) + 5K^2/2\right\} \,,$$

where the equality holds by using the rewriting of the complexities done in the end of Appendix F.1.

Let $A = 8H_{i_\mu(\varepsilon_1)}$ and $B = 8H_{i_\mu(\varepsilon_1)}\log\left(K(K+1)/2\right) + 5K^2/2$. Using a proof similar to Lemma 51, applying Lemma 49 yields that

$$n > E_{\varepsilon_1,\varepsilon_0,\mu}(\delta) \iff n > 2A\log n + A\log(1/\delta) + B$$

$$\iff \frac{n}{2A} - \log\left(\frac{n}{2A}\right) > \frac{1}{2}\log(1/\delta) + \frac{B}{2A} + \log(2A)$$

$$\iff n > 2A\overline{W}_{-1}\left(\frac{1}{2}\log(1/\delta) + \frac{B}{2A} + \log(2A)\right),$$

Let us define $h_2(\log(1/\delta), A, B) = 2A\overline{W}_{-1}\left(\frac{1}{2}\log(1/\delta) + \frac{B}{2A} + \log(2A)\right)$ and, for all $i \in [C_\mu - 1]$, we define

$$U_{i,\delta}(\mu,\varepsilon_0) = h_2\left(\log(1/\delta), 8H_i(\mu,\varepsilon_0), 8H_i(\mu,\varepsilon_0)\log\left(K(K+1)/2\right) + 5K^2/2\right).$$

Therefore, we have shown that, for all $\varepsilon_1 \geq 0$,

$$\{\exists n > U_{i_\mu(\varepsilon_1),\delta}(\mu,\varepsilon_0), \; \hat{\imath}_n \notin \mathcal{I}_{\varepsilon_1}(\mu)\} \subseteq \mathcal{E}_{n,\delta}^{\complement}$$

Using that $\mathbb{P}_\nu(\mathcal{E}_{n,\delta}^{\complement}) \leq \delta$, we can conclude that, for all $\varepsilon_1 \geq 0$,

$$\mathbb{P}_\nu\left(\forall n > U_{i_\mu(\varepsilon_1),\delta}(\mu,\varepsilon_0), \; \hat{\imath}_n \in \mathcal{I}_{\varepsilon_1}(\mu)\right) \geq 1 - \delta.$$

$\square$

### F.3 Cumulative regret of induced policy

Given a stream of recommendations $(\hat{\imath}_n)_{n>K}$, an external agent aiming at minimizing its cumulative regret can simply pull the arm $\hat{\imath}_n$ at time $n$. Since the EB-TCa algorithm has anytime guarantees on the simple regret of this recommendation, the regret of the induced policy is constant (Corollary 2).

**Corollary 2.** *Let $\varepsilon_0 > 0$. Let $(\hat{\imath}_n)_{n>K}$ be the recommendations of the EB-TC$_{\varepsilon_0}$ algorithm with fixed proportions $\beta = 1/2$. An agent pulling $\hat{\imath}_n$ at time $n$ has constant induced regret, i.e.*

$$\forall \nu \in \mathcal{D}^K, \; \forall T > K, \quad \sum_{n=K+1}^{T} \mathbb{E}_\nu[\mu_\star - \mu_{\hat{\imath}_n}] = \mathcal{O}\left(\frac{K^3\Delta_{\max}}{\min\{\Delta_{\min},\varepsilon_0\}^2}\left(\log\frac{K}{\min\{\Delta_{\min},\varepsilon_0\}^2}\right)^2\right).$$

*Proof.* Using that $H_1(\mu,\varepsilon_0) = \max_{i\in[C_\mu-1]} H_i(\mu,\varepsilon_0)$, a direct upper bound based on Corollary 1 yields that, for all $n \geq D_\mu$,

$$\mathbb{E}_\nu[\mu_\star - \mu_{\hat{\imath}_n}] \leq \frac{K(K+1)}{2}e^2(2+\log n)^2\Delta_{\max}\frac{n-5K^2/2}{8H_1(\mu,\varepsilon_0)}\exp\left(-\frac{n-5K^2/2}{8H_1(\mu,\varepsilon_0)}\right).$$

Let $c > 0$. Let $f(c,x) = (c+\log(x))^2 x\exp(-x)$ for all $x \geq 1$. Let $A > 0$, $B > 0$. By integration, for all $C \geq A + B$, we obtain

$$\sum_{n=C}^{T} f(c,(n-B)/A) \leq A\int_1^{+\infty} f(c,x)\mathrm{d}x = A\left(c^2 + 2cC_1 + C_2\right).$$

where $C_\alpha = \int_1^{+\infty}\log(x)^\alpha\exp(-x)\mathrm{d}x < +\infty$ for all $\alpha \geq 0$. For $n \geq 5K^2$, we have $n \leq 2(n-5K^2/2)$. Recall that $D_\mu = 8H_1(\mu,\varepsilon_0)h_2\left(8H_1(\mu,\varepsilon_0), 5K^2/2, 2+\log\left(K(K+1)/2\right)\right) + 5K^2/2$ with $h_2$ defined in Lemma 51. For all $n < 5K^2/2 + D_\mu$, we have trivially that $\mathbb{E}_\nu[\mu_\star - \mu_{\hat{\imath}_n}] \leq \Delta_{\max}$. Therefore, we obtain

$$\sum_{n=K+1}^{T}\mathbb{E}_\nu[\mu_\star - \mu_{\hat{\imath}_n}] \leq (5K^2/2 + D_\mu)\Delta_{\max} + \frac{K(K+1)}{2}e^2\Delta_{\max}$$

$$\sum_{n=5K^2+8H_1(\mu,\varepsilon_0)}^{T} f\left(2+\log\left(16H_1(\mu,\varepsilon_0)\right), \frac{n-5K^2/2}{8H_1(\mu,\varepsilon_0)}\right)$$

$$\leq (5K^2/2 + D_\mu)\Delta_{\max} + \frac{K(K+1)}{2}e^2\Delta_{\max}$$

$$8H_1(\mu,\varepsilon_0)\left(\left(2+\log\left(16H_1(\mu,\varepsilon_0)\right)\right)^2 + 2C_1\left(2+\log\left(16H_1(\mu,\varepsilon_0)\right)\right) + C_2\right).$$

Studying the function $h_2(x, y, z)$, it is direct to see that $h_2(x, y, z) =_{x \to +\infty} o(\log(x)^\alpha)$ for all $\alpha > 0$, hence $\Delta_{\max} D_\mu$ is dominated by the other term. Therefore, by studying the dominating term when $\min\{\Delta_{\min}, \varepsilon_0\} \to 0$, our upper bound scales as

$$\mathcal{O}\left(K^2 \Delta_{\max} H_1(\mu, \varepsilon_0)(\log H_1(\mu, \varepsilon_0))^2\right) = \mathcal{O}\left(\frac{K^3 \Delta_{\max}}{\min\{\Delta_{\min}, \varepsilon_0\}^2}\left(\log \frac{K}{\min\{\Delta_{\min}, \varepsilon_0\}^2}\right)^2\right),$$

where we used that $H_1(\mu, \varepsilon_0) = K(2\Delta_{\min}^{-1} + 3\varepsilon_0^{-1})^2 = \mathcal{O}\left(\frac{K}{\min\{\Delta_{\min}, \varepsilon_0\}^2}\right)$. $\qquad\square$

The idea of decoupling exploration and exploitation when minimizing the regret in the multi-armed bandits literature was introduced by [2]. At each round, the agent is allowed to choose one arm to explore and one arm to exploit at every round. While the agent suffers the loss from the exploited arm, it observes the one of the explored arm without cost. In the stochastic regime for instances having a unique best arm, the Decoupled-Tsallis-INF algorithm [33] achieves a constant regret $\mathcal{O}(K/\Delta_{\min})$. Therefore, in this setting, our constant regret in Corollary 2 does have the best achievable scaling in $K$ and $\Delta_{\min}$.

**Beyond synchronized policy**    When the external agent plays $\hat{\imath}_n$ at time $n$, it is said to be *synchronized* with the recommendation rule. In all generality, one could pull arm $\hat{\imath}_{\mathcal{L}(n)}$ at time $n$, where $\mathcal{L} : \mathbb{N} \to \mathbb{N}$ is a non-decreasing function, and obtain similar guarantees as Corollary 2. The agent can be slower than the recommendation rule (e.g. offline hyper-parameter optimization) hence $\mathcal{L}$ can be rapidly increasing, e.g. $\mathcal{L}(n) = Bn + K$ where $B > 1$. The recommendation rule can be slower than the agent (e.g. pulling the same arm multiple times) hence $\mathcal{L}$ can be piecewise constant with small steps, e.g. $\mathcal{L}(n) = \lceil n/B \rceil + K$. The communication between the agent and the recommendation rule can happen infrequently (e.g. paying to access the recommendation) hence $\mathcal{L}$ can be piecewise constant with large steps, e.g. $\mathcal{L}(n) = (\lceil n/B \rceil)^\alpha + K$ where $\alpha > 1$.

## G    Concentration results

The proof of Lemma 3 is given in Appendix G.1. Appendix G.2 gathers concentration results to control the empirical means and gaps.

### G.1    Proof of Lemma 3

Proving that a GLR stopping rule ensures the algorithm to be $(\varepsilon_1, \delta)$-PAC is done by leveraging concentration results. In particular, we build upon Theorem 9 of [27] which is restated below. While Theorem 9 was only stated for Gaussian distributions, it is direct to notice that the result also holds for sub-Gaussian distributions with variance proxy $\sigma^2 = 1$ (as the authors mentioned).

**Lemma 38** (Theorem 9 of [27]). *Consider a sub-Gaussian bandit $\nu$ with means $\mu \in \mathbb{R}^K$. Let $S \subseteq [K]$ and $x > 0$.*

$$\mathbb{P}_\nu\left[\exists n \in \mathbb{N}, \sum_{k \in S} \frac{N_{n,k}}{2}(\mu_{n,k} - \mu_k)^2 > \sum_{k \in S} 2\log\left(4 + \log(N_{n,k})\right) + |S|\mathcal{C}_G\left(\frac{x}{|S|}\right)\right] \le e^{-x}$$

*where $\mathcal{C}_G$ is defined in [27] by $\mathcal{C}_G(x) = \min_{\lambda \in ]1/2, 1]} \frac{g_G(\lambda) + x}{\lambda}$ and*

$$g_G(\lambda) = 2\lambda - 2\lambda \log(4\lambda) + \log \zeta(2\lambda) - \frac{1}{2}\log(1 - \lambda), \tag{23}$$

*where $\zeta$ is the Riemann $\zeta$ function and $\mathcal{C}_G(x) \approx x + \log(x)$.*

Since $\hat{\imath}_n = i^\star(\mu_n)$, standard manipulations yield that for all $i \ne \hat{\imath}_n$

$$\frac{(\mu_{n,\hat{\imath}_n} - \mu_{n,i} + \varepsilon_1)^2}{1/N_{n,\hat{\imath}_n} + 1/N_{n,i}} = \inf_{u \in \mathbb{R}}\left(N_{n,\hat{\imath}_n}(\mu_{n,\hat{\imath}_n} - u)^2 + N_{n,i}(\mu_{n,i} - u - \varepsilon_1)^2\right)$$

$$= \inf_{y \ge x + \varepsilon_1}\left(N_{n,\hat{\imath}_n}(\mu_{n,\hat{\imath}_n} - x)^2 + N_{n,i}(\mu_{n,i} - y)^2\right).$$

Let $i^\star = i^\star(\mu)$. Using the stopping rule (3) and the above manipulations, we obtain

$$\mathbb{P}_\nu\left(\tau_{\varepsilon_1,\delta} < +\infty, \hat{\imath}_{\tau_{\varepsilon_1,\delta}} \notin \mathcal{I}_{\varepsilon_1}(\mu)\right)$$

$$\leq \mathbb{P}_\nu\left(\exists n \in \mathbb{N}, \exists i \notin \mathcal{I}_{\varepsilon_1}(\mu), \ i = i^\star(\mu_n),\right.$$

$$\left. \min_{k \neq i} \inf_{y \geq x + \varepsilon_1} \left(N_{n,\hat{\imath}_n}(\mu_{n,\hat{\imath}_n} - x)^2 + N_{n,i}(\mu_{n,i} - y)^2\right) \geq 2c(n-1,\delta)\right)$$

$$\leq \mathbb{P}_\nu\left(\exists n \in \mathbb{N}, \exists i \notin \mathcal{I}_{\varepsilon_1}(\mu), \ i = i^\star(\mu_n),\right.$$

$$\left. \frac{N_{n,i}}{2}(\mu_{n,i} - \mu_i)^2 + \frac{N_{n,i^\star}}{2}(\mu_{n,i^\star} - \mu_{i^\star})^2 \geq c(n-1,\delta)\right)$$

$$\leq \sum_{i \notin \mathcal{I}_{\varepsilon_1}(\mu)} \mathbb{P}_\nu\left(\exists n \in \mathbb{N}, \ \frac{N_{n,i}}{2}(\mu_{n,i} - \mu_i)^2 + \frac{N_{n,i^\star}}{2}(\mu_{n,i^\star} - \mu_{i^\star})^2 \geq c(n-1,\delta)\right),$$

where the second inequality is obtained with $(k, x, y) = (i^\star, \mu_i, \mu_{i^\star})$ since $i \notin \mathcal{I}_{\varepsilon_1}(\mu)$, and the third by union bound. By concavity of $x \mapsto \log(4 + \log(x))$ and $N_{n,i^\star} + N_{n,i} \leq \sum_{k \in [K]} N_{n,k} = n - 1$, we obtain

$$\forall i \notin \mathcal{I}_{\varepsilon_1}(\mu), \quad \log(4 + \log N_{n,i^\star}) + \log(4 + \log N_{n,i}) \leq 2\log(4 + \log((n-1)/2))$$

Combining the above with Lemma 38 for all $i \notin \mathcal{I}_{\varepsilon_1}(\mu)$ and using that $|\mathcal{I}_{\varepsilon_1}(\mu)| \leq K - 1$, we obtain

$$\mathbb{P}_\nu\left(\tau_\delta < +\infty, \hat{\imath}_{\tau_\delta} \notin \mathcal{I}_{\varepsilon_1}(\mu)\right) \leq \sum_{i \notin \mathcal{I}_{\varepsilon_1}(\mu)} \frac{\delta}{K-1} \leq \delta.$$

## G.2 Sequence of concentration events

Lemma 39 is a standard concentration result for sub-Gaussian distribution, hence we omit the proof.

**Lemma 39.** *Let $X$ be an observation from a sub-Gaussian distribution with mean $0$ and variance proxy $\sigma^2$. Then, for all $\delta \in (0, 1]$,*

$$\mathbb{P}_X\left(|X| \geq \sigma\sqrt{2\log(1/\delta)}\right) \leq \delta.$$

Lemma 40 gives a sequence of concentration events under which the empirical means are close to their true values.

**Lemma 40.** *Let $\delta \in (0, 1]$ and $s \geq 0$. For all $n > K$, let $f_1(x, \delta) = \log(1/\delta) + (1 + s)\log x$ and*

$$\mathcal{E}_{n,\delta}^1 := \left\{\forall k \in [K], \forall t \leq n, \ |\mu_{t,k} - \mu_k| < \sqrt{\frac{2f_1(n,\delta)}{N_{t,k}}}\right\}. \tag{24}$$

*Then, for all $n > K$, $\mathbb{P}_\nu((\mathcal{E}_{n,\delta}^1)^\complement) \leq \frac{K\delta}{n^s}$.*

*Proof.* Let $(X_s)_{s \in [n]}$ be i.i.d. observations from one sub-Gaussian distribution with mean $0$ and variance proxy $\sigma^2 = 1$. Then, $\frac{1}{m}\sum_{i=1}^m X_i$ is sub-Gaussian with mean $0$ and variance proxy $\sigma^2 = 1/m$. By union bound over $[K]$ and over $m \in [n]$, we obtain

$$\mathbb{P}_\nu\left(\exists k \in [K], \exists t \leq n, \ |\mu_{t,k} - \mu_k| \geq \sqrt{\frac{2f_1(n,\delta)}{N_{t,k}}}\right)$$

$$\leq \sum_{k \in [K]} \sum_{m \in [n]} \mathbb{P}\left(\left|\frac{1}{m}\sum_{s \in [m]} X_s\right| \geq \sqrt{\frac{2f_1(n,\delta)}{m}}\right)$$

$$\leq \delta \sum_{k \in [K]} \sum_{m \in [n]} n^{-(1+s)} = K\delta n^{-s},$$

where we used that $\mu_{t,k} - \mu_k = \frac{1}{N_{t,k}}\sum_{s=1}^t \mathbb{1}(I_s = k) X_{s,k}$ and concentration results for sub-Gaussian observations (Lemma 39). $\quad\square$

Lemma 41 gives a sequence of concentration events under which the empirical gaps are close to their true values.

**Lemma 41.** *Let $\delta \in (0, 1]$ and $s \geq 0$. For all $n > K$, let $f_2(x, \delta) = \log(1/\delta) + (2 + s) \log(x)$ and*

$$\mathcal{E}_{n,\delta}^2 := \left\{ \forall (i,k) \in [K]^2 \text{ s.t. } i \neq k, \forall t \leq n, \frac{|(\mu_{t,i} - \mu_{t,k}) - (\mu_i - \mu_k)|}{\sqrt{1/N_{t,i} + 1/N_{t,k}}} < \sqrt{2f_2(n,\delta)} \right\}. \quad (25)$$

*Then, for all $n > K$, $\mathbb{P}_\nu((\mathcal{E}_{n,\delta}^2)^{\complement}) \leq \frac{K(K-1)\delta}{2n^s}$.*

*Proof.* Let $(X_s)_{s \in [n]}$ and $(Y_s)_{s \in [n]}$ be two streams of i.i.d. observations from two sub-Gaussian distributions with mean 0 and variance proxy $\sigma^2 = 1$. Then, $\frac{1}{m_1} \sum_{i=1}^{m_1} X_i - \frac{1}{m_2} \sum_{i=1}^{m_2} Y_i$ is sub-Gaussian with mean 0 and variance proxy $\sigma^2 = \frac{1}{m_1} + \frac{1}{m_2}$. By union bound, we obtain

$$\mathbb{P}_\nu \left( \exists (i,k) \in [K]^2 \text{ s.t. } i \neq k, \exists t \leq n, \left| \frac{(\mu_{t,i} - \mu_{t,k}) - (\mu_i - \mu_k)}{\sqrt{1/N_{t,i} + 1/N_{t,k}}} \right| \leq \sqrt{2f_2(n,\delta)} \right)$$

$$\leq \sum_{(i,k) \in [K]^2, i \neq k} \sum_{(m_1, m_2) \in [n]^2} \mathbb{P} \left( \left| \frac{1}{m_1} \sum_{i=1}^{m_1} X_i - \frac{1}{m_2} \sum_{i=1}^{m_2} Y_i \right| \leq -\sqrt{2f_2(n,\delta)(1/m_1 + 1/m_2)} \right)$$

$$\leq \delta \sum_{(i,k) \in [K]^2, i \neq k} \sum_{(m_1, m_2) \in [n]^2} n^{-(2+s)} = \frac{K(K-1)}{2} \delta n^{-s},$$

where we used that $(\mu_{t,i^\star} - \mu_{i^\star}) - (\mu_{t,k} - \mu_k) = \frac{1}{N_{t,i^\star}} \sum_{s=1}^t \mathbb{1}(I_s = i^\star) X_{s,i^\star} - \frac{1}{N_{t,k}} \sum_{s=1}^t \mathbb{1}(I_s = k) X_{s,k}$ and concentration results for sub-Gaussian observations (Lemma 39). $\square$

**Tighter concentration** Lemma 42 provides concentration results on the empirical means, which are tighter than the one obtained in Lemma 40.

**Lemma 42.** *Let $\delta \in (0, 1]$ and $s \geq 0$. Let $\overline{W}_{-1}$ defined in Lemma 49. For all $n > K$, let*

$$\tilde{f}_1(n, \delta) = \frac{1}{2} \overline{W}_{-1}(2 \log(1/\delta) + 2s \log n + 2 \log(2 + \log n) + 2), \quad (26)$$

*and*

$$\tilde{\mathcal{E}}_{n,\delta}^1 = \left\{ \forall k \in [K], \forall t \leq n, |\mu_{t,k} - \mu_k| < \sqrt{\frac{2\tilde{f}_1(n,\delta)}{N_{t,k}}} \right\}. \quad (27)$$

*Then, for all $n > K$, $\mathbb{P}_\nu((\tilde{\mathcal{E}}_{n,\delta}^1)^{\complement}) \leq \frac{K\delta}{n^s}$.*

*Proof.* Let $(X_s)_{s \in [n]}$ be i.i.d. observations from one sub-Gaussian distribution with mean 0 and variance proxy $\sigma^2 = 1$. Let $S_t = \sum_{s \in [t]} X_s$. To derived concentration result, we use peeling.

Let $\eta > 0$, $\gamma > 0$ and $D = \lceil \frac{\log(n)}{\log(1+\eta)} \rceil$. For all $i \in [D]$, let $N_i = (1 + \eta)^{i-1}$. For all $i \in [D]$, we define the family of priors $f_{N_i, \gamma}(x) = \sqrt{\frac{\gamma N_i}{2\pi}} \exp\left( -\frac{x^2 \gamma N_i}{2} \right)$ with weights $w_i = \frac{1}{D}$ and process

$$\overline{M}(t) = \sum_{i \in [D]} w_i \int f_{N_i, \gamma}(x) \exp\left( x S_t - \frac{1}{2} x^2 t \right) dx,$$

which satisfies $\overline{M}(0) = 1$. It is direct to see that $M(t) = \exp\left( x S_t - \frac{1}{2} x^2 t \right)$ is a non-negative supermartingale since sub-Gaussian distributions with mean 0 and variance proxy $\sigma^2 = 1$ satisfy

$$\forall \lambda \in \mathbb{R}, \quad \mathbb{E}_X[\exp(sX)] \leq \exp(\lambda^2/2).$$

By Tonelli's theorem, then $\overline{M}(t)$ is also a non-negative supermartingale of unit initial value.

Let $i \in [D]$ and consider $t \in [N_i, N_{i+1})$. For all $x$,

$$f_{N_i, \gamma}(x) \geq \sqrt{\frac{N_i}{t}} f_{t, \gamma}(x) \geq \frac{1}{\sqrt{1 + \eta}} f_{t, \gamma}(x)$$

Direct computations shows that

$$\int f_{t, \gamma}(x) \exp\left(x S_t - \frac{1}{2} x^2 t\right) dx = \frac{1}{\sqrt{1 + \gamma^{-1}}} \exp\left(\frac{S_t^2}{2(1 + \gamma)t}\right) \ .$$

Minoring $\overline{M}(t)$ by one of the positive term of its sum, we obtain

$$\overline{M}(t) \geq \frac{1}{D} \frac{1}{\sqrt{(1 + \gamma^{-1})(1 + \eta)}} \exp\left(\frac{S_t^2}{2(1 + \gamma)t}\right) \ ,$$

Using Ville's maximal inequality for non-negative supermartingale, we have that with probability greater than $1 - \delta$, $\log \overline{M}(t) \leq \log(1/\delta)$. Therefore, with probability greater than $1 - \delta$, for all $i \in [D]$ and $t \in [N_i, N_{i+1})$,

$$\frac{S_t^2}{t} \leq (1 + \gamma)\left(2 \log(1/\delta) + 2 \log D + \log(1 + \gamma^{-1}) + \log(1 + \eta)\right) \ .$$

Since this upper bound is independent of $t$, we can optimize it and choose $\gamma$ as in Lemma 43.

**Lemma 43** (Lemma A.3 in [9]). *For $a, b \geq 1$, the minimal value of $f(\eta) = (1 + \eta)(a + \log(b + \frac{1}{\eta}))$ is attained at $\eta^\star$ such that $f(\eta^\star) \leq 1 - b + \overline{W}_{-1}(a + b)$. If $b = 1$, then there is equality.*

Therefore, with probability greater than $1 - \delta$, for all $i \in [D]$ and $t \in [N_i, N_{i+1})$,

$$
\begin{aligned}
\frac{S_t^2}{t} &\leq \overline{W}_{-1}\left(1 + 2 \log(1/\delta) + 2 \log D + \log(1 + \eta)\right) \\
&\leq \overline{W}_{-1}\left(1 + 2 \log(1/\delta) + 2 \log\left(\log(1 + \eta) + \log n\right) - 2 \log\log(1 + \eta) + \log(1 + \eta)\right) \\
&= \overline{W}_{-1}\left(2 \log(1/\delta) + 2 \log(2 + \log n) + 3 - 2 \log 2\right)
\end{aligned}
$$

The second inequality is obtained since $D \leq 1 + \frac{\log n}{\log(1 + \eta)}$. The last equality is obtained for the choice $\eta^\star = e^2 - 1$, which minimizes $\eta \mapsto \log(1 + \eta) - 2 \log(\log(1 + \eta))$. Since $[n] \subseteq \bigcup_{i \in [D]} [N_i, N_{i+1})$ and $N_{t,k}(\mu_{t,k} - \mu_k) = \sum_{s \in [N_{t,k}]} X_{s,k}$ (unit-variance), this yields

$$\mathbb{P}\left(\exists m \leq n, \left|\frac{1}{m} \sum_{s=1}^{m} X_s\right| \geq \sqrt{\frac{1}{m} \overline{W}_{-1}\left(2 \log(1/\delta) + 2 \log(2 + \log(n)) + 3 - 2 \log 2\right)}\right) \leq \delta \ .$$

Since $3 - 2 \log 2 \leq 2$ and $\overline{W}_{-1}$ is increasing, taking $\delta n^{-s}$ yields

$$\mathbb{P}_\nu\left(\exists t \leq n, \sqrt{N_{t,k}} |\mu_{t,k} - \mu_k| \geq \sqrt{2 \tilde{f}_1(n, \delta)}\right) \leq \delta n^{-s} \ .$$

Doing a union bound over arms yields the result. $\qquad\square$

Lemma 44 provides concentration results on the empirical gaps, which are tighter than the ones obtained in Lemma 41.

**Lemma 44.** *Let $\delta \in (0, 1]$ and $s \geq 0$. Let $\overline{W}_{-1}$ defined in Lemma 49. For all $n > K$, let*

$$\tilde{f}_2(n, \delta) = \overline{W}_{-1}\left(\log(1/\delta) + s \log n + 2 \log(2 + \log n) + 2\right) \ , \tag{28}$$

*and*

$$\tilde{\mathcal{E}}_{n,\delta}^2 := \left\{ \forall (i, k) \in [K]^2 \text{ s.t. } i \neq k, \ \forall t \leq n, \ \frac{|(\mu_{t,i} - \mu_{t,k}) - (\mu_i - \mu_k)|}{\sqrt{1/N_{t,i} + 1/N_{t,k}}} < \sqrt{2 \tilde{f}_2(n, \delta)} \right\} \ . \tag{29}$$

*Then, for all $n > K$, $\mathbb{P}_\nu\left((\tilde{\mathcal{E}}_{n,\delta}^2)^\complement\right) \leq \frac{K(K-1)}{2} \frac{\delta}{n^s}$.*

*Proof.* Without loss of generality, we show the result for arms $(i,k) = (1,2)$. Let $(X_s)_{s\in[n]}$ and $(Y_s)_{s\in[n]}$ be i.i.d. observations from two sub-Gaussian distribution with mean $0$ and variance proxy $\sigma^2 = 1$. Let $S_{t,k} = \sum_{s\in[t]} \mathbb{1}\,(I_s = k)\,X_s$ for all $k \in \{1,2\}$. To derived concentration result, we use peeling as in Lemma 42.

Let $\eta > 0$, $\gamma > 0$ and $D = \lceil \frac{\log(n)}{\log(1+\eta)} \rceil$. For all $i \in [D]$, let $N_i = (1+\eta)^{i-1}$. For all $i \in [D]$, we define the family of priors $f_{N_i,\gamma}(x) = \sqrt{\frac{\gamma N_i}{2\pi}} \exp\left(-\frac{x^2 \gamma N_i}{2}\right)$ with weights $w_i = \frac{1}{D}$ and process

$$\forall k \in \{1,2\}, \quad \overline{M}_k(t) = \sum_{i\in[D]} w_i \int f_{N_i,\gamma}(x) \exp\left(xS_{t,k} - \frac{1}{2}x^2 N_{t,k}\right)\, dx\,,$$

which satisfies $\overline{M}_k(0) = 1$ for all $k \in \{1,2\}$. As in the proof of Lemma 42, we obtain that $\overline{M}_1(t)$ and $\overline{M}_2(t)$ are non-negative supermartingale such that

$$\forall k \in \{1,2\}, \quad \overline{M}_k(t) \geq \frac{1}{D} \frac{1}{\sqrt{(1+\gamma^{-1})(1+\eta)}} \exp\left(\frac{S_{t,k}^2}{2(1+\gamma)N_{t,k}}\right)\,.$$

where we used $(i_1, i_2) \in [D]^2$ and consider $N_{t,k} \in [N_{i_k}, N_{i_k+1})$ for all $k \in \{1,2\}$. Let us define $\overline{M}(t) = \overline{M}_1(t_1)\overline{M}_2(t_2)$. Then, we have that $\overline{M}(t)$ is a non-negative supermartingale such that

$$\overline{M}(t) \geq \frac{1}{D^2} \frac{1}{(1+\gamma^{-1})(1+\eta)} \exp\left(\frac{1}{2(1+\gamma)} \sum_{k\in\{1,2\}} \frac{S_{t,k}^2}{N_{t,k}}\right)$$

Using Ville's maximal inequality for non-negative supermartingale, we have that with probability greater than $1-\delta$, $\log \overline{M}(t) \leq \log(1/\delta)$. Therefore, with probability greater than $1-\delta$, for all $(i_1, i_2) \in [D]^2$ and $N_{t,k} \in [N_{i_k}, N_{i_k+1})$ for all $k \in \{1,2\}$,

$$\sum_{k\in\{1,2\}} \frac{S_{t,k}^2}{N_{t,k}} \leq 2(1+\gamma)\left(\log(1/\delta) + 2\log D + \log(1+\gamma^{-1}) + \log(1+\eta)\right)\,.$$

Since this upper bound is independent of $t$, we can optimize it and choose $\gamma$ as in Lemma 43.

Therefore, with probability greater than $1-\delta$, for all $(i_1, i_2) \in [D]^2$ and $N_{t,k} \in [N_{i_k}, N_{i_k+1})$ for all $k \in \{1,2\}$,

$$\sum_{k\in\{1,2\}} \frac{S_{t,k}^2}{N_{t,k}} \leq 2\overline{W}_{-1}\left(\log(1/\delta) + 2\log D + \log(1+\eta) + 1\right)$$

$$\leq 2\overline{W}_{-1}\left(\log(1/\delta) + 2\log(2 + \log n) + 2\right)$$

The second inequality is obtained as in Lemma 42 by using that $D \leq 1 + \frac{\log n}{\log(1+\eta)}$, $\overline{W}_{-1}$ increasing, taking $\eta^\star = e^2 - 1$ and using $3 - 2\log 2 \leq 2$.

Since $[n] \subseteq \bigcup_{i\in[D]}[N_i, N_{i+1})$, $N_{t,k} \leq n$ and $N_{t,k}(\mu_{t,k} - \mu_k) = S_{t,k}$ (unit-variance), this yields

$$\mathbb{P}\left(\exists t \leq n, \sum_{k\in\{1,2\}} \frac{N_{t,k}}{2}(\mu_{t,k} - \mu_k)^2 \geq \overline{W}_{-1}\left(\log(1/\delta) + 2\log(2 + \log n) + 2\right)\right) \leq \delta\,.$$

Let $C(n,\delta) := \overline{W}_{-1}\left(\log(1/\delta) + 2\log(2 + \log n) + 2\right)$. In the following, we consider that we are under the event,

$$\mathcal{E}_{n,\delta}(1,2) := \left\{\forall t \leq n, \sum_{k\in\{1,2\}} \frac{N_{t,k}}{2}(\mu_{t,k} - \mu_k)^2 < C(n,\delta)\right\}\,,$$

which has probability at least $1-\delta$. Since it satisfies the above constraint, the quantity $\mu_{t,1} - \mu_1 - (\mu_{t,2} - \mu_2)$ is upper bounded by

$$\max_{x\in\mathbb{R}^2} \{\mu_{t,1} - \mu_{t,2} - x_1 + x_2\} \quad \text{subject to} \quad \sum_{k\in\{1,2\}} \frac{N_{t,1}}{2}(\mu_{t,k} - x_k)^2 \leq C(n,\delta)\,.$$

Introducing a Lagrange multiplier $\alpha$, the above optimization problem is equivalent to

$$\min_{\alpha \geq 0} \max_{x \in \mathbb{R}^2} \left\{ \mu_{t,1} - \mu_{t,2} - x_1 + x_2 + \alpha \left( C(n,\delta) - \sum_{k \in \{1,2\}} \frac{N_{t,k}}{2} (\mu_{t,k} - x_k)^2 \right) \right\} .$$

Solving for $x$ by cancelling the derivative yields that $x_1 = \mu_{t,1} - \frac{1}{\alpha N_{t,1}}$ and $x_2 = \mu_{t,2} + \frac{1}{\alpha N_{t,2}}$. Therefore, we obtain as solution

$$\min_{\alpha \geq 0} \left\{ \alpha C(n,\delta) + \frac{1}{2\alpha} \sum_{k \in \{1,2\}} \frac{1}{N_{t,k}} \right\} = \sqrt{2 C(n,\delta) \sum_{k \in \{1,2\}} \frac{1}{N_{t,k}}} ,$$

where the last equality is obtained by solving the optimization since the derivative is null at $\alpha^\star = \sqrt{\sum_{k \in \{1,2\}} \frac{1}{2C(n,\delta)N_{t,k}}}$. By symmetry, we obtain the same result to upper bound $\mu_{t,2} - \mu_2 - (\mu_{t,1} - \mu_1)$. Therefore, under $\mathcal{E}_{n,\delta}(1,2)$, we have shown that

$$\frac{|\mu_{t,1} - \mu_1 - (\mu_{t,2} - \mu_2)|}{1/N_{t,1} + 1/N_{t,2}} \leq \sqrt{2\overline{W}_{-1} \left( \log\left(1/\delta\right) + 2\log\left(2 + \log n\right) + 2 \right)} .$$

The same argument as above can be applied for all $(i,k) \in [K]^2$ such that $i \neq k$. A direct union bound yields

$$\mathbb{P}_\nu((\tilde{\mathcal{E}}_{n,\delta}^2)^{\complement}) \leq \sum_{(i,k) \in [K]^2, i \neq k} \mathbb{P}_\nu \left( \mathcal{E}_{n,\delta n^{-s}}(i,k)^{\complement} \right) \leq \frac{K(K-1)}{2} \frac{\delta}{n^s} ,$$

where we used the inclusion proven above for $(n, \delta/n^s)$. This concludes the proof. $\qquad \square$

**Global events** Lemma 45 combines the above sequences of concentration events.

**Lemma 45.** *Let $s > 1$. Let $(\mathcal{E}_{n,\delta}^1)_{n>K}$ and $(\mathcal{E}_{n,\delta}^2)_{n>K}$ as in (24) and (25). Let $(\tilde{\mathcal{E}}_{n,\delta}^1)_{n>K}$ and $(\tilde{\mathcal{E}}_{n,\delta}^2)_{n>K}$ as in (27) and (29). Let us define $\mathcal{E}_{n,\delta} := \mathcal{E}_{n,\delta}^1 \cap \mathcal{E}_{n,\delta}^2$ and $\tilde{\mathcal{E}}_{n,\delta} := \tilde{\mathcal{E}}_{n,\delta}^1 \cap \tilde{\mathcal{E}}_{n,\delta}^2$ for all $n > K$. Then,*

$$\sum_{n>K} \mathbb{P}_\nu(\mathcal{E}_{n,\delta}^{\complement}) \leq \frac{K(K+1)}{2}\zeta(s)\delta \quad and \quad \sum_{n>K} \mathbb{P}_\nu(\tilde{\mathcal{E}}_{n,\delta}^{\complement}) \leq \frac{K(K+1)}{2}\zeta(s)\delta .$$

*Proof.* It is direct to see that

$$\sum_{n>K} \mathbb{P}_\nu(\mathcal{E}_{n,\delta}^{\complement}) \leq \sum_{n>K} \frac{K\delta}{n^s} + \sum_{n>K} \frac{K(K-1)\delta}{2n^s} = \frac{K(K+1)}{2}\zeta(s)\delta .$$

The same proof can be applied to upper bound $\sum_{n>K} \mathbb{P}_\nu(\tilde{\mathcal{E}}_{n,\delta}^{\complement})$. $\qquad \square$

# H   Technicalities

Appendix H gathers existing and new technical results which are used for our proofs.

**Key technical result** Lemma 4 is the key technical ingredient on which our proofs rely on. It builds on a sequence of "bad" events such that, under each "bad" event, either the leader or the challenger was not often selected as leader or challenger, and shows that the number of times those "bad" events occur is small.

**Lemma** (Lemma 4). *Let $\delta \in (0,1]$ and $n > K$. Let $(A_{t,\delta}(n,\delta))_{n \geq t > K}$ be a sequence of events and $(D_i(n,\delta))_{i \in [K]}$ be positive thresholds satisfying that, for all $t \in [n] \setminus [K]$, under the event $A_{t,\delta}(n,\delta)$,*

$$\exists i_t \in [K], \quad T_t(i_t) \leq D_{i_t}(n,\delta) \quad and \quad T_{t+1}(i_t) = T_t(i_t) + 1 .$$

*Then, we have $\sum_{t=K+1}^n \mathbb{1}\left(A_{t,\delta}(n,\delta)\right) \leq \sum_{i \in [K]} D_i(n,\delta)$.*

*Proof.* Using the inclusion of events given by the assumption on $(A_{t,\delta}(n,\delta))_{n \geq t > K}$, we obtain

$$\sum_{t=K+1}^{n} \mathbb{1}\left(A_{t,\delta}(n,\delta)\right) \leq \sum_{t=K+1}^{n} \mathbb{1}\left(\exists k_t \in [K], \, T_t(k_t) \leq D_{k_t}(n,\delta), \, T_{t+1}(k_t) = T_t(k_t) + 1\right)$$

$$\leq \sum_{i \in [K]} \sum_{t=K+1}^{n} \mathbb{1}\left(T_t(i) \leq D_i(n,\delta), \, T_{t+1}(i) = T_t(i) + 1\right) \leq \sum_{i \in [K]} D_i(n,\delta).$$

The second inequality is obtained by union bound. The third inequality is direct since the number of times one can increase by one a quantity that is positive and bounded by $D_i(n,\delta)$ is at most $D_i(n,\delta)$. $\qquad\square$

**Tracking** Lemma 46 provide general results for the $K(K-1)$ tracking procedures both for IDS and fixed proportions, which includes the ones of Lemma 2. The main theoretical argument behind those results is based on applying Theorem 6 in [12].

**Lemma 46.** *For all $n > K$, $i \in [K]$, $j \neq i$, we have*

$$-1/2 \leq N_{n,j}^i - (1 - \bar{\beta}_n(i,j))T_n(i,j) \leq 1 \quad i.e. \quad -1 \leq (T_n(i,j) - N_{n,j}^i) - \bar{\beta}_n(i,j)T_n(i,j) \leq 1/2$$

*and*

$$\frac{N_{n,i}^i - (K-1)/2}{\max_{j \neq i} \bar{\beta}_n(i,j)} \leq \sum_{j \neq i} T_n(i,j) \leq \frac{N_{n,i}^i + K - 1}{\min_{j \neq i} \bar{\beta}_n(i,j)}.$$

*Let $\beta \in (0,1)$. For fixed proportions $\beta$, we have*

$$N_{n,i} \geq \min\{\beta, 1-\beta\}\left(T_n(i) - \frac{3(K-1)}{2}\right).$$

*Proof.* We have $K(K-1)$ independent two-arms C-Tracking between the challenger and the leader. Theorem 6 in [12] yields the result. The second inequality is a simple re-ordering.

To obtain the second part, direct manipulations yield that

$$\sum_{j \neq i} T_n(i,j) \leq \sum_{j \neq i} \frac{T_n(i,j) - N_{n,j}^i + 1}{\bar{\beta}_n(i,j)} \leq \frac{\sum_{j \neq i}(T_n(i,j) - N_{n,j}^i) + K - 1}{\min_{j \neq i} \bar{\beta}_n(i,j)},$$

which allows to conclude since $N_{n,i}^i = \sum_{j \neq i}(T_n(i,j) - N_{n,j}^i)$. The lower bound is obtained similarly.

The third is a direct consequence of the first part since

$$N_{n,i} = \sum_{j \neq i}(T_n(i,j) - N_{n,j}^i) + \sum_{j \neq i} N_{n,i}^j \leq \beta \sum_{j \neq i} T_n(i,j) + (1-\beta)\sum_{j \neq i} T_n(j,i) + 3(K-1)/2$$

$$\geq \beta \sum_{j \neq i} T_n(i,j) + (1-\beta)\sum_{j \neq i} T_n(j,i) - 3(K-1)/2.$$

Having shown

$$\max_{i \in [K]} \left| N_{n,i} - \left(\beta \sum_{j \neq i} T_n(i,j) + (1-\beta)\sum_{j \neq i} T_n(j,i)\right)\right| \leq 3(K-1)/2,$$

it is direct to conclude. $\qquad\square$

**Methodology** Lemma 47 is a standard result to upper bound the expected sample complexity of an algorithm, e.g. see Lemma 1 in [11]. This is a key method extensively used in the literature.

**Lemma 47.** *Let $(\mathcal{E}_n)_{n > K}$ be a sequence of events and $T(\delta) > K$ be such that for $n \geq T(\delta)$, $\mathcal{E}_n \subseteq \{\tau_\delta \leq n\}$. Then, $\mathbb{E}_\nu[\tau_\delta] \leq T(\delta) + \sum_{n > K} \mathbb{P}_\nu(\mathcal{E}_n^{\complement})$.*

*Proof.* Since the random variable $\tau_\delta$ is positive and $\{\tau_\delta > n\} \subseteq \mathcal{E}_n^{\complement}$ for all $n \geq T(\delta)$, we have

$$\mathbb{E}_\nu[\tau_\delta] = \sum_{n \geq 0} \mathbb{P}_\nu(\tau_\delta > n) \leq T(\delta) + \sum_{n \geq T(\delta)} \mathbb{P}_\nu(\mathcal{E}_n^{\complement}),$$

which concludes the proof by adding positive terms. $\square$

Lemma 48 is a key method to upper bound the probability of error of an algorithm.

**Lemma 48.** *Let $\varepsilon \geq 0$ and $(\mathcal{E}_{n,\delta})_{n > K, \delta \in (0,1)}$ be a sequence of events such that $\mathbb{P}_\nu(\mathcal{E}_{n,\delta}^{\complement}) \leq C\delta$ with $C > 0$. Suppose that $T_\varepsilon(\delta) > K$ is such that for $n > T_\varepsilon(\delta)$, $\mathcal{E}_{n,\delta} \subseteq \{\hat{\imath}_n \in \mathcal{I}_\varepsilon(\mu)\}$. Then,*

$$\mathbb{P}_\nu(\hat{\imath}_n \notin \mathcal{I}_\varepsilon(\mu)) \leq C \inf\{\delta \mid n > T_\varepsilon(\delta)\}.$$

*Proof.* Using the assumption, we have $\mathbb{P}_\nu(\hat{\imath}_n \notin \mathcal{I}_\varepsilon(\mu)) \leq \mathbb{P}_\nu(\mathcal{E}_{n,\delta}^{\complement}) \leq C\delta$ for all $n > T_\varepsilon(\delta)$. Taking the infimum yields the result. $\square$

**Inversion results**  Lemma 49 gathers properties on the function $\overline{W}_{-1}$, which is used in the literature to obtain concentration results.

**Lemma 49** ([22])**.** *Let $\overline{W}_{-1}(x) = -W_{-1}(-e^{-x})$ for all $x \geq 1$, where $W_{-1}$ is the negative branch of the Lambert W function. The function $\overline{W}_{-1}$ is increasing on $(1, +\infty)$ and strictly concave on $(1, +\infty)$. In particular, $\overline{W}'_{-1}(x) = \left(1 - \frac{1}{\overline{W}_{-1}(x)}\right)^{-1}$ for all $x > 1$. Then, for all $y \geq 1$ and $x \geq 1$,*

$$\overline{W}_{-1}(y) \leq x \quad \Longleftrightarrow \quad y \leq x - \log(x).$$

*Moreover, for all $x > 1$,*

$$x + \log(x) \leq \overline{W}_{-1}(x) \leq x + \log(x) + \min\left\{\frac{1}{2}, \frac{1}{\sqrt{x}}\right\}.$$

Lemma 50 provides an ordering on the thresholds for tighter concentration results. It leverages properties of $\overline{W}_{-1}$.

**Lemma 50.** *Let $s \geq 0$. Let $\overline{W}_{-1}$ defined in Lemma 49. For all $\delta \in (0, 1]$ and $n \geq 1$, let us denote by*

$$\tilde{f}_1(n, \delta) = \frac{1}{2}\overline{W}_{-1}(2\log(1/\delta) + 2s\log n + 2\log(2 + \log n) + 2),$$

$$\tilde{f}_2(n, \delta) = \overline{W}_{-1}\left(\log(1/\delta) + s\log n + 2\log(2 + \log n) + 2\right).$$

*Then, we have $\tilde{f}_1(n, \delta) \leq \tilde{f}_2(n, \delta)$.*

*Proof.* Let $a > 0$ and $b \geq 1$. Let us define $f(x) = \frac{1}{x}\overline{W}_{-1}(ax + b)$. Using Lemma 49, we have

$$f'(x) = -\frac{1}{x^2}\left(\overline{W}_{-1}(ax + b) - \frac{ax}{1 - \overline{W}_{-1}(ax + b)^{-1}}\right).$$

Therefore, we obtain $f'(x) \leq 0$ if and only if $\overline{W}_{-1}(ax + b) \geq ax + 1$. Since $\overline{W}_{-1}(ax + b) \geq ax + b$ and $b \geq 1$, the function $f(x)$ is decreasing for all $x \geq 1$. Therefore, $\frac{1}{2}\overline{W}_{-1}(2x + b) \leq \overline{W}_{-1}(a + b)$. Applying this result with $a = \log(1/\delta) + s\log n$ and $b = 2\log(2 + \log n) + 2 \geq 1$ for all $n \geq 1$, we obtain $\tilde{f}_1(n, \delta) \leq \tilde{f}_2(n, \delta)$. $\square$

Lemma 51 is an inversion result to upper bound a time which is implicitly defined. It is a direct consequence of Lemma 49.

**Lemma 51.** *Let $\overline{W}_{-1}$ defined in Lemma 49. Let $A > 0$, $B > 0$ such that $B/A + \log A > 1$ and*

$$C(A, B) = \sup\{x \mid x < A\log x + B\},$$

$$C(A, B, D) = \sup\{x \mid x < A\overline{W}_{-1}(2\log(2 + \log x) + D) + B\}.$$

*Then, $C(A, B) < h_1(A, B)$ with $h_1(z, y) = z\overline{W}_{-1}(y/z + \log z)$ and $C(A, B) < Ah_2(A, B, D) + B$ where*

$$h_2(x, y, z) = \inf\{u \mid u - \log u - 2\log(2 + \log(xu + y)) \geq z\}.$$

*Proof.* Since $B/A + \log A > 1$, we have $C(A, B) \geq A$, hence

$$C(A, B) = \sup\{x \mid x < A\log(x) + B\} = \sup\{x \geq A \mid x < A\log(x) + B\}.$$

Using Lemma 49 yields that

$$x \geq A\log x + B \iff \frac{x}{A} - \log\left(\frac{x}{A}\right) \geq \frac{B}{A} + \log A \iff x \geq A\overline{W}_{-1}\left(\frac{B}{A} + \log A\right).$$

By changing the variable, we obtain

$$C(A, B, D) = B + A\sup\left\{y \mid y < \overline{W}_{-1}\left(2\log\left(2 + \log(Ay + B)\right) + D\right)\right\}$$

Likewise, Lemma 49 yields that

$$y \geq \overline{W}_{-1}\left(2\log\left(2 + \log(Ay + B)\right) + D\right) \iff y - \log y - 2\log\left(2 + \log(Ay + B)\right) \geq D.$$

$\square$

Lemma 52 is an inversion result to upper bound a probability which is implicitly defined based on times that are implicitly defined.

**Lemma 52.** *Let $\overline{W}_{-1}$ defined in Lemma 49. Let $A > 0$, $B > 0$, $C > 0$, $E > 0$, $\alpha > 0$, $\beta > 0$ and*

$$D_{A,B,C}(\delta) = \sup\{x \mid x \leq A(\log(1/\delta) + C\log x) + B\},$$

$$D_{A,B,C,E,\alpha,\beta}(\delta) = \sup\left\{x \mid x \leq \frac{A}{\alpha}\overline{W}_{-1}\left(\alpha\left(\log(1/\delta) + C\log(\beta + \log x) + E\right)\right) + B\right\}.$$

*Then,*

$$\inf\{\delta \mid x > D_{A,B,C}(\delta)\} \leq x^C \exp\left(-\frac{x - B}{A}\right),$$

$$\inf\{\delta \mid x > D_{A,B,C,E,\alpha,\beta}(\delta)\} \leq e^E\left(\alpha\frac{x - B}{A}\right)^{1/\alpha}(\beta + \log x)^C \exp\left(-\frac{x - B}{A}\right).$$

*Proof.* Direct manipulations yield that

$$x > D_{A,B,C}(\delta) \iff x > A(\log(1/\delta) + C\log x) + B \iff \delta < x^C \exp\left(-\frac{x - B}{A}\right).$$

Likewise, using Lemma 49, we obtain

$$x > D_{A,B,C,E,\alpha,\beta}(\delta) \iff \alpha\frac{x - B}{A} > \overline{W}_{-1}\left(\alpha\left(\log(1/\delta) + C\log(\beta + \log x) + E\right)\right)$$

$$\iff \frac{x - B}{A} - \frac{1}{\alpha}\log\left(\alpha\frac{x - B}{A}\right) > \log(1/\delta) + C\log(\beta + \log x) + E$$

$$\iff \delta < e^E\left(\alpha\frac{x - B}{A}\right)^{1/\alpha}(\beta + \log x)^C \exp\left(-\frac{x - B}{A}\right).$$

$\square$

# I   Multiplicative setting

Let $\nu \in \mathcal{D}^K$ with mean parameters $\mu \in (\mathbb{R}_+^\star)^K$. Compared to the additive setting, we only consider strictly positive mean parameters. In many applications, this assumption is natural. In other applications, there is a known lower bound on the value of the true mean parameters, hence we can simply translate the problem by adding this lower bound. We also note that the problem of identifying which arms are above the threshold 0 is the thresholding bandit problem, which has been extensively studied in the literature. Overall, while this assumption might seem restrictive, we think it is rather mild. Since the means are strictly positive, the multiplicative error $\varepsilon$ has also to be positive and strictly smaller than 1. Given a multiplicative error $\varepsilon \in [0, 1)$, the set of multiplicative $\varepsilon$-good arms are denoted by $\mathcal{I}_\varepsilon^{\mathrm{mul}}(\mu) := \{i \in [K] \mid \Delta_i \leq \mu_\star\varepsilon\}$. We will refer to this problem as multiplicative $\varepsilon$-BAI.

First, we discuss the characteristic times involved in the fixed-confidence multiplicative $\varepsilon$-BAI (Appendix I.1). Second, for $\varepsilon_0 \in (0, 1)$, we present the EB-TC$_{\varepsilon_0}^{\mathrm{m}}$ with IDS or fixed $\beta$ proportions (Appendix I.2). Finally, we prove that EB-TC$_{\varepsilon_0}^{\mathrm{m}}$ is asymptotically (resp. $\beta$-)optimal when using IDS (resp. fixed $\beta$) when combined with the appropriate GLR stopping rule (Appendix I.3).

## I.1 Characteristic times

Let $\nu \in \mathcal{D}^K$ with mean parameters $\mu \in (\mathbb{R}_+^\star)^K$. We assume in the following that there exists a unique best arm, i.e. $i^\star(\mu) = \{i^\star\}$. Let $\varepsilon \in [0,1)$. The $(\beta\text{-})$characteristic times for the fixed confidence multiplicative $\varepsilon$-BAI setting with Gaussian bandits $\mathcal{N}(\mu,1)$ are defined as $T_\varepsilon^{\mathrm{mul}}(\mu) = \min_{i \in \mathcal{I}_\varepsilon^{\mathrm{mul}}(\mu), \beta \in (0,1)} T_{\varepsilon,\beta}^{\mathrm{mul}}(\mu,i)$ and $T_{\varepsilon,\beta}^{\mathrm{mul}}(\mu) = \min_{i \in \mathcal{I}_\varepsilon^{\mathrm{mul}}(\mu)} T_{\varepsilon,\beta}^{\mathrm{mul}}(\mu,i)$ where

$$2T_{\varepsilon,\beta}^{\mathrm{mul}}(\mu)^{-1}(\mu,i) = \max_{w \in \triangle_K, w_i = \beta} \min_{j \neq i} \frac{(\mu_i - (1-\varepsilon)\mu_j)^2}{1/\beta + (1-\varepsilon)^2/w_j}, \tag{30}$$

We denote by $w_\varepsilon^{\mathrm{mul}}(\mu,i)$ and $w_{\varepsilon,\beta}^{\mathrm{mul}}(\mu,i)$ their maximizer, which we refer to as the $(\beta\text{-})$optimal allocation for multiplicative $\varepsilon$-BAI.

Even though the multiplicative $\varepsilon$-BAI is less studied, it is direct to see that it has similar properties as $T_0(\mu)$. Due to the factor $(1-\varepsilon)^2$ in the denominator, it is not possible to simply use a reduction to a BAI problem as we did for the additive $\varepsilon$-BAI problem (see Lemma 9). Since the properties listed in Appendix C are obtained by studying the optimization problem defining $T_0(\mu)$, we will have the same properties for $T_\varepsilon^{\mathrm{mul}}(\mu)$ by accounting for the extra multiplicative factor $(1-\varepsilon)^2$. We list here the properties on which our proof relies:

- $T_\varepsilon^{\mathrm{mul}}(\mu) = \min_{\beta \in (0,1)} T_{\varepsilon,\beta}^{\mathrm{mul}}(\mu,i^\star)$ and $T_{\varepsilon,\beta}^{\mathrm{mul}}(\mu) = T_{\varepsilon,\beta}^{\mathrm{mul}}(\mu,i^\star)$, meaning the unique best arm corresponds to the arm which is the easiest to verify multiplicative $\varepsilon$-BAI.

- $w_\varepsilon^{\mathrm{mul}}(\mu) = \{w_\varepsilon^{\mathrm{mul}}\}$ and $w_{\varepsilon,\beta}^{\mathrm{mul}}(\mu) = \{w_{\varepsilon,\beta}^{\mathrm{mul}}\}$, meaning there is a unique $(\beta\text{-})$optimal allocation which satisfies $\min_{i \in [K]} (w_\varepsilon^{\mathrm{mul}})_i > 0$ and $\min_{i \in [K]} (w_{\varepsilon,\beta}^{\mathrm{mul}})_i > 0$.

- There is equality at the equilibrium of the transportation costs, meaning for all $i \neq i^\star$

$$\frac{(\mu_{i^\star} - (1-\varepsilon)\mu_i)^2}{(w_\varepsilon^{\mathrm{mul}})_{i^\star}^{-1} + (1-\varepsilon)^2 (w_\varepsilon^{\mathrm{mul}})_i^{-1}} = 2T_\varepsilon^{\mathrm{mul}}(\mu)^{-1} \text{ and } \frac{(\mu_{i^\star} - (1-\varepsilon)\mu_i)^2}{\beta^{-1} + (1-\varepsilon)^2 (w_{\varepsilon,\beta}^{\mathrm{mul}})_i^{-1}} = 2T_{\varepsilon,\beta}^{\mathrm{mul}}(\mu)^{-1}, \tag{31}$$

and the multiplicative overall balance equation is satisfied

$$\sum_{i \neq i^\star} \left( \frac{(w_\varepsilon^{\mathrm{mul}})_i}{(w_\varepsilon^{\mathrm{mul}})_{i^\star}} \right)^2 = (1-\varepsilon_0)^2. \tag{32}$$

For the sake of space, we omit the proofs and defer the reader to Appendix C for ideas on how to prove them.

## I.2 Anytime Top Two algorithm

Let $\varepsilon_0 \in (0,1)$. The EB-TC$_{\varepsilon_0}^{\mathrm{m}}$ algorithm, detailed in Figure 3, has the same structure as EB-TC$_{\varepsilon_0}$. There are two differences. First, the IDS proportions are $\beta_n(i,j) = N_{n,j}/((1-\varepsilon_0)^2 N_{n,i} + N_{n,j})$ when considering multiplicative $\varepsilon$-BAI. Second, the challenger will be based on transportation costs for multiplicative $\varepsilon$-BAI, namely

$$C_n^{\mathrm{TC}_{\varepsilon_0}^{\mathrm{m}}} \in \arg\min_{i \neq B_n^{\mathrm{EB}}} \frac{\mu_{n,B_n^{\mathrm{EB}}} - (1-\varepsilon_0)\mu_{n,i}}{\sqrt{1/N_{n,B_n^{\mathrm{EB}}} + (1-\varepsilon_0)^2/N_{n,i}}}. \tag{33}$$

**Stopping rule for fixed-confidence multiplicative $\varepsilon$-BAI** Similarly, the fixed-confidence setting requires a stopping rule, which will be different since we tackle multiplicative $\varepsilon$-BAI. Given an error/confidence pair $(\varepsilon, \delta) \in [0,1) \times (0,1)$, the GLR$_\varepsilon^{\mathrm{m}}$ stopping rule [15] prescribes to stop at the time

$$\tau_{\varepsilon,\delta}^{\mathrm{mul}} = \inf \left\{ n > K \mid \min_{i \neq \hat{i}_n} \frac{\mu_{n,\hat{i}_n} - (1-\varepsilon)\mu_{n,i}}{\sqrt{1/N_{n,\hat{i}_n} + (1-\varepsilon)^2/N_{n,i}}} \geq \sqrt{2c(n-1,\delta)} \right\}, \tag{34}$$

where $c : \mathbb{N} \times (0,1) \to \mathbb{R}_+$ is a threshold function. Lemma 53 gives a threshold ensuring that the GLR$_\varepsilon^{\mathrm{m}}$ stopping rule is $(\varepsilon, \delta)$-PAC for all $\varepsilon \in [0,1)$ and $\delta \in (0,1)$, independently of the sampling rule.

Figure 3: EB-TC$_{\varepsilon_0}^{\mathrm{m}}$ algorithm with **fixed** or **IDS** proportions.

1: **Input:** slack $\varepsilon_0 \in (0,1)$, proportion $\beta \in (0,1)$ (only for fixed proportions).
2: Set $\hat{\imath}_n \in \arg\max_{i \in [K]} \mu_{n,i}$, $B_n^{\mathrm{EB}} = \hat{\imath}_n$ and $C_n^{\mathrm{TC}_{\varepsilon_0}^{\mathrm{m}}} \in \arg\max_{i \neq B_n^{\mathrm{EB}}} \frac{\mu_{n,B_n^{\mathrm{EB}}} - (1-\varepsilon_0)\mu_{n,i}}{\sqrt{1/N_{n,B_n^{\mathrm{EB}}} + (1-\varepsilon_0)^2/N_{n,i}}}$.
3: Set **[fixed]** $\bar{\beta}_{n+1}(i,j) = \beta$ or **[IDS]** $\beta_n(i,j) = N_{n,j}/((1-\varepsilon_0)^2 N_{n,i} + N_{n,j})$ and update $\bar{\beta}_{n+1}(i,j)$.
4: Set $I_n = C_n^{\mathrm{TC}_{\varepsilon_0}^{\mathrm{m}}}$ if $N_{n,C_n^{\mathrm{TC}_{\varepsilon_0}^{\mathrm{m}}}}^{B_n^{\mathrm{EB}}} \leq (1 - \bar{\beta}_{n+1}(B_n^{\mathrm{EB}}, C_n^{\mathrm{TC}_{\varepsilon_0}^{\mathrm{m}}}))T_{n+1}(B_n^{\mathrm{EB}}, C_n^{\mathrm{TC}_{\varepsilon_0}^{\mathrm{m}}})$, otherwise set $I_n = B_n^{\mathrm{EB}}$.
5: **Output:** next arm to sample $I_n$ and next recommendation $\hat{\imath}_n$.

**Lemma 53.** *Let $\varepsilon \in [0,1)$ and $\delta \in (0,1)$. Given any sampling rule, using the threshold* (4) *with the stopping rule* (34) *with error/confidence pair $(\varepsilon, \delta)$ yields a $(\varepsilon, \delta)$-PAC algorithm for sub-Gaussian distributions.*

*Proof.* The proof of Lemma 53 is actually almost identical to the one of Lemma 3. The only change is at the beginning of the proof. Since $\hat{\imath}_n = i^\star(\mu_n)$, standard manipulations yield that for all $i \neq \hat{\imath}_n$

$$\frac{(\mu_{n,\hat{\imath}_n} - (1-\varepsilon)\mu_{n,i})^2}{1/N_{n,\hat{\imath}_n} + (1-\varepsilon)^2/N_{n,i}} = \inf_{u \in \mathbb{R}} \left\{ N_{n,\hat{\imath}_n}(\mu_{n,\hat{\imath}_n} - u)^2 + N_{n,i}(\mu_{n,i} - u/(1-\varepsilon))^2 \right\}$$
$$= \inf_{y \geq x/(1-\varepsilon)} \left\{ N_{n,\hat{\imath}_n}(\mu_{n,\hat{\imath}_n} - x)^2 + N_{n,i}(\mu_{n,i} - y)^2 \right\}.$$

Let $i^\star = i^\star(\mu)$. Using the stopping rule (34) and the above manipulations, we obtain

$$\mathbb{P}_\nu \left( \tau_{\varepsilon,\delta}^{\mathrm{mul}} < +\infty, \hat{\imath}_{\tau_{\varepsilon,\delta}^{\mathrm{mul}}} \notin \mathcal{I}_\varepsilon^{\mathrm{mul}}(\mu) \right)$$
$$\leq \mathbb{P}_\nu \left( \exists n \in \mathbb{N}, \exists i \notin \mathcal{I}_\varepsilon^{\mathrm{mul}}(\mu), i = i^\star(\mu_n), \right.$$
$$\left. \min_{k \neq i} \inf_{y \geq x/(1-\varepsilon)} \left( N_{n,\hat{\imath}_n}(\mu_{n,\hat{\imath}_n} - x)^2 + N_{n,i}(\mu_{n,i} - y)^2 \right) \geq 2c(n-1, \delta) \right)$$
$$\leq \mathbb{P}_\nu \left( \exists n \in \mathbb{N}, \exists i \notin \mathcal{I}_\varepsilon^{\mathrm{mul}}(\mu), i = i^\star(\mu_n), \right.$$
$$\left. \frac{N_{n,i}}{2}(\mu_{n,i} - \mu_i)^2 + \frac{N_{n,i^\star}}{2}(\mu_{n,i^\star} - \mu_{i^\star})^2 \geq c(n-1, \delta) \right)$$
$$\leq \sum_{i \notin \mathcal{I}_\varepsilon^{\mathrm{mul}}(\mu)} \mathbb{P}_\nu \left( \exists n \in \mathbb{N}, \frac{N_{n,i}}{2}(\mu_{n,i} - \mu_i)^2 + \frac{N_{n,i^\star}}{2}(\mu_{n,i^\star} - \mu_{i^\star})^2 \geq c(n-1, \delta) \right),$$

where the second inequality is obtained with $(k, x, y) = (i^\star, \mu_i, \mu_{i^\star})$ since $i \notin \mathcal{I}_\varepsilon^{\mathrm{mul}}(\mu)$, and the third by union bound. Then, we can conclude as in Appendix G.1 that

$$\mathbb{P}_\nu \left( \tau_{\varepsilon,\delta}^{\mathrm{mul}} < +\infty, \hat{\imath}_{\tau_{\varepsilon,\delta}^{\mathrm{mul}}} \notin \mathcal{I}_\varepsilon^{\mathrm{mul}}(\mu) \right) \leq \delta.$$

$\square$

**Asymptotic expected sample complexity** While Theorem 8 holds for all sub-Gaussian distributions, it is particularly interesting for Gaussian ones, in light of Lemma 1. It shows that, for multiplicative $\varepsilon$-BAI, EB-TC$_{\varepsilon_0}^{\mathrm{m}}$ is asymptotically optimal for Gaussian bandits when using IDS proportions and asymptotically $\beta$-optimal when using fixed proportions $\beta$.

**Theorem 8.** *Let $\varepsilon_0 \in (0,1)$, $\beta \in (0,1)$ and $\delta \in (0,1)$. Using the threshold* (4) *in the stopping rule* (34) *with error/confidence $(\varepsilon_0, \delta)$, the EB-TC$_{\varepsilon_0}^{\mathrm{m}}$ algorithm is $(\varepsilon_0, \delta)$-PAC and it satisfies that, for all $\nu \in \mathcal{D}^K$ such that $|i^\star(\mu)| = 1$,*

$$\textbf{[IDS]} \quad \limsup_{\delta \to 0} \frac{\mathbb{E}_\nu[\tau_{\varepsilon_0,\delta}]}{\log(1/\delta)} \leq T_{\varepsilon_0}^{\mathrm{mul}}(\mu) \quad and \quad \textbf{[fixed $\beta$]} \quad \limsup_{\delta \to 0} \frac{\mathbb{E}_\nu[\tau_{\varepsilon_0,\delta}]}{\log(1/\delta)} \leq T_{\varepsilon_0,\beta}^{\mathrm{mul}}(\mu).$$

## I.3 Proof of Theorem 8

The proof is very similar to the one of Theorem 1, hence we will only highlight key differences. We defer the reader to Appendix D for a detailed discussion on the proof method and intuition on the different steps of the proof, see also [36, 21, 40].

In the following, we consider a slack $\varepsilon_0$ for EB-TC$_{\varepsilon_0}^{\mathrm{m}}$ which matches the error $\varepsilon$ considered by the GLR$_\varepsilon^{\mathrm{m}}$ stopping rule (34), i.e. $\varepsilon_0 = \varepsilon$. For conciseness, we denote $w^\star = w_\varepsilon^{\mathrm{mul}}$ and $w_\beta^\star = w_{\varepsilon,\beta}^{\mathrm{mul}}$.

Let $\gamma > 0$. Let us define the *convergence times* $T_{\varepsilon_0,\gamma}$ and $T_{\varepsilon_0,\beta,\gamma}$ as in (10) and (14), i.e.

$$T_{\varepsilon_0,\gamma} := \inf\left\{ T \geq 1 \mid \forall n \geq T, \ \max_{i \neq i^\star} \left| \frac{N_{n,i}}{N_{n,i^\star}} - \frac{w_i^\star}{w_{i^\star}^\star} \right| \leq \gamma \right\},$$

$$T_{\varepsilon_0,\beta,\gamma} := \inf\left\{ T \geq 1 \mid \forall n \geq T, \ \left\| \frac{N_n}{n-1} - w_\beta^\star \right\|_\infty \leq \gamma \right\}.$$

Lemma 54 gives a sufficient condition for asymptotic optimality and asymptotic $\beta$-optimality. The proof is the same as for Lemmas 13 and 25, hence we omit it.

**Lemma 54.** *Let $\varepsilon_0 \in (0,1)$ and $\delta \in (0,1)$. Assume that there exists $\gamma_1(\mu) > 0$ such that for all $\gamma \in (0, \gamma_1(\mu)]$, $\mathbb{E}_\nu[T_{\varepsilon_0,\gamma}] < +\infty$. Using the threshold (4) in the stopping rule (3) with error $\varepsilon_0$ yields an algorithm such that, for all $\nu \in \mathcal{D}^K$ such that $|i^\star(\mu)| = 1$,*

$$\limsup_{\delta \to 0} \frac{\mathbb{E}_\nu[\tau_{\varepsilon_0,\delta}]}{\log(1/\delta)} \leq T_{\varepsilon_0}^{\mathrm{mul}}(\mu).$$

*Assume that there exists $\gamma_1(\mu) > 0$ such that for all $\gamma \in (0, \gamma_1(\mu)]$, $\mathbb{E}_\nu[T_{\varepsilon_0,\beta,\gamma}] < +\infty$. Using the threshold (4) in the stopping rule (3) with error $\varepsilon_0$ yields an algorithm such that, for all $\nu \in \mathcal{D}^K$ such that $|i^\star(\mu)| = 1$,*

$$\limsup_{\delta \to 0} \frac{\mathbb{E}_\nu[\tau_{\varepsilon_0,\delta}]}{\log(1/\delta)} \leq T_{\varepsilon_0,\beta}^{\mathrm{mul}}(\mu).$$

### I.3.1 Sufficient exploration

Lemma 55 shows that the transportation cost is strictly positive and increases linearly, it bares similarity with Lemma 14.

**Lemma 55.** *Let $S_n^L$ and $\mathcal{I}_n^\star$ as in (11). There exists $L_0$ with $\mathbb{E}_\mu[(L_0)^\alpha] < +\infty$ for all $\alpha > 0$ such that if $L \geq L_0$, for all $n$ such that $S_n^L \neq \emptyset$, for all $(i,j) \in \mathcal{I}_n^\star \times S_n^L$ such that $i \neq j$, we have*

$$\frac{\mu_{n,i} - (1-\varepsilon_0)\mu_{n,j}}{\sqrt{1/N_{n,i} + (1-\varepsilon_0)^2/N_{n,j}}} \geq \sqrt{L} D_\mu,$$

*where $D_\mu > 0$ is a problem dependent constant.*

*Proof.* Using Lemma 12 and $\min\{N_{n,i}, N_{n,j}\} \geq L$, we obtain

$$\mu_{n,i} - (1-\varepsilon_0)\mu_{n,j} \geq \mu_i - (1-\varepsilon_0)\mu_j - W_\mu \left( \sqrt{\frac{\log(e + N_{n,i})}{N_{n,i}}} + (1-\varepsilon_0)\sqrt{\frac{\log(e + N_{n,j})}{N_{n,j}}} \right)$$

$$\geq \varepsilon_0 \min_{i \in [K]} \mu_i - (2 - \varepsilon_0) W_\mu \sqrt{\frac{\log(e+L)}{L}} \geq \varepsilon_0 \min_{i \in [K]} \mu_i / 2,$$

where the last inequality is obtained for $L \geq L_0 = L_1 - e$ which is defined as

$$L_1 = \sup\left\{ x \mid x < \frac{4(2-\varepsilon_0)^2 W_\mu^2}{\varepsilon_0^2 \min_{i \in [K]} \mu_i^2} \log(x) + e \right\} < h_1\left( \frac{4(2-\varepsilon_0)^2 W_\mu^2}{\varepsilon_0^2 \min_{i \in [K]} \mu_i^2}, e \right),$$

The last inequality is obtained by using Lemma 51, and we recall that $h_1$ is defined in Lemma 51. Since $h_1(x,e) \sim_{x \to +\infty} x \log x$, we have $\mathbb{E}_\mu[(L_0)^\alpha] < +\infty$ for all $\alpha > 0$ by using Lemma 12 (polynomial in $W_\mu$). Then, we have

$$\frac{\mu_{n,i} - (1-\varepsilon_0)\mu_{n,j}}{\sqrt{1/N_{n,i} + (1-\varepsilon_0)^2/N_{n,j}}} \geq \frac{\varepsilon_0 \min_{i \in [K]} \mu_i / 2}{\sqrt{1/N_{n,i} + (1-\varepsilon_0)^2/N_{n,j}}} \geq \sqrt{L} \frac{\varepsilon_0 \min_{i \in [K]} \mu_i}{2\sqrt{1 + (1-\varepsilon_0)^2}}.$$

Setting $D_\mu = \frac{\varepsilon_0 \min_{i \in [K]} \mu_i}{2\sqrt{1 + (1-\varepsilon_0)^2}}$ yields the result. $\qquad\square$

Lemma 56 gives an upper bound on the transportation costs between a sampled enough arm and an under-sampled one it bares similarity with Lemma 15.

**Lemma 56.** *Let $S_n^L$ as in (11). There exists $L_1$ with $\mathbb{E}_\mu[(L_1)^\alpha] < +\infty$ for all $\alpha > 0$ such that for all $L \geq L_1$ and all $n \in \mathbb{N}$,*

$$\forall (i,j) \in S_n^L \times \overline{S_n^L}, \quad \frac{\mu_{n,i} - (1-\varepsilon_0)\mu_{n,j}}{\sqrt{1/N_{n,i} + (1-\varepsilon_0)^2/N_{n,j}}} \leq \frac{\sqrt{L}}{1-\varepsilon_0}(D_1 + 4W_\mu),$$

*where $D_1 > 0$ is a problem dependent constant and $W_\mu$ is the random variables defined in Lemma 12.*

*Proof.* Using Lemma 12 and $N_{n,i} \geq L \geq N_{n,j} \geq 1$, we obtain

$$\frac{\mu_{n,i} - (1-\varepsilon_0)\mu_{n,j}}{\sqrt{1/N_{n,i} + (1-\varepsilon_0)^2/N_{n,j}}} \leq \frac{\sqrt{N_{n,j}}}{1-\varepsilon_0}|\mu_{n,i} - (1-\varepsilon_0)\mu_{n,j}|$$

$$\leq \frac{\sqrt{L}}{1-\varepsilon_0}\left(|\mu_i - (1-\varepsilon_0)\mu_j| + W_\mu\left(\sqrt{\frac{\log(e + N_{n,i})}{N_{n,i}}} + (1-\varepsilon_0)\sqrt{\frac{\log(e + N_{n,j})}{N_{n,j}}}\right)\right)$$

$$\leq \frac{\sqrt{L}}{1-\varepsilon_0}\left(|\mu_i - (1-\varepsilon_0)\mu_j| + (2-\varepsilon_0)W_\mu\sqrt{\log(e+1)}\right) \leq \frac{\sqrt{L}}{1-\varepsilon_0}(D_1 + 4W_\mu),$$

for $D_1 = \max_{i \neq j}|\mu_i - (1-\varepsilon_0)\mu_j|$. $\qquad\square$

Lemma 57 show that the desired property for the TC challenger, we omit the proof since it is the same as for Lemma 17.

**Lemma 57.** *There exists $L_1$ with $\mathbb{E}_\nu[L_3] < +\infty$ such that if $L \geq L_3$, for all $n$ (at most polynomial in $L$) such that $U_n^L \neq \emptyset$, $B_n^{EB} \notin V_n^L$ implies $C_n^{TC_{\varepsilon_0}^m} \in V_n^L$.*

Combining Lemma 16 and Lemma 57 yields Lemma 58. Importantly, Lemma 58 holds for the EB-TC$_{\varepsilon_0}$ algorithm with fixed proportions $\beta \in (0,1)$ and IDS proportions. Since the proof is the same as for Lemmas 18 and 26, we omit it.

**Lemma 58.** *There exist $N_1$ with $\mathbb{E}_\nu[N_1] < +\infty$ such that for all $n \geq N_1$ and all $i \in [K]$, $N_{n,i} \geq \sqrt{n/K}$ and $B_n^{EB} = i^\star$.*

### I.3.2   Empirical overall balance

As in [40], the key to obtain asymptotic optimality is to show that the empirical proportion satisfy the empirical overall balance equation. Compared to them, the novelty of Lemma 59 is that we use IDS proportions with $K(K-1)$ tracking procedures to select between the leader and the challenger instead of sampling. The proof of Lemma 59 is highly similar to the one of Lemma 19.

**Lemma 59.** *Let $\gamma > 0$. There exists $N_2$ with $\mathbb{E}_\nu[N_2] < +\infty$ such that for all $n \geq N_2$*

$$\left|\left(\frac{N_{n,i^\star}}{n-1}\right)^2 - \sum_{i \neq i^\star}\left(\frac{1}{1-\varepsilon_0}\frac{N_{n,i}}{n-1}\right)^2\right| \leq \gamma.$$

*Proof.* Let $N_1$ as in Lemma 58. We proceed as in Lemma 19. Let us define

$$G_n = \left(\sum_{t=N_1}^{n-1}\beta_t(i^\star, C_t)\right)^2 - \sum_{j \neq i^\star}\left(\frac{1}{1-\varepsilon_0}\sum_{t=N_1}^{n-1}\mathbb{1}(C_t = j)(1 - \beta_t(i^\star, j))\right)^2,$$

Direct manipulations yield that

$$\frac{1}{2}(G_{n+1} - G_n) = \beta_n(i^\star, C_n)\sum_{t=N_1}^{n-1}\beta_t(i^\star, C_t) - \frac{1 - \beta_n(i^\star, C_n)}{(1-\varepsilon_0)^2}\sum_{t=N_1}^{n-1}\mathbb{1}(C_t = C_n)(1 - \beta_t(i^\star, C_n))$$

$$+ \beta_n(i^\star, C_n)^2 - \frac{1}{(1-\varepsilon_0)^2}(1 - \beta_n(i^\star, C_n))^2.$$

Using that $\beta_n(i^\star, C_n)N_{n,i^\star} = (1 - \beta_n(i^\star, C_n))N_{n,C_n}/(1 - \varepsilon_0)^2$ since $\beta_n(i^\star, C_n) = N_{n,C_n}/((1 - \varepsilon_0)^2 N_{n,i^\star} + N_{n,C_n})$ and the inequalities proven in Lemma 19, we can exhibit deterministic constants $A, B > 0$ (depending on $K$ and $\varepsilon_0$) such that

$$|G_{n+1} - G_n| \le AN_1 + B \quad \text{hence} \quad |G_n| \le (n - 1 - N_1)(AN_1 + B) + (1 - \varepsilon_0)^2 .$$

where the second inequality is obtained by telescopic sum (as in Lemma 19) and using that $G_{N_1+1} = \beta_{N_1}(i^\star, C_{N_1})^2 - (1 - \beta_{N_1}(i^\star, C_{N_1}))^2/(1 - \varepsilon_0)^2$.

As in Lemma 19, it is possible to exhibit deterministic constants $C, D > 0$ (depending on $K$ and $\varepsilon_0$) such that

$$\left| \left( \frac{N_{n,i^\star}}{n - 1} \right)^2 - \sum_{i \ne i^\star} \left( \frac{1}{1 - \varepsilon_0} \frac{N_{n,i}}{n - 1} \right)^2 - \frac{G_n}{(n - 1)^2} \right| \le \frac{CN_1 + D}{n - 1} .$$

Combining both results, we can show

$$\left| \left( \frac{N_{n,i^\star}}{n - 1} \right)^2 - \sum_{i \ne i^\star} \left( \frac{1}{1 - \varepsilon_0} \frac{N_{n,i}}{n - 1} \right)^2 \right| \le \frac{(A + C)N_1 + B + D}{n - 1} + \frac{(1 - \varepsilon_0)^2}{(n - 1)^2} .$$

Therefore, we have shown the desired result for $n \ge N_2$ defined as

$$N_2 = \inf \left\{ n > 2 \mid \frac{(A + C)N_1 + B + D}{n - 1} + \frac{(1 - \varepsilon_0)^2}{(n - 1)^2} \le \gamma \right\} ,$$

which satisfies $\mathbb{E}_\nu[N_2] < +\infty$ since it is a linear function of $N_1$. $\qquad\square$

As in [40], using Lemma 59 allows to bound the empirical proportion allocated to the unique best arm $N_{n,i^\star}/(n - 1)$ away from 0 (Lemma 60).

**Lemma 60.** *There exists $N_3$ with $\mathbb{E}_\nu[N_3] < +\infty$ such that for all $n \ge N_3$*

$$c_L \le \frac{N_{n,i^\star}}{n - 1} \le c_U \quad \text{where} \quad c_U = \sqrt{\frac{(1 - \varepsilon_0)^2/2 + 1}{(1 - \varepsilon_0)^2 + 1}} \quad \text{and} \quad c_L = \frac{1 - c_U}{(1 - \varepsilon_0)\sqrt{2(K - 1)}} .$$

*Proof.* Let $N_2$ as in Lemma 59 for $\frac{\gamma}{(1 - \varepsilon_0)^4}$. Using Lemma 59, we obtain for all $n \ge N_2$

$$\gamma \ge (1 - \varepsilon_0)^2 \left( \frac{N_{n,i^\star}}{n - 1} \right)^2 - \sum_{i \ne i^\star} \left( \frac{N_{n,i}}{n - 1} \right)^2 \ge (1 - \varepsilon_0)^2 \left( \frac{N_{n,i^\star}}{n - 1} \right)^2 - \left( 1 - \frac{N_{n,i^\star}}{n - 1} \right)^2$$

$$= ((1 - \varepsilon_0)^2 - 1) \left( \frac{N_{n,i^\star}}{n - 1} \right)^2 + 2\frac{N_{n,i^\star}}{n - 1} - 1$$

$$\ge ((1 - \varepsilon_0)^2 + 1) \left( \frac{N_{n,i^\star}}{n - 1} \right)^2 - 1 .$$

where we used that $\frac{N_{n,i^\star}}{n-1} \le 1$. Taking $\gamma = (1 - \varepsilon_0)^2/2$ yields the upper bound.

Let $\tilde{N}_2$ as in Lemma 59 for $\frac{\gamma}{(K-1)(1-\varepsilon_0)^2}$. Using Lemma 59, we obtain for all $n \ge \max\{N_2, \tilde{N}_2\}$

$$-\gamma \le (K - 1)(1 - \varepsilon_0)^2 \left( \frac{N_{n,i^\star}}{n - 1} \right)^2 - (K - 1) \sum_{i \ne i^\star} \left( \frac{N_{n,i}}{n - 1} \right)^2$$

$$\le (K - 1)(1 - \varepsilon_0)^2 \left( \frac{N_{n,i^\star}}{n - 1} \right)^2 - \left( 1 - \frac{N_{n,i^\star}}{n - 1} \right)^2$$

$$\le (K - 1)(1 - \varepsilon_0)^2 \left( \frac{N_{n,i^\star}}{n - 1} \right)^2 - (1 - c_U)^2$$

Taking $\gamma = (1 - c_U)^2/2$ yields the lower bound. Note that

$$c_L \le c_U \quad \Longleftrightarrow \quad 1 \le \frac{(1 - \varepsilon_0)^2/2 + 1}{(1 - \varepsilon_0)^2 + 1} \left( 2(K - 1)(1 - \varepsilon_0)^2 + 2\sqrt{2(K - 1)}(1 - \varepsilon_0) + 1 \right) ,$$

where the last condition is trivially true. Taking $N_3 = \max\{N_2, \tilde{N}_2\}$ yields the result. $\qquad\square$

Lemma 61 is a rescaled version of the empirical overall balance equation which is proven by simply combining Lemma 59 and Lemma 60.

**Lemma 61.** *Let $\gamma > 0$. There exists $N_4$ with $\mathbb{E}_\nu[N_4] < +\infty$ such that for all $n \geq N_4$*

$$\left| (1 - \varepsilon_0)^2 - \sum_{i \neq i^\star} \left( \frac{N_{n,i}}{N_{n,i^\star}} \right)^2 \right| \leq \gamma \, .$$

*Proof.* Let $N_3$ and $c_L$ as in Lemma 60. Let $N_2$ as in Lemma 59 for $\gamma c_L^2 / (1 - \varepsilon_0)^2$. Direct manipulation shows that, for all $n \geq N_4 = \max\{N_2, N_3\}$,

$$\left| 1 - \sum_{i \neq i^\star} \left( \frac{1}{1 - \varepsilon_0} \frac{N_{n,i}}{N_{n,i^\star}} \right)^2 \right| = \left( \frac{n-1}{N_{n,i^\star}} \right)^2 \left| \left( \frac{N_{n,i^\star}}{n-1} \right)^2 - \sum_{i \neq i^\star} \left( \frac{1}{1 - \varepsilon_0} \frac{N_{n,i}}{n-1} \right)^2 \right|$$

$$\leq \frac{1}{c_L^2} \gamma c_L^2 = \frac{\gamma}{(1 - \varepsilon_0)^2} \, .$$

This concludes the result. $\qquad\square$

### I.3.3 Convergence towards optimal allocation

As in [40], we show that a challenger will not be sampled next when the ratio of its empirical proportion compared to the one of $i^\star$ overshoots the ratio of the optimal allocations (Lemma 62).

**Lemma 62.** *Let $\gamma > 0$. There exists $N_5$ with $\mathbb{E}_\nu[N_5] < +\infty$ such that for all $n \geq N_5$ and all $i \neq i^\star$,*

$$\frac{N_{n,i}}{N_{n,i^\star}} \geq \frac{w_i^\star}{w_{i^\star}^\star} + \gamma \quad \implies \quad C_n^{TC_{\varepsilon_0}^m} \neq i \, .$$

*Proof.* Let $\gamma > 0$. Let $i \neq i^\star$ such that

$$\frac{N_{n,i}}{N_{n,i^\star}} \geq \frac{w_i^\star}{w_{i^\star}^\star} + \gamma \, .$$

Let $N_4$ as in Lemma 61. As in Lemma 22, we can show for all $n \geq N_4$

$$\exists j \notin \{i^\star, i\}, \quad \frac{N_{n,j}}{N_{n,i^\star}} \leq \frac{w_j^\star}{w_{i^\star}^\star} \, .$$

Let $N_1$ as in Lemma 18. Using that $B_n^{\mathrm{EB}} = i^\star$ for all $n \geq N_1$ and the definition of $C_n^{TC_{\varepsilon_0}^m}$, we known that

$$\frac{\mu_{n,i^\star} - (1 - \varepsilon_0)\mu_{n,i}}{\sqrt{1/N_{n,i^\star} + (1 - \varepsilon_0)^2/N_{n,i}}} > \frac{\mu_{n,i^\star} - (1 - \varepsilon_0)\mu_{n,j}}{\sqrt{1/N_{n,i^\star} + (1 - \varepsilon_0)^2/N_{n,j}}} \quad \implies \quad C_n^{TC_{\varepsilon_0}^m} \neq i \, .$$

To conclude the proof, it is sufficient to show that the ratio of the two quantities is strictly larger than one. For all $n \geq \max\{N_1, N_4\}$, we obtain

$$\frac{\mu_{n,i^\star} - (1 - \varepsilon_0)\mu_{n,i}}{\mu_{n,i^\star} - (1 - \varepsilon_0)\mu_{n,j}} \sqrt{\frac{1 + (1 - \varepsilon_0)^2 N_{n,i^\star}/N_{n,j}}{1 + (1 - \varepsilon_0)^2 N_{n,i^\star}/N_{n,i}}}$$

$$\geq \frac{\mu_{n,i^\star} - (1 - \varepsilon_0)\mu_{n,i}}{\mu_{n,i^\star} - (1 - \varepsilon_0)\mu_{n,j}} \sqrt{\frac{1 + (1 - \varepsilon_0)^2 w_{i^\star}^\star/w_j^\star}{1 + (1 - \varepsilon_0)^2 (w_i^\star/w_{i^\star}^\star + \gamma)^{-1}}} \, .$$

Let $\tilde{\gamma} > 0$. Using Lemmas 12 and 18, we have, for all $n \geq N_1$ and all $k \neq i^\star$,

$$\left| \frac{\mu_{n,i^\star} - (1 - \varepsilon_0)\mu_{n,k}}{\mu_{i^\star} - (1 - \varepsilon_0)\mu_k} - 1 \right| \leq \frac{W_\mu \left( (1 - \varepsilon_0) \sqrt{\frac{\log(e + N_{n,k})}{N_{n,k}}} + \sqrt{\frac{\log(e + N_{n,i^\star})}{N_{n,i^\star}}} \right)}{\mu_{i^\star} - (1 - \varepsilon_0)\mu_k}$$

$$\leq \frac{(2 - \varepsilon_0)W_\mu}{\min_{k \neq i^\star}(\mu_{i^\star} - (1 - \varepsilon_0)\mu_k)} \sqrt{\frac{\log(e + \sqrt{n/K})}{\sqrt{n/K}}} \leq \tilde{\gamma}$$

for all $n \geq N_6 = K(X_0 - e)^2 + 1$ which is defined as

$$X_0 = \sup \left\{ x \geq 1 \mid x < \frac{(2-\varepsilon_0)^2 W_\mu^2}{\tilde{\gamma}^2 \min_{k \neq i^\star}(\mu_{i^\star} - (1-\varepsilon_0)\mu_k)^2} \log x + e \right\}$$

$$< h_1 \left( \frac{(2-\varepsilon_0)^2 W_\mu^2}{\tilde{\gamma}^2 \min_{k \neq i^\star}(\mu_{i^\star} - (1-\varepsilon_0)\mu_k)^2}, e \right) .$$

where the last inequality is obtained by using Lemma 51 with $h_1$ defined therein. Since $h_1(x,y) \sim_{x \to +\infty} x \log x$, we have $\mathbb{E}_\nu[N_6] < +\infty$ since it polynomial in $W_\mu$ (by using Lemma 12). Let $\kappa > 0$. We have shown that, for all $n \geq \max\{N_1, N_4, N_6\}$,

$$\frac{\mu_{n,i^\star} - (1-\varepsilon_0)\mu_{n,i}}{\mu_{n,i^\star} - (1-\varepsilon_0)\mu_{n,j}} \sqrt{\frac{1 + (1-\varepsilon_0)^2 N_{n,i^\star}/N_{n,j}}{1 + (1-\varepsilon_0)^2 N_{n,i^\star}/N_{n,i}}}$$

$$\geq \frac{\mu_{i^\star} - (1-\varepsilon_0)\mu_i}{\mu_{i^\star} - (1-\varepsilon_0)\mu_j} \sqrt{\frac{1 + (1-\varepsilon_0)^2 w_{i^\star}^\star/w_j^\star}{1 + (1-\varepsilon_0)^2 (w_i^\star/w_{i^\star}^\star + \gamma)^{-1}}} \frac{1 - \tilde{\gamma}}{1 + \tilde{\gamma}}$$

$$= \sqrt{\frac{1 + (1-\varepsilon_0)^2 w_{i^\star}^\star/w_i^\star}{1 + (1-\varepsilon_0)^2 (w_i^\star/w_{i^\star}^\star + \gamma)^{-1}}} \frac{1 - \tilde{\gamma}}{1 + \tilde{\gamma}} .$$

where the equality uses that the transportation costs are equal at the equilibrium, i.e. (31). Taking $\tilde{\gamma}$ small enough, we have that shown that, for all $n \geq \max\{N_1, N_4, N_6\}$,

$$\frac{N_{n,i}}{N_{n,i^\star}} \geq \frac{w_i^\star}{w_{i^\star}^\star} + \gamma \quad \implies \quad \frac{\mu_{n,i^\star} - (1-\varepsilon_0)\mu_{n,i}}{\mu_{n,i^\star} - (1-\varepsilon_0)\mu_{n,j}} \sqrt{\frac{1 + (1-\varepsilon_0)^2 N_{n,i^\star}/N_{n,j}}{1 + (1-\varepsilon_0)^2 N_{n,i^\star}/N_{n,i}}} > 1 ,$$

hence $C_n^{\mathrm{TC}_{\varepsilon_0}^m} \neq i$. This concludes the proof. $\qquad \square$

Lemma 63 is obtained with the same proof as Lemma 23.

**Lemma 63.** *Let $\gamma > 0$. There exists $N_6$ with $\mathbb{E}_\nu[N_6] < +\infty$ such that for all $n \geq N_6$ and all $i \neq i^\star$,*

$$\frac{N_{n,i}}{N_{n,i^\star}} \leq \frac{w_i^\star}{w_{i^\star}^\star} + \gamma .$$

Lemma 64 is obtained with the same proof as Lemma 24.

**Lemma 64.** *Let $\varepsilon_0 \in (0,1)$, $\gamma > 0$ and $T_{\varepsilon_0, \gamma}$ introduced before (adaptation of (10)). Under the EB-TC$_{\varepsilon_0}^m$ sampling rule with IDS proportions, we have $\mathbb{E}_\nu[T_{\varepsilon_0, \gamma}] < +\infty$.*

Combining Lemmas 54 and 64 yields the first of Theorem 8.

### I.3.4 Convergence towards $\beta$-optimal allocation

Combining Lemmas 58 and 27 with the proof of Lemma F.9 in [20] yields Lemma 65.

**Lemma 65.** *Let $\gamma > 0$. There exists $N_5$ with $\mathbb{E}_\nu[N_5] < +\infty$ such that for all $n \geq N_5$ and all $i \neq i^\star$,*

$$\frac{N_{n,i}}{n-1} \geq w_{\beta,i}^\star + \gamma \quad \implies \quad C_n^{TC_{\varepsilon_0}^m} \neq i .$$

Lemma 66 is obtained with the same proof as Lemmas 29 and 30.

**Lemma 66.** *Let $\varepsilon_0 \in (0,1)$, $\beta \in (0,1)$, $\gamma > 0$ and $T_{\varepsilon_0, \beta, \gamma}$ introduced before (adaptation of (14)). Under the EB-TC$_{\varepsilon_0}^m$ sampling rule with fixed proportions $\beta$, we have $\mathbb{E}_\nu[T_{\varepsilon_0, \beta, \gamma}] < +\infty$.*

Combining Lemmas 54 and 66 yields the second part of Theorem 8.

## J  Implementation details and additional experiments

After presenting the implementations details in Appendix J.1, we display supplementary experiments in Appendix J.2.

### J.1 Implementation details

**Top Two sampling rules** Given a fixed $\beta$, most Top Two algorithms use randomization to select $I_n \in \{B_n, C_n\}$, namely they sample $B_n$ with probability $\beta$, otherwise sample $C_n$. Instead of randomization, TTUCB [20] uses $K$ tracking procedures to select $I_n \in \{B_n, C_n\}$. Namely, they choose $I_n = B_n$ if $N_{n,B_n}^{B_n} \le \beta \sum_{i \ne B_n} T_{n+1}(B_n, i)$, and $I_n = C_n$ otherwise. In comparison, EB-TC$_{\varepsilon_0}$ relies on $K(K-1)$ tracking procedures both for fixed proportion $\beta$ and IDS proportions [40]. In [40], they consider IDS proportions to define $\beta_n$ adaptively. For Gaussian with known homoscedastic variance, their update mechanism is

$$\beta_n = \frac{N_{n,B_n} d_{\mathrm{KL}}(\mu_{n,B_n}, u_n(B_n, C_n))}{N_{n,B_n} d_{\mathrm{KL}}(\mu_{n,B_n}, u_n(B_n, C_n)) + N_{n,C_n} d_{\mathrm{KL}}(\mu_{n,C_n}, u_n(B_n, C_n))} = \frac{N_{n,C_n}}{N_{n,B_n} + N_{n,C_n}},$$

where the second equality is obtained by direct computations which uses that

$$u_n(i,j) = \inf_{x \in \mathbb{R}} \left[ N_{n,i} d_{\mathrm{KL}}(\mu_{n,i}, x) + N_{n,j} d_{\mathrm{KL}}(\mu_{n,j}, x) \right] = \frac{N_{n,i}\mu_{n,i} + N_{n,j}\mu_{n,j}}{N_{n,i} + N_{n,j}}.$$

[8] proposed a new methodology called Balancing Optimal Large Deviations (BOLD) to select $I_n \in \{B_n, C_n\}$ adaptively. While their approach aims at minimizing the probability of incorrect selection (i.e. fixed-budget setting), it is possible to adapt their idea to minimize the expected sample complexity (i.e. fixed-confidence setting) by "swapping" the arguments of the KL divergence. Namely, the BOLD procedure can be rewritten in our setting as selecting $I_n = B_n$ if

$$\sum_{i \ne B_n} \frac{d_{\mathrm{KL}}(\mu_{n,B_n}, u_n(B_n, i))}{d_{\mathrm{KL}}(\mu_{n,i}, u_n(B_n, i))} > 1 ,$$

and $I_n = C_n$ otherwise. For Gaussian with known homoscedastic variance, their approach recovers the heuristic "adaptive Welch divergence" algorithm of proposed in [37], i.e. $I_n = B_n$ if $N_{n,B_n}^2 < \sum_{i \ne B_n} N_{n,i}^2$, and $I_n = C_n$ otherwise. The analysis of [8] is focused on asymptotic guarantees in probability. It is an interesting direction for future research to show that the BOLD procedure also yields asymptotically optimal algorithms.

TTTS [34] uses a TS (Thompson Sampling) leader and a RS (Re-Sampling) challenger based on a sampler $\Pi_n$. For Gaussian bandits, the sampler $\Pi_n$ is the posterior distribution $\bigtimes_{i \in [K]} \mathcal{N}(\mu_{n,i}, 1/N_{n,i})$ given the improper prior $\Pi_1 = (\mathcal{N}(0, +\infty))^K$. The TS leader is $B_n^{\mathrm{TS}} \in \arg\max_{i \in [K]} \theta_i$ where $\theta \sim \Pi_n$. The RS challenger samples vector of realizations $\theta \in \Pi_n$ until $B_n \notin \arg\max_{i \in [K]} \theta_i$, then it is defined as $C_n^{\mathrm{RS}} \arg\max_{i \in [K]} \theta_i$ for this specific vector of realization. When the posterior $\Pi_n$ and the leader $B_n$ have almost converged towards the Dirac distribution on $\mu$ and the best arm $i^\star(\mu)$ respectively, the event $B_n \notin \arg\max_{i \in [K]} \theta_i$ becomes very rare. The experiments in [21] reveals that computing the RS challenger can require more than millions of re-sampling steps. Therefore, the RS challenger can become computationally intractable even for Gaussian distribution where sampling from $\Pi_n$ can be done more efficiently.

T3C [36] combines the TS leader and the TC challenger. EB-TCI [21] combines the EB leader with the TCI challenger defined as

$$C_n^{\mathrm{TCI}} = \arg\min_{i \ne B_n} \mathbb{1}\left(\mu_{n,B_n} > \mu_{n,i}\right) \frac{(\mu_{n,B_n} - \mu_{n,i})^2}{2(1/N_{n,B_n} + 1/N_{n,i})} + \log(N_{n,i}) .$$

TTUCB [20] combined the TC challenger with a UCB leader which is defined as

$$B_n^{\mathrm{UCB}} = \arg\max_{i \in [K]} \{\mu_{n,i} + \sqrt{g(n)/N_{n,i}}\} ,$$

where $\sqrt{g(n)/N_{n,i}}$ is a bonus coping for uncertainty. TS-KKT [40] uses the TS leader and a challenger based on transportation costs with a different penalization than the one used in [21], namely

$$C_n) \in \arg\min_{i \ne B_n} \left\{ \frac{(\mu_{n,B_n} - \mu_{n,i})^2}{2(1/N_{n,B_n} + 1/N_{n,i})} - \frac{\rho}{n} \log\left(1/N_{n,B_n} + 1/N_{n,i}\right) \right\} ,$$

where $\rho$ is a parameter that needs to be selected beforehand. The authors highlight that the choice of $\rho$ has an important impact on the empirical performance.

While the above Top Two algorithms are BAI algorithms, we can adapt them straightforwardly to tackle the $\varepsilon$-BAI setting by (1) using the stopping rule as in (3) and (2) using transportation costs for $\varepsilon$-BAI, i.e. the ones in (2). The modification (1) is inspired by [19] and the modification (2) by [22]. As an example, the modified TCI challenger can be written as

$$C_n^{\text{TCI}} = \arg\min_{i \neq B_n} \mathbb{1}\left(\mu_{n,B_n} > \mu_{n,i}\right) \frac{(\mu_{n,B_n} - \mu_{n,i} + \varepsilon)^2}{2(1/N_{n,B_n} + 1/N_{n,i})} + \log(N_{n,i}).$$

**Fixed-confidence BAI algorithms**   At each time $n$, Track-and-Stop (TaS) [15] computes the optimal allocation for the current empirical mean, $w_n = w^\star(\mu_n)$. Given $w_n \in \triangle_K$, it uses a tracking procedure to obtain an arm $I_n$ to sample. On top of this tracking a forced exploration is used to enforce convergence towards the optimal allocation for the true unknown parameters. The optimization problem defining $w^\star(\mu)$ can be rewritten as solving an equation $\psi_\mu(r) = 0$, where

$$\forall r \in (1/\min_{i \neq i^\star}(\mu_{i^\star} - \mu_i)^2, +\infty), \ \psi_\mu(r) = \sum_{i \neq i^\star} \frac{1}{\left(r(\mu_{i^\star} - \mu_i)^2 - 1\right)^2} - 1$$

The function $\psi_\mu$ is decreasing, and satisfies $\lim_{r \to +\infty} \psi_\mu(r) = -1$ and $\lim_{y \to 1/\min_{i \neq i^\star}(\mu_{i^\star} - \mu_i)^2} F_\mu(y) = +\infty$. For the practical implementation of the optimal allocation, we use the approach of [15] and perform binary searches to compute the unique solution of $\psi_\mu(r) = 0$. A faster implementation based on Newton's iterates was proposed by [3] after proving that $\psi_\mu$ is convex. While this improvement holds only for Gaussian distributions, the binary searches can be used for more general distributions. Similarly, one can adapt TaS to tackle $\varepsilon$-BAI as done in [16], i.e. by tracking $w_n = w_\varepsilon(\mu_n)$. Using Lemma 9, it is direct to see that one can use the same implementation to compute it efficiently.

DKM [11] view $T^\star(\mu)^{-1}$ as a min-max game between the learner and the nature, and design saddle-point algorithms to solve it sequentially. At each time $n$, a learner outputs an allocation $w_n$, which is used by the nature to compute the worst alternative mean parameter $\lambda_n$. Then, the learner is updated based on optimistic gains based on $\lambda_n$. Similarly, one can adapt DKM to tackle $\varepsilon$-BAI by using a modified alternative mean parameter. This coincides with the L$\varepsilon$BAI algorithm [19] when used on unstructured multi-armed bandits.

FWS [39] alternates between forced exploration and Frank-Wolfe (FW) updates. Similarly, FWS can be adapted to tackle $\varepsilon$-BAI

LUCB [24] samples and stops based on upper/lower confidence indices for a bonus function $g$. For Gaussian distributions, it rewrites for all $i \in [K]$ as

$$U_{n,i} = \mu_{n,i} + \sqrt{\frac{2c(n-1,\delta)}{N_{n,i}}} \quad \text{and} \quad L_{n,i} = \mu_{n,i} - \sqrt{\frac{2c(n-1,\delta)}{N_{n,i}}}.$$

At each time $n$, it samples $\hat{\imath}_n$ and $\arg\max_{i \neq \hat{\imath}_n} U_{n,i}$. For $\varepsilon$-BAI, LUCB stops when $L_{n,\hat{\imath}_n} + \varepsilon \geq \max_{i \neq \hat{\imath}_n} U_{n,i}$.

**Fixed-budget BAI algorithms**   We consider Successive Reject (SR) [1] and Sequential Halving (SH) [25]. SR eliminates one arm with the worst empirical mean at the end of each phase, and SH eliminated half of them but drops past observations between each phase. Within each phase, both algorithms use a round-robin uniform sampling rule on the remaining active arms. It is possible to convert the fixed-budget SH algorithm into an anytime algorithm by using the doubling trick. It considers a sequences of algorithms that are run with increasing budgets $(T_k)_{k \geq 1}$, with $T_{k+1} = 2T_k$ and $T_1 = 2K\lceil \log_2 K \rceil$, and recommend the answer outputted by the last instance that has finished to run. It is well know that the "cost" of doubling is to have a multiplicative factor 4 in front of the hardness constant. The first two-factor is due to the fact that we forget half the observations. The second two-factor is due to the fact that we use the recommendation from the last instance of SH that has finished. The doubling version of SR and SH are named Doubling SR (DSR) and Doubling SH (DSH).

**Reproducibility**   Our code is implemented in `Julia 1.9.0`, and the plots are generated with the `StatsPlots.jl` package. Other dependencies are listed in the `Readme.md`. The `Readme.md` file also provides detailed julia instructions to reproduce our experiments, as well as a `script.sh` to run them all at once. The general structure of the code (and some functions) is taken from the

 Our experiments are conducted on an institutional cluster with 4 Intel Xeon Gold 5218R CPU with 20 cores per CPU and an x86_64 architecture.

## J.2 Supplementary experiments

For the sake of space, we only presented a small subset of our experiments in Section 5, hence we provide additional empirical evidence below. In Appendix J.2.1, we assess the impact of the choice of the slack $\varepsilon_0$ on the empirical performance of EB-TC$_{\varepsilon_0}$ with IDS or with fixed proportion $\beta = 1/2$. Appendix J.2.2 complements Section 5 by providing more comparison between EB-TC$_{\varepsilon_0}$ and existing algorithms. In Appendix J.2.3, we study the BAI problem and show the empirical performance of the EB-TC$_{(\varepsilon_n)_{n>K}}$ with fixed proportion $\beta = 1/2$ and using varying slack parameters $(\varepsilon_n)_{n>K}$ (as described in Sections 3 and 4). As in Section 5, we consider $\delta = 0.01$ and use the heuristic threshold $c(n, \delta) = \log((1 + \log n)/\delta)$, which yields an empirical error which is several orders of magnitude lower than $\delta$.

### J.2.1 Numerical simulations on EB-TC$_{\varepsilon_0}$

Appendix J.2.1 is meant to be a sensitivity analysis of the EB-TC$_{\varepsilon_0}$ algorithm. It will allow us to numerically substantiate our recommendation to the practitioner on how to set the slack $\varepsilon_0$, and why using IDS yields better empirical performance. For slacks $\varepsilon_0 \in \{0.15, 0.1, 0.05\}$, we will consider the EB-TC$_{\varepsilon_0}$ algorithms with IDS proportions and fixed proportions $\beta = 1/2$.

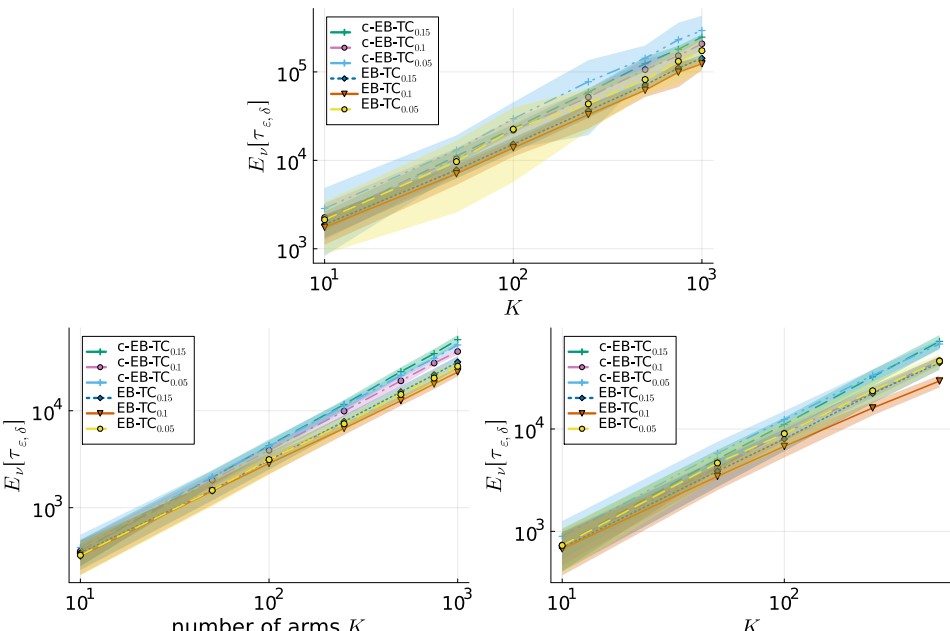

Figure 4: Empirical stopping time with stopping rule (3) using $(\varepsilon, \delta) = (0.1, 0.01)$ on (a)"sparse" instances", (b) $\alpha = 0.3$" instances and (c) "$\alpha = 0.6$" instances. "c-" denotes fixed $\beta = 1/2$, without refers to IDS.

**Large sets of arms** First, we evaluate the impact of larger number of arms (up to $K = 1000$). The "$\alpha = 0.6$" scenario of [18] sets $\mu_i = 1 - ((i-1)/(K-1))^\alpha$ for all $i \in [K]$. The "sparse" scenario of [18] sets $\mu_1 = 1/4$ and $\mu_i = 0$ otherwise. We average on 100 runs, and the standard deviations are displayed.

In Figure 4, we see that all instances of EB-TC$_{\varepsilon_0}$ with IDs or fixed $\beta = 1/2$ have the same scaling with the dimension $K$. Figure 4 also reveals that using IDS instead of fixed $\beta = 1/2$ consistently yields smaller empirical stopping time. Therefore, IDS should be preferred to choosing fixed $\beta = 1/2$. Finally, Figure 4 shows that better empirical performance are obtained when using a slack $\varepsilon_0$ which matches the error $\varepsilon$, i.e. choosing $\varepsilon_0 = \varepsilon$ to tackle $\varepsilon$-BAI. While choosing $\varepsilon_0 > \varepsilon$ only slightly damage the empirical performance, taking $\varepsilon_0 < \varepsilon$ is clearly detrimental to the empirical performance.

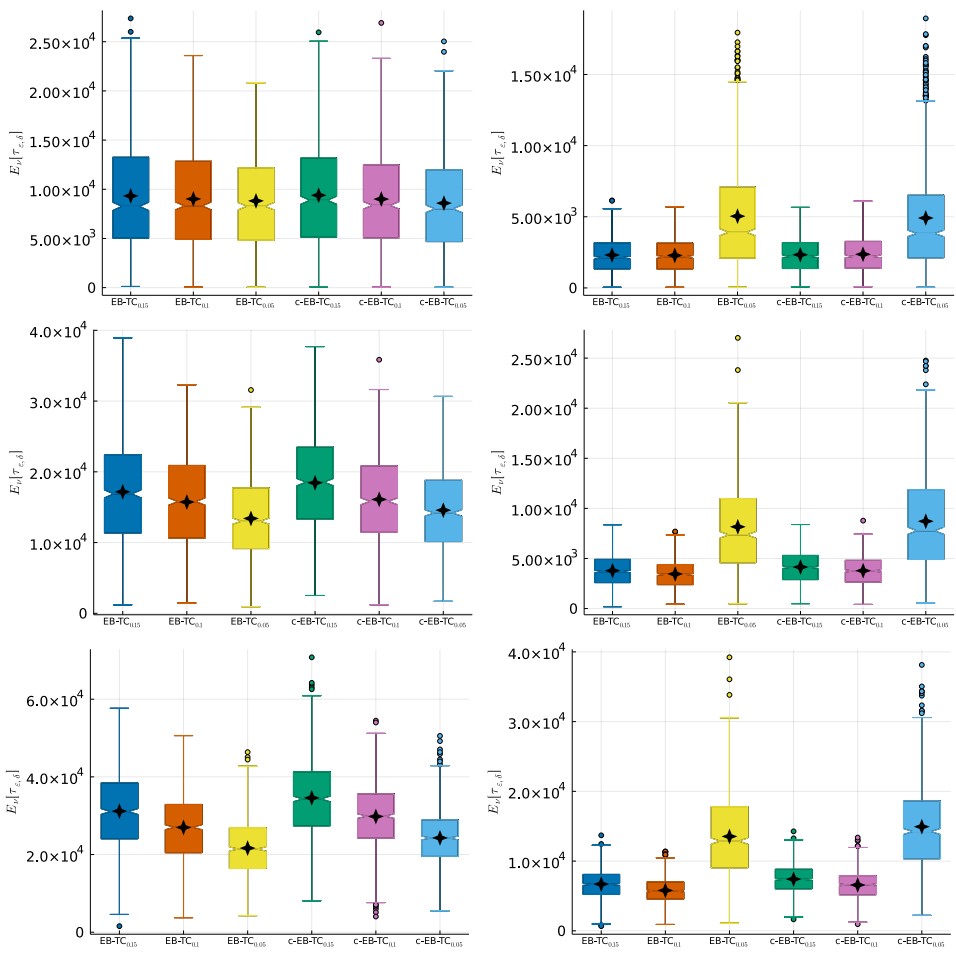

Figure 5: Empirical stopping time on random instances with stopping rule (3) using (left) $(\varepsilon, \delta) = (0.05, 0.01)$ and (right) $(\varepsilon, \delta) = (0.1, 0.01)$ with (top) $K = 5$, (middle) $K = 10$ and (bottom) $K = 20$. "c-" denotes fixed $\beta = 1/2$, without refers to IDS.

**Random instances** We further investigate the phenomenon described above on random instances. For $\varepsilon \in \{0.05, 0.1\}$, we assess the performance on 1000 random Gaussian instances with $K = 5$ (resp. $K = 10$ and $K = 10$) such that $\mu_1 = 1$, $\mu_i \sim \mathcal{U}([1 - \varepsilon, 1])$ for $i \in \{2\}$ (resp. $i \in \{2, 3\}$ and $i \in \{2, 3, 4\}$) and $\mu_i \sim \mathcal{U}([0, 1 - \varepsilon))$ for $i \geq 3$ (resp. $i \geq 4$ and $i \geq 6$), hence $\mathcal{I}_\varepsilon(\mu) = [2]$ (resp. $\mathcal{I}_\varepsilon(\mu) = [3]$ and $\mathcal{I}_\varepsilon(\mu) = [5]$). We display the boxplots of the empirical stopping time on 1000 runs.

In Figure 5, the tendencies glimpsed by inspecting Figure 4 are more apparent. Figure 5 shows that better empirical performance are obtained when using a slack $\varepsilon_0$ which matches the error $\varepsilon$, i.e. choosing $\varepsilon_0 = \varepsilon$ to tackle $\varepsilon$-BAI. We also see that the empirical gain of taking $\varepsilon_0 = \varepsilon$ is increasing with the dimension $K$. Figure 5 shows that $\varepsilon_0 > \varepsilon$ slightly damage the empirical performance: the larger the missmatch $\varepsilon_0 - \varepsilon$ is, the worse the performances are. Moreover, Figure 5 highlights how choosing $\varepsilon_0 < \varepsilon$ is highly detrimental to the empirical performance. Finally, we see that there is a mild advantage in using IDS over fixed $\beta = 1/2$. Note that the mild empirical gain of adaptive proportions is a known phenomenon which was studied empirically in [20], and larger gains can be observed for "two-groups" instances. Intuitively, the characteristic times $T_\varepsilon(\mu)$ and $T_{\varepsilon, 1/2}(\mu)$ are very close to each other on average, and the difference is the largest when all the sub-optimal arms have the same mean. Therefore, on average, using IDS has mild gain compared to using fixed $\beta = 1/2$. This worst-case scenario of the "two-groups" instances is studied in Figure 7.

Table 3: Specific instances from [16] with corresponding $\varepsilon$.

| $\varepsilon$ | Instance | $|\mathcal{I}_\varepsilon(\mu)|$ | Arms | | | | | |
|---|---|---|---|---|---|---|---|---|
| | | | 1 | 2 | 3 | 4 | 5 | 6 |
| 0.1 | $\mu_1$ | 1 | 0.7 | 0.55 | 0.5 | 0.4 | 0.2 | - |
| 0.15 | $\mu_2$ | 3 | 0.8 | 0.75 | 0.7 | 0.6 | 0.5 | 0.4 |
| 0.1 | $\mu_3$ | 3 | 0.6 | 0.6 | 0.55 | 0.45 | 0.3 | 0.2 |

**Specific instances** We continue the experimental validation of our the phenomenon described above by considering the three specific instances from the experimental section of [16] which are described in Table 3. Instance $\mu_3$ corresponds to the specific instances studied in Section 5. It is particularly interesting since it has two best arms, a third $\varepsilon$-good arm which is equally distant from $\varepsilon$ than the bad arm with largest mean, and has two additional bad arms. Instance $\mu_2$ has also three $\varepsilon$-good arms, which are equally spaced, and three bad arms. Instance $\mu_1$ has only one $\varepsilon$-good arm, and four bad arms. We display the boxplots of the empirical stopping time on 1000 runs, and the empirical simple regret on 10000 runs (with associated standard deviation).

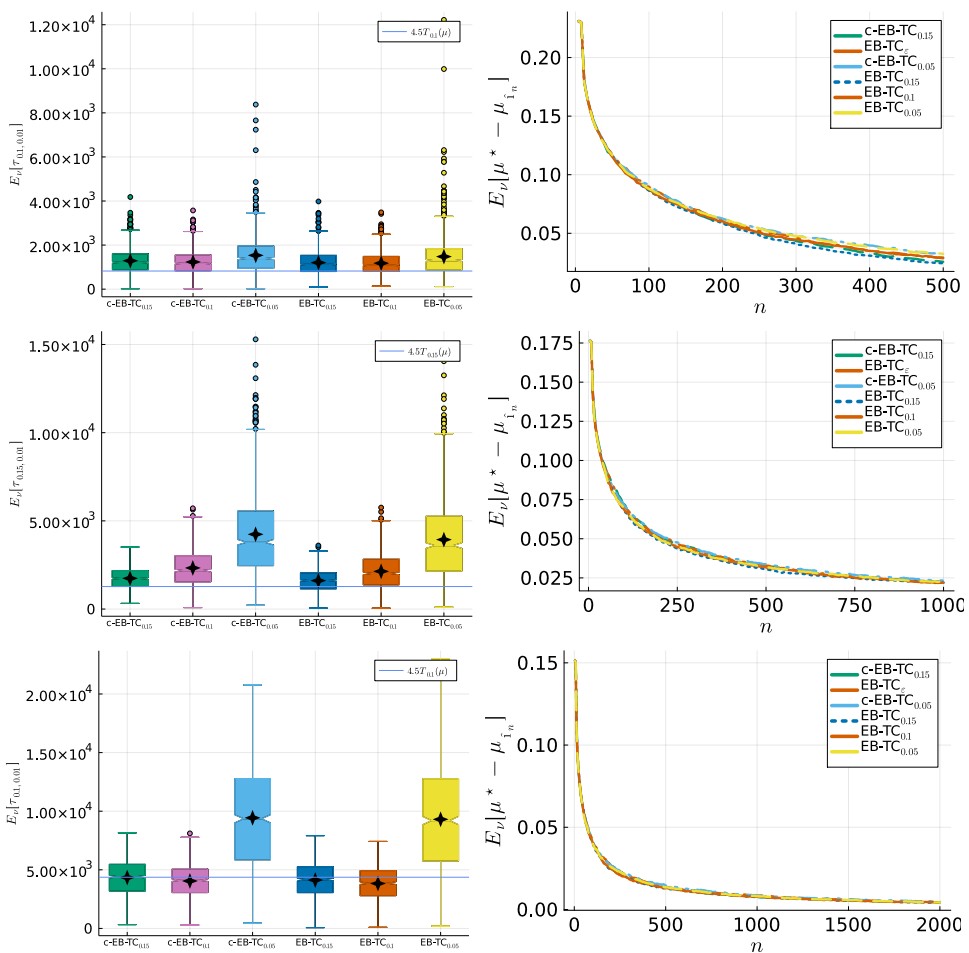

Figure 6: (Left) Empirical stopping time for the stopping rule (3) using $\delta = 0.01$ and (right) empirical simple regret on instances (top) $\mu_1$ with $\varepsilon = 0.1$, (middle) $\mu_2$ with $\varepsilon = 0.15$ and (bottom) $\mu_3$ with $\varepsilon = 0.1$. "c-" denotes fixed $\beta = 1/2$, without refers to IDS.

The left column of Figure 6 confirms our previous observations: taking $\varepsilon = \varepsilon_0$ yields the best performance, taking $\varepsilon_0 > \varepsilon$ slightly damage performance and taking $\varepsilon_0 < \varepsilon$ highly damage performance.

Figure 6(middle left) shows that the the larger the missmatch $\varepsilon - \varepsilon_0$ is, the worse the performances are.

The right column of Figure 6 reveals that the choice of $\varepsilon_0$ has few impact on the empirical simple regret, at least for reasonably chosen $\varepsilon_0$. Likewise, there is few differences between IDS proportions and fixed $\beta = 1/2$. The good empirical performance of IDS proportions as regards the simple regret hints that EB-TC$_{\varepsilon_0}$ with IDS has (most likely) similar theoretical guarantees as the ones obtained in Section 4 for EB-TC$_{\varepsilon_0}$ with fixed $\beta = 1/2$ (Theorem 3 and Corollary 1). This is an interesting open problem that we leave for future work.

**Two-groups instances** We conclude our sensitivity analysis on EB-TC$_{\varepsilon_0}$ by considering the "two-groups" instances $\mu \in \{0.6, 0.4\}^{10}$ for varying number of best arms, i.e. $|i^\star(\mu)| \in [8]$. Our aim is to assess the impact of having multiple best arms on the empirical stopping time. We average on 1000 runs, and the standard deviations are displayed.

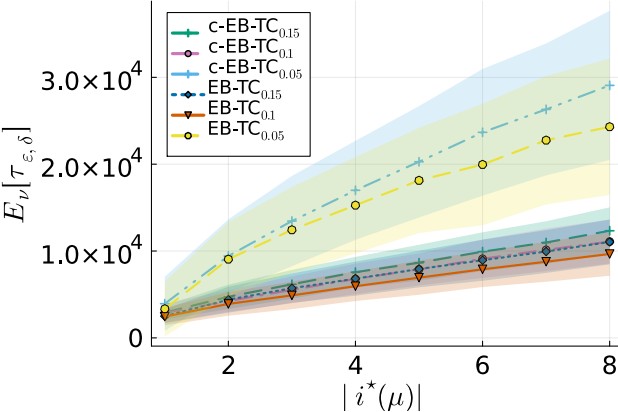

Figure 7: Empirical stopping time with stopping rule (3) using $(\varepsilon, \delta) = (0.1, 0.01)$ on instances $\mu \in \{0.6, 0.4\}^{10}$ for varying $|i^\star(\mu)|$. "c-" denotes fixed $\beta = 1/2$, without refers to IDS.

On this specific instances where the number of arms is fixed, it would be intuitive to think that the problem is easier when there are more best arms. However, this is actually the opposite since there less bad arms (which were easy to detect) and more good arms which need to be estimated well enough. While Figure 7 shows how poor the performances are when taking $\varepsilon_0 < \varepsilon$, it also shows that the impact of having multiple best arms is rather mild for slacks $\varepsilon_0 \geq \varepsilon$. We will see in Figure 12 that EB-TC$_{\varepsilon_0}$ is the most resilient algorithm with respect to multiple arms, meaning that its slope is the smallest.

"Two-groups" instances are interesting since they are the instances for which the ratio $T_{\varepsilon,1/2}(\mu)/T_\varepsilon(\mu)$ is the largest. Using Lemma 9 and the theoretical results of [20] (see their Lemma C.6), we conjecture that $T_{\varepsilon,1/2}(\mu) \leq r_K T_\varepsilon(\mu)$ with $r_K = 2K/(1 + \sqrt{K-1})^2$ and that the equality occurs for "two-groups" instances. Since $r_{10} = 5/4$, using fixed $\beta = 1/2$ should be roughly 25% slower than using IDS. While Figure 7 confirms that using IDS yields better empirical performance than using fixed $\beta = 1/2$, the empirical ratio between their performance is lower than the one suggested by the theory.

Table 4: Comparison between the original sampling rule and the modified sampling rule for BAI algorithms when combined with the GLR$_\varepsilon$ stopping rule (3) using $(\varepsilon, \delta) = (0.1, 0.01)$. We display the empirical stopping time (and standard deviation) on random instances with $K \in \{5, 10, 20\}$.

| $K$ | T3C | $\varepsilon$-T3C | EB-TCI | $\varepsilon$-EB-TCI |
|---|---|---|---|---|
| 5 | 9138 ($\pm$7988) | 2259 ($\pm$1243) | 22505 ($\pm$66624) | 2290 ($\pm$1202) |
| 10 | 17418 ($\pm$9726) | 3793 ($\pm$1524) | 58086 ($\pm$135655) | 3975 ($\pm$1509) |
| 20 | 30771 ($\pm$13038) | 6631 ($\pm$1992) | 115861 ($\pm$195797) | 6905 ($\pm$1879) |

### J.2.2 Comparison with modified BAI algorithms

Appendix J.2.2 is meant to provide further empirical evidence that EB-TC$_{\varepsilon_0}$ outperforms its competitors on different tasks and for different types of instances. Overall, we will use the same instances as used in Appendix J.2.1 since they provide a wide range of interesting problems. Throughout this section, we will be using EB-TC$_{\varepsilon_0}$ with slack $\varepsilon_0 = 0.1$. While IDS proportions are used for the experiments on the empirical stopping time, we use fixed $\beta = 1/2$ for the empirical simple regret.

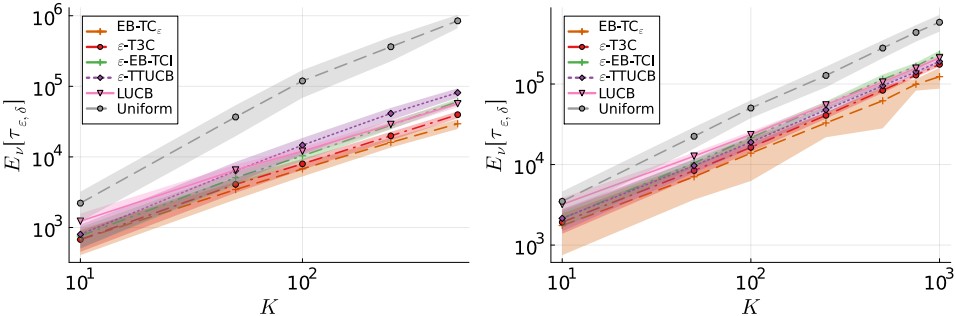

Figure 8: Empirical stopping time on (a) "$\alpha = 0.6$" instances and (b) "sparse" instances for varying $K$ and stopping rule (3) using $(\varepsilon, \delta) = (0.1, 0.01)$. The BAI algorithms T3C, EB-TCI and TTUCB are modified to be $\varepsilon$-BAI ones.

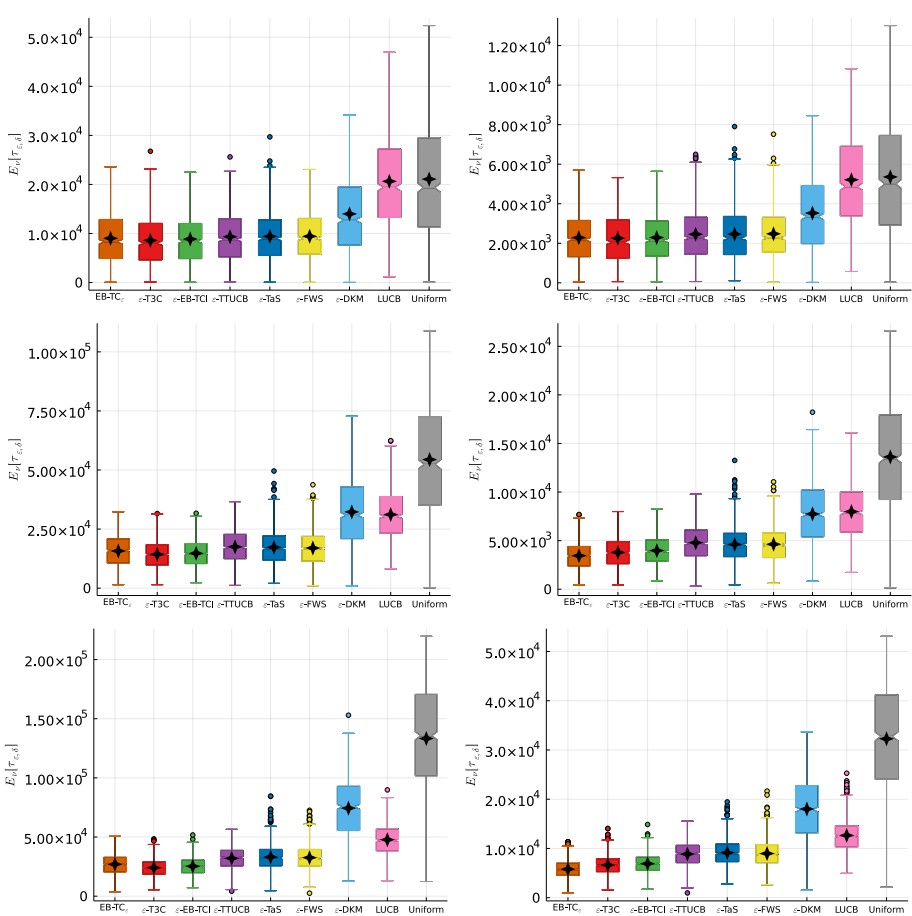

Figure 9: Empirical stopping time on random instances with stopping rule (3) using (left) $(\varepsilon, \delta) = (0.05, 0.01)$ and (right) $(\varepsilon, \delta) = (0.1, 0.01)$ with (top) $K = 5$, (middle) $K = 10$ and (bottom) $K = 20$. The BAI algorithms T3C, EB-TCI, TTUCB, TaS, FWS, DKM are modified for $\varepsilon$-BAI.

As benchmarks we consider modified BAI which are tailored to tackle $\varepsilon$-BAI, and we combine them with the $\text{GLR}_\varepsilon$ stopping rule. One can still wonder whether the unmodified sampling rule would perform well for $\varepsilon$-BAI when combined with the $\text{GLR}_\varepsilon$ stopping rule. While we could hope that modifying the stopping rule is enough, our experiments reveal that it is not the case. If we don't adapt the sampling rule as well, Table 4 shows that the empirical stopping time can be more than ten times larger than their modified version.

**Large sets of arms**  To complement Figure 2(a), we consider the "$\alpha = 0.6$" instances and "sparse" instances for varying $K$. We average on 100 runs, and the standard deviations are displayed.

Figure 8 confirms the observations made in Figure 2(a). We see that EB-TC$_\varepsilon$ performs on par with the $\varepsilon$-T3C, and outperforms the other algorithms. It also highlights that the regularization ensured by the TC$\varepsilon$ challenger is sufficient to ensure enough exploration. As a consequence, other exploration mechanism are superfluous when using Top Two algorithms for $\varepsilon$-BAI (e.g. using TS/UCB leader or TCI challenger).

**Random instances**  For $\varepsilon \in \{0.05, 0.1\}$, we assess the performance on 1000 random Gaussian instances with $K \in \{5, 10, 20\}$ as described above. We display the boxplots of the empirical stopping time on 1000 runs.

Figure 9 confirms that EB-TC$_{\varepsilon_0}$ performs on par with the state-of-the-art $\varepsilon$-BAI algorithms and greatly outperform $\varepsilon$-DKM, LUCB and uniform sampling. Interestingly, for larger sets of arms ($K = 20$), EB-TC$_{\varepsilon_0}$ appears to be more robust than its competitors, closely followed by $\varepsilon$-T3C and $\varepsilon$-EB-TCI. The performance $\varepsilon$-TTUCB, $\varepsilon$-TaS and $\varepsilon$-FWS is slighlty worse, while the one of $\varepsilon$-DKM is greatly impacted. LUCB seems relatively robust to larger sets of arms, but it is still significantly worse than EB-TC$_{\varepsilon_0}$. It is even the best when considering slack $\varepsilon_0 = \varepsilon$ in the bottom row of Figure 9. The slightly worse (but still competitive) empirical performance of EB-TC$_{\varepsilon_0}$ in the top row of Figure 9 is explained by the slack $\varepsilon_0 = 0.1$ which differs from the error $\varepsilon = 0.05$ considered in the $\text{GLR}_\varepsilon$ stopping rule.

**Specific instances**  To complement Figure 2(b), we consider the two other instances from Table 3. We display the boxplots of the empirical stopping time on 1000 runs, and the empirical simple regret on 10000 runs (with associated standard deviation).

The left column of Figure 10 validates the previous conclusions as regards the good empirical performance of EB-TC$_{\varepsilon_0}$ compare to existing $\varepsilon$-BAI algorithms. We also see that it seems to perform even better when there are multiple $\varepsilon$-good arms and multiple best arms, i.e. $|i^\star(\mu)| > 1$. The right column of Figure 10 corroborates the observations made in Figure 2(b): EB-TC$_{\varepsilon_0}$ outperforms uniform sampling, as well as DSR and DSH.

**Two groups**  We consider the "two-groups" instances $\mu \in \{0.6, 0.4\}^{10}$ for varying number of best arms, i.e. $|i^\star(\mu)| \in [8]$. We average on 1000 runs, and the standard deviations are displayed.

Figure 11 reveals that EB-TC$_{\varepsilon_0}$ is the most robust to increased number of best arms, i.e. its slope is the smallest. The good performance of EB-TC$_{\varepsilon_0}$ is closely followed by other Top Two algorithms. This confirms that additional regularization mechanisms (e.g. TS/UCB leader or TCI challenger) are superfluous since the TC$\varepsilon$ challenger is enough to ensure sufficient exploration. $\varepsilon$-TaS and $\varepsilon$-FWS have slightly worse slope, and the one of $\varepsilon$-DKM seems to become less steep after a large initial increase. The highest slope is achieved by LUCB. Quite surprisingly, uniform sampling seems to reach a plateau after which it is scaling better than EB-TC$_{\varepsilon_0}$ (in terms of slope), while still having worse empirical performance. It would be interesting to quantify theoretically the good scaling of EB-TC$_{\varepsilon_0}$ with respect to increased number of best arms.

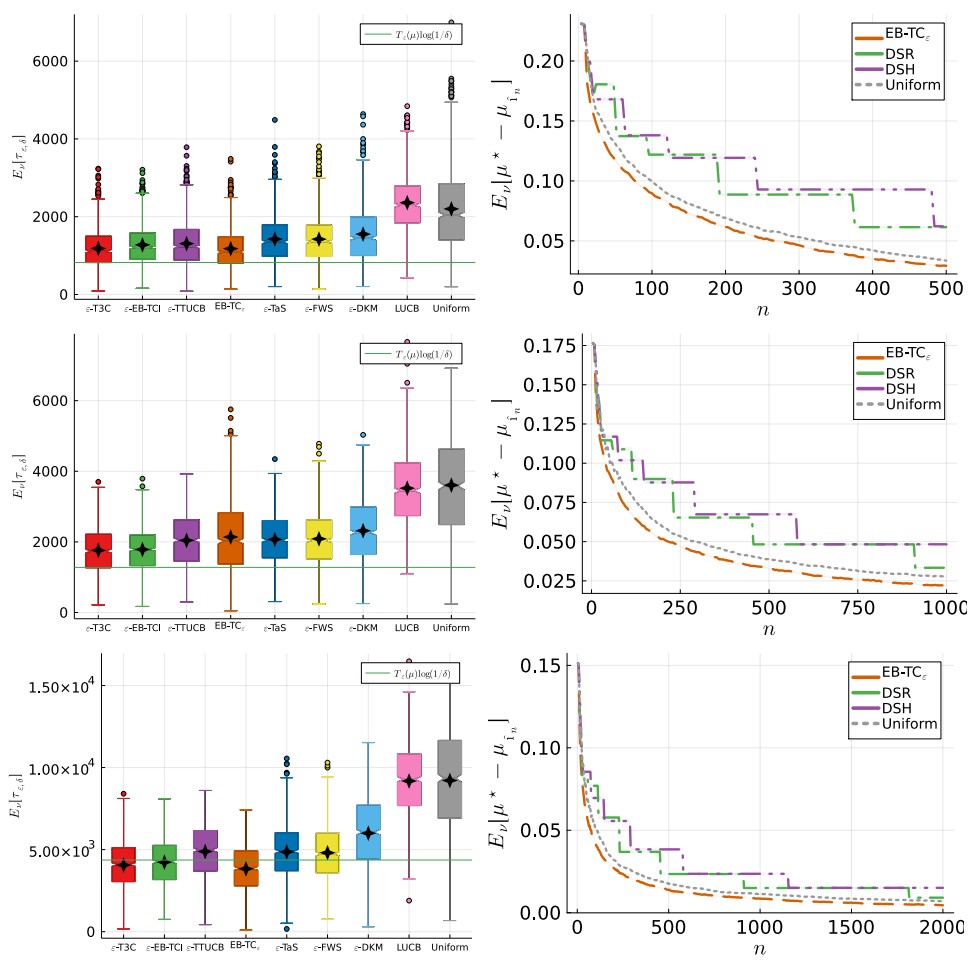

Figure 10: (Left) Empirical stopping time for the stopping rule (3) using $\delta = 0.01$ and (right) empirical simple regret on instances (top) $\mu_1$ with $\varepsilon = 0.1$, (middle) $\mu_2$ with $\varepsilon = 0.15$ and (bottom) $\mu_3$ with $\varepsilon = 0.1$. The BAI algorithms T3C, EB-TCI, TTUCB, TaS, FWS, DKM are modified to be $\varepsilon$-BAI ones.

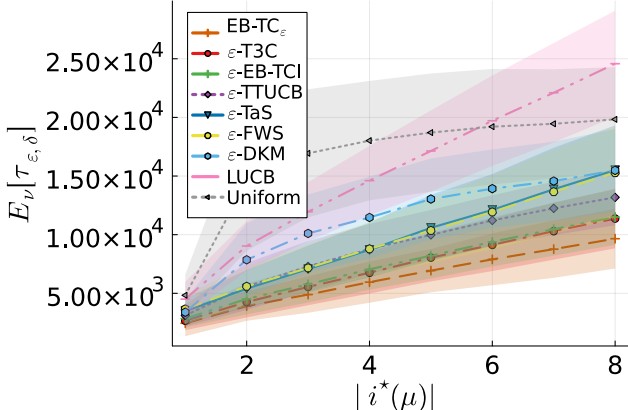

Figure 11: Empirical stopping time with stopping rule (3) using $(\varepsilon, \delta) = (0.1, 0.01)$ on instances $\mu \in \{0.6, 0.4\}^{10}$ for varying $|i^\star(\mu)|$. The BAI algorithms T3C, EB-TCI, TTUCB, TaS, FWS, DKM are modified to be $\varepsilon$-BAI ones.

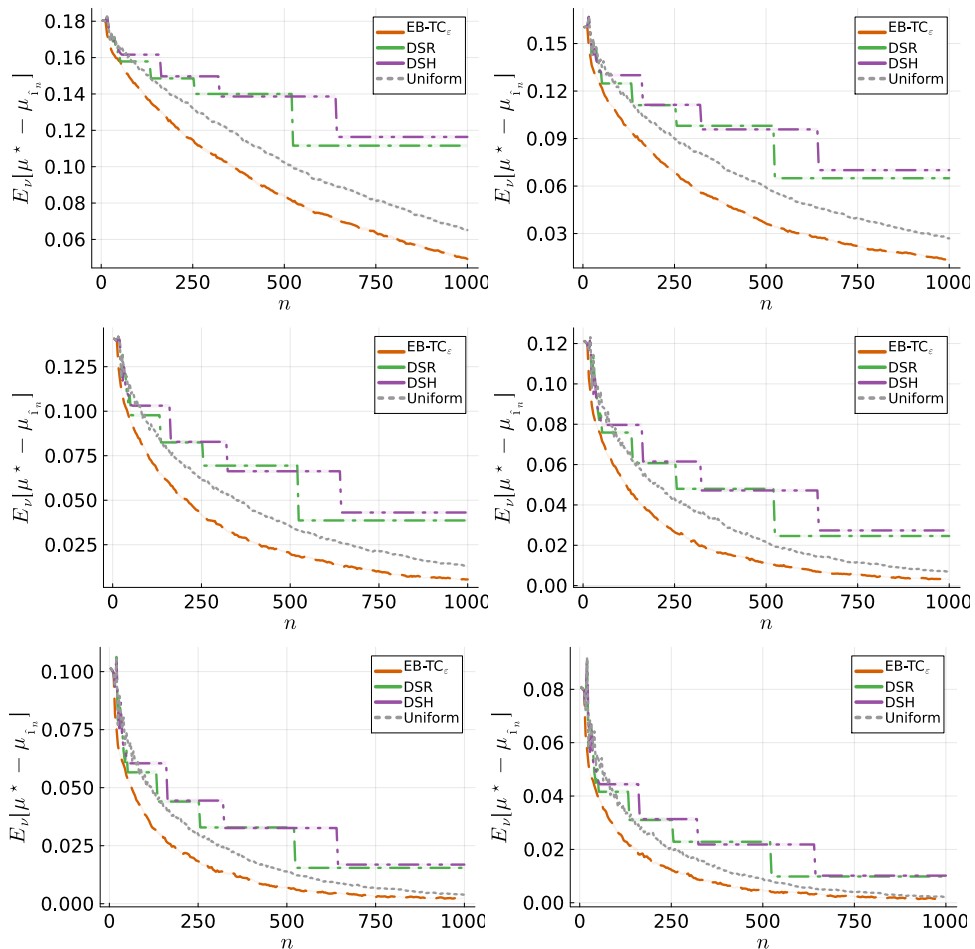

Figure 12: Empirical simple regret on instances $\mu \in \{0.6, 0.4\}^{10}$ for $|i^{\star}(\mu)| \in [6]$, i.e. (top left) $|i^{\star}(\mu)| = 1$ and (bottom right) $|i^{\star}(\mu)| = 6$.

Figure 12 provides additional empirical validation that EB-TC$_{\varepsilon_0}$ outperforms uniform sampling as well as DSR and DSH in terms of empirical simple regret. While the empirical stopping time increased with the number of best arms, Figure 12 confirms the intuition that the problem gets easier with respect to the simple regret. First, the range of empirical simple regret becomes smaller with increased number of best arms. This was expected as we have more chances to recommend one of the best arms at early stage even though this recommendation is close to random. Second, we observe a steeper decrease of the empirical simple regret with increased number of best arms. Likewise, this is a natural phenomenon since the collected data will reveal that there multiple good options very fast.

Quite surprisingly, EB-TC$_{\varepsilon_0}$ is still better than DSH when the number of best arms increases. To understand why it is surprising, we recall that the exponential decrease of DSH's probability of error is linear with a rate proportional to a hardness constant $H_{\mathrm{DSH}}(\mu)$. By definition, $H_{\mathrm{DSH}}(\mu)$ gets small when there are multiple best arms, hence we would expect to observe lower empirical simple regret (i.e. a "speed-up"). In contrast, our hardness constant $H_1(\mu, \varepsilon_0) = K(2\Delta_{\min}^{-1} + 3\varepsilon_0^{-1})^2$ does not enjoy such property. Therefore, we would expect that DSH will outperform EB-TC$_{\varepsilon_0}$ when the number of best arms increases. Theoretically proving that EB-TC$_{\varepsilon_0}$ has a better scaling than the one given in Theorem 3 is still an open problem. Solving it would allow to better understand its good empirical performance on this task.

**Fixed-budget performance** In addition, we compare the performance of EB-TC$_{\varepsilon_0}$ with $\varepsilon_0$ and $\beta = 1/2$ to the one of SR and SH. Since SR and SH are fixed-budget algorithms, we ran a different instance for each budget $T$. Therefore, this comparison gives an unfair advantage to the fixed-budget

algorithms which fully leverage the knowledge of $T$ but have no theoretical guarantees at time $n \neq T$ (even at time $T \pm 1$). As a result, the empirical performances of SR and SH are better than the ones of DSR and DSH. We consider the instances from Table 3, as well as "two-groups" instances.

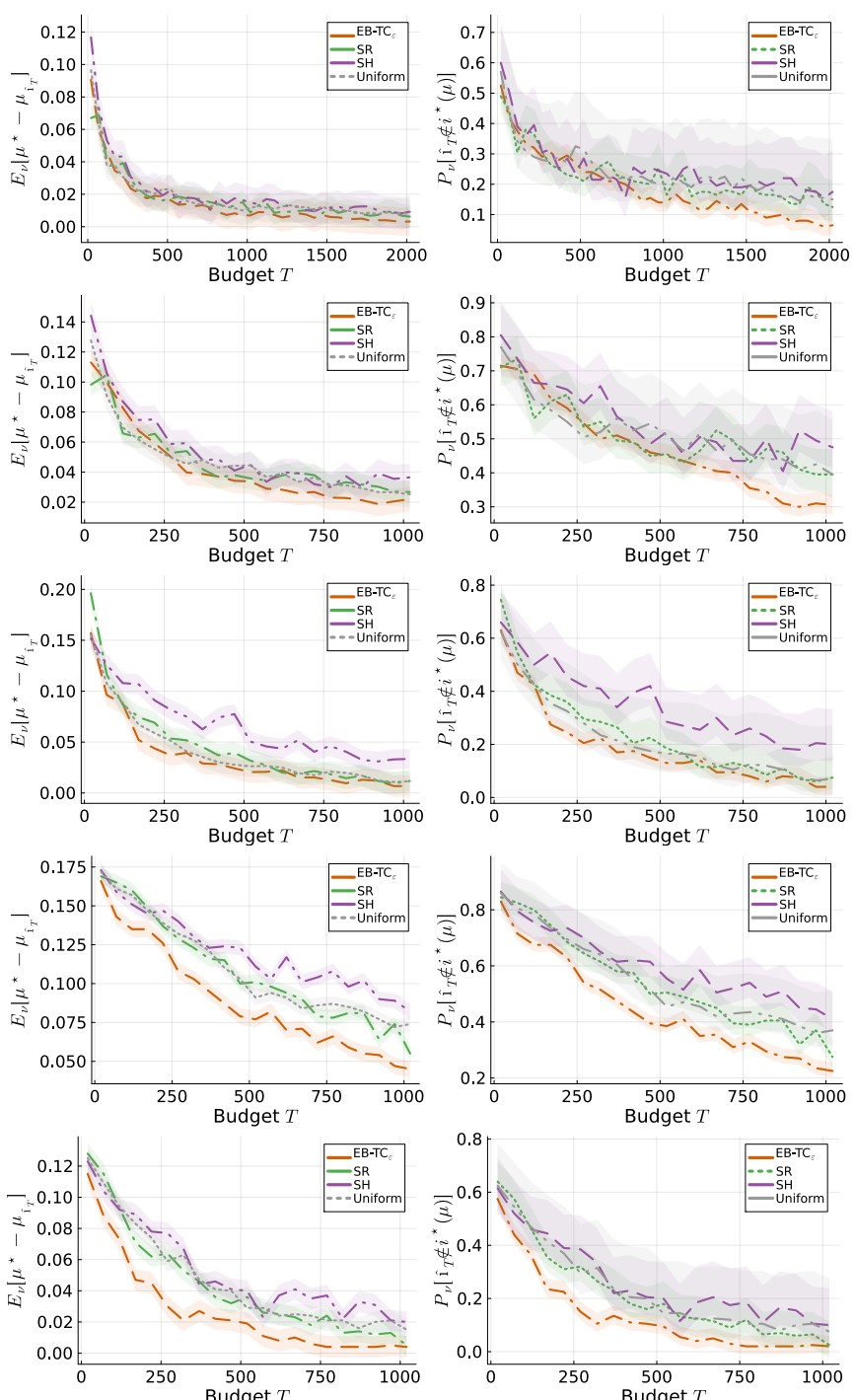

Figure 13: (Left) Empirical simple regret and (right) empirical error on instances (top to bottom) $\mu_3 = (0.6, 0.6, 0.55, 0.45, 0.3, 0.2)$, $\mu_2 = (0.8, 0.75, 0.7, 0.6, 0.5, 0.4)$, $\mu_1 = (0.7, 0.55, 0.5, 0.4, 0.2)$ and $\mu \in \{0.6, 0.4\}^{10}$ with $|i^\star(\mu)| \in \{1, 3\}$. Average over 200 runs. We ran a different instance of SR and SH for each budget $T$.

Overall, according to Figure 13, EB-TC$_{\varepsilon_0}$ seems to perform on par with SR and SH in all instances considered, and to outperform SH for some instances. In our experiments, we consider SH in which the samples are dropped between each phase, as analyzed theoretically in [1]. This loss of information explains why SR has better empirical performance than SH. To the best of our knowledge, there is no theoretical analysis for the heuristic SH where all the samples are preserved, even though it has better empirical performance.

### J.2.3 Varying slack parameter

In Appendix J.2.3, we study the BAI problem. Those experiments have two goals: (1) sensitivity analysis of EB-TC$_{(\varepsilon_n)_n}$ described in Section 3 and (2) performance assessment of EB-TC$_{\varepsilon_0}$ on BAI problems. We consider fixed proportions $\beta = 1/2$, and use $\varepsilon_0 = 0.1$ for EB-TC$_{\varepsilon_0}$. For the rate of decrease of $(\varepsilon_n)_n$, we will be considering two archetypal choices: (a) polynomial by taking $\varepsilon_n = n^{-\alpha/2}$ and (b) polylogarithmic by taking $\varepsilon_n = \log(n)^{-\alpha/2}$. We compare empirically those two rates of decrease for different choices of $\alpha$, namely $\alpha \in \{0.05, 0.1, 0.5\}$. To tackle BAI, we consider the GLR$_0$ stopping rule (3) with $(\varepsilon, \delta) = (0, 0.01)$ and the heuristic threshold $c(n, \delta) = \log((1 + \log n)/\delta)$. Even though this choice is not sufficient to prove $(0, \delta)$-PAC, it yields an empirical error which is several orders of magnitude lower than $\delta$. As benchmark, we use the unmodified BAI algorithms.

**Random instances** We assess the performance on 1000 random Gaussian instances with $K \in \{5, 10, 20\}$ such that $\mu_1 = 1$ and $\mu_i \sim \mathcal{U}([0.5, 08])$ for $i \neq 1$. We display the boxplots of the empirical stopping time on 1000 runs.

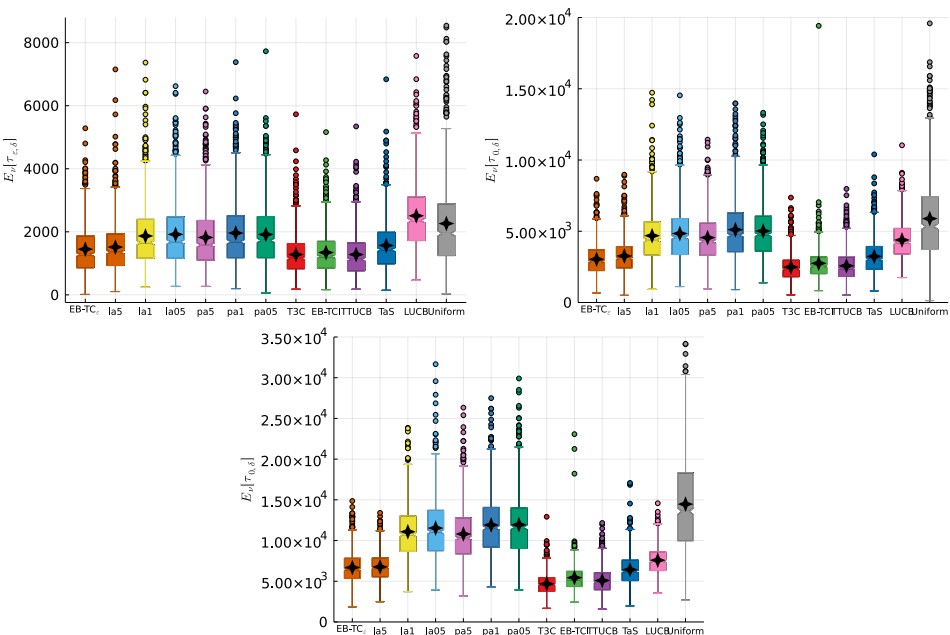

Figure 14: Empirical stopping time for the stopping rule (3) using $(\varepsilon, \delta) = (0, 0.01)$ on random instances with (a) $K = 5$, (b) $K = 10$ and (c) $K = 20$. "l" denotes $\varepsilon_n = \log(n)^{-\alpha/2}$, "p" denotes $\varepsilon_n = n^{-\alpha/2}$. "a··" refers to $\alpha \in \{0.5, 0.1, 0.05\}$.

The most stricking feature of Figure 14 is that it reveals how poor the performance of EB-TC$_{(\varepsilon_n)_n}$ can be for many choices of $(\varepsilon_n)_n$. For the considered $\alpha$, we observe that polynomial decrease is always bad since it yields roughly the same empirical performance as uniform sampling. This can be explained by the fact that this decrease is too fast, hence EB-TC$_{(\varepsilon_n)_n}$ will ressemble EB-TC$_0$ which is know to have poor empirical performance [21]. For the considered $\alpha$, we see that polylogarithmic decrease is often bad, except for $\alpha = 0.5$. In that case, it performs on par with TaS, slightly worse than standard Top Two algorithms for BAI and better than LUCB and uniform sampling. This can be explained by the fact that EB-TC$_{(\varepsilon_n)_n}$ is similar to uniform sampling when the decrease is too slow. The above sensitivity analysis truly shows how difficult it is to choose $(\varepsilon_n)_n$ beforehand. The

best "trade-off" between a slow decrease (yet not too slow) highly depends on the complexity of the unknown instance. While $\text{EB-TC}_{(\varepsilon_n)_n}$ reaches asymptotic optimality for BAI, we recommend the practitioner to use other Top Two algorithms which enjoy the same theoretical guarantees and better empirical performance. When one needs to have a deterministic algorithms, EB-TCI seems to be the best. Without the deterministic constraint, T3C has great performance.

In Figure 14, we also see that, when combined with the $\text{GLR}_0$ stopping rule, $\text{EB-TC}_{\varepsilon_0}$ has good empirical performance for BAI. It outperforms $\text{EB-TC}_{(\varepsilon_n)_n}$, performs on par with TaS and is only slightly worse than standard Top Two algorithms for BAI. Experiments conducted in Appendix J.2.1 already provide experimentally confirmation that $\text{EB-TC}_{\varepsilon_0}$ has still good empirical performance on $\varepsilon$-BAI problems when $\varepsilon_0 \geq \varepsilon$. Theoretically, Theorem 1 provide some intuition on why this is true. It shows that the gap between the asymptotic lower bound $T_0(\mu)$ and the asymptotic upper bound of $\text{EB-TC}_{\varepsilon_0}$ is bounded by

$$\frac{T_{\varepsilon_0, 1/2}(\mu)}{T_0(\mu)} \left(1 + \frac{\varepsilon_0}{\Delta_{\min}}\right)^2 \leq 4 \frac{\sum_{i \neq i^\star}(\Delta_i + \varepsilon_0)^{-2}}{\sum_{i \neq i^\star} \Delta_i^{-2}} \left(1 + \frac{\varepsilon_0}{\Delta_{\min}}\right)^2 \approx \begin{cases} 4 & \text{if } \varepsilon_0 \gg \Delta_{\min} \\ 4 & \text{if } \varepsilon_0 \ll \Delta_{\min} \end{cases},$$

where we used Lemmas 6, 9 and 7.

**Specific instances** We consider the two instances $\mu_1$ and $\mu_2$ from Table 3 with $\varepsilon = 0$. Since most theoretical guarantees on BAI algorithms assume that there is a unique best arm, we don't study $\mu_3$. We display the boxplots of the empirical stopping time on 1000 runs.

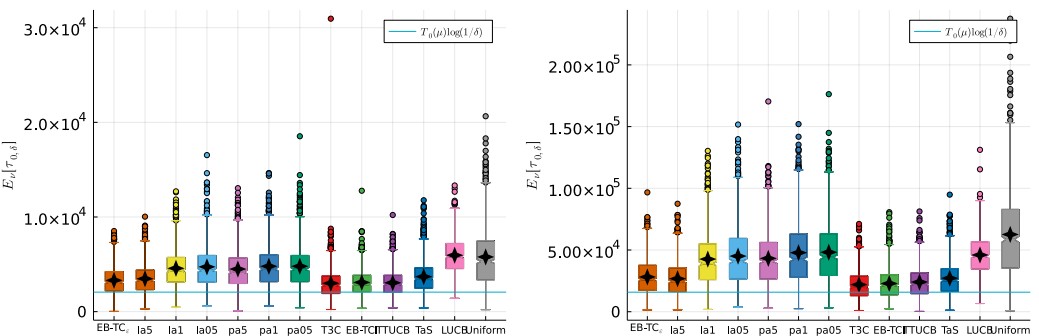

Figure 15: Empirical stopping time for the stopping rule (3) using $(\varepsilon, \delta) = (0, 0.01)$ on instances (a) $\mu_1$ and (b) $\mu_2$. "l" denotes $\varepsilon_n = \log(n)^{-\alpha/2}$, "p" denotes $\varepsilon_n = n^{-\alpha/2}$.

Figure 15 confirms the empirical observations from Figure 14. Overall, $\text{EB-TC}_{(\varepsilon_n)_n}$ performs poorly and $\text{EB-TC}_{\varepsilon_0}$ has good empirical performance for BAI.

