# OpenReview forum: "An $\varepsilon$-Best-Arm Identification Algorithm for Fixed-Confidence and Beyond"
_NeurIPS.cc/2023/Conference — NeurIPS 2023 poster_

### Official Review · Reviewer_kKhf · 2023-07-04

**Soundness:** 3 good
**Presentation:** 3 good
**Contribution:** 2 fair
**Rating:** 6
**Confidence:** 3

**Summary:**

The paper gives a new epsilon-best arm identification algorithm which is simple and anytime, just amounting to balancing between the empirical best arm and roughly speaking, the arm with the best chance to be superior to it by epsilon. Its performance is analyzed mainly for fixed confidence, and it also attains decent (not particularly optimal) bounds for fixed budget. The results assume a unique best arm but allow repeated mean values among the other arms.



**Strengths:**

The fixed confidence bounds are very good (sharp for Gaussians) and the algorithm is nice and simple. The epsilon parameter can be mismatched, though this degrades the bounds a bit.

**Weaknesses:**

No new minimax regret bounds are obtained. As I understand it the paper is basically giving a simpler algorithm with some nice guarantees. Which is fine but not the most revolutionary.



**Questions:**

Maybe ``\beta-optimal'' could be defined more carefully, and the ratio bound of T_{\eps} vs T_{\eps,\beta} could be stated near the definitions.

Just before line 200, it would be nice to cite a few of these other works on Top Two algorithms.

---

> ### Author Rebuttal · Authors · 2023-08-07
>
> We thank the reviewer for the feedback and questions. We hope to address them in the following.
>
> We first answer the points raised in the “weaknesses” section.
>
> **Minimax regret bounds.**
> Given that we are considering the $\varepsilon$-BAI setting, we are not sure to understand what the reviewer meant by “minimax regret bounds” as those bounds are only meaningful in the regret minimization literature. In this work, we consider pure exploration and focus on problem-dependent bounds. A (nearly) minimax optimal algorithm for minimizing simple regret is uniform sampling as proved in [5] so there is not much more to say in this setting. We consider the fixed-confidence setting, in which we compare ourselves to the tight asymptotic problem-dependent lower bound given in Lemma 1, and we compare ourselves to known problem-dependent results in the finite-confidence and anytime regimes.
>
> **Simple algorithm with nice guarantees: revolutionary or not ?**
> We truly believe that it is a valuable contribution to study simple algorithms that are (1) easy to understand and implement, (2) empirically competitive (3) robust to different formulations of the pure exploration task (see also our answer to reviewer B3zV). Our work is the first to propose a single algorithm that is simultaneously asymptotically optimal in the fixed-confidence $\varepsilon$-BAI setting (Theorem 1), has finite-confidence guarantees (Theorem 2), and has also anytime guarantees on the probability of error at any level $\varepsilon$ (Theorem 3), hence on the simple regret (Corollary 1). Such simple and versatile algorithms are the ones that will be most likely useful by practitioners. While we don’t claim to be revolutionary, we still consider it to be a significant contribution to the pure exploration literature.
>
> We now answer your other questions.
>
> **Definition $\beta$-optimality.**
> We will add explicitly the definition of ($\beta-$)optimal below Lemma 1 and add the worst-case inequality $T_{\epsilon,1/2}(\mu) \le 2 T_{\varepsilon}(\mu)$ there as well.
>
> **Citing other Top Two algorithms.**
> We provided multiple references to other works on Top Two algorithms throughout the paper (e.g. in Section 2.1). For the sake of completeness, we will add more references just before line 200.

---

> > ### Comment · Reviewer_kKhf · 2023-08-14
> >
> > Thanks for the reply. By "minimax" I just meant worst-case rather than Bayesian, sorry for the confusion.
> >
> > I agree that the contribution is nice and worthwhile and that simplicity is nice to have. My intent was just to clarify that if this were a super simple algorithm which also beat the state of the art significantly, it would merit an even higher score.
> >
> > I'll keep my score unchanged for now and don't have further questions.

---

### Official Review · Reviewer_PH2Y · 2023-07-06

**Soundness:** 3 good
**Presentation:** 3 good
**Contribution:** 3 good
**Rating:** 6
**Confidence:** 3

**Summary:**

This work proposed the EB-TC algorithm for $\varepsilon$-best arm identification in stochastic bandits.
EB-TC is a BAI algorithm that does not require the knowledge of confidence or horizon, and hence it can be applied in both fixed-confidence and fixed-budget settings. Beyond theoretical guarantees, this work also provided numerical experiments to evaluate the performance of EB-TC.



==========================

Thanks for the response. I would like to keep the score.

**Strengths:**

1. The paper is generally easy to follow.
2. EB-TC seems to be a nice BAI algorithm that does not need to know confidence or horizon as priori knowledge, which can be applied in both fixed-confidence and fixed-budget settings.
3. EB-TC can be viewed as a generalization of the Top Two Sampling rule which was designed in an earlier work.
4. The paper provides a clear comparison to existing works.

**Weaknesses:**

As discussed in Section 2.1, the EB-TC algorithm may not perform well to identify the optimal arm. Can this issue be solved?

**Questions:**

As mentioned in the Weakness section: what is the key obstacle such that EB-TC cannot work well for the identification of the optimal arm? I appreciate some explanation of this point. A (possibly) suboptimal theoretical bound is even better.

---

> ### Author Rebuttal · Authors · 2023-08-07
>
> We thank the reviewer for the feedback and question. We hope to address it in the following.
>
> **Performance of EB-TC$_{0}$ in BAI.**
> It is well known from prior work that EB-TC empirically performs poorly to identify the optimal arm for moderate $\delta$. This issue can be solved by adding an implicit exploration mechanism in the choice of the leader/challenger pair. For the choice of leader, we can use randomization (TS leader [33,31,35]) or optimism (UCB leader [19]). For the choice of the challenger, we can use randomization (RS challenger [33]) or penalization (TCI challenger [20], KKT challenger [38] or EI challenger [31]). A contribution of our work is precisely to show that for $\epsilon_0$-BAI with $\epsilon_0>0$, there is no need for an additional exploration mechanism since the slack $\varepsilon_0$ used by EB-TC$_{\epsilon_0}$ naturally induces sufficient exploration. Our experiments highlight that the extra exploration mechanisms discussed above are indeed superfluous for approximate best arm identification (see lines 343-345).
>
> **Understanding the lack of robustness of EB-TC$_{0}$.**
> To better understand the lack of robustness of EB-TC in BAI, let us consider 3-arms bandits with means $\mu = (x, 0, 0)$ where $x > 0$. For those instances, we call “unlucky first draws” any situation where after gathering a couple of samples $n_{0}$, we have $\mu_{n_{0},1} << 0$ and $\mu_{n_{0},2}$ $\approx$ $\mu_{n_{0},3} >> 0$. The probability of this event depends on $n_0$ and $x$, yet it is fixed and bounded away from $0$. Confronted with this situation, EB-TC will choose its leader/challenger pair among the two sub-optimal arms, and never sample the best arm. Since $\mu_{n_{0},1} << 0 = \mu_{2} = \mu_{3}$, the situation will not change after both sub-optimal arms have converged to their true value: (1) one of them will be leader and (2) the second one will be the challenger since the mean gap tends to $0$.
> Now, let’s consider a slightly modified $3$-arms bandit with mean parameter $\mu = (x, 0, -\kappa)$ where $x >> \kappa > 0$. Similarly as above, under the “unlucky first draws” event, EB-TC will choose its leader/challenger pair among the two sub-optimal arms, without sampling the best arm for a long time. In contrast to the above example, this instance has distinct means hence there is a different behavior after the convergence of the empirical means of the two suboptimal arms. After a very large number of samples, the leader will be the second arm and the challenger will be the first arm. Compared to above, the third arm cannot be the challenger since its TC will be roughly equal to $n \kappa^2/2$ which is larger than the TC of the first arm which is roughly $n_{0} \mu_{n_{0},1}^2$. By comparison between both TC, we see that the time necessary for the algorithm to stop being stuck and start sampling the arm $1$ is proportional to $1/\kappa^2$. Therefore, we conjecture it is possible to show a probabilistic lower bound on the sample complexity of EB-TC that involve the minimal gap between arms $\Delta_{\min} = \min_{i \ne j} |\mu_{i} - \mu_{j}|$ with a dependency of the order of $O(1/\Delta_{\min}^2)$.
> The above discussion explains why EB-TC is asymptotic optimality for BAI under the distinct mean assumption, but has very poor performance in the finite-confidence performance. For more details on the limitations of EB-TC, we refer to Appendix D.3 [20] in which the algorithm is introduced and where its lack of robustness is discussed.

---

> > ### Comment · Reviewer_PH2Y · 2023-08-20
> > **Thanks for your response.**

---

### Official Review · Reviewer_KTru · 2023-07-06

**Soundness:** 4 excellent
**Presentation:** 2 fair
**Contribution:** 4 excellent
**Rating:** 7
**Confidence:** 2

**Summary:**

This paper proposes an anytime $\epsilon$-best-arm identification algorithm for multi-armed bandits that can be employed in both fixed-confidence and fixed-budget setting. The algorithm is analyzed in asymptotic fixed-confidence, non-asymptotic fixed-confidence and fixed-budget settings. Meanwhile, the paper also provides empirical results in many different settings.

**Strengths:**

- The proposed algorithm is anytime in nature and thus widely applicable.
- The analysis for fixed $\beta$ is thorough and covers all asymptotic fixed-confidence, non-asymptotic fixed-confidence and fixed-budget settings.
- The empirical experiments are extensive.

**Weaknesses:**

- The analysis for the IDS variant is limited.
- This may be just due to my misunderstanding, but it seems the expected stopping time of the proposed algorithm with finite confidence $\delta>0$ will explode for BAI ($\epsilon=0$) problem.

### Suggestions on Writing
- It may be better to also give the formula of $\bar{\beta}_{n+1}(i, j)$ under the IDS variant in the algorithm summary.
- Do we simply have $T_{\epsilon_0, \beta}(\mu)=\min_{i\in\mathcal{I}\_{\epsilon_0}(\mu)}T_{\epsilon_0, \beta}(\mu, i)$ in the statement of Theorem 1? It seems this quantity is not formally defined in the main paper.
- It seems the statement of Theorem 1 does not mention that the result for IDS variant only holds for Gaussian distribution.
- If possible, it may be better to present results in the main paper more in terms of big-$\mathcal{O}$ instead of exact formula.

**Questions:**

- How does the proposed algorithm perform, compared with uniform sampling, SH and SR, in fixed-budget BAI ($\epsilon=0$) problem?
- Based on Theorem 1, the proposed algorithm is asymptotically optimal for 0-BAI when taking $\epsilon_0=0$. However, based on Theorem 2, it seems the expected stopping time just explode when taking $\epsilon_0=0$ with positive $\delta>0$. Is this a contradiction? Did I misunderstand anything? How does the algorithm perform in 0-BAI with finite $\delta>0$?

**Limitations:**

The limitations are addressed well in this paper.

---

> ### Author Rebuttal · Authors · 2023-08-07
>
> We thank the reviewer for the feedback and questions. We hope to address them in the following.
>
> We first answer the points raised in the “weaknesses” section.
>
> **Limited analysis of IDS.**
> Extending our other results (Theorems 2 and 3) to EB-TC$_{\varepsilon}$ with IDS proportions is an interesting direction for future research that we are exploring. Our experiments suggest that this algorithm enjoys smaller empirical stopping time and similar empirical simple regret. It would be valuable to quantify those improvements theoretically.
>
> **Theorem 1 holding for sub-Gaussian distributions.**
> Theorem 1 holds for all $\nu \in \mathcal D^{K}$ with mean $\mu$ such that $|i^\star(\mu)|=1$, where $\mathcal D$ is the set of $1$-sub-Gaussian distribution, also for its IDS version. As mentioned in lines 184-187, the results are only asymptotically ($\beta$-)optimal in the family of Gaussian distributions. For more general sub-Gaussian distributions, our asymptotic upper bound still holds, yet it is suboptimal compared to the best achievable lower bound.
>
> We now answer your other questions.
>
> **Expected stopping time for $\varepsilon_{0} = 0$.**
> Theorem 1 does not allow $\varepsilon_0 = 0$. With $\varepsilon_0 = 0$ we recover the EB-TC algorithm for BAI (see [20]), which is asymptotically $\beta$-optimal if all arms have distinct means, but can be arbitrarily bad if that’s not the case (see Appendix  D.3 in [20]). Compared to existing asymptotic results on Top Two algorithms, Theorem 1 is the first one removing the assumption that all arms have distinct means. The regularization of the $\varepsilon_0$ parameter allowed us to remove it with little modification to the original proof. To remove this assumption in BAI (i.e. $\varepsilon = 0$), we would require an additional mechanism to ensure sufficient exploration. While we conjecture that the TS or UCB leader and the TCI challenger would be enough, we are certain that EB-TC can get stuck for instances having all sub-optimal arms with the same mean. Intuitively, it cannot recover from unlucky first draws and the leader/challenger will alternate among the sub-optimal arms.
> Theorem 2 does not allow $\varepsilon_0 = 0$. In Theorem 2, the sample complexity indeed explodes when $\varepsilon_0 \to 0$. When the arms have distinct means, we emphasize the Theorem 2 and the asymptotic optimality of EB-TC are proved using very different tools, so that the fact that Theorem 2 cannot be used to prove asymptotic optimality when letting $\delta$ go to zero is not a contradiction. The theoretical intuition and empirical studies made in [20] are abounding towards the same observation: EB-TC can truly get stuck in a finite-confidence regime, hence have dramatically poor empirical performance.
>
> **Empirical performance in the fixed-budget setting.**
> EB-TC$\varepsilon$, uniform sampling, DSR and DSH are all anytime algorithms. The fixed-budget performance of an anytime algorithm is the same as its anytime performance. Therefore, our experiments provide empirical evidence of the superior performance of EB-TC$\varepsilon$ compared to uniform sampling, DSR and DSH as regards the simple regret (see Figures 2(b), 6, 10, 12). We displayed the simple regret metric since it is an aggregated metric with a more practical meaning.
> Comparing an anytime algorithm with a fixed-budget algorithm with budget $T$ gives an unfair advantage to the fixed-budget algorithms which fully leverage the knowledge of $T$ but have no theoretical guarantees at time $n \ne T$ (even at time $T \pm 1$). Nevertheless, we performed additional experiments to answer your question. In the separate pdf available in the “global” response, we compared EB-TC$\varepsilon$ with SR and SH on different instances. For SR and SH, we emphasize that each point corresponds to a different instance which leverages the knowledge of the budget $T$. As a result, the empirical performances of SR and SH are better than the ones of DSR and DSH. Overall, EB-TC$\varepsilon$ seems to perform on par with SR and SH in all instances considered, and to outperform SH for some instances. In our experiments, we consider SH in which the samples are dropped between each phase, as analyzed theoretically in [1]. This loss of information explains why SR has better empirical performance than SH. To the best of our knowledge, there is no theoretical analysis for the heuristic SH where all the samples are preserved, even though it has better empirical performance.
>
> **Miscellaneous.** Thank you for your suggestions. We will add the reference to the update of $\bar \beta_{n+1}(i,j)$ in Figure 1 and the formal definition of $T_{\epsilon, \beta}(\mu) = \min_{i \in \mathcal I_{\varepsilon}(\mu)} T_{\epsilon, \beta}(\mu, i)$.

---

> > ### Comment · Reviewer_KTru · 2023-08-11
> > **Response**
> >
> > Thank you very much for your rebuttal and most of my concerns have been well-addressed! As a follow-up question, it seems there is a hidden barrier between $\epsilon$-BAI problem and BAI problem such that the results of $\epsilon$-BAI problem cannot be naturally applied to BAI problem by simply setting $\epsilon=0$, which sounds quite counter-intuitive. Is that possible to explain this phenomenon from a high-level perspective? Do you think this hidden barrier between $\epsilon$-BAI and BAI is fundamental? Thanks!

---

> > > ### Author Response · Authors · 2023-08-11
> > >
> > > In the asymptotic fixed-confidence regime and for a fixed $\beta$, EB-TC$_\epsilon$ permits a smooth interpolation between the $\varepsilon$-BAI and the BAI problem, as the algorithm is asymptotically $\beta$-optimal in both cases, provided that all means are distincts (Theorem 1 and Appendix H in [21]). However, in non asymptotic regimes, we believe that there might indeed be some barrier between BAI and $\epsilon$-BAI.  In the finite-confidence regime, in both $\varepsilon$-BAI and even in BAI, it is still an open problem to derive a tight lower bound in the finite-confidence regime. Recent works [7, 36] have shown that the sample complexity in BAI is affected by strong moderate confidence terms (independent of $\delta$). While similar terms probably have their counterparts in $\varepsilon$-BAI, our intuition suggests that having $\varepsilon > 0$ may act as a regularizer preventing those terms to ``explode’’ (as they would have upper bounds scaling in $O(K/\epsilon^2)$).

---

> > > > ### Comment · Reviewer_KTru · 2023-08-11
> > > > **Response**
> > > >
> > > > Thank you very much for sharing your thoughts! For now, I don't have further questions and I'll keep my score unchanged.

---

### Official Review · Reviewer_6z6m · 2023-07-13

**Soundness:** 3 good
**Presentation:** 3 good
**Contribution:** 3 good
**Rating:** 7
**Confidence:** 3

**Summary:**

The paper considers the epsilon-BAI problem in the multi-armed bandit setting with sub-gaussian arm distributions. It proposes a top2 algorithm for the problem that works across different performace criteria: fixed-confidence, fixed-budget, as well as simple-regret. The proposed algorithm is asymptotically optimal (as delta-> 0) for the fixed confidence setting for gaussian bandits, where delta is the bound on probability of errors. The paper also provides finite delta analysis of the upper bound on the sample complexity in the fixed confidence setting. Another novel feature of the proposed algorithm is that it can be used in the fixed-budget setting, without a prior knowledge of the budget. The theoretical results are accompanied with extensive simulations and numerical studies.

**Strengths:**

I believe that the results of this work are novel. The proposed sampling rule, when coupled with either stopping rule or recommendation rule (or both) works across different BAI settings: fixed-confidence, fixed-budget, simple regret. The paper also discusses well and brings out the comparison with existing works and techniques for each of these problems. I especially like the discussions that are included after each result. I believe that the literature review is also thorough. The extensive numerical studies undertaken definitely add value to the theoretical work.

**Weaknesses:**

I believe that the paper is dense and too long for a conference submission with many details in the appendix. But this may be because of the space constraints.

Plots in Figures 6, 10, and 12 are very light. Also the legends are too small to read for most of the plots. In general, it will help to spread out the plots for readability. Since the plots are already in appendix, space isn't an issue.


**Questions:**

1. Could you clarify the definition of simple regret in the epsilon-BAI framework? In line 47, should there also be an indicator on chosen arm not being epsilon-best? In line 97-98, is Delta_{\hat{i}_n} missing in the integration?

2. The lower bound in Lemma 1 is specifically for Gaussian bandits. How does it compare with sub-Gaussian bandits? Is there some monotonicity? It will be good to add a discussion around this. If not, including an example (if known) could help.

3. In Figure 1 (EP-TC_eps algorithm with fixed or IDS) line 3, it will be good to refer to the equation for updating \bar{beta}_{n+1}(i,j) in the main text.

4. It will be helpful if the discussion in lines 150-151 could be elaborated. What are the K(K-1) different tracking rules?  Are these the different choices of beta that are maintained for each leader/challenger pair?

5. How should one choose epsilon_0 in practice?

6.  In lines 201-203, is that a requirement for asymp. optimality, or is it a requirement for proofs to work? Can one show that if number of optimal arms is greater than 1, then the algorithm doesn't converge (may be b/c of some discontinuity)?

7. Lines 235-237: Would using any-delta methods to construct UCB and LCB indexes in LUCB help in this regard?

8. In Lemma 9, Line 530, could you clarify what j is in mu_eps(i)_j ?

9. Is Lemma 12 a standard result? Adding a citation or proof would help.

10. What are the challenges in generalizing the proofs to beyond gaussian arms to exponential families? What specific properties of Gaussians are used in analysis that may not hold for exponential families? Can one pin down to the level that if we can prove this, this, and this about KL (for example) or other functions in exponential families, then we should get the bound?

Minor:
1. Line 207 satisfies these ...
2. Line 587: eeror --> error

---

> ### Author Rebuttal · Authors · 2023-08-07
>
>
> We thank the reviewer for the feedback and questions. We will follow your suggestion and improve the presentation of the plots in the appendix.
>
> **1) Simple regret in $\varepsilon$-BAI.**
> The simple regret considered in this work is the standard simple regret introduced by [5] in the pure exploration literature, which does not depend on any parameter. $\varepsilon$ from $\varepsilon$-BAI is only meaningful when considering probabilistic performance metrics, e.g. the probability of marking an $\varepsilon$-error. The simple regret is an aggregated performance metric (lines 97-98). Therefore, there is no missing indicator in line 47.
> There is no missing $\Delta$ in the formula in l. 97-98, which follows from integrating a positive function.
>
> **2) Lower bound in Lemma 1 for more general distributions than Gaussians.**
> Let $\nu$ be a $\sigma$-sub-Gaussian instance with means $\mu$. On $\nu$, Theorem 1 shows that there exists an algorithm (EB-TC$\varepsilon$) with asymptotic expected sample complexity upper bounded by $T_{\varepsilon}(\mu)$. Therefore, the characteristic time for the class of sub-Gaussian distributions (which does not have a form as “explicit” as (1)) is always smaller than the ones for Gaussian having the same means and variance $\sigma^2$.
>
> **4) Tracking procedures.**
> We define one tracking procedure per pair of leader/challenger $(i,j) \in [K]^2$ such that $i \ne j$, hence we have $K(K-1)$ tracking procedures. For the tracking associated with $(i,j)$, we maintain one average value $\bar \beta_{n}(i,j)$ of proportions. For fixed $\beta$, we have $\beta_{t}(i,j) = \beta$, hence $\bar \beta_{n}(i,j) = \beta$. When using IDS proportions, each tracking procedure has a different target. For example, the tracking procedure of pair $(i,j)$ targets the average proportions $\bar \beta_n (i,j)$ for the counters $N_{n,j}^{i}$ and $T_{n}(i,j)$. Those counters are independent of each other, meaning one cannot increment the counters of pair $(i,j)$ when using a different pair. At each round $n$, only one tracking rule is considered, i.e. the one of the pair $(i,j) = (B_n, C_n)$.
>
> **5) Choice of $\varepsilon_{0}$.**
> We discuss the choice of $\varepsilon_0$ between line 152 and 161, and perform experiments to support our claims (e.g. Figures 5, 6 and 7 in Appendix J.2.1). One should take $\varepsilon_0 = \varepsilon$ when considering fixed-confidence $\varepsilon$-BAI if $\varepsilon > 0$. As regards the simple regret, we recommend choosing $\varepsilon_0$ not too small. Of course for every instance $\mu$ there exists an optimal value of $\varepsilon_0$ minimizing the simple regret at some target time $T$, that depends on both $\mu$ and $T$. We did not investigate the scaling of this “oracle” value of $\varepsilon_0$ as it is not relevant in practice. Instead in our experiments we mostly used the value $\varepsilon_0 = 0.1$ across different benchmarks with means in $[0,1]$. For example Figure 6 reveals that the empirical simple regret stays similar when taking $\varepsilon_0 \in \{0.05,0.1,0.15\}$.
>
> **6) Unique best arm assumption.**
> We did not prove that the algorithm is not asymptotically optimal when the number of optimal arms is greater than 1, but we observed empirically that the leader in the sampling rule can oscillate between the multiple best arms. Due to this oscillating behavior, it is not clear whether the empirical proportion will converge towards one of the (multiple) vectors of optimal weights. We note that only a few algorithms in the literature manage to attain asymptotic optimality on instances for which there are multiple optimal allocations. We note that it follows from Theorem 2 that the asymptotic sample complexity is at most a factor 2|i_\star(\mu)| from the optimal one.
>
> **7) Using any-delta methods to construct UCB and LCB indexes in LUCB.**
> If by “any-delta” methods, you mean using UCB and LCB that are independent of $\delta$ (such as the one used in regret minimizing algorithms, only depending on the current time step $t$), no algorithm of this kind has ever been proved to be ($\varepsilon,\delta$)-PAC for best arm identification. In confidence-based BAI algorithms, the confidence intervals are calibrated such that the probability that there exists some $t$ and some arm $i$ such that $\mu_i$ is not in $[LCB_i(t),UCB_i(t)]$ is smaller than $\delta$. As such, the confidence bounds have to depend on $\delta$.
>
> **8) Lemma 9.**
> Typo in line 530: for all $j \ne i$, we have $\mu_j$ - $\epsilon$ = $\mu_{\epsilon} $(i)j, and $\mu_{i}$ = $\mu_{\varepsilon}(i)_{i}$.
>
> **9)  Lemma 12.**
> We will add that Lemma 12 was used by previous asymptotic analysis of Top Two algorithms, e.g. Lemma 3 in [31], Lemma 5 in [35] or Lemma 14 in [20].
>
> **10) Generalizing the proofs to exponential families.**
> It is possible to adapt the EB-TC$\varepsilon$ algorithm to a given single-parameter exponential family, by using the corresponding expression of the transportation cost [21] and the adaptive proportions given by IDS [38]. Obtaining asymptotic $\beta$-optimality in $\varepsilon$-BAI for more general exponential families should be doable (up to technicalities) by adapting the existing proofs for BAI, see e.g. [20]. However, even for $\epsilon=0$, it is still an open problem to (1) show asymptotic optimality with IDS and (2) obtain finite-confidence guarantees for distributions other than Gaussians. Our proofs leverage Gaussian-specific properties for the transportation costs: they can be expressed as the product between a function of the gap and a function of the allocation.
> Regarding the results presented in Section 4 (Theorem 3 and Corollary 1), it is actually not clear to what extent our theoretical/empirical results can be improved by adapting the algorithm to take into account the family of sub-Gaussian distributions itself. The simplicity of EB-TC$_{\varepsilon}$ is a key element that allows us to control the probability of error for all $\epsilon$ and all time $n$.

---

> > ### Comment · Reviewer_6z6m · 2023-08-11
> > **Response to rebuttal by authors**
> >
> > Thank you for your response. I acknowledge reading the entire thread of reviews and corresponding rebuttals. For now, I don't have further questions and I'll keep my score unchanged.

---

### Official Review · Reviewer_B3zV · 2023-07-14

**Soundness:** 3 good
**Presentation:** 2 fair
**Contribution:** 3 good
**Rating:** 5
**Confidence:** 2

**Summary:**

In this paper, the authors proposed EB-TC_epsilon, a novel sampling rule for epsilon-best arm identification. While the proposed algorithm is analyzed for approximate best arm identification, it is also an anytime sampling rule that can be used for fixed confidence, fixed budget, or anytime settings. The algorithm is analyzed for those settings and is empirically evaluated with numerical simulations.

**Strengths:**

1. The proposed algorithm is simple and easy to implement. Without modification, it works for approximate best arm identification, fixed confidence, fixed budget, or anytime settings.

2. The proposed algorithm is theoretically analyzed and empirically evaluated with numerical simulations. The results show that the proposed algorithm can achieve competitive performance compared to other methods.

**Weaknesses:**

1. The presentation of this paper can be improved. I found it hard to follow this paper in its current form, and I can not discuss its contribution with confidence.
   (a) The main results and theoretical discussions are mixed with related works. While it is nice to relate the proposed algorithms with previous works, it is hard to see the difference/improvement without clearly presenting previous results -- many of the previous algorithms are only cited by name and without further explanation.
   (b) Many of the high-level discussions and insights are also mixed with mathematical derivations, which makes it hard to follow. It would be nice to focus more on the main results and insights and present the details in the appendix.

2. The algorithm presentation is not well organized. The algorithm is introduced in line 120 and a crucial quantity $\bar \beta_{n+1}$ is not defined until line 140. It would be nice to define all the quantities before or when introducing the algorithm.

**Questions:**

What is the motivation for proposing one algorithm that works for approximate best arm identification and works, without modification, for fixed confidence, fixed budget, or anytime settings? There are other algorithms proposed specifically for different settings, and one might choose an algorithm based on the setting. It would be nice to see some discussion on this.

**Limitations:**

Yes.

---

> ### Author Rebuttal · Authors · 2023-08-07
>
>
> We thank the reviewer for the feedback and questions. We hope to address them in the following.
>
> We first answer the points raised in the “weaknesses” section.
>
> **Presentation of the results.**
> We choose to intertwine the presentation of our results and some related work in order to better contextualize our contributions and compare them with existing work. While Theorem 2 is already followed with a precise comparison with existing bounds (l. 230-237), after Theorem 1 we will remind the reader of other algorithms that are asymptotically optimal for $\varepsilon$-BAI in Gaussian bandits, emphasizing that none of them have non asymptotic guarantees of any kind. And after Theorem 3 we will explicitly state the expression of $H(\varepsilon)$ featured in the bound of [39].
>
> **Presentation of the algorithm.**
> We will add the definition of $\bar{beta}_{n+1}(i,j)$ explicitly in Figure 1. Figure 1 serves as a brief summary of the sampling rule. More details on the algorithms is given in Section 2, including the notation used in Figure 1.
>
> We now answer your other question.
>
> **Motivation for an $\varepsilon$-BAI algorithm having guarantees in fixed confidence, fixed budget, or anytime settings simultaneously.**
> Typical applications of (approximate) best arm identification are in A/B/n testing: identify a promising treatment in a clinical trial by giving treatments to patients, or a version of a website that generates sufficient revenue on a marketing platform. In case we opt for a fixed-confidence algorithm but for some practical reasons (no more funding to include more patients, company decision that the new version has to be deployed now) the test has to be stopped before the stopping condition is reached, it is desirable to still have guarantees for the recommended arm (treatment, webpage). Our anytime guarantees in that case say that despite stopping early, our error probability cannot be much worse than the error that would have been obtained using uniform sampling with that budget –and uniform sampling is still a common practical choice for A/B testing. In case we opt for a fixed-budget algorithm, state-of-the-art algorithms (SH,SR) are very sensitive to the budget T, which is used to decide the size of their different phases. These algorithms come with no guarantees on the error probability if early stopping occurs. If on the contrary we are given an additional testing budget (new funding secured due to promising results, more time before deploying the new version) it is not clear how to make good use of it, besides restarting these elimination-based algorithms. Our algorithm, conversely, would have guarantees whatever the budget ends up being, and we could even decide to couple it with a fixed-confidence stopping rule, for any confidence level $\delta$. We think that it is valuable to have a  simple sampling rule which is robust to some changes in the setting tackled, and also performs very well empirically in fixed-confidence, fixed-budget and anytime settings. While EB-TC$_\varepsilon$ is optimal in the asymptotic fixed-confidence setting when combined with an appropriate stopping rule, we acknowledge that we did not prove that it is state-of-the-art in the anytime (fixed-budget) setting, as the simple regret bound proved for Doubling SH is smaller. However, the practical story seems different, and Doubling SH comes with no guarantees in the fixed-confidence setting.

---

> > ### Comment · Reviewer_B3zV · 2023-08-20
> > **Response**
> >
> > Thank you for the response and the clarification on the motivation. I have raised my score to 5. Please consider adding this discussion on motivation in future revisions.

---

### Author Rebuttal · Authors · 2023-08-07

We attach additional experiments to better answer the questions of the reviewer 6z6m.

---

### Decision · Program_Chairs · 2023-09-21

**Decision:**

Accept (poster)

**Comment:**

Reviewers are very positive after rebuttal. A very novel contribution to epsilon best arm identification and algorithm is versatile and applicable to fixed confidence, fixed budget and anytime settings. This is a solid technical contribution to the bandit literature.